# Quantum Phases and Transitions in Spin Chains with Non-Invertible Symmetries

Arkya Chatterjee ⓘ *⁵, Ömer M. Aksoy ⓘ *ᵗ, Xiao-Gang Wen ⓘ

Department of Physics, Massachusetts Institute of Technology, Cambridge, Massachusetts 02139, USA

* These authors contributed equally to this work.
⁵ achatt@mit.edu    ᵗ omaksoy@mit.edu

## Abstract

Generalized symmetries often appear in the form of emergent symmetries in low energy effective descriptions of quantum many-body systems. Non-invertible symmetries are a particularly exotic class of generalized symmetries, in that they are implemented by transformations that do not form a group. Such symmetries appear generically in gapless states of quantum matter constraining the low-energy dynamics. To provide a UV-complete description of such symmetries, it is useful to construct lattice models that respect these symmetries exactly. In this paper, we discuss two families of one-dimensional lattice Hamiltonians with finite on-site Hilbert spaces: one with (invertible) $S_3$ symmetry and the other with non-invertible $\mathsf{Rep}(S_3)$ symmetry. Our models are largely analytically tractable and demonstrate all possible spontaneous symmetry breaking patterns of these symmetries. Moreover, we use numerical techniques to study the nature of continuous phase transitions between the different symmetry-breaking gapped phases associated with both symmetries. Both models have self-dual lines, where the models are enriched by so-called intrinsically non-invertible symmetries generated by Kramers-Wannier-like duality transformations. We provide explicit lattice operators that generate these non-invertible self-duality symmetries. We show that the enhanced symmetry at the self-dual lines is described by a 2+1d symmetry-topological-order (SymTO) of type $\mathsf{JK}_4 \boxtimes \overline{\mathsf{JK}}_4$. The condensable algebras of the SymTO determine the allowed gapped and gapless states of the self-dual $S_3$-symmetric and $\mathsf{Rep}(S_3)$-symmetric models.

# 1  Introduction

In quantum many-body physics, global internal symmetries are conventionally represented by unitary (or anti-unitary) operators acting on all the degrees of freedom constituting the system. The associated symmetry transformations can be composed following group-like multiplication rules. Symmetries allow the decomposition of quantum states into sectors that are dynamically decoupled, *i.e.*, under time-evolution generated by the Hamiltonian, states in a particular sector do not develop overlaps with those in other sectors. These sectors are labeled by symmetry charges, which correspond to irreducible representations of the symmetry group.

This conventional picture of symmetries has been generalized considerably in recent years. Such "generalized symmetries" have been studied from various perspectives in the mathematical physics, high energy physics, and condensed matter physics communities for several decades. However, attempts at a unified understanding are more recent. One class of generalized symmetries that is particularly exotic and will form the main focus of this work are the so-called non-invertible symmetries, which were first studied in rational conformal field theories in the form of topological defect lines [1,2].[1] The composition of these symmetry transformations are, in general, not described by a group, but by a fusion category (in the case that there are a finite number of them) [3].

Generalized symmetries may also be implemented by operators with support on higher co-dimension manifolds in spacetime [4,5], instead of co-dimension 1 manifolds. These lead to a different generalization, known as higher-form symmetries [6,7]. For finite symmetries in $d + 1$-dimensional spacetime, invertible and non-invertible 0- and higher-form symmetries (also known as algebraic higher symmetries [8]), are understood to be unified in the structure of a fusion $d$-category $\mathcal{C}$ [9]. Anomaly-free algebraic higher symmetries are classified by *local* fusion $d$-categories [8]. In this work, we will focus only on non-invertible 0-form symmetries in $1 + 1$d so that the relevant mathematical structure is that of ordinary fusion categories [10]. Symmetry operators in a fusion category $\mathcal{C}$ are labeled by the objects $\mathsf{a}$ of $\mathcal{C}$. All objects can be decomposed into a direct sum of finitely many simple objects,[2] so our labels can be allowed to take values in the set of simple objects without loss of generality. The composition of two symmetry operators $\widehat{W}_{\mathsf{a}}$ and $\widehat{W}_{\mathsf{b}}$, labeled by simple objects $\mathsf{a}$ and $\mathsf{b}$, can be decomposed in terms of simple objects as

$$\widehat{W}_{\mathsf{a}} \widehat{W}_{\mathsf{b}} = \sum_{\alpha} \mathsf{N}_{\mathsf{a}\,\mathsf{b}}^{\mathsf{c}} \widehat{W}_{\mathsf{c}}, \qquad (1.1)$$

where non-negative integers $\mathsf{N}_{\mathsf{a}\,\mathsf{b}}^{\mathsf{c}}$ are known as *fusion coefficients*.

Non-invertible symmetries have found various applications in the context of continuum QFTs – comprehensive lists of references can be found, for example, in Refs. [12,13]. Realizations of non-invertible symmetries in lattice models and associated phases of matter are far less understood; however, see the discussion on related prior work in the next subsection. While the infrared (IR) limit of many-body quantum systems are often described by effective QFTs, which can accommodate non-invertible symmetries, their naïve lattice regularization may break emergent symmetries of the continuum theory (see, *e.g.*, Ref. [14]). In this spirit, it is desirable to construct spin chains that respect generalized symmetries, putting them on the same footing as ordinary symmetries. To be precise, by "spin chains" here we mean local Hamiltonians acting on a Hilbert space that is a tensor product of finite-dimensional local Hilbert spaces.

---

[1]In this context, the word "defect" simply refers to the fact that these are extended operators in the theory.

[2]This is because fusion categories are semi-simple [11].

In this paper, we explore the possible gapped or gapless phases and continuous phase transitions realized in spin chains with non-invertible symmetries. In particular, we study the example of the smallest anomaly-free non-invertible symmetry category: $\mathsf{Rep}(S_3)$. Building up to that, in Sec. 2 we introduce a spin chain with $S_3$ symmetry constructed out of qubit and qutrit degrees of freedom. In Sec. 3, we show how gauging either the entire $S_3$ symmetry or its non-normal $\mathbb{Z}_2$ subgroup delivers a spin chain with $\mathsf{Rep}(S_3)$ symmetry. By studying appropriate limits, we identify fixed-point ground states corresponding to the four distinct $\mathsf{Rep}(S_3)$ spontaneous symmetry breaking (SSB) patterns. We explore the phase diagrams of both spin chains using tensor network algorithms to verify the analytical predictions. Section 4 is a synthesis of the salient aspects of our results from the point of view of the symmetry-topological-order (SymTO) framework. In Sec. 5, we discuss connections of our results with more abstract approaches, propose order parameters that detect SSB patterns in our models, and comment on an incommensurate gapless phase that our numerical calculations reveal. We close with some comments on directions for future exploration in Sec. 6.

## Relation to prior work

The literature on non-invertible symmetries has a long history. Topological defect lines in $1+1$d rational conformal field theories (CFTs) have been studied since the 1980s [1,2,15–18]. A general study of topological defects in topological quantum field theories (TQFTs) was carried out in [19–21]; see Ref. [22] for a recent review. A study of invertible defects of various dimensions in the context of general quantum field theories was carried out in great detail in Refs. [6,23] under the name of higher-form symmetries. It is interesting to note that, Refs. [4,5] earlier discussed lattice analogues of higher-form symmetry transformations in the context of topologically ordered phases of quantum matter. Finite non-invertible symmetries in $1+1$d and their anomalies were systematically studied in Refs. [10,24,25], and constraints on RG flows obtained in Ref. [3].

Parallel to these developments, (non-invertible) gravitational anomalies were classified by topological orders in one higher dimension in Ref. [26].[3] An isomorphic holographic decomposition of a quantum field theory was introduced in Ref. [27] to expose its hidden gravitational anomaly. It was later realized that a subclass of non-invertible gravitational anomalies are nothing but generalized symmetries [8,28–30]. The aforementioned isomorphic holographic decomposition can be re-interpreted as a holographic theory of generalized symmetry (see Fig. 9), which was described via the "sandwich" construction in Ref. [9]. This holographic description of symmetries was also discovered in the context of superstring theory [31]. In the holographic approach, symmetry data is stored in a non-invertible field theory (or a topological order) in one higher dimension such that the physical theory with generalized symmetries is realized as a boundary theory of the former. This idea has various names in different parts of the theoretical physics community: symmetry-topological-order (SymTO) correspondence [8,27,30],[4] symmetry-topological-field-theory (SymTFT) [31], topological symmetry [32], or topological holography [33–35].

The holographic approach has many applications. It leads to a classification of anomaly-free generalized symmetries using local fusion higher categories $\mathcal{C}^\vee$ that describe the fusion

---

[3]In this context, a gravitational anomaly is an obstruction to realizing a $d$-dimensional theory in a $d$ dimensional Hamiltonian lattice model on a tensor product Hilbert space. If such a theory is realizable on the boundary of a non-invertible, or invertible, topological order defined on a tensor product Hilbert space in $d+1$ dimensions, the gravitational anomaly is referred to as non-invertible, or invertible, respectively.

[4]SymTO was referred to as "categorical symmetry" in some early papers [8,29,30]. In current literature, categorical symmetry usually refers to non-invertible symmetry (referred to as algebraic higher symmetry in Ref. [8]). See Appendix B for a brief review of SymTO.

of corresponding charged operators, which may be of arbitrary dimensionality [8,30]. There is also an equivalent classification by local fusion higher categories $\mathcal{C}$ describing the fusion of the symmetry defects, instead of the charged operators. This approach also provides a classification of invertible anomalies[5] for generalized symmetries in any dimensions [8]. A related discussion on the classification of gravitational anomalies and anomalies of group-like symmetries can be found in Refs. [26,36]). Some anomalous non-invertible symmetries were also studied in Refs. [37,38]. Generalized symmetries, anomalous or not, are classified (up to holo-equivalence) by their SymTO in one higher dimension [8].

More importantly, the holographic approach allows the use of emergent generalized symmetries to constrain compatible gapless liquid states. In Refs. [39–42], the concept of topological Wick rotation was introduced to describe the canonical gapless liquids determined by a generalized symmetry, via the gapless boundary of the corresponding SymTO that has no anyon condensation. Such canonical gapless liquids for a SymTO were studied in Refs. [29,43–45] using the holographic modular bootstrap. The condensable algebras [46,47] of the SymTO classify the different allowed phases [44,48]. As long as the condensable algebra is non-Lagrangian, the corresponding state must be gapless [8,44]. The Lagrangian condensable algebras, on the other hand, classify the gapped states allowed by the SymTO. Such a classification includes SSB, symmetry-protected topological (SPT), and symmetry-enriched topological (SET) phases [8,30]. Related discussions about phases of $1+1$d systems can also be found in Refs. [49,50].

It is worthwhile to note here that generalized symmetries can also be viewed from the perspective of the algebra of a subset of all local operators. Given a set of symmetry transformations, the subset of local operators invariant under these transformations forms the algebra of local symmetric operators (also called a bond algebra [51,52]). One can turn this idea on its head and consider subsets of local operators as (indirectly) defining a generalized symmetry, provided the subset forms an algebra. Ref. [53] took this point view and showed that isomorphic algebras of local symmetric operators correspond one-to-one to topological orders in one higher dimension, by considering simple examples. The commutant algebra of the subset of local operators contains operators that implement (generalized) symmetry transformations [54,55]. The structure of commutant algebras is rich enough to include the above-mentioned non-invertible symmetries. Notably, this structure is less rigid than that of fusion (higher) categories since the fusion coefficients need not be non-negative integers. Making contact between the commutant algebraic approach and the topological defect approach of generalized symmetries is an interesting open question.

As an instance of generalized symmetries, higher-form symmetries have found various applications in condensed matter physics; see Ref. [56] for a recent review. For instance, it was found that even if microscopic lattice models do not have exact higher-form symmetries, they can appear as emergent [57], or even *exact emergent* [58,59], symmetries at low energies. In many ways, higher-form symmetries behave just like ordinary symmetries: they can be spontaneously broken leading to degenerate ground states or Goldstone bosons [5,60], depending on whether the symmetry is discrete or continuous; they can have 't Hooft anomalies themselves, or have mixed 't Hooft anomalies with crystalline symmetries leading to Lieb-Schultz-Mattis (LSM)-type theorems [61]; they can lead to new symmetry protected topological (SPT) phases [7,62]. A generic way to construct models with higher-form symmetries in $2+1$d and higher, is via gauging (some subgroup of) an ordinary symmetry [29,63].

Non-invertible symmetries also have a natural place in the condensed matter set-

---

[5]Invertible ('t Hooft) anomalies are those for which the anomaly theory in one higher dimension is an invertible topological field theory.

ting. For instance, Ref. [64] showed that these symmetries appear generically as emergent symmetries in SSB phases of ordinary symmetries. Another generic way to realize non-invertible symmetries is to start from a model with 0-form non-Abelian (finite) $G$ symmetry and gauge this symmetry. In the resulting gauge theory, in $d + 1$ spacetime dimensions, the Wilson loops obey the fusion rules dictated by the representations of $G$, forming the layer of $(d - 1)$-morphisms of a fusion $d$-category, $d$-$\mathsf{Rep}(G)$. The fusion rules of the Wilson loops are not group-like whenever $G$ is non-Abelian. This strategy was used to construct various lattice models with non-invertible symmetries [8, 29, 65, 66]. Another class of examples can be obtained through the so-called half-gauging scheme. Namely, if gauging an invertible symmetry of a theory $\mathcal{T}$ produce an isomorphic dual symmetry, a defect constructed by gauging this symmetry on one half of spacetime, the interface can be thought of as a non-invertible self-duality defect [67–69]. Building on this, it is also possible to construct new duality by half-gauging non-invertible symmetries [70, 71]. On a related note, statistical mechanical models with general fusion category symmetries were proposed and studied in Refs. [72, 73]. Recently, non-invertible self-duality symmetries in Hamiltonian lattice models have also been obtained by gauging internal symmetries that participate in a mixed anomaly with translation symmetry such as in the case of LSM anomalies [74–76].

There has been an exciting flurry of recent work [8, 30, 66, 76–83] exploring phases of matter with fusion category symmetries. A generalized Landau paradigm [44, 45, 84], classifying both gapped and gapless phases in systems with general fusion category symmetries in 1+1d has been formulated based on condensable algebras in the symmetry topological order (SymTO) in one higher dimension. Our work contributes to this rapidly developing literature by exploring simple examples in the spin chain context. The key results of this paper are summarized below.

## Summary of key results

(i) We show, through a microscopic calculation, that our spin chain (2.3) with $S_3$ symmetry is dual to the $\mathsf{Rep}(S_3)$-symmetric spin chain (3.13) by gauging a $\mathbb{Z}_2$ subgroup of $S_3$. As a consequence, phase diagrams of these spin chains can be mapped to each other in a one-to-one manner.

(ii) We find gapped phases realizing all four SSB patterns of both $S_3$ and $\mathsf{Rep}(S_3)$ symmetries which correspond to the four inequivalent module categories over the corresponding fusion categories [10]. We define order and disorder operators whose non-vanishing expectation values can be used to distinguish different SSB patterns. For the non-invertible $\mathsf{Rep}(S_3)$ symmetry, the SSB is detected by string order parameters as opposed to the invertible $S_3$ symmetry.

(iii) We show that for special subspaces in the parameter space, our spin chains are both invariant under an exact, intrinsic [85, 86], non-invertible self-duality symmetry. We provide the lattice operators that implement the respective self-duality symmetry in the form of a sequential circuit.[6] In particular, for the $\mathsf{Rep}(S_3)$-symmetric model (3.13), this circuit implements a self-duality symmetry associated with gauging $\mathsf{Rep}(S_3)$ by the algebra object $\mathbf{1} \oplus \mathbf{2}$.

(iv) For both spin chains, the four gapped phases meet at a multi-critical point that is symmetric under the respective non-invertible self-duality symmetry. For each multi-critical point, we identify three relevant perturbations, two of which break the

---

[6]Notably, Ref. [87] considered this class of sequential circuits as maps between distinct gapped phases. See also Refs. [74, 88] for closely related constructions.

non-invertible self-duality symmetry explicitly. In particular, for the $S_3$ spin chain, one of these relevant perturbations allows the realization of a (Landau-forbidden) direct continuous transition between SSB phases preserving $\mathbb{Z}_3$ and $\mathbb{Z}_2$ subgroups.

(v) We find an extended gapless region in the parameter space which is consistent with an incommensurate gapless phase. To better understand this gapless phase, we draw an analogy with an exactly solvable spin-1/2 chain with exact $\mathbb{Z}_2$ KW self-duality symmetry. This spin chain, also supports a gapless incommensurate phase with central charge $c = 1$ and an anomalous chiral U(1) symmetry[7] that emanates [89] from lattice translation symmetry.[8] This gapless phase is separated from the neighboring gapped phases by critical lines with dynamical critical exponent $z > 1$. We provide arguments on why this feature may be valid for incommensurate phases more generally. We also discover a new type of continuous transition in the incommensurate phase. The new continuous transition has the same number of gapless modes at and away from the transition point.[9] In our example, there is only one U(1) current $J_L$ for left-movers at the transition point, while there are two U(1) currents $J_L, J_R$ for for both left- and right-movers away from the transition point.

(vi) We identify the SymTO of our self-dual $S_3$ and $\mathsf{Rep}(S_3)$-symmetric spin chains to be the 2+1d $\mathrm{JK}_4 \boxtimes \overline{\mathrm{JK}}_4$ topological order. We also obtain possible phases and phase transitions of the self-dual spin chains from the allowed boundary conditions of the SymTO. In particular, we show that the self-dual spin chains do not allow gapped non-degenerate ground states, consistent with the fact that the non-invertible self-duality symmetries are anomalous.

**Note added:** While this manuscript was being completed, we became aware of potentially overlapping work in Refs. [92, 93]; we thank the authors for coordinating their arXiv submission with us.

## 2 $S_3$-symmetric spin chain

### 2.1 Definitions

We consider lattice $\Lambda$ in one spatial dimension with $|\Lambda| = L$ sites. We associate a tensor product Hilbert space $\mathcal{H}$ with lattice $\Lambda$, where the each site $i \in \Lambda$ supports an on-site Hilbert space $\mathcal{H}_i$ that is 12-dimensional. We label the orthonormal basis vectors spanning $\mathcal{H}_i$ by a $\mathbb{Z}_2 \times \mathbb{Z}_2 \times \mathbb{Z}_3$-valued triplet $(a_i, b_i, c_i)$, *i.e.*,

$$\mathcal{H} = \otimes_{i=1}^{L} \mathcal{H}_i, \quad \mathcal{H}_i = \mathrm{span}\{|a_i, b_i, c_i\rangle \,|\, a_i \in \mathbb{Z}_2, \, b_i \in \mathbb{Z}_2, \, c_i \in \mathbb{Z}_3\}. \tag{2.1}$$

On each local Hilbert space $\mathcal{H}_i$, we define $\mathbb{Z}_2$ (qubit) and $\mathbb{Z}_3$ (qutrit) clock operators that satisfy the algebras

$$\hat{\sigma}_i^z \, \hat{\sigma}_j^x = (-1)^{\delta_{ij}} \hat{\sigma}_j^x \, \hat{\sigma}_i^z, \qquad\qquad (\hat{\sigma}_i^z)^2 = (\hat{\sigma}_i^x)^2 = \hat{\mathbb{1}}, \tag{2.2a}$$

$$\hat{\tau}_i^z \, \hat{\tau}_j^x = (-1)^{\delta_{ij}} \, \hat{\tau}_j^x \, \hat{\tau}_i^z, \qquad\qquad (\hat{\tau}_i^z)^2 = (\hat{\tau}_i^x)^2 = \hat{\mathbb{1}}, \tag{2.2b}$$

---

[7]In the low-energy CFT description, only the left-movers carry the U(1) charge while the right-movers are described by two branches of Majorana fermion fields with different velocities.

[8]Along the self-dual line, the Majorana representation of translation symmetry carries an LSM anomaly since there are odd number of Majorana degrees of freedom per unit cell, see Refs. [74, 90, 91].

[9]The number of gapless mode can be measured experimentally, via the the thermal conductance of short clean sample.

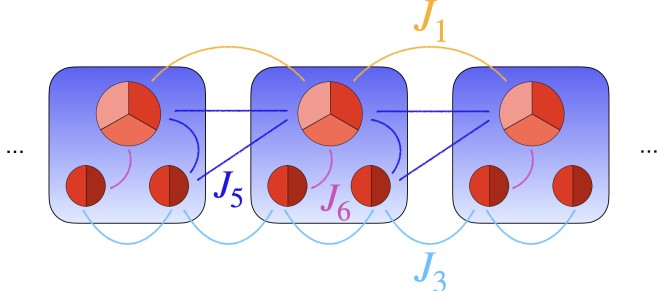

Figure 1: Schematic of the Hamiltonian (2.3) showing the couplings between qutrit (depicted by a tripartitioned disk) and qubit (depicted by a bipartitioned disk) degrees of freedom. Single-body terms $J_2$, $J_4$ are suppressed.

$$\widehat{Z}_i\,\widehat{X}_j = \omega^{\delta_{ij}}\widehat{X}_j\,\widehat{Z}_i, \qquad\qquad \left(\widehat{Z}_i\right)^3 = \left(\widehat{X}_i\right)^3 = \widehat{\mathbb{1}}, \qquad (2.2c)$$

with $\omega = \exp\{\mathrm{i}2\pi/3\}$ such that the operators $\hat{\sigma}_i^z$, $\hat{\tau}_i^z$, and $\widehat{Z}_i$ are diagonal in the basis (2.1), *i.e.*,

$$\hat{\sigma}_i^z\,|\boldsymbol{a},\boldsymbol{b},\boldsymbol{c}\rangle = (-1)^{a_i}\,|\boldsymbol{a},\boldsymbol{b},\boldsymbol{c}\rangle, \quad \hat{\tau}_i^z\,|\boldsymbol{a},\boldsymbol{b},\boldsymbol{c}\rangle = (-1)^{b_i}\,|\boldsymbol{a},\boldsymbol{b},\boldsymbol{c}\rangle, \quad \widehat{Z}_i\,|\boldsymbol{a},\boldsymbol{b},\boldsymbol{c}\rangle = \omega^{c_i}\,|\boldsymbol{a},\boldsymbol{b},\boldsymbol{c}\rangle. \qquad (2.2d)$$

We impose periodic boundary conditions on the Hilbert space $\mathcal{H}$ by identifying operators at site $i$ with those at site $i+L$

$$\widehat{X}_{i+L} \equiv \widehat{X}_i, \quad \widehat{Z}_{i+L} \equiv \widehat{Z}_i, \quad \hat{\sigma}_{i+L}^x \equiv \hat{\sigma}_i^x, \quad \hat{\sigma}_{i+L}^z \equiv \hat{\sigma}_i^z, \quad \hat{\tau}_{i+L}^x \equiv \hat{\tau}_i^x, \qquad \hat{\tau}_{i+L}^z \equiv \hat{\tau}_i^z. \quad (2.2e)$$

Our starting point is the Hamiltonian

$$\widehat{H}_{S_3} := \widehat{H}_{\mathrm{P}} + \widehat{H}_{\mathrm{I}} + \widehat{H}_{\mathrm{PI}}, \qquad (2.3a)$$

$$\widehat{H}_{\mathrm{P}} := -J_1 \sum_{i=1}^{L}\left(\widehat{Z}_i\,\widehat{Z}_{i+1}^\dagger + \widehat{Z}_i^\dagger\,\widehat{Z}_{i+1}\right) - J_2 \sum_{i=1}^{L}\left(\widehat{X}_i + \widehat{X}_i^\dagger\right), \qquad (2.3b)$$

$$\widehat{H}_{\mathrm{I}} := -J_3 \sum_{i=1}^{L}\left(\hat{\sigma}_i^z\,\hat{\tau}_i^z + \hat{\tau}_i^z\,\hat{\sigma}_{i+1}^z\right) - J_4 \sum_{i=1}^{L}\left(\hat{\sigma}_i^x + \hat{\tau}_i^x\right), \qquad (2.3c)$$

$$\widehat{H}_{\mathrm{PI}} := -J_5 \sum_{i=1}^{L}\mathrm{i}\,\hat{\tau}_i^z\left(\widehat{Z}_i\,\widehat{Z}_{i+1}^\dagger - \widehat{Z}_i^\dagger\,\widehat{Z}_{i+1}\right) - J_6 \sum_{i=1}^{L}\mathrm{i}\,\hat{\sigma}_i^z\left(\widehat{X}_i - \widehat{X}_i^\dagger\right), \qquad (2.3d)$$

with six positive coupling constants $J_i > 0$ for $i = 1,\cdots,6$. Hamiltonians $\widehat{H}_{\mathrm{P}}$ and $\widehat{H}_{\mathrm{I}}$ describe the quantum three-state Potts model on a chain of $L$ sites and transverse-field Ising model defined on $2L$ sites, respectively [10]. The last Hamiltonian $\widehat{H}_{\mathrm{PI}}$ then describes the coupling between qubits and qutrits. A schematic description of the couplings in Hamiltonian (2.3) is shown in Fig. 1.

Hamiltonian (2.3) is invariant under an $S_3$ symmetry generated by the unitary operators

$$\widehat{U}_r := \prod_{i=1}^{L}\widehat{X}_i, \qquad \widehat{U}_s := \prod_{i=1}^{L}\hat{\sigma}_i^x\,\hat{\tau}_i^x\,\widehat{C}_i, \qquad (2.4)$$

---

[10]The reason for choosing number of qubits to be twice that of qutrits will be clear in Sec. 2.3 when we discuss the self-dual points in the phase diagram of Hamiltonian (2.3).

where $\widehat{C}_i := \sum_{\alpha=0}^{2} \widehat{X}_i^{\alpha} \widehat{P}^{Z_i=\omega^{\alpha}}$ and $\widehat{P}^{Z_i=\omega^{\alpha}}$ is the projector onto the subspace of $\widehat{Z}_i = \omega^{\alpha}$. The operator $\widehat{C}_i$ implements the charge conjugation on the qutrits, $i.e.$, it maps $\widehat{X}_i \mapsto \widehat{X}_i^{\dagger}$ and $\widehat{Z}_i \mapsto \widehat{Z}_i^{\dagger}$. On the local operators, the $S_3$ symmetry generators, $\widehat{U}_r$ and $\widehat{U}_s$, implement the transformations

$$
\begin{aligned}
\widehat{U}_r \left( \widehat{X}_i \quad \widehat{Z}_i \quad \hat{\sigma}_i^x \quad \hat{\sigma}_i^z \quad \hat{\tau}_i^x \quad \hat{\tau}_i^z \right) \widehat{U}_r^{\dagger} &= \left( +\widehat{X}_i \quad \omega^2\, \widehat{Z}_i \quad +\hat{\sigma}_i^x \quad +\hat{\sigma}_i^z \quad +\hat{\tau}_i^x \quad +\hat{\tau}_i^z \right), \\
\widehat{U}_s \left( \widehat{X}_i \quad \widehat{Z}_i \quad \hat{\sigma}_i^x \quad \hat{\sigma}_i^z \quad \hat{\tau}_i^x \quad \hat{\tau}_i^z \right) \widehat{U}_s^{\dagger} &= \left( +\widehat{X}_i^{\dagger} \quad +\widehat{Z}_i^{\dagger} \quad +\hat{\sigma}_i^x \quad -\hat{\sigma}_i^z \quad +\hat{\tau}_i^x \quad -\hat{\tau}_i^z \right),
\end{aligned}
\tag{2.5}
$$

respectively. Any operator that commutes with $\widehat{U}_r$ and $\widehat{U}_s$ can be written as linear combinations of products of eight local operators. These are precisely those that appeared in the Hamiltonian (2.3). Accordingly we define the bond algebra [51, 52] of $S_3$-symmetric operators

$$
\begin{aligned}
\mathfrak{B}_{S_3} := \Big\langle \hat{\sigma}_i^z\, \hat{\tau}_i^z,\ \hat{\tau}_i^z\, \hat{\sigma}_{i+1}^z,\ \hat{\sigma}_i^x,\ \hat{\tau}_i^x,\ \left( \widehat{X}_i + \widehat{X}_i^{\dagger} \right),\ \left( \widehat{Z}_i\, \widehat{Z}_{i+1}^{\dagger} + \widehat{Z}_i^{\dagger}\, \widehat{Z}_{i+1} \right), \\
\hat{\sigma}_i^z \left( \widehat{X}_i - \widehat{X}_i^{\dagger} \right),\ \hat{\tau}_i^z \left( \widehat{Z}_i\, \widehat{Z}_{i+1}^{\dagger} - \widehat{Z}_i^{\dagger}\, \widehat{Z}_{i+1} \right) \Big| i \in \Lambda \Big\rangle.
\end{aligned}
\tag{2.6}
$$

We identify the $S_3$ symmetry as the commutant algebra of $\mathfrak{B}_{S_3}$, $i.e.$, algebra of all operators that commute with all elements of $\mathfrak{B}_{S_3}$.

In Sections 2.2, 3.1, and 3.2, we are going to construct the dual bond algebras $\mathfrak{B}_{S_3/\mathbb{Z}_3}$, $\mathfrak{B}_{S_3/\mathbb{Z}_2}$, and $\mathfrak{B}_{S_3/S_3}$, that are delivered by gauging the subgroups $\mathbb{Z}_3$, $\mathbb{Z}_2$, and $S_3$, respectively. As we shall see, the precise statement of the duality will then be expressed as isomorphisms between appropriately defined "symmetric" subalgebras of these bond algebras. Therein, for each dual bond algebra, we will identify the corresponding commutant algebras, $i.e.$, the corresponding dual symmetry structure.

## 2.2   Gauging $\mathbb{Z}_3$ subgroup: non-invertible self-duality symmetry

Gauging the $\mathbb{Z}_3$ subgroup is achieved in two steps. First, on the each link between sites $i$ and $i+1$, we introduce $\mathbb{Z}_3$ clock operators $\{\hat{x}_{i+1/2},\, \hat{z}_{i+1/2}\}$ that satisfy the algebra

$$
\begin{aligned}
\hat{z}_{i+1/2}\, \hat{x}_{j+1/2} &= \omega^{\delta_{ij}} \hat{x}_{j+1/2}\, \hat{z}_{i+1/2}, \quad \left( \hat{z}_{i+1/2} \right)^3 = \left( \hat{x}_{i+1/2} \right)^3 = \hat{\mathbb{1}}, \\
\hat{z}_{i+1/2+L} &= \hat{z}_{i+1/2}, \quad \hat{x}_{i+1/2+L} = \hat{x}_{i+1/2},
\end{aligned}
\tag{2.7a}
$$

where we imposed periodic boundary conditions. This enlarges the dimension of the Hilbert space (2.1) by a factor of $3^L$. Second, we define the Gauss operators on every site

$$
\widehat{G}_i^{\mathbb{Z}_3} := \hat{z}_{i-1/2}^{\dagger}\, \widehat{X}_i\, \hat{z}_{i+1/2}, \qquad \left[ \widehat{G}_i^{\mathbb{Z}_3} \right]^3 = \hat{\mathbb{1}}.
\tag{2.7b}
$$

Hereby, the link operators $\hat{z}_{i+1/2}$ and $\hat{x}_{i+1/2}$ take the roles of $\mathbb{Z}_3$-valued electric field and $\mathbb{Z}_3$-valued gauge field, respectively. The physical Hilbert space consists of those states for which the Gauss constraint $\widehat{G}_i^{\mathbb{Z}_3} = 1$ is satisfied. Imposing each of one of the $L$ Gauss constraints reduces the dimension of the extended Hilbert space by a factor of $1/3$. The $S_3$-symmetric algebra (2.6) is not invariant under local gauge transformations. By minimally coupling it to the gauge field $\hat{x}_{i+1/2}$, we define the gauge invariant extended algebra

$$
\begin{aligned}
\mathfrak{B}_{S_3/\mathbb{Z}_3}^{\mathrm{mc}} := \Big\langle \hat{\sigma}_i^z\, \hat{\tau}_i^z,\ \hat{\tau}_i^z\, \hat{\sigma}_{i+1}^z,\ \hat{\sigma}_i^x,\ \hat{\tau}_i^x,\ \left( \widehat{X}_i + \widehat{X}_i^{\dagger} \right),\ \left( \widehat{Z}_i\, \hat{x}_{i+1/2}\, \widehat{Z}_{i+1}^{\dagger} + \widehat{Z}_i^{\dagger}\, \hat{x}_{i+1/2}^{\dagger}\, \widehat{Z}_{i+1} \right), \\
\hat{\sigma}_i^z \left( \widehat{X}_i - \widehat{X}_i^{\dagger} \right),\ \hat{\tau}_i^z \left( \widehat{Z}_i\, \hat{x}_{i+1/2}\, \widehat{Z}_{i+1}^{\dagger} - \widehat{Z}_i^{\dagger}\, \hat{x}_{i+1/2}^{\dagger}\, \widehat{Z}_{i+1} \right) \Big| \widehat{G}_i^{\mathbb{Z}_3} = 1, \quad i \in \Lambda \Big\rangle.
\end{aligned}
\tag{2.8}
$$

It is convenient to do a basis transformation to impose the Gauss constraint explicitly. To this end, we apply a unitary operator $\widehat{U}$ which implements the transformation

$$
\begin{aligned}
\widehat{U}\,\hat{\sigma}_i^x\,\widehat{U}^\dagger &= \hat{\sigma}_i^x, & \widehat{U}\,\hat{\sigma}_i^z\,\widehat{U}^\dagger &= \hat{\sigma}_i^z, \\
\widehat{U}\,\hat{\tau}_i^x\,\widehat{U}^\dagger &= \hat{\tau}_i^x, & \widehat{U}\,\hat{\tau}_i^z\,\widehat{U}^\dagger &= \hat{\tau}_i^z, \\
\widehat{U}\,\widehat{X}_i\,\widehat{U}^\dagger &= \hat{z}_{i-1/2}\,\widehat{X}_i\,\hat{z}_{i+1/2}^\dagger, & \widehat{U}\,\widehat{Z}_i\,\widehat{U}^\dagger &= \widehat{Z}_i, \\
\widehat{U}\,\hat{x}_{i+1/2}\,\widehat{U}^\dagger &= \widehat{Z}_i^\dagger\,\hat{x}_{i+1/2}\,\widehat{Z}_{i+1}, & \widehat{U}\,\hat{z}_{i+1/2}\,\widehat{U}^\dagger &= \hat{z}_{i+1/2}.
\end{aligned}
\tag{2.9}
$$

In particular, this unitary simplifies the Gauss operator to $\widehat{U}\,\widehat{G}_i^{\mathbb{Z}_3}\,\widehat{U}^\dagger = \widehat{X}_i$. After the unitary transformation, we project down to the $\widehat{X}_i = 1$ subspace and relabel the link degrees of freedom by $i+1/2 \mapsto i+1$ for notational simplicity. This delivers the dual bond algebra

$$
\begin{aligned}
\mathfrak{B}_{S_3/\mathbb{Z}_3} &:= \widehat{U}\,\mathfrak{B}_{S_3/\mathbb{Z}_3}^{\mathrm{mc}}\,\widehat{U}^\dagger\Big|_{\widehat{X}_i=1} \\
&= \Big\langle \hat{\sigma}_i^z\,\hat{\tau}_i^z,\; \hat{\tau}_i^z\,\hat{\sigma}_{i+1}^z,\; \hat{\sigma}_i^x,\; \hat{\tau}_i^x,\; \left(\hat{z}_i\,\hat{z}_{i+1}^\dagger + \hat{z}_i^\dagger\,\hat{z}_{i+1}\right),\; \left(\hat{x}_i + \hat{x}_i^\dagger\right), \\
&\qquad \hat{\sigma}_i^z\left(\hat{z}_i\,\hat{z}_{i+1}^\dagger - \hat{z}_i^\dagger\,\hat{z}_{i+1}\right),\; \hat{\tau}_i^z\left(\hat{x}_{i+1} - \hat{x}_{i+1}^\dagger\right)\;\Big|\; i \in \Lambda \Big\rangle.
\end{aligned}
\tag{2.10}
$$

We note that the dual bond algebra contains the same type of terms as algebra (2.6) and, hence, is the algebra of $S_3^\vee$-symmetric operators.[11] The generators of dual $S_3^\vee$ symmetry are represented by the unitary operators [12]

$$
\widehat{U}_r^\vee := \prod_{i=1}^{L} \hat{x}_i, \qquad \widehat{U}_s^\vee := \prod_{i=1}^{L} \hat{\sigma}_i^x\,\hat{\tau}_i^x\,\hat{c}_i, \qquad \hat{c}_i := \sum_{\alpha=0}^{2} \hat{x}_i^\alpha\,\widehat{P}^{z_i=\omega^\alpha}.
\tag{2.11}
$$

We note that the duality as we described does not hold between entirety of algebras $\mathfrak{B}_{S_3}$ and $\mathfrak{B}_{S_3/\mathbb{Z}_3}$. On the one hand, because we imposed periodic boundary conditions on the operators $\{\hat{x}_i,\,\hat{z}_i\}$, the product of all Gauss operators is equal to the generator of global $\mathbb{Z}_3$ transformations, i.e.,

$$
\prod_{i=1}^{L} \widehat{G}_i^{\mathbb{Z}_3} = \widehat{U}_r = 1.
\tag{2.12a}
$$

On the other hand, since we imposed periodic boundary conditions on the operators $\left\{\widehat{X}_i,\,\widehat{Z}_i\right\}$, the image of the product $\prod_{i=1}^{L} \widehat{Z}_i\,\widehat{Z}_{i+1}$, which is the dual $\mathbb{Z}_3^\vee$ symmetry generator, must be equal to identity, i.e.,

$$
\prod_{i=1}^{L} \widehat{Z}_i\,\widehat{Z}_{i+1} \equiv \widehat{U}_r^\vee = 1.
\tag{2.12b}
$$

---

[11]We use the superscript $\vee$ to differentiate the dual $S_3^\vee$ symmetry of the dual algebra (2.10) from the $S_3$ symmetry of the algebra (2.6).

[12]The dual symmetry $\widehat{U}_s^\vee$ is obtained from the operator $\widehat{U}_s$ by demanding the covariance of the Gauss operator $\widehat{G}_i^{\mathbb{Z}_3}$, i.e., demanding

$$
\widehat{U}_s^{\mathrm{mc}}\,\widehat{G}_i^{\mathbb{Z}_3}\left(\widehat{U}_s^{\mathrm{mc}}\right)^\dagger = \left(\widehat{G}_i^{\mathbb{Z}_3}\right)^\dagger,
$$

where $\widehat{U}_s^{\mathrm{mc}}$ is an operator acting on the extended Hilbert space and contains both site and link degrees of freedom. The dual symmetry $\widehat{U}_s^\vee$ is then obtained by applying the unitary transformation (2.9) and projecting to the $\widehat{X}_i = 1$ subspace.

Therefore, the duality holds when both conditions (2.12a) and (2.12b). In other words, the isomorphism

$$\mathfrak{B}_{S_3}\Big|_{\widehat{U}_r=1} \cong \mathfrak{B}_{S_3/\mathbb{Z}_3}\Big|_{\widehat{U}_r^\vee=1},\tag{2.12c}$$

holds.[13]

Using the mapping between the two operator algebras $\mathfrak{B}_{S_3}$ and $\mathfrak{B}_{S_3/\mathbb{Z}_3}$, we obtain the Hamiltonian

$$
\begin{aligned}
\widehat{H}_{S_3^\vee} := & -J_1 \sum_{i=1}^L \left(\widehat{x}_i + \widehat{x}_i^\dagger\right) - J_2 \sum_{i=1}^L \left(\widehat{z}_i\,\widehat{z}_{i+1}^\dagger + \widehat{z}_i^\dagger\,\widehat{z}_{i+1}\right) \\
& -J_3 \sum_{i=1}^L \left(\widehat{\sigma}_i^z\,\widehat{\tau}_i^z + \widehat{\tau}_i^z\,\widehat{\sigma}_{i+1}^z\right) - J_4 \sum_{i=1}^L \left(\widehat{\sigma}_i^x + \widehat{\tau}_i^x\right) \\
& -J_5 \sum_{i=1}^L \mathrm{i}\,\widehat{\tau}_i^z \left(\widehat{x}_{i+1} - \widehat{x}_{i+1}^\dagger\right) - J_6 \sum_{i=1}^L \mathrm{i}\,\widehat{\sigma}_i^z \left(\widehat{z}_i\,\widehat{z}_{i+1}^\dagger - \widehat{z}_i^\dagger\,\widehat{z}_{i+1}\right).
\end{aligned}\tag{2.13}
$$

This Hamiltonian is unitarily equivalent to the Hamiltonian (2.3) under exchanging the couplings $J_1$ and $J_2$, and the couplings $J_5$ and $J_6$. The unitary transformation connecting the two Hamiltonians is a *half-translation* of the qubits implemented by the unitary operator

$$\widehat{\mathfrak{t}}_{\mathbb{Z}_2} \begin{pmatrix} \widehat{\tau}_i^x & \widehat{\tau}_i^z & \widehat{\sigma}_i^x & \widehat{\sigma}_i^z \end{pmatrix} \widehat{\mathfrak{t}}_{\mathbb{Z}_2}^\dagger = \begin{pmatrix} \widehat{\sigma}_{i+1}^x & \widehat{\sigma}_{i+1}^z & \widehat{\tau}_i^x & \widehat{\tau}_i^z \end{pmatrix},\tag{2.14}$$

This equivalence between the Hamiltonian (2.3) and (2.13) is the $\mathbb{Z}_3$ Kramers-Wannier (KW) duality due to gauging the $\mathbb{Z}_3$ subgroup of the $S_3$ symmetry group. When $J_1 = J_2$ and $J_5 = J_6$, both Hamiltonians (2.3) and (2.13) become self-dual under the KW duality. In this submanifold of parameter space, the KW duality becomes a genuine non-invertible symmetry of the Hamiltonian. Without loss of generality, we focus on the dual Hamiltonian (2.3). The full KW duality operator[14] takes the form [74, 87, 88]

$$\widehat{D}_{\mathrm{KW}} := \widehat{\mathfrak{t}}_{\mathbb{Z}_2}\,\widehat{P}^{U_r=1}\,\widehat{W} \left(\widehat{\mathfrak{H}}_1^\dagger\,\widehat{CZ}_{2,1}^\dagger\right) \left(\widehat{\mathfrak{H}}_2^\dagger\,\widehat{CZ}_{3,2}^\dagger\right) \cdots \left(\widehat{\mathfrak{H}}_{L-1}^\dagger\,\widehat{CZ}_{L,L-1}^\dagger\right),\tag{2.15a}$$

where (i) the unitary operator $\widehat{\mathfrak{t}}_{\mathbb{Z}_2}$ is the half-translation operator defined in Eq. (2.14) that is necessary to preserve the form of the Hamiltonian (2.3), (ii) the operator

$$\widehat{P}^{U_r=1} := \frac{1}{3} \sum_{\alpha=0}^2 \prod_{i=1}^L \widehat{X}_i^\alpha,\tag{2.15b}$$

is the projector to the $\widehat{U}_r = 1$ subspace, (iii) the unitary operator

$$\widehat{W} := \sum_{\alpha=0}^2 \widehat{Z}_L^\alpha\,\widehat{P}^{Z_1^\dagger\,Z_L=\omega^\alpha}, \quad \widehat{W}\,\widehat{X}_1\,\widehat{W}^\dagger = \widehat{Z}_L^\dagger\,\widehat{X}_1, \quad \widehat{W}\,\widehat{X}_L\,\widehat{W}^\dagger = \widehat{Z}_L\,\widehat{X}_L\,\widehat{Z}_L\,\widehat{Z}_1^\dagger,\tag{2.15c}$$

---

[13]We could have also gauged the $\mathbb{Z}_3$ symmetry of the bond algebra (2.6) in the presence of a $\mathbb{Z}_3$ twist. However, such twisted boundary conditions lead to a reduced $\mathbb{Z}_3$ symmetry due to the fact that $\mathbb{Z}_2$ elements of $S_3$ act nontrivially the $\mathbb{Z}_3$ twist. Here, we keep the periodic boundary conditions on both sides of the gauging duality to ensure both bond algebras $\mathfrak{B}_{S_3}$ and $\mathfrak{B}_{S_3/\mathbb{Z}_3}$ have full $S_3$ and $S_3^\vee$ symmetries, respectively.

[14]We should note that a closely-related duality defect operator was also formulated in terms of a Temperley-Lieb algebra in Ref. [94].

that contains the projector $\widehat{P}^{Z_1^\dagger Z_L = \omega^\alpha}$ to the $\widehat{Z}_1^\dagger \widehat{Z}_L = \omega^\alpha$ subspace and acts nontrivially only on operators $\widehat{X}_1$ and $\widehat{X}_L$, and finally (iv) the unitary operators $\widehat{\mathfrak{H}}_i^\dagger$ and $\widehat{CZ}_{i+1,i}^\dagger$ are Hadamard and control Z operators with their only nontrivial actions being

$$\widehat{\mathfrak{H}}_i^\dagger \begin{pmatrix} \widehat{X}_i \\ \widehat{Z}_i \end{pmatrix} \widehat{\mathfrak{H}}_i = \begin{pmatrix} \widehat{Z}_i \\ \widehat{X}_i^\dagger \end{pmatrix}, \qquad \widehat{CZ}_{i+1,i}^\dagger \begin{pmatrix} \widehat{X}_{i+1} \\ \widehat{X}_i \end{pmatrix} \widehat{CZ}_{i+1,i} = \begin{pmatrix} \widehat{Z}_i^\dagger \, \widehat{X}_{i+1} \\ \widehat{X}_i \, \widehat{Z}_{i+1}^\dagger \end{pmatrix}. \tag{2.15d}$$

As written in Eq. (2.15a), the KW duality operator can be thought as a sequential circuit [87] of control Z and Hadamard operators that are applied sequentially from site $L$ down to site 1.

The KW duality operator (2.15a) is non-invertible since it contains the projector $\widehat{P}^{U_r=1}$. It becomes unitary in the subspace $\widehat{U}_r = 1$, where the self-duality holds. Its action on the local operators can be read from the identities

$$\begin{aligned} \widehat{D}_{\mathrm{KW}} \, \widehat{X}_i &= \widehat{Z}_i \, \widehat{Z}_{i+1}^\dagger \, \widehat{D}_{\mathrm{KW}}, & \widehat{D}_{\mathrm{KW}} \, \widehat{Z}_i \, \widehat{Z}_{i+1}^\dagger &= \widehat{X}_{i+1} \, \widehat{D}_{\mathrm{KW}}, \\ \widehat{D}_{\mathrm{KW}} \begin{pmatrix} \hat{\sigma}_i^x & \hat{\sigma}_i^z \end{pmatrix} &= \begin{pmatrix} \hat{\tau}_i^x & \hat{\tau}_i^z \end{pmatrix} \widehat{D}_{\mathrm{KW}}, & \widehat{D}_{\mathrm{KW}} \begin{pmatrix} \hat{\tau}_i^x & \hat{\tau}_i^z \end{pmatrix} &= \begin{pmatrix} \hat{\sigma}_{i+1}^x & \hat{\sigma}_{i+1}^z \end{pmatrix} \widehat{D}_{\mathrm{KW}}. \end{aligned} \tag{2.16}$$

In the parameter space where self-duality holds, the symmetry algebra is appended to

$$\widehat{D}_{\mathrm{KW}} \, \widehat{U}_r = \widehat{U}_r \, \widehat{D}_{\mathrm{KW}} = \widehat{D}_{\mathrm{KW}}, \tag{2.17}$$

$$\widehat{D}_{\mathrm{KW}}^\dagger = \widehat{T}^\dagger \, \widehat{D}_{\mathrm{KW}}, \tag{2.18}$$

$$\widehat{D}_{\mathrm{KW}}^2 = \widehat{P}^{U_r=1} \, \widehat{T}, \tag{2.19}$$

where $\widehat{T}$ is the operator that implements translation by one lattice site for both qubits and qutrits. Hence, action of the operator $\widehat{D}_{\mathrm{KW}}$ can be thought of as a *half-translation* operator in the subspace $\widehat{U}_r = 1$. We note that the operator $\hat{\mathfrak{t}}_{\mathbb{Z}_2}$ in Eq. (2.14) implements this half-translation only for qubits. This operator exists owing to the fact that each unit cell contains two flavors of qubits for a single flavor of qutrit. Had we defined a 6 dimensional local Hilbert space which supports single flavor of $\mathbb{Z}_2$- and $\mathbb{Z}_3$-clock operators, KW self-duality would only hold when the couplings $J_5$ and $J_6$ are zero, *i.e.*, when qubits and qutrits are decoupled.

The symmetry algebra above includes eqn. (2.17) as a lattice analogue of the fusion rules of the $\mathbb{Z}_3$ Tambara-Yamagami fusion category symmetry. In the continuum limit, we expect that the $\mathbb{Z}_3$ symmetry generator $\widehat{U}_r$ flows to a $\mathbb{Z}_3$ topological line $\xi$, while both $\sqrt{3} \, \widehat{D}_{\mathrm{KW}}$ and its Hermitian conjugate flow to the continuum duality topological line $\mathcal{D}$. They satisfy the fusion rules

$$\xi \mathcal{D} = \mathcal{D} \xi = \mathcal{D}, \quad \mathcal{D}^2 = 1 + \xi + \xi^2, \quad \xi^3 = 1, \tag{2.20}$$

where the operator that implements single lattice site translation becomes an internal symmetry in the continuum limit. This interpretation follows the approach presented in Ref. [74].

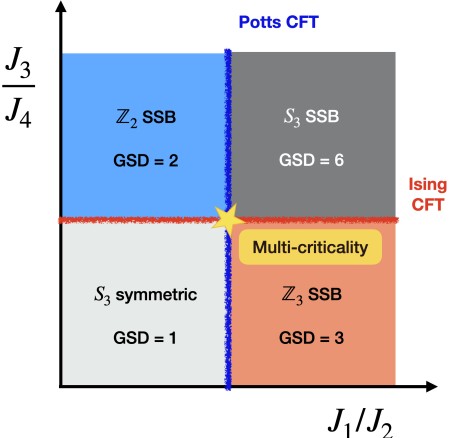

Figure 2: Phase diagram of Hamiltonian (2.21) based on analytical arguments for $\theta \approx 0$. The ground state degeneracy (GSD) for each of the SSB phases are labeled. The vertical and horizontal critical lines correspond to the Potts (6,5) and Ising (4,3) minimal model CFTs, respectively. They intersect at a multi-critical point belonging to the 3-state Potts $\boxtimes$ Ising universality class.

## 2.3   Phase diagram

To discuss the phase diagram of the Hamiltonian (2.3), we first reparameterize it as

$$
\widehat{H}_{S_3} = - J_1 \cos\theta \sum_{i=1}^{L} \left( \widehat{Z}_i \, \widehat{Z}_{i+1}^{\dagger} + \widehat{Z}_i^{\dagger} \, \widehat{Z}_{i+1} \right) - J_1 \sin\theta \sum_{i=1}^{L} \mathrm{i}\, \hat{\tau}_i^z \left( \widehat{Z}_i \, \widehat{Z}_{i+1}^{\dagger} - \widehat{Z}_i^{\dagger} \, \widehat{Z}_{i+1} \right)
$$

$$
- J_2 \cos\theta \sum_{i=1}^{L} \left( \widehat{X}_i + \widehat{X}_i^{\dagger} \right) - J_2 \sin\theta \sum_{i=1}^{L} \mathrm{i}\, \hat{\sigma}_i^z \left( \widehat{X}_i - \widehat{X}_i^{\dagger} \right) \tag{2.21}
$$

$$
- J_3 \sum_{i=1}^{L} \left( \hat{\sigma}_i^z \, \hat{\tau}_i^z + \hat{\tau}_i^z \, \hat{\sigma}_{i+1}^z \right) - J_4 \sum_{i=1}^{L} \left( \hat{\sigma}_i^x + \hat{\tau}_i^x \right) .
$$

In what follows, we will explore the phase diagram of this Hamiltonian as a function of dimensionless ratios $J_1/J_2$ and $J_3/J_4$, for the cases of $\theta = 0$, non-zero but small $\theta \approx 0$, and large $\theta \sim 0.7$.

### 2.3.1   Analytical arguments

When $\theta = 0$, the Hamiltonian (2.21) describes decoupled quantum Ising and 3-state Potts chains, for which the phase diagram is known. There are four gapped phases which correspond to four different symmetry breaking patterns for $S_3$. At four fixed-points, we can write the wave-functions exactly:

(i) When $J_1 = J_3 = 0$, there is only a single ground state

$$
|\mathrm{GS}_{S_3}\rangle := \bigotimes_{i=1}^{L} |\sigma_i^x = 1,\, \tau_i^x = 1,\, X_i = 1\rangle , \tag{2.22}
$$

which describes the $S_3$-disordered phase.

(ii) When $J_1 = J_4 = 0$, there are two degenerate ground states

$$
|\mathrm{GS}_{\mathbb{Z}_3}^{\pm}\rangle := \bigotimes_{i=1}^{L} |\sigma_i^z = \pm 1,\, \tau_i^z = \pm 1,\, X_i = 1\rangle , \tag{2.23}
$$

that describe the phase where qubits are ordered and $S_3$ symmetry is broken down to $\mathbb{Z}_3$.

(iii) When $J_2 = J_3 = 0$, there are three degenerate ground states

$$|\mathrm{GS}_{\mathbb{Z}_2}^\alpha\rangle := \bigotimes_{i=1}^{L} |\sigma_i^x = +1,\, \tau_i^x = +1,\, Z_i = \omega^\alpha\rangle, \tag{2.24}$$

with $\alpha = 0, 1, 2$, that describe the phase where qutrits are ordered. One each ground state, $S_3$ symmetry is broken down to a $\mathbb{Z}_2$ subgroup.

(iv) When $J_2 = J_4 = 0$, there are six degenerate ground states

$$|\mathrm{GS}_{\mathbb{Z}_1}^{\pm,\alpha}\rangle := \bigotimes_{i=1}^{L} |\sigma_i^z = \pm 1,\, \tau_i^z = \pm 1,\, Z_i = \omega^\alpha\rangle, \tag{2.25}$$

with $\alpha = 0, 1, 2$ that describe the $S_3$ ordered phase.

See Sec. 5.2 for the discussion of expectation values of the correlation functions and disorder operators in these ground states.

The lines $J_1/J_2 = 1$ and $J_3/J_4 = 1$ correspond to the transition points between the gapped phases 1 and 2, and 3 and 4, respectively. They are described by the 3-state Potts CFT and the Ising CFT, respectively. The 3-state Potts CFT is one of the $(6,5)$ minimal models with $c = 4/5$, while the Ising CFT is the $(4,3)$ minimal model wth $c = 1/2$. At $J_1/J_2 = 1$ and $J_3/J_4 = 1$, there is a multicritical point described by the stacking of the two CFTs, with total central charge $c = 13/10$.

If we turn on small $\theta \neq 0$, the gapped phases are expected to remain unaffected by the virtue of finiteness of the gap (in the thermodynamic limit). However, one may wonder what the fate of the critical lines and the multicritical point is under these perturbations. To understand what happens to the critical lines, we first argue that three out of the four critical lines are stable against small nonzero $\theta$ as follows. Along the line $J_3/J_4 = 1$, when $J_2 < J_1$, qutrits are ordered and gapped. This means that both $\widehat{X}_i - \widehat{X}_i^\dagger$ and $\widehat{Z}_i \widehat{Z}_{i+1}^\dagger - \widehat{Z}_i^\dagger \widehat{Z}_{i+1}$ vanish in the low-energy states below the qutrit excitation gap. The same line of thought holds when $J_1 < J_2$ for which qutrits are disordered and gapped. Similarly, along the line $J_1/J_2 = 1$, when $J_3 < J_4$, qubits are disordered and gapped. Both $\hat{\sigma}_i^z$ and $\hat{\tau}_i^z$ vanish in the low-energy states below the qubit excitation gap.

The situation is different when $J_1/J_2 = 1$ and $J_4 < J_3$ for which quibts order, *i.e.*, the terms with $\sin\theta$ coefficient are not trivially vanishing. On this line (excluding the multicritical point), the terms with $\sin\theta$ coefficient must flow to a primary or descendant operator $\mathcal{O}$ in 3-state Potts CFT that is odd under the charge conjugation symmetry. Using holographic modular bootstrap techniques [28,44], we identify that (see Appendix E.1) only possible relevant operators are those primaries with scaling dimension $\Delta_{\mathcal{O}} = 9/5$ and conformal spin $\pm 1$. Indeed, in Ref. [95], the terms with coefficients $\sin\theta$ in Hamiltonian (2.21) are identified with these primary operators. Even though these primaries are relevant, we expect that as long as the KW self-duality symmetry is preserved one cannot open a KW-symmetric gap owing to the fact that this non-invertible symmetry is anomalous. This is consistent with the results in Ref. [96] where it was shown that (see Fig. 2.(f) therein) the critical line with $J_1/J_2 = 1$ and $J_4 < J_3$ is stable for small $\theta$ and an incommensurate phase opens up for $\theta$ larger than a critical value. We confirm this numerically for our model as we shall see in the next section. In summary, we conclude that the phase diagram at perturbatively small $\theta$ has the same form as that for $\theta = 0$. The phase diagram at $\theta = 0$ plane is shown in Fig. 2. We verify our predictions for non-zero and small $\theta$ numerically in Fig. 3.

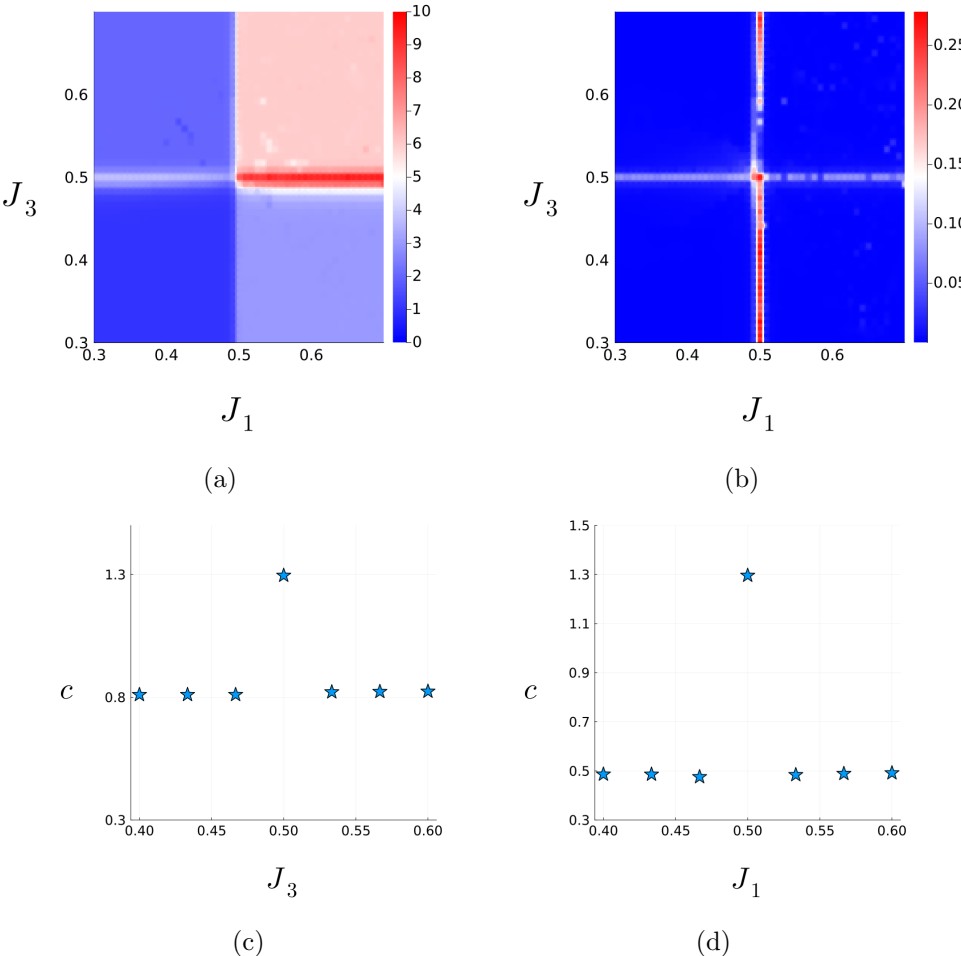

Figure 3: Numerical phase diagram obtained from the TEFR algorithm showing GSD (a) and central charge (b) as a function of $J_1$ and $J_3$. The effective system size for these plots is $L = 128$. Figs. (c) and (d) show the central charges computed from bipartite entanglement entropy scaling in the ground state obtained from DMRG, as discussed in the main text. We plot the central charges along a vertical ($J_1 = 0.5$) and a horizontal ($J_3 = 0.5$) slice of the phase diagram shown in Fig. (b), for a chain of $L = 100$ sites. We fix $\theta = 0.1$, with $J_2 = 1 - J_1$, $J_4 = 1 - J_3$ in all of these plots.

### 2.3.2   Numerical results

We mapped out the phase diagram of Hamiltonian (2.21) numerically, using the tensor entanglement filtering renormalization (TEFR) algorithm [97, 98]. The ground state degeneracies of each gapped phase and the central charges associated with continuous phase transitions were obtained using this algorithm. The results are shown in Fig. 3 as a function of $J_1$ and $J_3$ with $\theta = 0.1$. We chose a 2d slice in the full parameter space such that at every point of the phase diagram shown here, $J_2 = 1 - J_1$ and $J_4 = 1 - J_3$.

The TEFR algorithm does not give numerically precise values of central charge even though it is extremely precise at extracting ground state degeneracy.[15] In order to extract numerically precise central charges, we used the density matrix renormalization group (DMRG) algorithm from the iTensor library [99, 100]. On the phase diagram shown in

---

[15]The interested reader is referred to Appendix D for more details on this point.

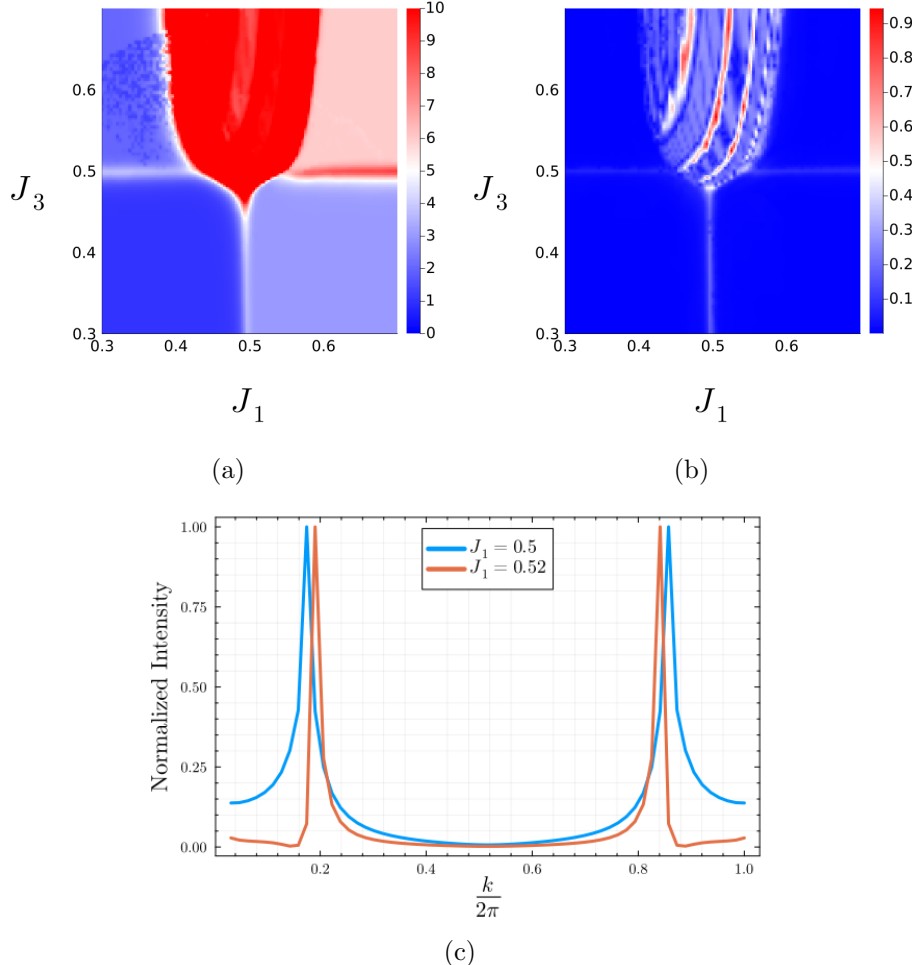

Figure 4: Numerical phase diagram obtained from the TEFR algorithm with fixed $\theta = 0.7 \approx \frac{2\pi}{9}$, showing GSD (Fig. (a)), and central charge (Fig. (b)) as a function of $J_1$ and $J_3$, setting $J_2 = 1 - J_1$ and $J_4 = 1 - J_3$ everywhere. The nominal GSD in this gapless region is much larger than that shown in Fig. (a); we have capped the maximum allowed values to 10 so that these plots may be easily compared with the ones in Fig. 3. In these figures, the effective system size is $L = 128$. Fig. (c) shows the absolute value of the Fourier transform of ground state expectation value $\langle \hat{\sigma}_i^z \rangle$ for $J_1 = 0.5$ and $J_1 = 0.52$ with fixed $J_3 = 0.5$ and $L = 101$ sites.

Fig. 3, we make two cuts, one horizontal and one vertical, and compute the central charges using finite size bipartite entanglement entropy scaling that is computed using DMRG calculation. Our results for finite chain of $L = 100$ sites are shown in Figs. 3c and 3d.

We find that the both $c = 1/2$ and $c = 4/5$ lines are stable against small values of $\theta$, in agreement with our argument in the previous section. For large values of $\theta$, a gapless region opens up in the Ising ordered regime, surrounding the $\mathbb{Z}_3$ KW symmetric line $J_1 = J_2$ in the parameter space of the Hamiltonian (2.21). From numerical estimation, we find the critical value of $\theta$ to be around $\theta_* \simeq \frac{\pi}{8}$. Heuristically, the gapless region appears first in the $\mathbb{Z}_2$ ordered phase as the terms in Hamiltonian (3.25) with $\sin\theta$ are proportional to $\hat{\sigma}_i^z$ and $\hat{\tau}_i^z$ operators which have vanishing expectation values when the $\mathbb{Z}_2$ subgroup of $S_3$ is unbroken. The multicritical point is engulfed by the gapless region beyond a certain value of $\theta \in (\pi/8, 2\pi/9)$. Several comments are due:

(i) Since the $\mathbb{Z}_3$ KW duality symmetry is anomalous, the only compatible phase without symmetry breaking is gapless [10]. This is compatible with the gapless region we numerically observe. Also, because of the KW duality, the gapless phase must be symmetrically placed about the $J_1 = 0.5 = J_2$ line of the phase diagram, consistent with the numerically obtained phase diagram in Fig. 4.

(ii) Extraction of the central charge in the gapless region is somewhat subtle. The phase diagram obtained from TEFR (Fig. 4b) shows fluctuations in $c$ throughout the gapless region. From DMRG calculations, we find that the ground state expectation value $\langle \hat{\sigma}_i^z \rangle$ shows oscillatory behavior around its mean value. In Fig. 4c, we plot the absolute value of Fourier transform of this expectation value (minus its mean) for two points in the gapless region of the parameter space for $L = 101$ sites. We find that the position of the peak value changes for $J_1 = 0.50$ and $J_1 = 0.52$ for fixed $J_3 = 0.5$, with corresponding periods being $63/11$ and $63/12$ lattice constants, respectively. This suggests a smooth variation of the oscillation period as a function of $J_1$ and $J_3$ in the thermodynamic limit. We conjecture that in this gapless region, the system realizes a incommensurate phase with central charge $c = 1$. In other words, the ground state contains low-energy states at a non-zero quasi-momentum which smoothly varies as a function of the couplings $J_1$ and $J_3$. This echoes the behavior of the self-dual deformed Ising model, which is exactly solvable, discussed in Sec. 5.3.

(iii) Results from the TEFR algorithm also show that the interface between the gapless region and the neighboring gapped phases has a vanishing central charge. Since the transition from a gapless phase to a gapped one is expected to not be a first-order transition, we conjecture that this must correspond to a continuous transition with a dynamical critical exponent $z > 1$.

### 2.3.3 Explicitly breaking KW self-duality symmetry

Hamiltonian (2.21) consists of terms such that the lines $J_1 = J_2$ is symmetric under the $\mathbb{Z}_3$ KW self-duality symmetry generated by the operator (2.15). As explained in the previous section, the multicritical point, $J_1 = J_2$ and $J_3 = J_4$, is described by 3-state Potts $\boxtimes$ Ising CFT with central charge $c = 13/10$. This multicritical point has three relevant perturbations, see Appendix E.2. Two of these are generated when $J_1 - J_2 \neq 0$ and $J_3 - J_4 \neq 0$, which gap out the qutrits and qubits, respectively. In the continuum description, these perturbations correspond to the "energy" primaries $\epsilon_{\mathrm{I}}$ and $\epsilon_{\mathrm{P}}$ of the Ising and 3-state Potts CFTs, respectively. The former is self-dual under the $\mathbb{Z}_3$ KW duality and gaps out the qubits. In contrast, the latter breaks KW duality symmetry explicitly and gaps out the qutrits.

The third and final relevant perturbation is given by the product of two energy primaries $\epsilon_{\mathrm{I}} \epsilon_{\mathrm{P}}$ which is odd under the $\mathbb{Z}_3$ self-duality symmetry. In the lattice model, this perturbation is generated by

$$\widehat{H}_\perp := -J_\perp \sum_{i=1}^{L} \left\{ \left( \hat{\tau}_i^z \, \hat{\sigma}_{i+1}^z - \hat{\sigma}_{i+1}^x \right) \left( \widehat{Z}_i \, \widehat{Z}_{i+1}^\dagger + \text{H.c.} \right) - \left( \hat{\tau}_i^z \, \hat{\sigma}_i^z - \hat{\tau}_i^x \right) \left( \widehat{X}_i + \text{H.c.} \right) \right\}, \quad (2.26)$$

which is odd under the $\mathbb{Z}_3$ KW duality symmetry (2.15).

From the TEFR algorithm, we find that adding this term has the effect of replacing the multicritical point by a critical line which separates the $S_3$ symmetric and the $S_3$ completely broken phases for positive $J_\perp$ and by a critical line which separates the $\mathbb{Z}_3$

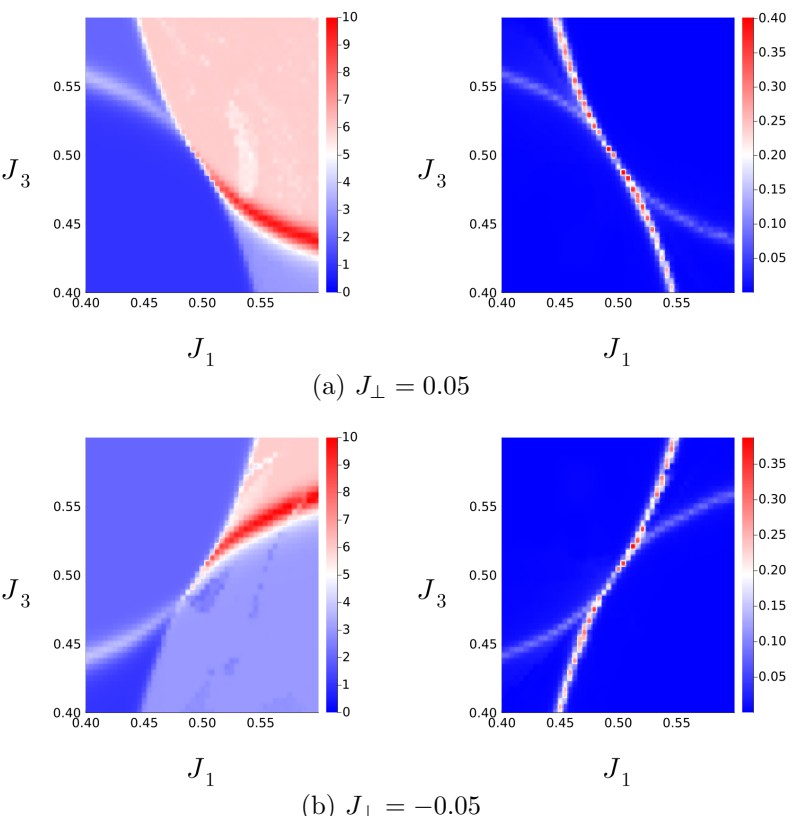

Figure 5: Numerical phase diagram from TEFR algorithm, with fixed $J_\perp$ and $\theta = 0$, showing GSD (left) and central charge(right) as heatmaps, in the $J_1, J_3$ plane (with $J_2 = 1 - J_1$ and $J_4 = 1 - J_3$ everywhere). The effective system size is $L = 128$. (a) For positive $J_\perp$, we find the multicritical point widens into a critical line between the $S_3$ symmetric and $S_3$ SSB phases. (b) For negative $J_\perp$, we find the multicritical point widens into a critical line between the phases which spontaneously break $S_3$ down to $\mathbb{Z}_3$ and $\mathbb{Z}_2$, instead.

symmetric and $\mathbb{Z}_2$ symmetric phases for negative $J_\perp$. This is shown in the GSD and central charge plots obtained from the TEFR algorithm in Fig. 5.

While it is rather hard to precisely determine the points on the critical lines, we know that the point with $J_1 = J_2 = 0.5$ and $J_3 = J_4 = 0.5$ must be critical. This is because when $\theta = 0$ the Hamiltonian (2.3) has an additional non-invertible $\mathbb{Z}_2$ KW self-duality symmetry along the $J_3 = J_4$ line. In the continuum limit, the perturbation (2.26) flows to $\epsilon_I \epsilon_P$, *i.e.*, product of energy primaries, which is odd under both $\mathbb{Z}_2$ and $Z_3$ KW self-duality symmetries.[16] Under both of these dualities, the point with $J_1 = J_2$ and $J_3 = J_4 = 0.5$ is invariant, and hence must be gapless in both phase diagrams with $J_\perp > 0$ and $J_\perp < 0$. At this point the DMRG results suggest a central charge of $c \approx 1.2$ which we believe to hold for the entire the critical line; a precise characterization of this transition in terms of an associated conformal field theory is beyond the scope of the present paper.

We note that the continuous phase transition which separates the $\mathbb{Z}_3$ symmetric and $\mathbb{Z}_2$ symmetric phases for $J_\perp < 0$ is a beyond Landau-Ginzburg (LG) transition as it is between phases that break different subgroups of the full symmetry group. Under the

---

[16]At the lattice level, the perturbation (2.26) is exactly odd only under the $\mathbb{Z}_3$ KW self-duality operator (2.15). While it explicitly breaks the $\mathbb{Z}_2$ KW self-duality symmetry too, it does not go to minus itself under $\mathbb{Z}_2$ KW self-duality transformation.

$\mathbb{Z}_3$ KW-duality symmetry, *i.e.*, when $J_\perp$ becomes positive, this beyond-LG transition is mapped to the ordinary LG-type transition between the $S_3$ symmetric and the $S_3$ SSB phases.

# 3    Rep($S_3$)-symmetric spin chain

In Sec. 2, we studied a spin chain with $S_3$ symmetry. We discussed how this $S_3$ is enriched by non-invertible $\mathbb{Z}_3$ Kramers-Wannier self-duality symmetry, at special points in the parameter space. Therein, the presence of KW duality symmetry ensures that the ground is either gapless or degenerate.[17] In this section, we are going to show that our $S_3$-symmetric spin chain is dual to another model which has non-invertible symmetries in its entire parameter space. As we shall see in Sec. 3.1, this duality follows from gauging a $\mathbb{Z}_2$ subgroup of $S_3$, which delivers a dual model with Rep($S_3$) fusion category symmetry. We discuss the gapped phases and phase transitions of this Rep($S_3$)-symmetric model in Sec. 3.3. As it was the case for Hamiltonian (2.3), we will see in Sec. 3.2, its dual with Rep($S_3$) symmetry also has an additional non-invertible symmetry at special points in the parameter space. We are going to describe how this additional symmetry is also associated with a gauging procedure which can be implemented through a sequential circuit.

## 3.1   Gauging $\mathbb{Z}_2$ subgroup: dual Rep($S_3$) symmetry

We shall gauge the $\mathbb{Z}_2$ subgroup of $S_3$. We follow the same prescription as in Sec. 2.2. As we shall see, as opposed to gauging $\mathbb{Z}_3$ subgroup, the dual symmetry will be the category Rep($S_3$), owing to the fact that $\mathbb{Z}_2$ is not a normal subgroup of $S_3$.

In order to gauge the $\mathbb{Z}_2$ symmetry, on each link between sites $i$ and $i+1$, we introduce $\mathbb{Z}_2$ clock operators $\left\{ \hat{\mu}^x_{i+1/2}, \hat{\mu}^z_{i+1/2} \right\}$ that satisfy the algebra

$$\hat{\mu}^z_{i+1/2}\, \hat{\mu}^x_{j+1/2} = (-1)^{\delta_{ij}}\hat{\mu}^x_{j+1/2}\, \hat{\mu}^z_{i+1/2},$$
$$\left( \hat{\mu}^z_{i+1/2} \right)^2 = \left( \hat{\mu}^x_{i+1/2} \right)^2 = \hat{\mathbb{1}}, \tag{3.1}$$
$$\hat{\mu}^z_{i+1/2+L} = \hat{\mu}^z_{i+1/2}, \quad \hat{\mu}^x_{i+1/2+L} = \hat{\mu}^x_{i+1/2},$$

where we have imposed periodic boundary conditions. Accordingly, we define the Gauss operator

$$\widehat{G}^{\mathbb{Z}_2}_i := \hat{\mu}^z_{i-1/2}\, \hat{\sigma}^x_i\, \hat{\tau}^x_i\, \widehat{C}_i\, \hat{\mu}^z_{i+1/2}, \qquad \left[ \widehat{G}^{\mathbb{Z}_2}_i \right]^2 = \hat{\mathbb{1}}. \tag{3.2}$$

The physical states are those in the subspace $\widehat{G}^{\mathbb{Z}_2}_i = 1$ for all $i = 1, \cdots, L$. By way of minimally coupling the bond algebra (2.6), we obtain the gauge invariant bond algebra [18]

$$\mathfrak{B}^{\mathrm{mc}}_{S_3/\mathbb{Z}_2} := \Big\langle \hat{\sigma}^z_i\, \hat{\tau}^z_i, \ \hat{\tau}^z_i\, \hat{\mu}^x_{i+1/2}\, \hat{\sigma}^z_{i+1}, \ \hat{\sigma}^x_i, \ \hat{\tau}^x_i, \ \left( \widehat{X}_i + \widehat{X}^\dagger_i \right), \ \left( \widehat{Z}^{\hat{\mu}^x_{i+1/2}}_i\, \widehat{Z}^\dagger_{i+1} + \widehat{Z}^{-\hat{\mu}^x_{i+1/2}}_i\, \widehat{Z}_{i+1} \right),$$
$$\hat{\sigma}^z_i \left( \widehat{X}_i - \widehat{X}^\dagger_i \right), \ \hat{\tau}^z_i\, \hat{\mu}^x_{i+1/2} \left( \widehat{Z}^{\hat{\mu}^x_{i+1/2}}_i\, \widehat{Z}^\dagger_{i+1} - \widehat{Z}^{-\hat{\mu}^x_{i+1/2}}_i\, \widehat{Z}_{i+1} \right) \Big| \widehat{G}^{\mathbb{Z}_2}_i = 1, \quad i \in \Lambda \Big\rangle. \tag{3.3}$$

---

[17]This is because this non-invertible symmetry is anomalous [10,38,101].

[18]Here, we introduce the short-hand notations

$$\widehat{Z}^{\hat{\mu}^x_{i+1/2}}_i \equiv \frac{1 + \hat{\mu}^x_{i+1/2}}{2}\, \widehat{Z}_i + \frac{1 - \hat{\mu}^x_{i+1/2}}{2}\, \widehat{Z}^\dagger_i, \qquad \widehat{Z}^{-\hat{\mu}^x_{i+1/2}}_i \equiv \frac{1 - \hat{\mu}^x_{i+1/2}}{2}\, \widehat{Z}_i + \frac{1 + \hat{\mu}^x_{i+1/2}}{2}\, \widehat{Z}^\dagger_i.$$

As it was in Sec. 2.2, we can simplify the Gauss constraint by applying the unitary transformation

$$\begin{aligned}
\widehat{U}\,\hat{\sigma}_i^x\,\widehat{U}^\dagger &= \hat{\mu}_{i-1/2}^z\,\hat{\sigma}_i^x\,\hat{\tau}_i^x\,\widehat{C}_i\,\hat{\mu}_{i+1/2}^z, & \widehat{U}\,\hat{\sigma}_i^z\,\widehat{U}^\dagger &= \hat{\sigma}_i^z, \\
\widehat{U}\,\hat{\tau}_i^x\,\widehat{U}^\dagger &= \hat{\tau}_i^x, & \widehat{U}\,\hat{\tau}_i^z\,\widehat{U}^\dagger &= \hat{\tau}_i^z\,\hat{\sigma}_i^z, \\
\widehat{U}\,\widehat{X}_i\,\widehat{U}^\dagger &= \widehat{X}_i^{\hat{\sigma}_i^z}, & \widehat{U}\,\widehat{Z}_i\,\widehat{U}^\dagger &= \widehat{Z}_i^{\hat{\sigma}_i^z}, \\
\widehat{U}\,\hat{\mu}_{i+1/2}^x\,\widehat{U}^\dagger &= \hat{\sigma}_i^z\,\hat{\mu}_{i+1/2}^x\,\hat{\sigma}_{i+1}^z, & \widehat{U}\,\hat{\mu}_{i+1/2}^z\,\widehat{U}^\dagger &= \hat{\mu}_{i+1/2}^z.
\end{aligned} \tag{3.4}$$

Under this unitary the Gauss operator simplifies $\widehat{U}\,\widehat{G}_i^{\mathbb{Z}_2}\,\widehat{U}^\dagger = \hat{\sigma}_i^x$. We apply this unitary to the minimally coupled algebra (3.3) and project onto the $\hat{\sigma}_i^x = 1$ sector. After shifting the link degrees of freedom by $i + 1/2 \mapsto i + 1$, we obtain the dual algebra

$$\begin{aligned}
\mathfrak{B}_{S_3/\mathbb{Z}_2} &:= \widehat{U}\,\mathfrak{B}_{S_3/\mathbb{Z}_2}^{\mathrm{mc}}\,\widehat{U}^\dagger\Big|_{\hat{\sigma}_i^x=1} \\
&= \Big\langle \hat{\tau}_i^z,\ \hat{\tau}_i^z\,\hat{\mu}_{i+1}^x,\ \hat{\mu}_i^z\,\hat{\tau}_i^x\,\widehat{C}_i\,\hat{\mu}_{i+1}^z,\ \hat{\tau}_i^x,\ \left(\widehat{X}_i + \widehat{X}_i^\dagger\right),\ \left(\widehat{Z}_i^{\hat{\mu}_{i+1}^x}\,\widehat{Z}_{i+1}^\dagger + \widehat{Z}_i^{-\hat{\mu}_{i+1}^x}\,\widehat{Z}_{i+1}\right), \\
&\qquad \left(\widehat{X}_i - \widehat{X}_i^\dagger\right),\ \hat{\tau}_i^z\,\hat{\mu}_{i+1}^x\,\left(\widehat{Z}_i^{\hat{\mu}_{i+1}^x}\,\widehat{Z}_{i+1}^\dagger - \widehat{Z}_i^{-\hat{\mu}_{i+1}^x}\,\widehat{Z}_{i+1}\right)\ \Big|\ i \in \Lambda \Big\rangle. 
\end{aligned} \tag{3.5}$$

What is the symmetry described by the dual algebra (3.5)? We claim that $\mathfrak{B}_{S_3/\mathbb{Z}_2}$ is the algebra of $\mathsf{Rep}(S_3)$-symmetric operators. The fusion category $\mathsf{Rep}(S_3)$ consists of three simple objects, $\mathbf{1}$, $\mathbf{1}'$, and $\mathbf{2}$, which are labeled by the three irreducible representations (irreps) of $S_3$. The object $\mathbf{1}$ is represented by the unitary identity operator $\widehat{W}_{\mathbf{1}} = \hat{\mathbb{1}}$, while the object $\mathbf{1}'$ is represented by the unitary operator

$$\widehat{W}_{\mathbf{1}'} := \prod_{i=1}^L \hat{\mu}_i^x, \tag{3.6}$$

which is the generator of $\mathbb{Z}_2$ dual symmetry associated with gauging the $\mathbb{Z}_2$ subgroup of $S_3$. Consistency in gauging with imposing periodic boundary conditions on the operator $\{\hat{\mu}_i^x,\ \hat{\mu}_i^z\}$ and the operators $\{\hat{\sigma}_i^x,\ \hat{\sigma}_i^z\}$ requires the conditions

$$\widehat{U}_s = 1, \qquad \widehat{W}_{\mathbf{1}'} = 1, \tag{3.7a}$$

to be satisfied, respectively. In other words, the duality holds between the subalgebras

$$\mathfrak{B}_{S_3}\Big|_{\widehat{U}_s=1} \cong \mathfrak{B}_{S_3/\mathbb{Z}_2}\Big|_{\widehat{W}_{\mathbf{1}'}=1}, \tag{3.7b}$$

where conditions in Eq. (3.7a) are satisfied.

Finally, we notice that gauging $\mathbb{Z}_2$ subgroup breaks the $\mathbb{Z}_3$ symmetry, since the former is not a normal subgroup. Under conjugation by $\widehat{U}_r$, the Gauss operator (3.2) transforms nontrivially

$$\widehat{U}_r\,\widehat{G}_i^{\mathbb{Z}_2}\,\widehat{U}_r^\dagger = \widehat{G}_i^{\mathbb{Z}_2}\,\widehat{X}_i. \tag{3.8}$$

Therefore, $\widehat{U}_r$ cannot be made gauge invariant by coupling to the gauge fields $\hat{\mu}_{i+1/2}^x$. However, the non-unitary and non-invertible operator

$$\widehat{U}_{r\oplus r^2} := \widehat{U}_r + \widehat{U}_r^\dagger = \prod_{i=1}^L \widehat{X}_i + \prod_{i=1}^L \widehat{X}_i^\dagger \tag{3.9}$$

commutes with all the generators of the algebra (2.6) and the global symmetry operator $\widehat{U}_s$ when periodic boundary conditions for all operators in the algebra (2.6). This is the

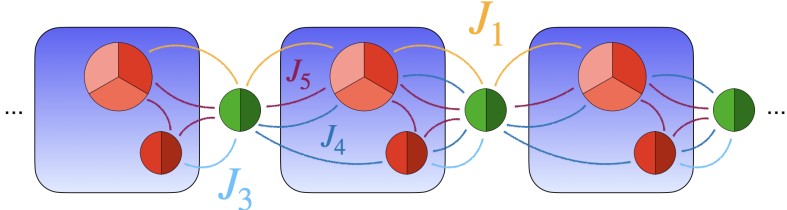

Figure 6: Schematic of the Hamiltonian (3.13) showing the couplings between qutrit (depicted by a tripartitioned disk) and qubit (depicted by a bipartitioned disk) degrees of freedom. Single-body terms $J_2, J_6$ are suppressed.

representation of direct sum $r \oplus r^2$ of simple objects $r$ and $r^2$ in the symmetry category $\mathsf{Vec}_{S_3}$. Since it commutes with $\widehat{U}_s$, $\widehat{U}_{r \oplus r^2}$ can be made gauge invariant. Minimally coupling the operator (3.9), and applying the unitary transformation (3.4) delivers the operator [19]

$$\widehat{W}_{\mathbf{2}} := \frac{1}{2} \left( 1 + \prod_{i=1}^{L} \hat{\mu}_i^x \right) \left[ \prod_{i=1}^{L} \widehat{X}_i^{\prod_{k=2}^i \hat{\mu}_k^x} + \widehat{X}_i^{-\prod_{k=2}^i \hat{\mu}_k^x} \right]. \tag{3.10}$$

This is the representation of simple object $\mathbf{2}$ in the category $\mathsf{Rep}(S_3)$. It can be expressed as a matrix product operator (MPO) with a 2-dimensional virtual index, as follows [20]

$$\widehat{W}_{\mathbf{2}} = \mathrm{Tr}\left[ \delta_{\alpha_1, \alpha_{L+1}} \prod_{j=1}^{L} \widehat{M}^{(j)}_{\alpha_j \alpha_{j+1}} \right], \quad \widehat{M}^{(j)} = \begin{pmatrix} \widehat{X}_j\, P^{\hat{\mu}_{j+1}^x=1} & \widehat{X}_j^\dagger\, P^{\hat{\mu}_{j+1}^x=-1} \\ \widehat{X}_j\, P^{\hat{\mu}_{j+1}^x=-1} & \widehat{X}_j^\dagger\, P^{,\hat{\mu}_{j+1}^x=1} \end{pmatrix} \tag{3.11}$$

Together with $\widehat{W}_{\mathbf{1}}$ and $\widehat{W}_{\mathbf{1}'}$, they satisfy the fusion rules of $\mathsf{Rep}(S_3)$, i.e.,

$$\widehat{W}_{\mathbf{1}'} \widehat{W}_{\mathbf{1}'} = \widehat{W}_{\mathbf{1}}, \qquad \widehat{W}_{\mathbf{1}'} \widehat{W}_{\mathbf{2}} = \widehat{W}_{\mathbf{2}}, \qquad \widehat{W}_{\mathbf{2}} \widehat{W}_{\mathbf{2}} = \widehat{W}_{\mathbf{1}} + \widehat{W}_{\mathbf{1}'} + \widehat{W}_{\mathbf{2}}. \tag{3.12}$$

We note that because of the projector, $\widehat{W}_{\mathbf{2}}$ is non-invertible. This projector to the $\prod_{i=1}^{L} \hat{\mu}_i^x = 1$ subspace ensures that there is no $\mathbb{Z}_2$ twist in the $S_3$-symmetric algebra (2.6). This is needed as $\widehat{U}_{r \oplus r^2}$ is not a symmetry of the algebra (2.6) when $\mathbb{Z}_2$-twisted boundary conditions are imposed.[21] In other words, in the presence of a $\mathbb{Z}_2$ twist, one expect the dual symmetry to be $\mathbb{Z}_2$ generated by $\widehat{W}_{\mathbf{1}'}$ instead of the full $\mathsf{Rep}(S_3)$.

We now use the duality mapping between the local operator algebras $\mathfrak{B}_{S_3}$ and $\mathfrak{B}_{S_3/\mathbb{Z}_2}$ (recall Eqs. (2.6) and (3.5)) to construct a local Hamiltonian with $\mathsf{Rep}(S_3)$ symmetry. Under this map, the image of Hamiltonian (2.3) is

$$
\begin{aligned}
\widehat{H}_{\mathsf{Rep}(S_3)} := &- J_1 \sum_{i=1}^{L} \left( \widehat{Z}_i^{\hat{\mu}_{i+1}^x} \widehat{Z}_{i+1}^\dagger + \widehat{Z}_i^{-\hat{\mu}_{i+1}^x} \widehat{Z}_{i+1} \right) - J_2 \sum_{i=1}^{L} \left( \widehat{X}_i + \widehat{X}_i^\dagger \right) \\
&- J_3 \sum_{i=1}^{L} \left( \hat{\tau}_i^z + \hat{\tau}_i^z\, \hat{\mu}_{i+1}^x \right) - J_4 \sum_{i=1}^{L} \left( \hat{\mu}_i^z\, \hat{\tau}_i^x\, \widehat{C}_i\, \hat{\mu}_{i+1}^z + \hat{\tau}_i^x \right) \\
&- J_5 \sum_{i=1}^{L} \mathrm{i}\, \hat{\tau}_i^z\, \hat{\mu}_{i+1}^x \left( \widehat{Z}_i^{\hat{\mu}_{i+1}^x} \widehat{Z}_{i+1}^\dagger - \widehat{Z}_i^{-\hat{\mu}_{i+1}^x} \widehat{Z}_{i+1} \right) - J_6 \sum_{i=1}^{L} \mathrm{i} \left( \widehat{X}_i - \widehat{X}_i^\dagger \right).
\end{aligned} \tag{3.13}
$$

---

[19] Each of the two terms in square brackets individually commutes with the Gauss operators associated with Gauss operators $\widehat{G}_2^{\mathbb{Z}_2}$ through $\widehat{G}_L^{\mathbb{Z}_2}$, while $\widehat{G}_1^{\mathbb{Z}_2}$ simply exchanges them so that the sum is still gauge invariant.

[20] See Appendix F for construction of $\mathsf{Rep}(G)$ operators in terms of MPOs that results from gauging a finite $G$ symmetry.

[21] A $g \in G$ twist reduces the full symmetry $G$ to the centralizer $\mathrm{C}_G(g)$ of $g$. Since $S_3$ is non-Abelian, imposing $\mathbb{Z}_3$- and $\mathbb{Z}_2$-twisted boundary conditions, reduce $S_3$ down to $\mathbb{Z}_3$ and $\mathbb{Z}_2$, respectively.

In what follows, we are going to study the phase diagram of this Hamiltonian and identify spontaneous symmetry breaking patterns for $\mathsf{Rep}(S_3)$ symmetry as well as the transitions between various ordered phases. Before doing so, we will briefly digress to discuss a duality that delivers a dual $\mathsf{Rep}(S_3^\vee)$ symmetric bond algebra.[22]

## 3.2 Another non-invertible self-duality symmetry

As we discussed in Sec. 2.2, $S_3$-symmetric Hamiltonian (2.3) enjoys a self-duality when $J_1 = J_2$ and $J_5 = J_6$, which is induced by gauging the $\mathbb{Z}_3$ subgroup. We hence expect the same self-duality to hold for $\mathsf{Rep}(S_3)$-symmetric Hamiltonian (3.13) too. To understand the self-duality of Hamiltonian (3.13), we will show that gauging the entire $S_3$ symmetry delivers another bond algebra with $\mathsf{Rep}(S_3^\vee)$ symmetry. This can be achieved by first gauging $\mathbb{Z}_3$ and then $\mathbb{Z}_2^\vee$ symmetry of dual $S_3^\vee$ symmetry. Starting from the bond algebra (2.10) and gauging the $\mathbb{Z}_2^\vee$ symmetry generated by $\widehat{U}_s^\vee$ defined in Eq. (2.11), we find the bond algebra

$$
\mathfrak{B}_{S_3/S_3} := \Big\langle \hat{\tau}_i^z, \ \hat{\tau}_i^z\,\hat{\mu}_{i+1}^x, \ \hat{\mu}_i^z\,\hat{\tau}_i^x\,\hat{c}_i\,\hat{\mu}_{i+1}^z, \ \hat{\tau}_i^x, \ \Big(\hat{z}_i^{\hat{\mu}_{i+1}^x}\,\hat{z}_{i+1}^\dagger + \hat{z}_i^{-\hat{\mu}_{i+1}^x}\,\hat{z}_{i+1}\Big), \ \Big(\hat{x}_i + \hat{x}_i^\dagger\Big),
$$
$$
\hat{\mu}_{i+1}^x\,\Big(\hat{z}_i^{\hat{\mu}_{i+1}^x}\,\hat{z}_{i+1}^\dagger - \hat{z}_i^{-\hat{\mu}_{i+1}^x}\,\hat{z}_{i+1}\Big), \ \hat{\tau}_i^z\,\hat{\mu}_{i+1}^x\,\Big(\hat{x}_{i+1} - \hat{x}_{i+1}^\dagger\Big) \ \Big| \ i \in \Lambda \Big\rangle, \qquad (3.14)
$$

that is dual to the algebra (2.10) under gauging the $\mathbb{Z}_2^\vee$ symmetry. We notice that this algebra has the same terms as algebra (3.5). Therefore, its commutant algebra is that of the category $\mathsf{Rep}(S_3^\vee)$. The simple objects in $\mathsf{Rep}(S_3^\vee)$ are represented by the operators

$$
\widehat{W}_{\mathbf{1}}^\vee := \hat{\mathbb{1}}, \quad \widehat{W}_{\mathbf{1}'}^\vee := \prod_{i=1}^L \hat{\mu}_i^x, \quad \widehat{W}_{\mathbf{2}}^\vee := \frac{1}{2}\left(1 + \prod_{i=1}^L \hat{\mu}_i^x\right)\left[\prod_{i=1}^L \hat{x}_i^{\prod_{k=2}^i \hat{\mu}_k^x} + \hat{x}_i^{-\prod_{k=2}^i \hat{\mu}_k^x}\right]. \quad (3.15)
$$

To find in which subalgebra the duality holds, we combine Eqs. (2.12) and (3.7) that describe the consistency conditions imposed by gauging $\mathbb{Z}_3$ and $\mathbb{Z}_2$ subgroups, respectively. We find that the duality induced by gauging the entire group $S_3$ holds between the subalgebras

$$
\mathfrak{B}_{S_3}\Big|_{\widehat{U}_s = \widehat{U}_r = 1} \cong \mathfrak{B}_{S_3/S_3}\Big|_{\widehat{W}_{\mathbf{1}'}^\vee = 1, \ \widehat{W}_{\mathbf{2}}^\vee = 2}. \qquad (3.16)
$$

This consistency condition says that the duality maps the $S_3$-singlet subalgebra of $\mathfrak{B}_{S_3}$ to the subalgebra of $\mathfrak{B}_{S_3/S_3}$ where the representation of each simple object is equal to its quantum dimension. The image of Hamiltonian (2.3) under gauging the entire $S_3$ symmetry is

$$
\begin{aligned}
\widehat{H}_{\mathsf{Rep}(S_3^\vee)} := &- J_1 \sum_{i=1}^L \Big(\hat{x}_i + \hat{x}_i^\dagger\Big) - J_2 \sum_{i=1}^L \Big(\hat{z}_i^{\hat{\mu}_{i+1}^x}\,\hat{z}_{i+1}^\dagger + \hat{z}_i^{-\hat{\mu}_{i+1}^x}\,\hat{z}_{i+1}\Big) \\
&- J_3 \sum_{i=1}^L \Big(\hat{\tau}_i^z + \hat{\tau}_i^z\,\hat{\mu}_{i+1}^x\Big) - J_4 \sum_{i=1}^L \Big(\hat{\mu}_i^z\,\hat{\tau}_i^x\,\widehat{C}_i\,\hat{\mu}_{i+1}^z + \hat{\tau}_i^x\Big) \\
&- J_5 \sum_{i=1}^L \mathrm{i}\,\hat{\tau}_i^z\,\hat{\mu}_{i+1}^x\,\Big(\hat{x}_{i+1} - \hat{x}_{i+1}^\dagger\Big) - J_6 \sum_{i=1}^L \mathrm{i}\,\hat{\mu}_{i+1}^x\,\Big(\hat{z}_i^{\hat{\mu}_{i+1}^x}\,\hat{z}_{i+1}^\dagger - \hat{z}_i^{-\hat{\mu}_{i+1}^x}\,\hat{z}_{i+1}\Big).
\end{aligned}
$$
$$
(3.17)
$$

---

[22]We use the superscript $\vee$ to differentiate the $\mathsf{Rep}(S_3)$ symmetry of the bond algebra (3.5) from the $\mathsf{Rep}(S_3^\vee)$ symmetry of the bond algebra (3.14).

This Hamiltonian is unitarily equivalent to the Hamiltonian (3.13) under the interchange $(J_1, J_5) \leftrightarrow (J_2, J_6)$. The unitary transformation connecting the two Hamiltonians is the unitary $\hat{\mathfrak{t}}_{\mathsf{Rep}(S_3)}$, whose definition and action on the $\mathsf{Rep}(S_3)$-symmetric bond algebra generators are described in Appendix C.2.

In Sec. 2.2, we gave the explicit form of an operator that performs the $\mathbb{Z}_3$ Kramers-Wannier duality transformation for the $S_3$-symmetric Hamiltonian (2.3). Gauging the $\mathbb{Z}_2$ subgroup of the original $S_3$ symmetry leads to the $\mathsf{Rep}(S_3)$ symmetry. So we would like to apply the same $\mathbb{Z}_2$ gauging map to $\widehat{D}_{\mathrm{KW}}$ to obtain the sequential quantum circuit that implements the duality under the $(J_1, J_5) \leftrightarrow (J_2, J_6)$ exchange. To that end, we follow how the individual operators (or, gates in the quantum circuit language) in (2.15) transform under this gauging map. The full gauged operator has the form

$$\widehat{D}_{\mathsf{Rep}(S_3)} := \hat{\mathfrak{t}}_{\mathsf{Rep}(S_3)} \, \widehat{P}_{\mathrm{reg}} \, \widehat{D}_0 \,, \tag{3.18}$$

where (i) the unitary $\widehat{D}_0$ is defined as

$$\widehat{D}_0 := \left( \sum_{\alpha=0}^{2} \widehat{Z}_L^{\alpha} \, \widehat{P}_{Z_1^{\mu_1^x} Z_L = \omega^\alpha} \right) \left( \prod_{j=1}^{L-1} \widehat{\mathfrak{H}}_j^\dagger \, \widehat{\mathrm{CZ}}_{j+1,j}^{-\hat{\mu}_{j+1}^x} \right) , \tag{3.19}$$

(ii) the projector $\widehat{P}_{\mathrm{reg}} := \frac{1}{6} \widehat{W}_{\mathrm{reg}}$ is defined in terms of the operator $\widehat{W}_{\mathrm{reg}}$, corresponding to the (non-simple) regular representation object $\mathrm{reg} = \mathbf{1} \oplus \mathbf{1'} \oplus 2\,\mathbf{2}$ of the $\mathsf{Rep}(S_3)$ fusion category, i.e.,

$$\widehat{W}_{\mathrm{reg}} := \widehat{W}_{\mathbf{1}} + \widehat{W}_{\mathbf{1'}} + 2\widehat{W}_{\mathbf{2}} \,,$$

and (iii) the unitary $\hat{\mathfrak{t}}_{\mathsf{Rep}(S_3)}$ is the operator obtained under the action of the $\mathbb{Z}_2$-gauging map on the *half-translation* operator $\hat{\mathfrak{t}}_{\mathbb{Z}_2}$ defined in Eq. (2.14). The explicit details of this operator are provided in Appendix C.2. The projector $\widehat{P}_{\mathrm{reg}}$ annihilates any state that is not $\mathsf{Rep}(S_3)$ symmetric since for any irrep $R = \mathbf{1}, \mathbf{1'}, \mathbf{2}$ of $S_3$, the identities

$$\widehat{P}_{\mathrm{reg}} \frac{1}{d_R} \widehat{W}_R = \widehat{P}_{\mathrm{reg}}, \qquad \widehat{P}_{\mathrm{reg}} \left( \hat{1} - \frac{1}{d_R} \widehat{W}_R \right) = 0,$$

hold. The duality operator $\widehat{D}_{\mathsf{Rep}(S_3)}$ acts on the $\mathsf{Rep}(S_3)$-symmetric bond algebra (see Appendix C.2 for details) as

$$\widehat{D}_{\mathsf{Rep}(S_3)} \begin{pmatrix} \widehat{X}_j + \mathrm{H.c.} \\ \widehat{Z}_j^{\hat{\mu}_{j+1}^x} \widehat{Z}_{j+1}^\dagger + \mathrm{H.c.} \\ \widehat{X}_j - \mathrm{H.c.} \\ \hat{\tau}_i^z \, \hat{\mu}_{i+1}^x \left( \widehat{Z}_i^{\hat{\mu}_{i+1}^x} \widehat{Z}_{i+1}^\dagger - \mathrm{H.c.} \right) \end{pmatrix} = \begin{pmatrix} \widehat{Z}_j^{\hat{\mu}_{j+1}^x} \widehat{Z}_{j+1}^\dagger + \mathrm{H.c.} \\ \widehat{X}_{j+1} + \mathrm{H.c.} \\ \hat{\tau}_j^z \hat{\mu}_{j+1}^x \left( \widehat{Z}_j^{\hat{\mu}_{j+1}^x} \widehat{Z}_{j+1}^\dagger - \mathrm{H.c.} \right) \\ \widehat{X}_{j+1} - \mathrm{H.c.} \end{pmatrix} \widehat{D}_{\mathsf{Rep}(S_3)}, \tag{3.20}$$

which implements the self-duality transformation $(J_1, J_5) \leftrightarrow (J_2, J_6)$ of Eq. (3.13). The operator $\widehat{D}_{\mathsf{Rep}(S_3)}$ is non-invertible since it contains the projector $\widehat{P}_{\mathrm{reg}}$. The operator $\widehat{D}_{\mathsf{Rep}(S_3)}$ obeys the algebraic relations [23]

$$\widehat{D}_{\mathsf{Rep}(S_3)} \widehat{W}_R = \widehat{W}_R \widehat{D}_{\mathsf{Rep}(S_3)} = d_R \, \widehat{D}_{\mathsf{Rep}(S_3)}, \tag{3.21a}$$

$$\left( \widehat{D}_{\mathsf{Rep}(S_3)} \right)^2 = \widehat{P}_{\mathrm{reg}} \left( \hat{\mathfrak{t}}_{\mathsf{Rep}(S_3)} \widehat{D}_0 \right)^2 = \widehat{P}_{\mathrm{reg}} \widehat{T}, \tag{3.21b}$$

---

[23] We use the fact that $\hat{\mathfrak{t}}_{\mathsf{Rep}(S_3)}$ and $\widehat{D}_0$ commute with $\widehat{P}_{\mathrm{reg}}$.

$$\left( \widehat{D}_{\mathsf{Rep}(S_3)} \right)^\dagger = \widehat{P}_{\mathrm{reg}} \left( \hat{\mathfrak{t}}_{\mathsf{Rep}(S_3)} \widehat{D}_0 \right)^\dagger = \widehat{T}^\dagger \widehat{D}_{\mathsf{Rep}(S_3)}, \qquad (3.21c)$$

where $d_R$ is the dimension of the irreducible representation $R$, and $\widehat{T}$ is the operator translating both qubits and qutrits by one lattice site. Let us note that, the second line in the above set of equations implies

$$\left( \sqrt{6} \widehat{D}_{\mathsf{Rep}(S_3)} \right)^2 = \left( \widehat{W}_{\mathbf{1}} + \widehat{W}_{\mathbf{1}'} + 2\widehat{W}_{\mathbf{2}} \right) \widehat{T}. \qquad (3.22)$$

Following the discussion for Eq. (2.20), an analogous calculation of the quantum dimension suggests

$$d^2_{\widehat{D}_{\mathsf{Rep}(S_3)}} = 1 + 1 + 2 \cdot 2 = 6 \implies d_{\widehat{D}_{\mathsf{Rep}(S_3)}} = \sqrt{6} \qquad (3.23)$$

The quantum dimension calculated above as well as the fusion rule in Eq. (3.22) are in tension with the category theoretic result [70] which suggests that a duality defect symmetry $\mathcal{D}$ arising from "half-gauging" by an algebra object $A$ must satisfy

$$\mathcal{D}^2 = A, \quad d_{\mathcal{D}} = \sqrt{\langle A \rangle}. \qquad (3.24)$$

In our case, the self-duality symmetry generated by $\widehat{D}_{\mathsf{Rep}(S_3)}$ corresponds to a duality defect associated with gauging by the algebra object $A = \mathbf{1} \oplus \mathbf{2}$, as we argue in Sec. 5.1. Therefore we should expect the quantum dimension to be $\sqrt{3}$. This highlights an important subtlety in calculating quantum dimension of self-duality symmetries on the lattice when considering self-duality symmetries associated with gauging of non-invertible symmetries by general algebra objects. A more careful way to compute the quantum dimension, as well as the fusion rules, involves unitary operators that move non-invertible symmetry defects in a lattice Hamiltonian, as is done in Ref. [76].

## 3.3 Phase diagram

To discuss the phase diagram of the Hamiltonian (3.13), we first reparameterize it as

$$\begin{aligned}
\widehat{H}_{\mathsf{Rep}(S_3)} = &- J_1 \cos\theta \sum_{i=1}^L \left( \widehat{Z}_i^{\hat{\mu}_{i+1}^x} \widehat{Z}_{i+1}^\dagger + \mathrm{H.c.} \right) - J_1 \sin\theta \sum_{i=1}^L \mathrm{i}\, \hat{\tau}_i^z\, \hat{\mu}_{i+1}^x \left( \widehat{Z}_i^{\hat{\mu}_{i+1}^x} \widehat{Z}_{i+1}^\dagger - \mathrm{H.c.} \right) \\
&- J_2 \cos\theta \sum_{i=1}^L \left( \widehat{X}_i + \widehat{X}_i^\dagger \right) - J_2 \sin\theta \sum_{i=1}^L \mathrm{i} \left( \widehat{X}_i - \widehat{X}_i^\dagger \right) \\
&- J_3 \sum_{i=1}^L \left( \hat{\tau}_i^z + \hat{\tau}_i^z\, \hat{\mu}_{i+1}^x \right) - J_4 \sum_{i=1}^L \left( \hat{\mu}_i^z\, \hat{\tau}_i^x\, \widehat{C}_i\, \hat{\mu}_{i+1}^z + \hat{\tau}_i^x \right).
\end{aligned}$$
$$(3.25)$$

As in Sec. 2.3, we will explore the phase diagram of this Hamiltonian as a function of dimensionless ratios $J_1/J_2$ and $J_3/J_4$, for the cases of $\theta = 0$, non-zero but small $\theta \approx 0$, and large $\theta \sim 0.7$.

### 3.3.1 Analytical arguments

When studying the phase diagram of (3.25), we can utilize its duality to the Hamiltonian (2.21), under the $\mathbb{Z}_2$-gauging map. When $\theta = 0$, we again identify four gapped phases that correspond to four distinct symmetry breaking patterns for $\mathsf{Rep}(S_3)$ as follows.

(i) When $J_1 = J_4 = 0$, Hamiltonian (3.25) becomes

$$\widehat{H}_{\mathsf{Rep}(S_3);2,3} := -\sum_{i=1}^{L} \left[ J_2 \left( \widehat{X}_i + \widehat{X}_i^\dagger \right) + J_3 \left( \hat{\tau}_i^z + \hat{\tau}_i^z \, \hat{\mu}_{i+1}^x \right) \right]. \tag{3.26}$$

The qubits and qutrits are decoupled and all terms in the Hamiltonian pairwise commute. There is a single nondegenerate gapped ground state

$$|\mathrm{GS}_{\mathsf{Rep}(S_3)}\rangle := \bigotimes_{i=1}^{L} |\tau_i^z = 1, \, \mu_i^x = 1, \, X_i = 1\rangle, \tag{3.27}$$

which is symmetric under the entire $\mathsf{Rep}(S_3)$ symmetry. At this point, it is instructive to note that there is a subtle distinction between states symmetric under invertible and non-invertible symmetries that is implicitly used in the above discussion. Namely, the state $|\mathrm{GS}_{\mathsf{Rep}(S_3)}\rangle$ transforms as

$$\widehat{W}_{\mathbf{1}'} |\mathrm{GS}_{\mathsf{Rep}(S_3)}\rangle = |\mathrm{GS}_{\mathsf{Rep}(S_3)}\rangle, \quad \widehat{W}_{\mathbf{2}} |\mathrm{GS}_{\mathsf{Rep}(S_3)}\rangle = 2 \, |\mathrm{GS}_{\mathsf{Rep}(S_3)}\rangle \tag{3.28}$$

where the factor of 2 reflects the quantum dimension of the non-invertible symmetry $\widehat{W}_{\mathbf{2}}$.[24] This is the lattice analogue of the field theory result that the vacuum expectation value (vev) of a non-invertible topological line defect is its quantum dimension. More generally, we say that a state spontaneously breaks a non-invertible symmetry if its expectation value is vanishing.

The $S_3$-symmetric Hamiltonian (2.21) has twofold degenerate ground states (2.23) when $J_1 = J_4$ and $\theta = 0$. Under the duality mapping, the unique ground state (3.27) is the image of the symmetric linear combination of these ground states, i.e., $|\mathrm{GS}_{\mathbb{Z}_2}^+\rangle + |\mathrm{GS}_{\mathbb{Z}_2}^-\rangle$. This is because the duality only holds in the subspace specified in Eq. (3.7).

(ii) When $J_2 = J_4 = 0$, Hamiltonian (3.25) becomes

$$\widehat{H}_{\mathsf{Rep}(S_3);1,3} := -\sum_{i=1}^{L} \left[ J_1 \left( \widehat{Z}_i^{\hat{\mu}_{i+1}^x} \, \widehat{Z}_{i+1}^\dagger + \mathrm{H.c.} \right) + J_3 \left( \hat{\tau}_i^z + \hat{\tau}_i^z \, \hat{\mu}_{i+1}^x \right) \right]. \tag{3.29}$$

Qubits are again in the disordered phase which pins their value to $\hat{\tau}_i^z = 1$ and $\hat{\mu}_i^x = 1$ subspace. This means that the $J_1$ term simply reduces to the classical 3-state Potts model. There are three degenerate ground states

$$|\mathrm{GS}_{\mathbf{1}'}^\alpha\rangle := \bigotimes_{i=1}^{L} |\tau_i^z = 1, \, \mu_i^x = 1, \, Z_i = \omega^\alpha\rangle. \tag{3.30}$$

---

[24]To ensure the consistency with the fusion rules of $\mathsf{Rep}(S_3)$, the numerical pre-factor in Eq. (3.28) is essential. On the one hand,

$$\widehat{W}_{\mathbf{2}}^2 |\mathrm{GS}_{\mathsf{Rep}(S_3)}\rangle = \widehat{W}_{\mathbf{2}} \left( \widehat{W}_{\mathbf{2}} |\mathrm{GS}_{\mathsf{Rep}(S_3)}\rangle \right) = 4 \, |\mathrm{GS}_{\mathsf{Rep}(S_3)}\rangle,$$

while on the other,

$$\widehat{W}_{\mathbf{2}}^2 |\mathrm{GS}_{\mathsf{Rep}(S_3)}\rangle = \left( \widehat{W}_{\mathbf{1}} + \widehat{W}_{\mathbf{1}'} + \widehat{W}_{\mathbf{2}} \right) |\mathrm{GS}_{\mathsf{Rep}(S_3)}\rangle = (1 + 1 + 2) \, |\mathrm{GS}_{\mathsf{Rep}(S_3)}\rangle = 4 \, |\mathrm{GS}_{\mathsf{Rep}(S_3)}\rangle.$$

For a general non-invertible symmetry $\sigma$, its eigenvalue corresponding to a symmetric state must match its quantum dimension $d_\sigma$.

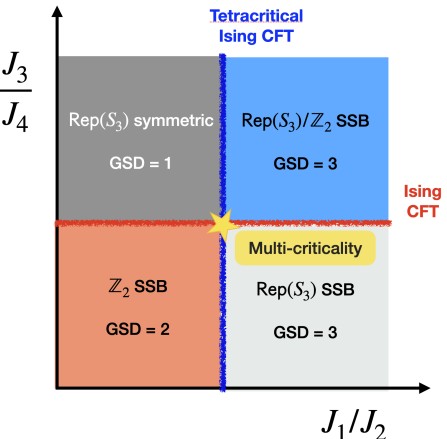

Figure 7: Phase diagram of Hamiltonian (3.25) based on analytical arguments at $\theta = 0$. The critical points are guesses based on duality arguments and various simple limits.

These ground states preserve the $\mathbb{Z}_2$ subgroup generated by $\widehat{W}_{\mathbf{1}'}$ while they break the non-invertible $\widehat{W}_{\mathbf{2}}$ symmetry. Under the latter each ground state is mapped to equal superposition of the the other two, i.e.,

$$
\begin{aligned}
\widehat{W}_{\mathbf{2}} \, |\mathrm{GS}^1_{\mathbf{1}'}\rangle &= |\mathrm{GS}^2_{\mathbf{1}'}\rangle + |\mathrm{GS}^3_{\mathbf{1}'}\rangle, \\
\widehat{W}_{\mathbf{2}} \, |\mathrm{GS}^2_{\mathbf{1}'}\rangle &= |\mathrm{GS}^3_{\mathbf{1}'}\rangle + |\mathrm{GS}^1_{\mathbf{1}'}\rangle, \\
\widehat{W}_{\mathbf{2}} \, |\mathrm{GS}^3_{\mathbf{1}'}\rangle &= |\mathrm{GS}^1_{\mathbf{1}'}\rangle + |\mathrm{GS}^2_{\mathbf{1}'}\rangle.
\end{aligned}
\tag{3.31}
$$

Note that the expectation value of $\widehat{W}_{\mathbf{2}}$ is zero in any of these ground states. We call this phase $\mathsf{Rep}(S_3)/\mathbb{Z}_2$ SSB phase.

The $S_3$-symmetric Hamiltonian (2.21) has sixfold degenerate ground states (2.25) when $J_2 = J_4$ and $\theta = 0$. Under the duality mapping, each ground state $|\mathrm{GS}^\alpha_{\mathbf{1}'}\rangle$ is the image of the linear combinations $|\mathrm{GS}^{+;\alpha}_{\mathbb{Z}_1}\rangle + |\mathrm{GS}^{-;\alpha}_{\mathbb{Z}_1}\rangle$ that are in the subspace (3.7).

(iii) When $J_1 = J_3 = 0$, the Hamiltonian (3.25) becomes

$$
\widehat{H}_{\mathsf{Rep}(S_3);2,4} := -\sum_{i=1}^{L} \left[ J_2 \left( \widehat{X}_i + \widehat{X}_i^\dagger \right) + J_4 \left( \hat{\mu}_i^z \, \hat{\tau}_i^x \, \widehat{C}_i \, \hat{\mu}_{i+1}^z + \hat{\tau}_i^x \right) \right].
\tag{3.32}
$$

First, we note that all terms in the Hamiltonian pairwise commute. Therefore we can set $\hat{\tau}_i^x = 1$. Second, we can minimize the $J_2$ term by setting $\widehat{X}_i = 1$, for which $\widehat{C}_i = 1$ too. This leaves us with the $J_4$ that is reduced to $J_4 \, \hat{\mu}_i^z \, \hat{\mu}_{i+1}^z$. This term favors twofold degenerate ground states for $\hat{\mu}$ degrees of freedom, i.e.,

$$
|\mathrm{GS}^\pm_{\mathbf{2}}\rangle := \bigotimes_{i=1}^{L} |\tau_i^x = 1, \, \mu_i^z = \pm 1, \, X_i = 1\rangle.
\tag{3.33}
$$

These ground states break the entire $\mathsf{Rep}(S_3)$ symmetry. First, the two ground states are mapped to each other under the $\mathbb{Z}_2$ symmetry generated by $\widehat{W}_{\mathbf{1}'}$. Second, one verifies

$$
\widehat{W}_{\mathbf{2}} \, |\mathrm{GS}^\pm_{\mathbf{2}}\rangle = |\mathrm{GS}^+_{\mathbf{2}}\rangle + |\mathrm{GS}^-_{\mathbf{2}}\rangle,
\tag{3.34}
$$

i.e., both ground states are mapped to the same linear combination under $\widehat{W}_{\mathbf{2}}$. This is to say that the vev of $\widehat{W}_{\mathbf{2}}$ is 1 in both of the ground states. While this does not match the quantum dimension of by $\widehat{W}_{\mathbf{2}}$, we say that the non-invertible $\widehat{W}_{\mathbf{2}}$ symmetry is not spontaneously broken. For this reason we call this phase $\mathbb{Z}_2$ SSB phase.[25] We provide further motivation for this interpretation in Sec. 5.2 where we computed expectation values of order and disorder operators in ground states (3.33).

The $S_3$-symmetric Hamiltonian (2.21) has a non-degenerate ground state (2.22) when $J_1 = J_3$ and $\theta = 0$. Under the duality mapping, the *cat state*, $|\text{GS}_{\mathbf{2}}^+\rangle + |\text{GS}_{\mathbf{2}}^-\rangle$ is the image of this non-degenerate ground state $|\text{GS}_{S_3}\rangle$.

(iv) When $J_2 = J_3 = 0$, the Hamiltonian (3.25) becomes

$$\widehat{H}_{\text{Rep}(S_3);1,4} := -\sum_{i=1}^{L} \left\{ J_1 \left( \widehat{Z}_i^{\hat{\mu}_{i+1}^x} \widehat{Z}_{i+1}^\dagger + \text{H.c.} \right) + J_4\, \hat{\mu}_i^z\, \widehat{C}_i\, \hat{\mu}_{i+1}^z \right\} + \text{const}. \quad (3.35)$$

Again, the two set of operators commute, so we can simultaneously diagonalize the operators and minimize their eigenvalues. There are three degenerate ground states. Two of them are quite simple, because they are obtained by setting $\widehat{Z}_i = 1$ for all sites. As in the discussion of Eq. (3.32), such states are eigenvalue $+1$ eigenstates of the charge conjugation operators $\widehat{C}_i$. The second term of Eq. (3.35) simply becomes an Ising-like term for the qubits, which favors a twofold degenerate ground state manifold spanned by

$$|\text{GS}_{\mathbf{1}}^\pm\rangle := \bigotimes_{i=1}^{L} |\tau_i^x = 1, \mu_i^z = \pm 1, Z_i = 1\rangle. \quad (3.36a)$$

The third degenerate ground state is

$$|\text{GS}_{\mathbf{1}}^3\rangle := \frac{1}{2^{L/2}} \sum_{\{s_i = \pm 1\}} \bigotimes_{i=1}^{L} |\tau_i^x = 1, \mu_i^x = s_i s_{i-1}, Z_i = \omega^{s_i}\rangle. \quad (3.36b)$$

The assignment of the $\hat{\mu}^x$ eigenvalues ensures that the $J_1$ term of Eq. (3.35) is minimized in each summand of (3.36b), while the superposition over different $\{s_i\}$ configurations ensures that the $J_4$ term is minimized.[26] All three of these states have the minimum possible energy associated with minimizing the eigenvalue of each of the two set of commuting terms. Under the action of $\text{Rep}(S_3)$ symmetry these ground states transform as

$$\widehat{W}_{\mathbf{1}'} |\text{GS}_{\mathbf{1}}^+\rangle = |\text{GS}_{\mathbf{1}}^-\rangle, \quad \widehat{W}_{\mathbf{1}'} |\text{GS}_{\mathbf{1}}^-\rangle = |\text{GS}_{\mathbf{1}}^+\rangle, \quad \widehat{W}_{\mathbf{1}'} |\text{GS}_{\mathbf{1}}^3\rangle = |\text{GS}_{\mathbf{1}}^3\rangle, \quad (3.37a)$$

and

$$\widehat{W}_{\mathbf{2}} |\text{GS}_{\mathbf{1}}^3\rangle = |\text{GS}_{\mathbf{1}}^+\rangle + |\text{GS}_{\mathbf{1}}^-\rangle + |\text{GS}_{\mathbf{1}}^3\rangle, \quad \widehat{W}_{\mathbf{2}} \left( |\text{GS}_{\mathbf{1}}^+\rangle + |\text{GS}_{\mathbf{1}}^-\rangle \right) = 2\, |\text{GS}_{\mathbf{1}}^3\rangle. \quad (3.37b)$$

We interpret this as the $\text{Rep}(S_3)$ SSB pattern as the vev $\widehat{W}_{\mathbf{2}}$ is vanishing in ground states $|\text{GS}_{\mathbf{1}}^\pm\rangle$. We provide further motivation for this interpretation in Sec. 5.2 where

---

[25]See also Ref. [82] where the same terminology is used.

[26]Even though it is not immediately obvious that $|\text{GS}_{\mathbf{1}}^3\rangle$ is a short-range entangled state, in fact, it is related to a product state by the action of a finite depth local unitary circuit, $|\text{GS}_{\mathbf{1}}^3\rangle = \prod_{j=1}^{L} \left( \widehat{C\mu^z}_{j,j}\widehat{C\mu^z}_{j,j+1} \right) \bigotimes_{i=1}^{L} \left[ |\tau_i^x = 1, \mu_i^x = 1\rangle \otimes \frac{1}{\sqrt{2}} \left( |Z_i = \omega\rangle + |Z_i = \omega^*\rangle \right) \right]$, where $\widehat{C\mu^z}_{i,j}$ is a kind of CZ operator that acts as the identity operator if $Z_i = \omega$ and as $\hat{\mu}_j^z$ if $Z_i = \omega^*$.

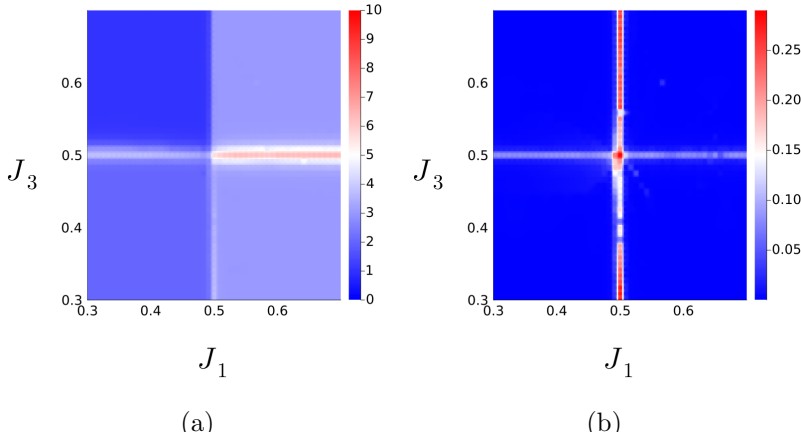

$$(a) \qquad\qquad (b)$$

Figure 8: Numerical phase diagram showing GSD (a) and central charge (b) as heatmaps, as a function of $J_1$ and $J_3$, with $J_2 = 1 - J_1$, $J_4 = 1 - J_3$ everywhere and with fixed $\theta = 0.1$. The effective system size is $L = 128$.

we computed expectation values of order and disorder operators in ground states (3.36).

The $S_3$-symmetric Hamiltonian (2.21) has a threefold degenerate ground states (2.24) when $J_2 = J_3$ and $\theta = 0$. Under the duality mapping, the linear combination $\left|\mathrm{GS}_\mathbf{1}^+\right\rangle + \left|\mathrm{GS}_\mathbf{1}^-\right\rangle$ is the image of the linear combination $\left|\mathrm{GS}_{\mathbb{Z}_2}^1\right\rangle + \left|\mathrm{GS}_{\mathbb{Z}_2}^2\right\rangle$ while the ground state $\left|\mathrm{GS}_\mathbf{1}^3\right\rangle$ is the image of $\left|\mathrm{GS}_{\mathbb{Z}_2}^0\right\rangle$.

In conclusion, we have identified 4 fixed-point gapped ground states of the $\mathsf{Rep}(S_3)$-symmetric Hamiltonian (3.25). On general grounds (see Sec. 5), this symmetry category is indeed expected to have 4 gapped phases. Therefore, we find consistency between our lattice model and general category theoretic arguments. Again, for the gapped phases, turning on small non-zero $\theta$ makes no difference.

The continuous phase transitions between these gapped phase can also be obtained from those between the gapped phases of $S_3$-symmetric Hamiltonian (2.21). More precisely, in the language of conformal field theory, the gauging procedure we performed in Sec. 3.1 corresponds to the orbifold construction. Namely, for the Ising and 3-state Potts CFTs, gauging the $\mathbb{Z}_2$ subgroup of $S_3$ can be achieved by orbifolding the Ising symmetry and charge conjugation symmetry of these CFTs, respectively. Under orbifolding the central charge of the CFT does not change while the local operator content does [102–106]. In particular, under the $\mathbb{Z}_2$ orbifold operation, the Ising CFT is isomorphic to itself, while the Potts CFT is mapped to tetracritical Ising (TCI).[27] As a result, we expect the same reasoning behind the stability of the critical lines and multicritical point to small non-zero $\theta$ to hold for the $\mathsf{Rep}(S_3)$-symmetric Hamiltonian (3.25) as well. In the following section, we verify these expectations by providing numerical evidence obtained through the TEFR and DMRG algorithms.

### 3.3.2 Numerical results

As we did in Sec. 2.3.2, we implement the TEFR algorithm to obtain the phase diagram of the Hamiltonian (3.25). We extract the ground state degeneracies and the central charges using the approach described in Sec. 2.3.2. For simplicity, we only focus on the

---

[27]It is known that the TCI CFT has $\mathsf{Rep}(S_3)$ topological defect lines [3].

case of small $\theta \approx 0$ limit. The results are shown in Fig. 8. We find, as expected, four gapped phases of the $\mathsf{Rep}(S_3)$-symmetric Hamiltonian (3.25) along with continuous phase transitions separating them from each other. We confirm using DMRG that the central charges at the continuous transition lines matches those for the phase diagram of $S_3$-symmetric Hamiltonian (2.3). In contrast, the ground state degeneracies of four gapped phases differ as they follow the $\mathsf{Rep}(S_3)$ SSB patterns. The duality between the gapped ground states then holds only in the symmetric subspaces (3.7).

In fact, the above reasoning also holds in the large $\theta$ limit of Hamiltonian (3.13). Just as it was the case for the Hamiltonian (2.3) with $S_3$ symmetry, around $\theta_* \sim \frac{\pi}{8}$ an extended gapless region opens up in the phase diagram. Similarly, we can add a term that breaks the non-invertible self-duality symmetry implemented by $\widehat{D}_{\mathsf{Rep}(S_3)}$ (recall Eq. (3.18)). This can be achieved by dualizing the perturbation (2.26). Under gauging the $\mathbb{Z}_2$ subgroup of $S_3$ perturbation (2.26) is mapped to

$$
\begin{aligned}
\widehat{H}_\perp := &- J_\perp \sum_{i=1}^{L} \left( \hat{\tau}_i^z \, \hat{\mu}_{i+1}^x - \hat{\mu}_{i+1}^z \, \hat{\tau}_{i+1}^x \, \widehat{C}_{i+1} \, \hat{\mu}_{i+2}^z \right) \left( \widehat{Z}_i^{\hat{\mu}_{i+1}^x} \, \widehat{Z}_{i+1}^\dagger + \widehat{Z}_i^{-\hat{\mu}_{i+1}^x} \, \widehat{Z}_{i+1} \right) \\
&+ J_\perp \sum_{i=1}^{L} (\hat{\tau}_i^z - \hat{\tau}_i^x) \left( \widehat{X}_i + \widehat{X}_i^\dagger \right),
\end{aligned}
\tag{3.38}
$$

which is odd under the non-invertible $\widehat{D}_{\mathsf{Rep}(S_3)}$ symmetry. When this term is added to the $\mathsf{Rep}(S_3)$-symmetric Hamiltonian (3.13), depending on the sign of $J_\perp$, shape of the phase diagram matches either that in Fig. 5b or Fig. 5a. This allows the direct continuous phase transitions between $\mathsf{Rep}(S_3)$-symmetric and $\mathsf{Rep}(S_3)$ SSB phases or between $\mathsf{Rep}(S_3)/\mathbb{Z}_2$ SSB and $\mathbb{Z}_2$ SSB phases.

# 4 Self-dual spin chains and their SymTO description

Emergent symmetries are an important characteristic feature of so-called gapless liquid states. These symmetries often take the form of generalized symmetries $e.g.$, higher-form, higher-group, non-invertible; each of these types may also be ('t Hooft) anomalous. A complete understanding of gapless phases, therefore, requires a general theory of emergent generalized symmetries. The SymTO framework [28] is a proposal for such a theory; it attempts to classify gapless states in terms of topological orders in one higher dimension.

In Secs. 2 and 3, we introduced spin chains which respect non-invertible self-duality symmetries. To understand how these non-invertible symmetries constrain the low energy dynamics of the lattice model, we need to use the symmetry-topological-order correspondence, and find out which SymTOs describe them. In what follows, we use this correspondence to first understand $S_3$ and $\mathsf{Rep}(S_3)$ symmetries. We will then obtain the SymTO that corresponds to the enhancement of these symmetries by the non-invertible self-duality symmetries.

## 4.1 SymTO of $S_3$ symmetry

The symmetry data of a 1+1d bosonic system with $S_3$ symmetry can be encapsulated completely in its SymTO [29], which is the $S_3$ quantum double $\mathcal{D}(S_3)$, $i.e.$, $S_3$ topological order in 2+1d. From the SymTO point of view, it has been argued [8, 44] that the

---

[28]See Appendix B for a review of SymTO.

|   $\mathcal{D}(S_3)$   |   $s$   |   $d$   |
|:---:|:---:|:---:|
| $\mathbf{1}$ | $0$ | $1$ |
| $\mathbf{1}'$ | $0$ | $1$ |
| $\mathbf{2}$ | $0$ | $2$ |
| $r$ | $0$ | $2$ |
| $r_1$ | $\frac{1}{3}$ | $2$ |
| $r_2$ | $\frac{2}{3}$ | $2$ |
| $s$ | $0$ | $3$ |
| $s_1$ | $\frac{1}{2}$ | $3$ |

(a)

$$
S = \begin{pmatrix}
1 & 1 & 2 & 2 & 2 & 2 & 3 & 3 \\
1 & 1 & 2 & 2 & 2 & 2 & -3 & -3 \\
2 & 2 & 4 & -2 & -2 & -2 & 0 & 0 \\
2 & 2 & -2 & 4 & -2 & -2 & 0 & 0 \\
2 & 2 & -2 & -2 & -2 & 4 & 0 & 0 \\
2 & 2 & -2 & -2 & 4 & -2 & 0 & 0 \\
3 & -3 & 0 & 0 & 0 & 0 & 3 & -3 \\
3 & -3 & 0 & 0 & 0 & 0 & -3 & 3
\end{pmatrix}
$$

(b)

|  | $\mathbf{1}$ | $\mathbf{1}'$ | $\mathbf{2}$ | $r$ | $r_1$ | $r_2$ | $s$ | $s_1$ |
|:---:|:---:|:---:|:---:|:---:|:---:|:---:|:---:|:---:|
| $\mathbf{1}$ | $\mathbf{1}$ | $\mathbf{1}'$ | $\mathbf{2}$ | $r$ | $r_1$ | $r_2$ | $s$ | $s_1$ |
| $\mathbf{1}'$ | $\mathbf{1}'$ | $\mathbf{1}$ | $\mathbf{2}$ | $r$ | $r_1$ | $r_2$ | $s_1$ | $s$ |
| $\mathbf{2}$ | $\mathbf{2}$ | $\mathbf{2}$ | $\mathbf{1}\oplus\mathbf{1}'\oplus\mathbf{2}$ | $r_1\oplus r_2$ | $r\oplus r_2$ | $r\oplus r_1$ | $s\oplus s_1$ | $s\oplus s_1$ |
| $r$ | $r$ | $r$ | $r_1\oplus r_2$ | $\mathbf{1}\oplus\mathbf{1}'\oplus r$ | $\mathbf{2}\oplus r_2$ | $\mathbf{2}\oplus r_1$ | $s\oplus s_1$ | $s\oplus s_1$ |
| $r_1$ | $r_1$ | $r_1$ | $r\oplus r_2$ | $\mathbf{2}\oplus r_2$ | $\mathbf{1}\oplus\mathbf{1}'\oplus r_1$ | $\mathbf{2}\oplus r$ | $s\oplus s_1$ | $s\oplus s_1$ |
| $r_2$ | $r_2$ | $r_2$ | $r\oplus r_1$ | $\mathbf{2}\oplus r_1$ | $\mathbf{2}\oplus r$ | $\mathbf{1}\oplus\mathbf{1}'\oplus r_2$ | $s\oplus s_1$ | $s\oplus s_1$ |
| $s$ | $s$ | $s_1$ | $s\oplus s_1$ | $s\oplus s_1$ | $s\oplus s_1$ | $s\oplus s_1$ | $\mathbf{1}\oplus\mathbf{2}\oplus r\oplus r_1\oplus r_2$ | $\mathbf{1}'\oplus\mathbf{2}\oplus r\oplus r_1\oplus r_2$ |
| $s_1$ | $s_1$ | $s$ | $s\oplus s_1$ | $s\oplus s_1$ | $s\oplus s_1$ | $s\oplus s_1$ | $\mathbf{1}'\oplus\mathbf{2}\oplus r\oplus r_1\oplus r_2$ | $\mathbf{1}\oplus\mathbf{2}\oplus r\oplus r_1\oplus r_2$ |

(c)

Table 1: (a) Topological spin $s$ and quantum dimension $d$ of the eight anyons of SymTO $\mathcal{D}(S_3)$. (b) $S$-matrix of these excitations which encodes mutual braiding statistics. (c) Fusion rules of these excitations.

gapped phases allowed by $S_3$ symmetry are in one-to-one correspondence with the gapped boundaries of the SymTO $\mathcal{D}(S_3)$.[29]

The $\mathcal{D}(S_3)$ SymTO has eight anyons whose topological spins $s$, quantum dimensions $d$, $S$-matrix, and fusion rules are given in Table 1. The anyons labeled $s$ and $r$ carry the gauge fluxes for their associated conjugacy class.[30] The anyon $\mathbf{1}'$ carries the 1-dimensional irrep of $S_3$ and can be viewed as a $\mathbb{Z}_2$ charge. From the $S$ matrix, we see that $\mathbf{1}'$ and $s$ have mutual $\pi$ statistics, consistent with $s$ being the $\mathbb{Z}_2$ flux.

There are four Lagrangian condensable algebras of $\mathcal{D}(S_3)$, which correspond to four maximal subsets of bosonic anyons in $\mathcal{D}(S_3)$ with trivial mutual statistics between them. We can condense all the anyons in a Lagrangian condensable algebra on 1+1d boundary of the 2+1d SymTO, which gives rise to a gapped boundary [19]. In turn, such condensable algebras correspond to gapped phases for systems with $S_3$ symmetry. The four Lagrangian condensable algebras of $\mathcal{D}(S_3)$ are denoted as follows:

(i) $\mathcal{A}_1 = \mathbf{1} \oplus \mathbf{1}' \oplus 2\,\mathbf{2}$ corresponds to an $S_3$ SSB phase, *i.e.*, $S_3$ ferromagnet

(ii) $\mathcal{A}_2 = \mathbf{1} \oplus \mathbf{1}' \oplus 2\,r$ corresponds to a $\mathbb{Z}_2$ SSB phase

(iii) $\mathcal{A}_3 = \mathbf{1} \oplus \mathbf{2} \oplus s$ corresponds to a $\mathbb{Z}_3$ SSB phase

(iv) $\mathcal{A}_4 = \mathbf{1} \oplus r \oplus s$ corresponds to an $S_3$-symmetric phase, *i.e.*, $S_3$ paramagnet

---

[29]Similar statements have also appeared elsewhere in the literature. [33, 82, 101]
[30]See Appendix A for notations.

## 4.2 SymTO of $\mathsf{Rep}(S_3)$ symmetry

As we exemplified in Sec. 3, the Morita equivalent $S_3$ and $\mathsf{Rep}(S_3)$ symmetries have the property that Hamiltonians with these symmetries have identical spectra when restricted to the respective symmetric sub-Hilbert space; this latter aspect was emphasized as "holo-equivalence" in Ref. [8]. As a result, $S_3$-symmetric models and $\mathsf{Rep}(S_3)$-symmetric models have identical phase diagrams. However, the corresponding phases and phase transitions may be given different names due to the difference in symmetry labels. The Morita equivalence between $S_3$ and $\mathsf{Rep}(S_3)$ symmetries follows from the fact that they have the same SymTO. The SymTO can be calculated by computing the algebra of the associated patch operators; some related examples were discussed in Refs. [53, 107].

The four Lagrangian condensable algebras of $\mathcal{D}(S_3)$ also give rise to four gapped phases for $\mathsf{Rep}(S_3)$ symmetric models. In terms of the Morita equivalent $\mathsf{Rep}(S_3)$ symmetry, $\mathcal{A}_1$ and $\mathcal{A}_4$ correspond to $\mathsf{Rep}(S_3)$-symmetric and $\mathsf{Rep}(S_3)$ SSB phases, respectively. The Lagrangian algebras $\mathcal{A}_2$ and $\mathcal{A}_3$ are more subtle; guided by the phase diagram of the lattice models introduced in Sec. 3, we find that they correspond to $\mathsf{Rep}(S_3)/\mathbb{Z}_2$ and $\mathbb{Z}_2$ SSB phases, respectively.[31]

## 4.3 SymTO of the self-duality symmetry

In Ref. [44], the authors also highlighted the importance of an automorphism of $\mathcal{D}(S_3)$ associated with the permutation of the anyons $\mathbf{2}$ and $r$. This automorphism of the SymTO suggests a non-invertible self-duality symmetry of the boundary theory.

Here, we would like stress an important difference between $\mathcal{D}(S_3)$ SymTO and $\mathcal{D}(S_3)$ SymTFT. In $\mathcal{D}(S_3)$ SymTFT, the automorphism $\mathbf{2} \leftrightarrow r$ implies a symmetry of SymTFT. In contrast, $\mathcal{D}(S_3)$ SymTO is just an 2+1d $S_3$ lattice gauge theory with matter. Thus in general, the SymTO (*i.e.*, the lattice gauge theory) does not have any symmetry. This corresponds to the fact that our $S_3$ and $\mathsf{Rep}(S_3)$ lattice models, in general, do not have the self-dual symmetry, and their SymTO is the $\mathcal{D}(S_3)$ SymTO, *without the $\mathbf{2} \leftrightarrow r$ automorphism symmetry* $\mathbb{Z}_2^{\mathbf{2}\leftrightarrow r}$.

The presence of $\mathbf{2} \leftrightarrow r$ automorphism in the $\mathcal{D}(S_3)$ SymTFT implies that we can fine tune the $\mathcal{D}(S_3)$ SymTO so that it has the automorphism symmetry $\mathbb{Z}_2^{\mathbf{2}\leftrightarrow r}$. This, in turn, implies that we can fine tune the $S_3$ and $\mathsf{Rep}(S_3)$ symmetric lattice models, so that these fine-tuned models have an additional self-duality symmetry. Such an existence of lattice self-duality symmetry was assumed in Ref. [45]; in Secs. 2 and 3, we explicitly constructed this lattice self-duality symmetry, and confirmed this conjecture.

The fine-tuned self-dual lattice models have a larger symmetry which include both the self-duality symmetry and either the $S_3$ or the $\mathsf{Rep}(S_3)$ symmetries. Thus, the self-dual lattice models must have a larger SymTO. Such a larger SymTO can be obtained by gauging the $\mathbb{Z}_2^{\mathbf{2}\leftrightarrow r}$ automorphism symmetry in $\mathcal{D}(S_3)$ SymTO. In Ref. [108], this guaging procedure was carried out and the larger SymTO is found to be either $\mathrm{SU}(2)_4 \boxtimes \overline{\mathrm{SU}(2)}_4$ or $\mathrm{JK}_4 \boxtimes \overline{\mathrm{JK}}_4$ topological order. Note that there still remains a two-fold ambiguity, coming from the possibility of the stacking a $\mathbb{Z}_2^{\mathbf{2}\leftrightarrow r}$ SPT state to the SymTO, before the $\mathbb{Z}_2^{\mathbf{2}\leftrightarrow r}$ gauging. The anyon data for the $\mathrm{SU}(2)_4$ and $\mathrm{JK}_4$ topological orders are shown in Table 2. From the fusion rule $e \otimes e = \mathbf{1}$, we see that $e$ carries a $\mathbb{Z}_2$ gauge charge. From the $S$-matrix, we see that $m$ (and $m_1$) carries the corresponding $\mathbb{Z}_2$ gauge flux. On an appropriate boundary of the SymTO, $e$ would reduce to a $\mathbb{Z}_2$ symmetry charge while $m$ (and $m_1$) would reduce to the corresponding domain walls.

---

[31]Here, we match the Lagrangian algebras with the gapped phases of the Hamiltonian (3.17) with $\mathsf{Rep}(S_3^\vee)$ symmetry.

| $\mathrm{SU(2)}_4$ | $s$ | $d$ |
|---|---|---|
| $\mathbf{1}$ | $0$ | $1$ |
| $e$ | $0$ | $1$ |
| $m$ | $\frac{1}{8}$ | $\sqrt{3}$ |
| $m_1$ | $\frac{5}{8}$ | $\sqrt{3}$ |
| $q$ | $\frac{1}{3}$ | $2$ |

| $\mathrm{JK}_4$ | $s$ | $d$ |
|---|---|---|
| $\mathbf{1}$ | $0$ | $1$ |
| $e$ | $0$ | $1$ |
| $m$ | $\frac{1}{8}$ | $\sqrt{3}$ |
| $m_1$ | $\frac{5}{8}$ | $\sqrt{3}$ |
| $q$ | $\frac{2}{3}$ | $2$ |

(a)

$$
S_{\mathrm{SU(2)}_4} = \begin{pmatrix}
1 & 1 & \sqrt{3} & \sqrt{3} & 2 \\
1 & 1 & -\sqrt{3} & -\sqrt{3} & 2 \\
\sqrt{3} & -\sqrt{3} & \sqrt{3} & -\sqrt{3} & 0 \\
\sqrt{3} & -\sqrt{3} & -\sqrt{3} & \sqrt{3} & 0 \\
2 & 2 & 0 & 0 & -2
\end{pmatrix}
\qquad
S_{\mathrm{JK}_4} = \begin{pmatrix}
1 & 1 & \sqrt{3} & \sqrt{3} & 2 \\
1 & 1 & -\sqrt{3} & -\sqrt{3} & 2 \\
\sqrt{3} & -\sqrt{3} & -\sqrt{3} & \sqrt{3} & 0 \\
\sqrt{3} & -\sqrt{3} & \sqrt{3} & -\sqrt{3} & 0 \\
2 & 2 & 0 & 0 & -2
\end{pmatrix}
$$

(b)

| | $\mathbf{1}$ | $e$ | $m$ | $m_1$ | $q$ |
|---|---|---|---|---|---|
| $\mathbf{1}$ | $\mathbf{1}$ | $e$ | $m$ | $m_1$ | $q$ |
| $e$ | $e$ | $\mathbf{1}$ | $m_1$ | $m$ | $q$ |
| $m$ | $m$ | $m_1$ | $\mathbf{1} \oplus q$ | $e \oplus q$ | $m \oplus m_1$ |
| $m_1$ | $m_1$ | $m$ | $e \oplus q$ | $\mathbf{1} \oplus q$ | $m \oplus m_1$ |
| $q$ | $q$ | $q$ | $m \oplus m_1$ | $m \oplus m_1$ | $\mathbf{1} \oplus e \oplus q$ |

(c)

Table 2: (a) Topological spin $s$ and quantum dimension $d$ of the five anyons of $\mathrm{SU(2)}_4$ and $\mathrm{JK}_4$ topological orders. (b) $S$-matrices of these excitations for the two topological orders which encodes mutual braiding statistics. (c) Fusion rules of these five anyons which is identical for $\mathrm{SU(2)}_4$ and $\mathrm{JK}_4$ topological orders.

Later, we will show that the generalized symmetries in our self-dual $S_3$-symmetric and self-dual $\mathsf{Rep}(S_3)$-symmetric models are both described by the $\mathrm{JK}_4 \boxtimes \overline{\mathrm{JK}_4}$ SymTO. Therefore, we momentarily concentrate on $\mathrm{JK}_4 \boxtimes \overline{\mathrm{JK}_4}$ SymTO. The $\mathrm{JK}_4 \boxtimes \overline{\mathrm{JK}_4}$ and $\mathcal{D}(S_3)$ SymTOs can have a gapped domain wall between them, which describes the breaking of the $\mathbb{Z}_2^{\mathbf{2}\leftrightarrow r}$ self-duality symmetry. This reduces the $\mathrm{JK}_4 \boxtimes \overline{\mathrm{JK}_4}$ SymTO to $\mathcal{D}(S_3)$ SymTO. Such a domain wall is created by the condensation of $e\bar{e}$ in $\mathrm{JK}_4 \boxtimes \overline{\mathrm{JK}_4}$ SymTO (and no condensation in the $\mathcal{D}(S_3)$ SymTO). More precisely, the domain wall is described by the following condensable algebra

$$
\mathcal{A} = (\mathbf{1},\mathbf{1},\mathbf{1}) \oplus (e,\bar{e},\mathbf{1}) \oplus (\mathbf{1},\bar{e},\mathbf{1}') \oplus (e,\mathbf{1},\mathbf{1}') \oplus (q,\bar{q},\mathbf{2}) \oplus (q,\bar{q},r) \oplus (q,\mathbf{1},r_1) \oplus (q,\bar{e},r_1)
$$
$$
\oplus (\mathbf{1},\bar{q},r_2) \oplus (e,\bar{q},r_2) \oplus (m,\bar{m},s) \oplus (m_1,\bar{m}_1,s) \oplus (m,\bar{m}_1,s_1) \oplus (m_1,\bar{m},s_1) \quad (4.1)
$$

in the topological order $\mathrm{JK}_4 \boxtimes \overline{\mathrm{JK}_4} \boxtimes \overline{\mathcal{D}}(S_3) = \mathrm{JK}_4 \boxtimes \overline{\mathrm{JK}_4} \boxtimes \mathcal{D}(S_3)$. This condensable algebra allows us to relate the anyons in $\mathcal{D}(S_3)$ SymTO to the anyons in $\mathrm{JK}_4 \boxtimes \overline{\mathrm{JK}_4}$ SymTO. The term $(\mathbf{1},\bar{e},\mathbf{1}')$ in $\mathcal{A}$ means that the anyon $\mathbf{1}'$ in $\mathcal{D}(S_3)$ SymTO and the anyon $\mathbf{1} \otimes \bar{e} = \bar{e}$ in $\mathrm{JK}_4 \boxtimes \overline{\mathrm{JK}_4}$ SymTO can condense on the domain wall. In other words, after going through the domain wall, the $\bar{e}$ anyon in $\mathrm{JK}_4 \boxtimes \overline{\mathrm{JK}_4}$ SymTO turns into the $\mathbf{1}'$ anyon in $\mathcal{D}(S_3)$ SymTO. Similarly, the term $(e,\mathbf{1},\mathbf{1}')$ in $\mathcal{A}$ means that, after going through the domain wall, the $e$ anyon in $\mathrm{JK}_4 \boxtimes \overline{\mathrm{JK}_4}$ SymTO turns into the $\mathbf{1}'$ anyon in $\mathcal{D}(S_3)$ SymTO. Thus the $e$ anyon and the $\bar{e}$ anyon in $\mathrm{JK}_4 \boxtimes \overline{\mathrm{JK}_4}$ SymTO carries the $\mathbb{Z}_2$-charge of the $S_3 = \mathbb{Z}_3 \rtimes \mathbb{Z}_2$ symmetry. The corresponding $\mathbb{Z}_2$-flux is carried by $m\bar{m}$ anyon in $\mathrm{JK}_4 \boxtimes \overline{\mathrm{JK}_4}$ SymTO, which

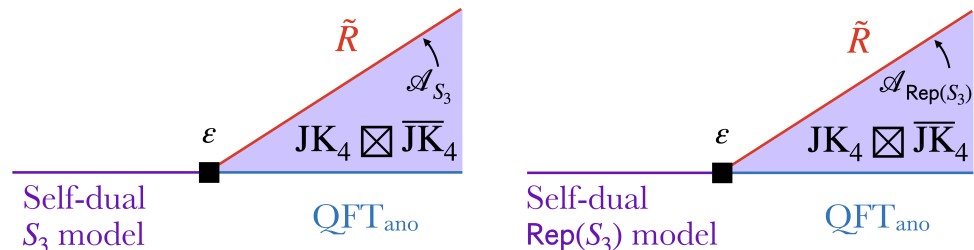

Figure 9: The isomorphic holographic decomposition reveals the generalized symmetry in self-dual $S_3$-symmetric models and in self-dual $\mathsf{Rep}(S_3)$-symmetric models, *i.e.*, reveal the SymTO $\mathcal{M} = \mathrm{JK}_4 \boxtimes \overline{\mathrm{JK}}_4$ and the fusion category $\widetilde{\mathcal{R}}$ for the symmetry defects: (Left) The bulk $\mathrm{JK}_4 \boxtimes \overline{\mathrm{JK}}_4$ SymTO and the $\mathcal{A}_{S_3}$-condensation induced topological boundary $\widetilde{\mathcal{R}}$ describes the $S_3$-symmetry plus the self-duality symmetry in the self-dual $S_3$-symmetric models. (Right) The bulk $\mathrm{JK}_4 \boxtimes \overline{\mathrm{JK}}_4$ SymTO and the $\mathcal{A}_{\mathsf{Rep}(S_3)}$-condensation induced topological boundary $\widetilde{\mathcal{R}}$ describes the $\mathsf{Rep}(S_3)$-symmetry plus the self-duality symmetry in the self-dual $\mathsf{Rep}(S_3)$-symmetric models.

has a $\pi$-mutual statistics with both $e$ and $\bar{e}$ anyons. This is also consistent with the term $(m, \bar{m}, s)$ in $\mathcal{A}$, which implies that after going through the domain wall, the $m\bar{m}$ anyon in $\mathrm{JK}_4 \boxtimes \overline{\mathrm{JK}}_4$ SymTO turns into the $s$ anyon (the $\mathbb{Z}_2$-flux) in $\mathcal{D}(S_3)$ SymTO. Thus, the string operator that creates a pair of $m\bar{m}$-anyons generates the $\mathbb{Z}_2$ (of $S_3$) transformation.

The Abelian anyon $e\bar{e}$ in $\mathrm{JK}_4 \boxtimes \overline{\mathrm{JK}}_4$ SymTO does not carry the $\mathbb{Z}_2$ charge of the $S_3$. But it carries the $\mathbb{Z}_2^{\mathbf{2} \leftrightarrow r}$ charge of the self-duality symmetry. This is consistent with the fact that the condensation $\mathcal{A} = \mathbf{1} \oplus e\bar{e}$ breaks the self-dual symmetry and reduces the $\mathrm{JK}_4 \boxtimes \overline{\mathrm{JK}}_4$ SymTO to $\mathcal{D}(S_3)$ SymTO [44, 45, 84]

$$(\mathrm{JK}_4 \boxtimes \overline{\mathrm{JK}}_4)_{/\mathbf{1} \oplus e\bar{e}} = \mathcal{D}(S_3). \tag{4.2}$$

To summarize, the anyons in $\mathrm{JK}_4 \boxtimes \overline{\mathrm{JK}}_4$ and $\mathcal{D}(S_3)$ SymTOs have the following identification under the condensation of $\mathbf{1} \oplus e\bar{e}$

$$e\bar{e} \to \mathbf{1}, \quad e \to \mathbf{1}', \quad \bar{e} \to \mathbf{1}', \quad m\bar{m} \to s, \quad q\bar{q} \to \mathbf{2} \oplus r, \quad q \to r_2, \quad \bar{q} \to r_1. \tag{4.3}$$

Since $e\bar{e}$ has $\pi$-mutual statistics with $m$ and $\bar{m}$ in $\mathrm{JK}_4 \boxtimes \overline{\mathrm{JK}}_4$ SymTO, anyons $m$ and $\bar{m}$ are $\mathbb{Z}_2^{\mathbf{2} \leftrightarrow r}$-flux for the self-duality symmetry. In other words, the string operator that creates a pair of $m$-anyons generates the self-duality transformation. Similarly, the string operator that creates a pair of $\bar{m}$-anyons also generates the self-duality transformation. The two transformations differ by a $\mathbb{Z}_2$ transformation. Because $m$ has quantum dimension $\sqrt{3}$, the transformation generated by the string operator of $m$ is an intrinsic non-invertible symmetry.[32] The non-integral quantum dimension also implies that the symmetry is anomalous,[33] since the anomaly-free generalized symmetry always have integral quantum dimensions [8, 10, 30].

In Refs. [8, 27, 30], an isomorphic holographic decomposition $(\epsilon, \widetilde{\mathcal{R}})$ of a model is introduced to expose the symmetry and the SymTO in the model (see Fig. 9):

$$\mathrm{model} \overset{(\epsilon, \widetilde{\mathcal{R}})}{\cong} \widetilde{\mathcal{R}} \boxtimes_{\mathrm{JK}_4 \boxtimes \overline{\mathrm{JK}}_4} \mathrm{QFT}_{\mathrm{ano}}, \tag{4.4}$$

---

[32]A (generalized) symmetry is called intrinsically non-invertible [85, 86] if all its Morita equivalent symmetries are non-invertible. This is to say that a symmetry with non-integral quantum dimension cannot be Morita equivalent to (sum of) simple objects with integral quantum dimension.

[33]By definition, a (generalized) symmetry is anomaly-free if it allows gapped non-degenerate ground state on all closed space.

where the boundary $\widetilde{\mathcal{R}}$ and the bulk $\mathrm{JK}_4 \boxtimes \overline{\mathrm{JK}}_4$ are assumed to have infinite energy gap. A similar picture was also obtained later in Refs. [31,32]; also see Ref. [45] and Appendix B for a short review. The isomorphic holographic decomposition (4.4) has the following physical meaning. The model is exactly simulated by the composite system $\widetilde{\mathcal{R}} \boxtimes_{\mathrm{JK}_4 \boxtimes \overline{\mathrm{JK}}_4} \mathrm{QFT}_{\mathrm{ano}}$. For example, the model and the composite system have the identical energy spectrum below the energy gaps of the boundary $\widetilde{\mathcal{R}}$ and the bulk. The local low-energy properties of the model are captured by a quantum field theory $\mathrm{QFT}_{\mathrm{ano}}$ [34] (which has a gravitational anomaly characterized by the $\mathrm{JK}_4 \boxtimes \overline{\mathrm{JK}}_4$ SymTO), while the fully gapped boundary $\widetilde{\mathcal{R}}$ and the bulk $\mathrm{JK}_4 \boxtimes \overline{\mathrm{JK}}_4$ SymTO cover the global properties of the model (such as ground state degeneracy).

Using the SymTO correspondence described above, we find that the $\mathrm{JK}_4 \boxtimes \overline{\mathrm{JK}}_4$ SymTO has only two Morita equivalent symmetries, characterized by two different choices of the gapped boundary $\widetilde{\mathcal{R}}$ in Fig. 9. The two Lagrangian condensable algebras that give rise to these two choices of $\widetilde{\mathcal{R}}$ are:

$$
\begin{aligned}
\mathcal{A}_{S_3} &= (\mathbf{1},\mathbf{1}) \oplus (\mathbf{1},\bar{e}) \oplus (e,\mathbf{1}) \oplus (e,\bar{e}) \oplus 2(q,\bar{q}), \\
\mathcal{A}_{\mathsf{Rep}(S_3)} &= (\mathbf{1},\mathbf{1}) \oplus (e,\bar{e}) \oplus (m,\bar{m}) \oplus (m_1,\bar{m}_1) \oplus (q,\bar{q}).
\end{aligned}
\tag{4.5}
$$

From Eq. (4.3), we see that the $\mathcal{A}_{S_3}$ condensation condenses the $S_3$-charges $\mathbf{1}'$ and $\mathbf{2}$, as well as the $\mathbb{Z}_2^{\mathbf{2} \leftrightarrow r}$-charge of the self-duality symmetry. This suggests that the $\mathcal{A}_{S_3}$ condensation in the $\widetilde{\mathcal{R}}$ boundary leads to the $S_3$ symmetry together with the self-duality symmetry of $\mathrm{QFT}_{\mathrm{ano}}$, $i.e.$, the symmetry of the self-dual $S_3$ symmetric model studied in Section 2 (see Fig. 9 (left)). Following the same logic, the $\mathcal{A}_{\mathsf{Rep}(S_3)}$ condensation condenses the $S_3$ fluxes $s$ and $r$, as well as the $\mathbb{Z}_2^{\mathbf{2} \leftrightarrow r}$-charge of the self-duality symmetry. This suggests that the $\mathcal{A}_{\mathsf{Rep}(S_3)}$ condensation in the $\widetilde{\mathcal{R}}$ boundary leads to the $\mathsf{Rep}(S_3)$ symmetry together with the self-duality symmetry of $\mathrm{QFT}_{\mathrm{ano}}$, $i.e.$, the symmetry of the self-dual $\mathsf{Rep}(S_3)$ symmetric model studied in Section 3 (see Fig. 9(right)).

The SymTO identified here classifies the gapped phases of the self-dual $S_3$ symmetric model in terms of possible gapped boundaries $\mathrm{QFT}_{\mathrm{ano}}$, induced by Lagrangian condensable algebras in (4.5). The self-dual $S_3$ symmetric model has only two possible gapped phases. The ground state degeneracy of a gapped phase is given by the inner product of the integer vectors associated with the Lagrangian algebras [35] giving rise to the $\widetilde{\mathcal{R}}$ and $\mathrm{QFT}_{\mathrm{ano}}$ boundaries [45]. So if we choose the $\mathcal{A}_{\mathsf{Rep}(S_3)}$ condensation to describe the gapped $\mathrm{QFT}_{\mathrm{ano}}$ boundary, the associated ground state degeneracy is

$$
\mathrm{GSD} = (\mathcal{A}_{S_3}, \mathcal{A}_{\mathsf{Rep}(S_3)}) = 4 \,.
\tag{4.6}
$$

This phase carries the degeneracies of $\mathbb{Z}_3$ SSB phase (with GSD = 3) and the $S_3$-symmetric phase (with GSD= 1). This gapped phase describes a first-order quantum phase transition between these two phases. The second gapped phase of the self-dual $S_3$ symmetric model corresponds to a $\mathcal{A}_{S_3}$-condensed $\mathrm{QFT}_{\mathrm{ano}}$ boundary. The corresponding ground state degeneracy is

$$
\mathrm{GSD} = (\mathcal{A}_{S_3}, \mathcal{A}_{S_3}) = 8.
\tag{4.7}
$$

This phase carries the degeneracies of $\mathbb{Z}_2$ SSB phase (with GSD = 2) and the $S_3$ SSB phase (with GSD= 6). It describes a first-order quantum phase transition between these two

---

[34]This is referred to as "physical boundary" in the SymTFT literature.

[35]This integer vector $\vec{n}$ for a particular Lagrangian algebra $\mathcal{A}$ is given by its decomposition into simple anyon types as $\mathcal{A} = \bigoplus_{a \in \mathcal{M}} n_a\, a$. In our context, $\mathcal{M}$ is the SymTO. We will use the notation $(\mathcal{A}_1, \mathcal{A}_2)$ to denote the inner product $(\vec{n}_1, \vec{n}_2)$ of the integer vectors associated with the Lagrangian algebras $\mathcal{A}_1$ and $\mathcal{A}_2$.

phases. We note that the self-dual $S_3$ symmetric model does not have any gapped phase with a non-degenerate ground state. This is consistent with the fact that it is anomalous.

We can repeat the same analysis for the self-dual $\mathsf{Rep}(S_3)$ symmetric model that is given by Fig. 9(right). Here, the $\widetilde{\mathcal{R}}$ boundary is given by $\mathcal{A}_{\mathsf{Rep}(S_3)}$ condensation. The self-dual $\mathsf{Rep}(S_3)$ symmetric model also has only two gapped phases, with ground state degeneracies

$$\text{GSD} = (\mathcal{A}_{S_3}, \mathcal{A}_{\mathsf{Rep}(S_3)}) = 4, \qquad \text{GSD} = (\mathcal{A}_{\mathsf{Rep}(S_3)}, \mathcal{A}_{\mathsf{Rep}(S_3)}) = 5. \tag{4.8}$$

These gapped phases describe the first-order transitions between $\mathsf{Rep}(S_3)$-symmetric phase (with GSD=1) and $\mathsf{Rep}(S_3)/\mathbb{Z}_2$ SSB phase (with GSD = 3), and that between $\mathbb{Z}_2$ SSB phase (with GSD=2) and $\mathsf{Rep}(S_3)$ SSB phase (with GSD = 3), respectively. The self-dual $\mathsf{Rep}(S_3)$ symmetric model also does not have any gapped phases with non-degenerate ground state, since the non-inverible self-duality symmetry is anomalous.

So far, we have used the Lagrangian algebras of the SymTO to classify all possible gapped phases for systems with the corresponding symmetry. These classify the ways in which the SymTO can be "maximally broken". Non-Lagrangian condensable algebras lead to a non-trivial *unbroken SymTO*, and the associated phase via SymTO correspondence must be gapless. Such gapless states are described by the **1**-condensed boundaries of the unbroken SymTOs [44], *i.e.*, the boundaries induced by the minimal condensable algebra $\mathcal{A} = \mathbf{1}$. We find that two of the **1**-condensed boundaries of the $\text{JK}_4 \boxtimes \overline{\text{JK}}_4$ SymTO are described by the $(6,5)$ minimal model. The first one is described by the multi-component partition function [29, 44]

$$Z^{\text{JK}_4 \boxtimes \overline{\text{JK}}_4}_{\mathbf{1};\mathbf{1}} = \chi^{m6 \times \overline{m6}}_{1,0;1,0} + \chi^{m6 \times \overline{m6}}_{10,\frac{7}{5};10,-\frac{7}{5}}, \qquad Z^{\text{JK}_4 \boxtimes \overline{\text{JK}}_4}_{\mathbf{1};\bar{e}} = \chi^{m6 \times \overline{m6}}_{1,0;5,-3} + \chi^{m6 \times \overline{m6}}_{10,\frac{7}{5};6,-\frac{2}{5}},$$

$$Z^{\text{JK}_4 \boxtimes \overline{\text{JK}}_4}_{\mathbf{1};\bar{m}} = \chi^{m6 \times \overline{m6}}_{1,0;2,-\frac{1}{8}} + \chi^{m6 \times \overline{m6}}_{10,\frac{7}{5};9,-\frac{21}{40}}, \qquad Z^{\text{JK}_4 \boxtimes \overline{\text{JK}}_4}_{\mathbf{1};\bar{m}_1} = \chi^{m6 \times \overline{m6}}_{1,0;4,-\frac{13}{8}} + \chi^{m6 \times \overline{m6}}_{10,\frac{7}{5};7,-\frac{1}{40}},$$

$$Z^{\text{JK}_4 \boxtimes \overline{\text{JK}}_4}_{\mathbf{1};\bar{q}} = \chi^{m6 \times \overline{m6}}_{1,0;3,-\frac{2}{3}} + \chi^{m6 \times \overline{m6}}_{10,\frac{7}{5};8,-\frac{1}{15}}, \qquad Z^{\text{JK}_4 \boxtimes \overline{\text{JK}}_4}_{e;\mathbf{1}} = \chi^{m6 \times \overline{m6}}_{5,3;1,0} + \chi^{m6 \times \overline{m6}}_{6,\frac{2}{5};10,-\frac{7}{5}},$$

$$Z^{\text{JK}_4 \boxtimes \overline{\text{JK}}_4}_{e;\bar{e}} = \chi^{m6 \times \overline{m6}}_{5,3;5,-3} + \chi^{m6 \times \overline{m6}}_{6,\frac{2}{5};6,-\frac{2}{5}}, \qquad Z^{\text{JK}_4 \boxtimes \overline{\text{JK}}_4}_{e;\bar{m}} = \chi^{m6 \times \overline{m6}}_{5,3;2,-\frac{1}{8}} + \chi^{m6 \times \overline{m6}}_{6,\frac{2}{5};9,-\frac{21}{40}},$$

$$Z^{\text{JK}_4 \boxtimes \overline{\text{JK}}_4}_{e;\bar{m}_1} = \chi^{m6 \times \overline{m6}}_{5,3;4,-\frac{13}{8}} + \chi^{m6 \times \overline{m6}}_{6,\frac{2}{5};7,-\frac{1}{40}}, \qquad Z^{\text{JK}_4 \boxtimes \overline{\text{JK}}_4}_{e;\bar{q}} = \chi^{m6 \times \overline{m6}}_{5,3;3,-\frac{2}{3}} + \chi^{m6 \times \overline{m6}}_{6,\frac{2}{5};8,-\frac{1}{15}},$$

$$Z^{\text{JK}_4 \boxtimes \overline{\text{JK}}_4}_{m;\mathbf{1}} = \chi^{m6 \times \overline{m6}}_{2,\frac{1}{8};1,0} + \chi^{m6 \times \overline{m6}}_{9,\frac{21}{40};10,-\frac{7}{5}}, \qquad Z^{\text{JK}_4 \boxtimes \overline{\text{JK}}_4}_{m;\bar{e}} = \chi^{m6 \times \overline{m6}}_{2,\frac{1}{8};5,-3} + \chi^{m6 \times \overline{m6}}_{9,\frac{21}{40};6,-\frac{2}{5}},$$

$$Z^{\text{JK}_4 \boxtimes \overline{\text{JK}}_4}_{m;\bar{m}} = \chi^{m6 \times \overline{m6}}_{2,\frac{1}{8};2,-\frac{1}{8}} + \chi^{m6 \times \overline{m6}}_{9,\frac{21}{40};9,-\frac{21}{40}}, \qquad Z^{\text{JK}_4 \boxtimes \overline{\text{JK}}_4}_{m;\bar{m}_1} = \chi^{m6 \times \overline{m6}}_{2,\frac{1}{8};4,-\frac{13}{8}} + \chi^{m6 \times \overline{m6}}_{9,\frac{21}{40};7,-\frac{1}{40}},$$

$$Z^{\text{JK}_4 \boxtimes \overline{\text{JK}}_4}_{m;\bar{q}} = \chi^{m6 \times \overline{m6}}_{2,\frac{1}{8};3,-\frac{2}{3}} + \chi^{m6 \times \overline{m6}}_{9,\frac{21}{40};8,-\frac{1}{15}}, \qquad Z^{\text{JK}_4 \boxtimes \overline{\text{JK}}_4}_{m_1;\mathbf{1}} = \chi^{m6 \times \overline{m6}}_{4,\frac{13}{8};1,0} + \chi^{m6 \times \overline{m6}}_{7,\frac{1}{40};10,-\frac{7}{5}},$$

$$Z^{\text{JK}_4 \boxtimes \overline{\text{JK}}_4}_{m_1;\bar{e}} = \chi^{m6 \times \overline{m6}}_{4,\frac{13}{8};5,-3} + \chi^{m6 \times \overline{m6}}_{7,\frac{1}{40};6,-\frac{2}{5}}, \qquad Z^{\text{JK}_4 \boxtimes \overline{\text{JK}}_4}_{m_1;\bar{m}} = \chi^{m6 \times \overline{m6}}_{4,\frac{13}{8};2,-\frac{1}{8}} + \chi^{m6 \times \overline{m6}}_{7,\frac{1}{40};9,-\frac{21}{40}},$$

$$Z^{\text{JK}_4 \boxtimes \overline{\text{JK}}_4}_{m_1;\bar{m}_1} = \chi^{m6 \times \overline{m6}}_{4,\frac{13}{8};4,-\frac{13}{8}} + \chi^{m6 \times \overline{m6}}_{7,\frac{1}{40};7,-\frac{1}{40}}, \qquad Z^{\text{JK}_4 \boxtimes \overline{\text{JK}}_4}_{m_1;\bar{q}} = \chi^{m6 \times \overline{m6}}_{4,\frac{13}{8};3,-\frac{2}{3}} + \chi^{m6 \times \overline{m6}}_{7,\frac{1}{40};8,-\frac{1}{15}},$$

$$Z^{\text{JK}_4 \boxtimes \overline{\text{JK}}_4}_{q;\mathbf{1}} = \chi^{m6 \times \overline{m6}}_{3,\frac{2}{3};1,0} + \chi^{m6 \times \overline{m6}}_{8,\frac{1}{15};10,-\frac{7}{5}}, \qquad Z^{\text{JK}_4 \boxtimes \overline{\text{JK}}_4}_{q;\bar{e}} = \chi^{m6 \times \overline{m6}}_{3,\frac{2}{3};5,-3} + \chi^{m6 \times \overline{m6}}_{8,\frac{1}{15};6,-\frac{2}{5}},$$

$$Z^{\text{JK}_4 \boxtimes \overline{\text{JK}}_4}_{q;\bar{m}} = \chi^{m6 \times \overline{m6}}_{3,\frac{2}{3};2,-\frac{1}{8}} + \chi^{m6 \times \overline{m6}}_{8,\frac{1}{15};9,-\frac{21}{40}}, \qquad Z^{\text{JK}_4 \boxtimes \overline{\text{JK}}_4}_{q;\bar{m}_1} = \chi^{m6 \times \overline{m6}}_{3,\frac{2}{3};4,-\frac{13}{8}} + \chi^{m6 \times \overline{m6}}_{8,\frac{1}{15};7,-\frac{1}{40}},$$

$$Z^{\text{JK}_4 \boxtimes \overline{\text{JK}}_4}_{q;\bar{q}} = \chi^{m6 \times \overline{m6}}_{3,\frac{2}{3};3,-\frac{2}{3}} + \chi^{m6 \times \overline{m6}}_{8,\frac{1}{15};8,-\frac{1}{15}}, \tag{4.9}$$

while the second one is described by

$$Z^{\text{JK}_4 \boxtimes \overline{\text{JK}}_4}_{\mathbf{1};\mathbf{1}} = \chi^{m6 \times \overline{m6}}_{1,0;1,0} + \chi^{m6 \times \overline{m6}}_{10,\frac{7}{5};10,-\frac{7}{5}}, \qquad Z^{\text{JK}_4 \boxtimes \overline{\text{JK}}_4}_{\mathbf{1};\bar{e}} = \chi^{m6 \times \overline{m6}}_{1,0;5,-3} + \chi^{m6 \times \overline{m6}}_{10,\frac{7}{5};6,-\frac{2}{5}},$$

$$Z_{\mathbf{1};\bar{m}}^{\mathrm{JK}_4 \boxtimes \overline{\mathrm{JK}}_4} = \chi_{5,3;2,-\frac{1}{8}}^{m6 \times \overline{m6}} + \chi_{6,\frac{2}{5};9,-\frac{21}{40}}^{m6 \times \overline{m6}}, \qquad Z_{\mathbf{1};\bar{m}_1}^{\mathrm{JK}_4 \boxtimes \overline{\mathrm{JK}}_4} = \chi_{5,3;4,-\frac{13}{8}}^{m6 \times \overline{m6}} + \chi_{6,\frac{2}{5};7,-\frac{1}{40}}^{m6 \times \overline{m6}},$$

$$Z_{\mathbf{1};\bar{q}}^{\mathrm{JK}_4 \boxtimes \overline{\mathrm{JK}}_4} = \chi_{1,0;3,-\frac{2}{3}}^{m6 \times \overline{m6}} + \chi_{10,\frac{7}{5};8,-\frac{1}{15}}^{m6 \times \overline{m6}}, \qquad Z_{e;\mathbf{1}}^{\mathrm{JK}_4 \boxtimes \overline{\mathrm{JK}}_4} = \chi_{5,3;1,0}^{m6 \times \overline{m6}} + \chi_{6,\frac{2}{5};10,-\frac{7}{5}}^{m6 \times \overline{m6}},$$

$$Z_{e;\bar{e}}^{\mathrm{JK}_4 \boxtimes \overline{\mathrm{JK}}_4} = \chi_{5,3;5,-3}^{m6 \times \overline{m6}} + \chi_{6,\frac{2}{5};6,-\frac{2}{5}}^{m6 \times \overline{m6}}, \qquad Z_{e;\bar{m}}^{\mathrm{JK}_4 \boxtimes \overline{\mathrm{JK}}_4} = \chi_{1,0;2,-\frac{1}{8}}^{m6 \times \overline{m6}} + \chi_{10,\frac{7}{5};9,-\frac{21}{40}}^{m6 \times \overline{m6}},$$

$$Z_{e;\bar{m}_1}^{\mathrm{JK}_4 \boxtimes \overline{\mathrm{JK}}_4} = \chi_{1,0;4,-\frac{13}{8}}^{m6 \times \overline{m6}} + \chi_{10,\frac{7}{5};7,-\frac{1}{40}}^{m6 \times \overline{m6}}, \qquad Z_{e;\bar{q}}^{\mathrm{JK}_4 \boxtimes \overline{\mathrm{JK}}_4} = \chi_{5,3;3,-\frac{2}{3}}^{m6 \times \overline{m6}} + \chi_{6,\frac{2}{5};8,-\frac{1}{15}}^{m6 \times \overline{m6}},$$

$$Z_{m;\mathbf{1}}^{\mathrm{JK}_4 \boxtimes \overline{\mathrm{JK}}_4} = \chi_{2,\frac{1}{8};5,-3}^{m6 \times \overline{m6}} + \chi_{9,\frac{21}{40};6,-\frac{2}{5}}^{m6 \times \overline{m6}}, \qquad Z_{m;\bar{e}}^{\mathrm{JK}_4 \boxtimes \overline{\mathrm{JK}}_4} = \chi_{2,\frac{1}{8};1,0}^{m6 \times \overline{m6}} + \chi_{9,\frac{21}{40};10,-\frac{7}{5}}^{m6 \times \overline{m6}},$$

$$Z_{m;\bar{m}}^{\mathrm{JK}_4 \boxtimes \overline{\mathrm{JK}}_4} = \chi_{4,\frac{13}{8};4,-\frac{13}{8}}^{m6 \times \overline{m6}} + \chi_{7,\frac{1}{40};7,-\frac{1}{40}}^{m6 \times \overline{m6}}, \qquad Z_{m;\bar{m}_1}^{\mathrm{JK}_4 \boxtimes \overline{\mathrm{JK}}_4} = \chi_{4,\frac{13}{8};2,-\frac{1}{8}}^{m6 \times \overline{m6}} + \chi_{7,\frac{1}{40};9,-\frac{21}{40}}^{m6 \times \overline{m6}},$$

$$Z_{m;\bar{q}}^{\mathrm{JK}_4 \boxtimes \overline{\mathrm{JK}}_4} = \chi_{2,\frac{1}{8};3,-\frac{2}{3}}^{m6 \times \overline{m6}} + \chi_{9,\frac{21}{40};8,-\frac{1}{15}}^{m6 \times \overline{m6}}, \qquad Z_{m_1;\mathbf{1}}^{\mathrm{JK}_4 \boxtimes \overline{\mathrm{JK}}_4} = \chi_{4,\frac{13}{8};5,-3}^{m6 \times \overline{m6}} + \chi_{7,\frac{1}{40};6,-\frac{2}{5}}^{m6 \times \overline{m6}},$$

$$Z_{m_1;\bar{e}}^{\mathrm{JK}_4 \boxtimes \overline{\mathrm{JK}}_4} = \chi_{4,\frac{13}{8};1,0}^{m6 \times \overline{m6}} + \chi_{7,\frac{1}{40};10,-\frac{7}{5}}^{m6 \times \overline{m6}}, \qquad Z_{m_1;\bar{m}}^{\mathrm{JK}_4 \boxtimes \overline{\mathrm{JK}}_4} = \chi_{2,\frac{1}{8};4,-\frac{13}{8}}^{m6 \times \overline{m6}} + \chi_{9,\frac{21}{40};7,-\frac{1}{40}}^{m6 \times \overline{m6}},$$

$$Z_{m_1;\bar{m}_1}^{\mathrm{JK}_4 \boxtimes \overline{\mathrm{JK}}_4} = \chi_{2,\frac{1}{8};2,-\frac{1}{8}}^{m6 \times \overline{m6}} + \chi_{9,\frac{21}{40};9,-\frac{21}{40}}^{m6 \times \overline{m6}}, \qquad Z_{m_1;\bar{q}}^{\mathrm{JK}_4 \boxtimes \overline{\mathrm{JK}}_4} = \chi_{4,\frac{13}{8};3,-\frac{2}{3}}^{m6 \times \overline{m6}} + \chi_{7,\frac{1}{40};8,-\frac{1}{15}}^{m6 \times \overline{m6}},$$

$$Z_{q;\mathbf{1}}^{\mathrm{JK}_4 \boxtimes \overline{\mathrm{JK}}_4} = \chi_{3,\frac{2}{3};1,0}^{m6 \times \overline{m6}} + \chi_{8,\frac{1}{15};10,-\frac{7}{5}}^{m6 \times \overline{m6}}, \qquad Z_{q;\bar{e}}^{\mathrm{JK}_4 \boxtimes \overline{\mathrm{JK}}_4} = \chi_{3,\frac{2}{3};5,-3}^{m6 \times \overline{m6}} + \chi_{8,\frac{1}{15};6,-\frac{2}{5}}^{m6 \times \overline{m6}},$$

$$Z_{q;\bar{m}}^{\mathrm{JK}_4 \boxtimes \overline{\mathrm{JK}}_4} = \chi_{3,\frac{2}{3};2,-\frac{1}{8}}^{m6 \times \overline{m6}} + \chi_{8,\frac{1}{15};9,-\frac{21}{40}}^{m6 \times \overline{m6}}, \qquad Z_{q;\bar{m}_1}^{\mathrm{JK}_4 \boxtimes \overline{\mathrm{JK}}_4} = \chi_{3,\frac{2}{3};4,-\frac{13}{8}}^{m6 \times \overline{m6}} + \chi_{8,\frac{1}{15};7,-\frac{1}{40}}^{m6 \times \overline{m6}},$$

$$Z_{q;\bar{q}}^{\mathrm{JK}_4 \boxtimes \overline{\mathrm{JK}}_4} = \chi_{3,\frac{2}{3};3,-\frac{2}{3}}^{m6 \times \overline{m6}} + \chi_{8,\frac{1}{15};8,-\frac{1}{15}}^{m6 \times \overline{m6}}. \tag{4.10}$$

The various terms in each component of the partition function are conformal characters of the (6,5) minimal model. The expression $\chi_{a,h_a;\,b,-h_b}^{m6 \times \overline{m6}}$ is a short-hand notation for the product of the left moving chiral conformal character associated with the primary operator labeled by $a$ (set by an arbitrary indexing convention) with conformal weight $(h_a, 0)$, and the right moving chiral conformal character associated with the primary operator labeled by $b$ with conformal weight $(0, h_b)$. The superscript $m6 \times \overline{m6}$ indicates that both the left and right moving chiral conformal characters are picked from the same (6,5) minimal model.

Note that in the above multi-component "SymTO-resolved" partition function, the $\mathbf{1}$ sector contains all the primary operators that respect the symmetry. We see from the term $\chi_{10,\frac{7}{5};10,-\frac{7}{5}}^{m6 \times \overline{m6}}$ in $Z_{\mathbf{1};\mathbf{1}}^{\mathrm{JK}_4 \boxtimes \overline{\mathrm{JK}}_4}$ that the scaling dimensions of the symmetric operators to be $7/5 + 7/5 + 2n > 2$ with a non-negative integer $n$ for both gapless states. Such symmetric operators are then irrelevant. Therefore, both of the above two gapless states are in fact gapless phases with no relevant perturbation that respects the symmetries dictated by $\mathrm{JK}_4 \boxtimes \overline{\mathrm{JK}}_4$ SymTO.

We remark that the above calculation was also performed for the other candidate $\mathrm{SU}(2)_4 \boxtimes \overline{\mathrm{SU}(2)}_4$ SymTO. We find that for systems with $\mathrm{SU}(2)_4 \boxtimes \overline{\mathrm{SU}(2)}_4$ SymTO, all gapless states that are described by (6,5) minimal model contain at least one relevant perturbation that respects the $\mathrm{SU}(2)_4 \boxtimes \overline{\mathrm{SU}(2)}_4$ SymTO. This contradicts our numerical calculations in Secs. 2 and 3 where we found a stable gapless phase described by (6,5) minimal model in the presence of $S_3$ (respectively, $\mathsf{Rep}(S_3)$) and self-duality symmetry. We conclude that the SymTO in our self-dual $S_3$-symmetric model and self-dual $\mathsf{Rep}(S_3)$-symmetric model is given by $\mathrm{JK}_4 \boxtimes \overline{\mathrm{JK}}_4$ and not by $\mathrm{SU}(2)_4 \boxtimes \overline{\mathrm{SU}(2)}_4$.

The operators that break the self-duality symmetry live in the $e\bar{e}$ sector. From the partition function $Z_{e;\bar{e}}^{\mathrm{JK}_4 \boxtimes \overline{\mathrm{JK}}_4}$, we find that the scaling dimensions of the operators breaking the self-duality symmetry to be $2/5 + 2/5 + 2n$ or $3 + 3 + 2n$ (with a non-negative integer $n$), for both gapless states. Thus, the two gapless states have only one relevant operator that

break the self-duality symmetry but not the $S_3$ symmetry. They can be identified with the upper and lower vertical lines that meet at the multi-critical point in phase diagrams in Figs. 3 and 8 which indeed have only one such relevant perturbation. In fact, SSB of the self-duality symmetry due to the $e\bar{e}$ condensation can be seen from the following relation between $\mathrm{JK}_4 \boxtimes \overline{\mathrm{JK}}_4$-SymTO-resolved partition functions (4.9) and (4.10), and $\mathcal{D}(S_3)$-SymTO-resolved partition function (E.1):

$$
\begin{aligned}
Z_{\mathbf{1}}^{\mathcal{D}(S_3)} &= Z_{\mathbf{1};\mathbf{1}}^{\mathrm{JK}_4 \boxtimes \overline{\mathrm{JK}}_4} + Z_{e;\bar{e}}^{\mathrm{JK}_4 \boxtimes \overline{\mathrm{JK}}_4}, &\qquad Z_{\mathbf{1}'}^{\mathcal{D}(S_3)} &= Z_{e;\mathbf{1}}^{\mathrm{JK}_4 \boxtimes \overline{\mathrm{JK}}_4} + Z_{\mathbf{1};\bar{e}}^{\mathrm{JK}_4 \boxtimes \overline{\mathrm{JK}}_4}, \\
Z_{s}^{\mathcal{D}(S_3)} &= Z_{m;\bar{m}}^{\mathrm{JK}_4 \boxtimes \overline{\mathrm{JK}}_4} + Z_{m_1;\bar{m}_1}^{\mathrm{JK}_4 \boxtimes \overline{\mathrm{JK}}_4}, &\qquad \cdots
\end{aligned}
\tag{4.11}
$$

where the sectors in $\mathrm{JK}_4 \boxtimes \overline{\mathrm{JK}}_4$ SymTO connected by $e\bar{e}$ are combined into a single sector in $\mathcal{D}(S_3)$ SymTO.

The above two gapless states with self-duality symmetry are very similar. The only difference is that $Z_s^{\mathcal{D}(S_3)}$ splits differently when we add the self-duality symmetry as

$$
\begin{aligned}
Z_s^{\mathcal{D}(S_3)} &= \underbrace{\chi_{2,\frac{1}{8};2,-\frac{1}{8}}^{m6\times\overline{m6}} + \chi_{9,\frac{21}{40};9,-\frac{21}{40}}^{m6\times\overline{m6}}}_{Z_{m;\bar{m}}^{\mathrm{JK}_4 \boxtimes \overline{\mathrm{JK}}_4}} + \underbrace{\chi_{4,\frac{13}{8};4,-\frac{13}{8}}^{m6\times\overline{m6}} + \chi_{7,\frac{1}{40};7,-\frac{1}{40}}^{m6\times\overline{m6}}}_{Z_{m_1;\bar{m}_1}^{\mathrm{JK}_4 \boxtimes \overline{\mathrm{JK}}_4}}, \\
Z_s^{\mathcal{D}(S_3)} &= \underbrace{\chi_{4,\frac{13}{8};4,-\frac{13}{8}}^{m6\times\overline{m6}} + \chi_{7,\frac{1}{40};7,-\frac{1}{40}}^{m6\times\overline{m6}}}_{Z_{m;\bar{m}}^{\mathrm{JK}_4 \boxtimes \overline{\mathrm{JK}}_4}} + \underbrace{\chi_{2,\frac{1}{8};2,-\frac{1}{8}}^{m6\times\overline{m6}} + \chi_{9,\frac{21}{40};9,-\frac{21}{40}}^{m6\times\overline{m6}}}_{Z_{m_1;\bar{m}_1}^{\mathrm{JK}_4 \boxtimes \overline{\mathrm{JK}}_4}}.
\end{aligned}
\tag{4.12}
$$

In addition to the two Lagrangian condensable algebras (4.5), $\mathcal{M} = \mathrm{JK}_4 \boxtimes \overline{\mathrm{JK}}_4$ SymTO also has six non-Lagrangian condensable algebras, listed below. The condensation of these six algebras reduces the SymTO $\mathcal{M} = \mathrm{JK}_4 \boxtimes \overline{\mathrm{JK}}_4$ to a smaller SymTO $\mathcal{M}_{/\mathcal{A}}$. The reduced, unbroken SymTO can be identified from its total quantum dimension $D_{\mathcal{M}_{/\mathcal{A}}} = \sqrt{\sum_i d_i^2}$ and topological spins of the anyons, as well as confirmed by the presence of a domain wall between $\mathcal{M}$ and $\mathcal{M}_{/\mathcal{A}}$ that has no condensation of $\mathcal{M}_{/\mathcal{A}}$ anyons:

(i) Condensable algebra $\mathcal{A}_1 = (\mathbf{1}, \mathbf{1}) \oplus (e, \bar{e}) \oplus (q, \bar{q})$:
SymTO $\mathcal{M}_{/\mathcal{A}_1}$ contains topological spins $(0)$ and has $D_{\mathcal{M}_{/\mathcal{A}_1}} = 2$.
We conclude that $\mathcal{M}_{/\mathcal{A}_1} = \mathcal{D}(\mathbb{Z}_2)$.

(ii) Condensable algebra $\mathcal{A}_2 = (\mathbf{1}, \mathbf{1}) \oplus (\mathbf{1}, \bar{e}) \oplus (e, \mathbf{1}) \oplus (e, \bar{e})$:
SymTO $\mathcal{M}_{/\mathcal{A}_2}$ contains topological spins $\left(0, \frac{1}{3}, \frac{2}{3}\right)$ and has $D_{\mathcal{M}_{/\mathcal{A}_2}} = 3$.
We conclude that $\mathcal{M}_{/\mathcal{A}_2} = \mathcal{D}(\mathbb{Z}_3)$.

(iii) Condensable algebra $\mathcal{A}_3 = (\mathbf{1}, \mathbf{1}) \oplus (e, \bar{e})$:
SymTO $\mathcal{M}_{/\mathcal{A}_3}$ contains topological spins $\left(0, \frac{1}{3}, \frac{1}{2}, \frac{2}{3}\right)$ and has $D_{\mathcal{M}_{/\mathcal{A}_3}} = 6$.
We conclude that $\mathcal{M}_{/\mathcal{A}_3} = \mathcal{D}(S_3)$.

(iv) Condensable algebra $\mathcal{A}_4 = (\mathbf{1}, \mathbf{1}) \oplus (e, \mathbf{1})$:
SymTO $\mathcal{M}_{/\mathcal{A}_4}$ contains topological spins $\left(0, \frac{1}{24}, \frac{1}{3}, \frac{3}{8}, \frac{13}{24}, \frac{2}{3}, \frac{7}{8}\right)$ and has $D_{\mathcal{M}_{/\mathcal{A}_4}} = 6$.
We conclude that $\mathcal{M}_{/\mathcal{A}_4} = \overline{\mathcal{Z}}_3 \boxtimes \overline{\mathrm{JK}}_4$.

(v) Condensable algebra $\mathcal{A}_5 = (\mathbf{1}, \mathbf{1}) \oplus (\mathbf{1}, \bar{e})$:
SymTO $\mathcal{M}_{/\mathcal{A}_5}$ contains topological spins $\left(0, \frac{1}{8}, \frac{1}{3}, \frac{11}{24}, \frac{5}{8}, \frac{2}{3}, \frac{23}{24}\right)$ and has $D_{\mathcal{M}_{/\mathcal{A}_5}} = 6$.
We conclude that $\mathcal{M}_{/\mathcal{A}_5} = \mathrm{JK}_4 \boxtimes \mathcal{Z}_3$.

(vi) Condensable algebra $\mathcal{A}_6 = (\mathbf{1}, \mathbf{1})$:
$\mathcal{M}_{/\mathcal{A}_6} = \mathrm{JK}_4 \boxtimes \overline{\mathrm{JK}}_4$.

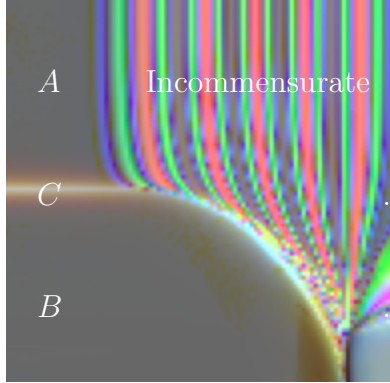
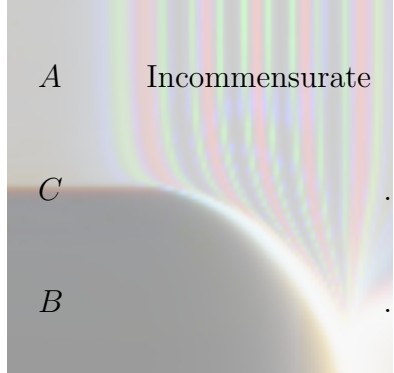

Figure 10: (Left) The central charge $c$ and (Right) GSD from tensor network numerical calculation, for the self-dual $S_3$-symmetric model with $J_1 = J_2 = 1/2$ and $J_4 = 1 - J_3$. The horizontal axis is $\theta \in [0, \pi/2]$. The vertical axis is $J_3 \in [0, 1]$. Here GSD is obtained from the partition function: $\text{GSD} \equiv Z^2(L, L)/Z(L, 2L)$ where $Z(L_x, L_t)$ is the partition function for spacetime of size $L_x \times L_t$. For gapped quantum systems, GSD happen to be the ground state degeneracy. The central charge $c$ is also obtained from the partition function, which has a form $Z(L, L_\infty) = e^{-L_\infty[\epsilon L - \frac{2\pi cv}{24L} + o(L^{-1})]}$ (which determines $cv$) and $Z(L_\infty, L) = e^{-L_\infty[\epsilon L - \frac{2\pi c/v}{24L} + o(L^{-1})]}$ (which determines $c/v$), when $L_\infty \gg L$. This way, the central charge $c$ is defined even for non-critical states. The red-channel of the colored image is for lattice of size 64, green-channel for size 128, and blue-channel for size 256. (Left) The color intensity $[0, 1]$ corresponds to central charge $c \in [0, 2]$. (Right) The color intensity $[0, 1]$ corresponds to $1 - \frac{1}{\text{GSD}}$.

Here $\mathcal{Z}_3$ is the Abelian topological order described by the $K$-matrix $\begin{pmatrix} 2 & 1 \\ 1 & 2 \end{pmatrix}$, where the fusion of the anyons form a $\mathbb{Z}_3$ group. These condensable algebras describe the possible spontaneous symmetry breaking patterns of the $\text{JK}_4 \boxtimes \overline{\text{JK}}_4$ SymTO, where the unbroken SymTO is given by $\mathcal{M}_{/\mathcal{A}} = (\text{JK}_4 \boxtimes \overline{\text{JK}}_4)_{/\mathcal{A}}$. Because the unbroken SymTO is non-trivial, those non-maximal SymTO broken states are gapless [40, 44], and are given by the **1**-condensed boundaries of $\mathcal{M}_{/\mathcal{A}} = (\text{JK}_4 \boxtimes \overline{\text{JK}}_4)_{/\mathcal{A}}$. In turn, such **1**-condensed boundaries are some possible gapless states of our self-dual $S_3$-symmetric or $\text{Rep}(S_3)$-symmetric models.

To see the actual gapless states of our self-dual $S_3$-symmetric model, we have calculated the phase diagram for the model (2.21) (see Fig. 10). We find gapless phases A and B (as indicated by the non-zero central charge $c$) and gapless incommensurate phase (as indicated by the "striped" non-zero central charge $c$), as well as continuous phase transitions between them. To understand such a phase diagram, in the following, we list the gapless states that we found using holographic modular bootstrap [29, 44, 109] for the self-dual $S_3$-symmetric model (2.21). We only list gapless phases (which have no symmetric relevant operator), and gapless critical points at continuous phase transitions (which have only one symmetric relevant operator). We group those gapless states by the condensable algebras $\mathcal{A}_i$, whose condensation lead to the corresponding gapless states. Those gapless states have a reduced SymTO $\mathcal{M}_{/\mathcal{A}_i}$ as discussed above:

$\mathcal{A}_1 = (\mathbf{1}, \mathbf{1}) \oplus (e, \bar{e}) \oplus (q, \bar{q})$, $\mathcal{M}_{/\mathcal{A}_1} = \mathcal{D}(\mathbb{Z}_2)$:
Gapless phase: $m4 \times m4 \times \overline{U1}$ (see (H.6) for the SymTO-resolved partition function)
Critical point: $m4 \times \overline{m4}$ (see (H.7))

$\mathcal{A}_2 = (\mathbf{1}, \mathbf{1}) \oplus (\mathbf{1}, \bar{e}) \oplus (e, \mathbf{1}) \oplus (e, \bar{e}), \quad \mathcal{M}_{/\mathcal{A}_2} = \mathcal{D}(\mathbb{Z}_3)$:
Critical point: $m6 \times \overline{m6}$ (see (H.9))

$\mathcal{A}_3 = (\mathbf{1}, \mathbf{1}) \oplus (e, \bar{e}), \quad \mathcal{M}_{/\mathcal{A}_3} = \mathcal{D}(S_3)$:
Critical point: $m6 \times \overline{m6}$ (see (H.8)).

$\mathcal{A}_4 = (\mathbf{1}, \mathbf{1}) \oplus (\mathbf{1}, \bar{e}), \quad \mathcal{M}_{/\mathcal{A}_4} = \overline{\mathbb{Z}}_3 \boxtimes \overline{\mathrm{JK}}_4$:
Gapless phase: $m6 \times \overline{m6}$ (see (H.5))
Critical point: $m7 \times \overline{m7}$ (see (H.12))

$\mathcal{A}_5 = (\mathbf{1}, \mathbf{1}) \oplus (e, \mathbf{1}), \quad \mathcal{M}_{/\mathcal{A}_5} = \mathrm{JK}_4 \boxtimes \mathbb{Z}_3$:
Gapless phase: $m6 \times \overline{m6}$ (see (H.4))
Critical point: $m7 \times \overline{m7}$ (see (H.13))

$\mathcal{A}_6 = (\mathbf{1}, \mathbf{1}), \quad \mathcal{M}_{/\mathcal{A}_6} = \mathrm{JK}_4 \boxtimes \overline{\mathrm{JK}}_4$:
Gapless phase: $m6 \times \overline{m6}_1$ (see (4.9)), $m6 \times \overline{m6}_2$ (see (4.10))
Critical point: $m7 \times \overline{m7}_1$ (see (H.10)), $m7 \times \overline{m7}_2$ (see (H.11)), $m4 \times m6 \times \overline{m4} \times \overline{m6}_1$ (see (H.14)), $m4 \times m6 \times \overline{m4} \times \overline{m6}_2$ (see (H.15)), $m4 \times m6 \times \overline{m4} \times \overline{m6}_3$ (see (H.16)), $m4 \times m6 \times \overline{m4} \times \overline{m6}_4$ (see (H.17)),

Here $m4$, $m6$, $m7$ represent $(4,3)$, $(6,5)$, $(7,6)$ chiral minimal model CFT. $U1$ represents chiral compact boson CFT. For example, $m4 \times m4 \times \overline{U1}$ is a gapless state whose right-movers are described by $m4 \times m4$ CFT and whose left-movers are described by $\overline{U1}$ CFT.

The SymTO-resolved partition function $\mathbf{Z}$ for $m4 \times m4 \times \overline{U1}$ state is given by (H.6). From the $(\mathbf{1}, \mathbf{1})$ component of the partition function

$$Z_{\mathbf{1};\mathbf{1}}^{JK_4 \boxtimes \overline{JK}_4} = \chi_{1,0;1,0;1,0}^{m4 \times m4 \times \overline{U1}_4} + \chi_{1,0;3,\frac{1}{2};3,-\frac{1}{2}}^{m4 \times m4 \times \overline{U1}_4} + \chi_{3,\frac{1}{2};1,0;3,-\frac{1}{2}}^{m4 \times m4 \times \overline{U1}_4} + \chi_{3,\frac{1}{2};3,\frac{1}{2};1,0}^{m4 \times m4 \times \overline{U1}_4} \tag{4.13}$$

we conclude that the $m4 \times m4 \times \overline{U1}$ state has two $JK_4 \boxtimes \overline{JK}_4$ symmetric relevant perturbations (highlighted in red). However, if we want to use $m4 \times m4 \times \overline{U1}$ state to describe the incommensurate state in the phase diagram Fig. 10, we also need to include the lattice translation symmetry. In this case, the crystal momentum of a many-body state become the $U(1)$ charge of the $\overline{U1}$ CFT. The primary field for the conformal character $\chi_{1,0;3,\frac{1}{2};3,-\frac{1}{2}}^{m4 \times m4 \times \overline{U1}_4}$ carries a non-zero $U(1)$ charge, $i.e.$, a non-zero crystal momentum. Such a primary field is not symmetric under the translation symmetry. As a result, the $m4 \times m4 \times \overline{U1}$ state has no $JK_4 \boxtimes \overline{JK}_4$ symmetric and translation symmetric relevant perturbations. So it is a stable gapless phase.

Let us point out that the $m4 \times m4 \times \overline{U1}$ state is nothing but the incommensurate state discussed in Section 5.3. Both states have the same unbroken SymTO $\mathcal{M}_{/\mathcal{A}} = \mathcal{D}(\mathbb{Z}_2)$, and both states are described by the same CFTs.

In the phase diagram shown in Fig. 10, we identify the gapless phase-B as the $m6 \times \overline{m6}_1$ phase (see (4.9)) or $m6 \times \overline{m6}_2$ phase (see (4.10)). Both phases have the full SymTO $JK_4 \boxtimes \overline{JK}_4$. Such phases for the self-dual $S_3$ symmetric model have the following modular invariant partition function, which is obtained via the inner product of the vector-valued partition function $\mathbf{Z}$ and the Lagrangian condensable Algebra $\mathcal{A}_{S_3}$

$$\begin{aligned} Z = (\mathbf{Z}, \mathcal{A}_{S_3}) &= Z_{\mathbf{1};\mathbf{1}}^{JK_4 \boxtimes \overline{JK}_4} + Z_{\mathbf{1};\bar{e}}^{JK_4 \boxtimes \overline{JK}_4} + Z_{e;\mathbf{1}}^{JK_4 \boxtimes \overline{JK}_4} + Z_{e;\bar{e}}^{JK_4 \boxtimes \overline{JK}_4} + 2Z_{q;\bar{q}}^{JK_4 \boxtimes \overline{JK}_4} \\ &= \chi_{1,0;1,0}^{m6 \times \overline{m6}} + \chi_{10,\frac{7}{5};10,-\frac{7}{5}}^{m6 \times \overline{m6}} + \chi_{1,0;5,-3}^{m6 \times \overline{m6}} + \chi_{10,\frac{7}{5};6,-\frac{2}{5}}^{m6 \times \overline{m6}} + \chi_{5,3;1,0}^{m6 \times \overline{m6}} + \chi_{6,\frac{2}{5};10,-\frac{7}{5}}^{m6 \times \overline{m6}} \\ &\quad + \chi_{5,3;5,-3}^{m6 \times \overline{m6}} + \chi_{6,\frac{2}{5};6,-\frac{2}{5}}^{m6 \times \overline{m6}} + 2(\chi_{3,\frac{2}{3};3,-\frac{2}{3}}^{m6 \times \overline{m6}} + \chi_{8,\frac{1}{15};8,-\frac{1}{15}}^{m6 \times \overline{m6}}) \end{aligned} \tag{4.14}$$

We also identify the gapless phase-A as $m6 \times \overline{m6}$ phase with $\mathcal{A}_4$-condensation (see (H.5)) or $m6 \times \overline{m6}$ phase with $\mathcal{A}_5$-condensation (see (H.4)). The two phases for the self-dual $S_3$ symmetric model have the following modular invariant partition function

$$
\begin{aligned}
Z = (\mathbf{Z}, \mathcal{A}_{S_3}) &= Z^{JK_4 \boxtimes \overline{JK}_4}_{\mathbf{1};\mathbf{1}} + Z^{JK_4 \boxtimes \overline{JK}_4}_{\mathbf{1};\bar{e}} + Z^{JK_4 \boxtimes \overline{JK}_4}_{e;\mathbf{1}} + Z^{JK_4 \boxtimes \overline{JK}_4}_{e;\bar{e}} + 2 Z^{JK_4 \boxtimes \overline{JK}_4}_{q;\bar{q}} \\
&= 2 \big( \chi^{m6 \times \overline{m6}}_{1,0;1,0} + \chi^{m6 \times \overline{m6}}_{5,3;1,0} + \chi^{m6 \times \overline{m6}}_{6,\frac{2}{5};10,-\frac{7}{5}} + \chi^{m6 \times \overline{m6}}_{10,\frac{7}{5};10,-\frac{7}{5}} + \chi^{m6 \times \overline{m6}}_{1,0;5,-3} + \chi^{m6 \times \overline{m6}}_{5,3;5,-3} \\
&\quad + \chi^{m6 \times \overline{m6}}_{6,\frac{2}{5};6,-\frac{2}{5}} + \chi^{m6 \times \overline{m6}}_{10,\frac{7}{5};6,-\frac{2}{5}} + 2 \chi^{m6 \times \overline{m6}}_{3,\frac{2}{3};3,-\frac{2}{3}} + 2 \chi^{m6 \times \overline{m6}}_{8,\frac{1}{15};8,-\frac{1}{15}} \big)
\end{aligned} \tag{4.15}
$$

Since the conformal character for the identity operator $\chi^{m6 \times \overline{m6}}_{1,0;1,0}$ appears twice, the $m6 \times \overline{m6}$ phase with $\mathcal{A}_4$ or $\mathcal{A}_5$ condensation has 2-fold degenerate ground states on a ring, as one can see from Fig. 10(Right). In fact, such phases spontaneously breaks the $\mathbb{Z}_2$ symmetry, due to the $e$ or $\bar{e}$ condensation, which leads to the 2-fold degenerate ground states. Let us note here that such phases were termed "gapless SSB" phases in Ref. [110].

We see that the gapless phase-A has a $\mathbb{Z}_2$ SSB while the gapless phase-B has no symmetry breaking, despite both phases being described by $m6 \times \overline{m6}$ CFT. The phase transition between phase-A and phase-B is a $\mathbb{Z}_2$ symmetry breaking transition. Thus the critical point of the transition is described by $m4 \times \overline{m4}$ Ising CFT on top of $m6 \times \overline{m6}$ CFT, which is one of the four gapless critical states: $m4 \times m6 \times \overline{m4} \times \overline{m6}_i$, $i = 1, 2, 3, 4$, listed above.

# 5  Discussion

Let us recap and give a detailed discussion of the main lessons from Secs. 2 and 3. We collect our key results under three directions. Sec. 5.1 reviews the web of dualities we have obtained by gauging various subgroups of $S_3$. In Sec. 5.2, we describe symmetry-breaking patterns in terms of patch operators, and compute their expectation values in the gapped fixed-point ground states. We argue that these can be used to detect ordered and disordered phases of models with general fusion category symmetries. Finally, in Sec. 5.3 we study a Hamiltonian of spin-1/2 degrees of freedom that is exactly solvable, has an exact non-invertible self-duality symmetry in a certain parameter regime, and supports a gapless incommensurate phase in its phase diagram. We use this model to draw analogies with the gapless regions in the phase diagrams of $S_3$- and $\mathsf{Rep}(S_3)$-symmetric models and better understand the latter two.

## 5.1  Gauging-induced dualities

In Secs. 2 and 3, we presented several dualities that are induced by gauging subgroups of $S_3$ symmetry. In Fig. 11, we summarize the corresponding web of dualities. The corners of the diagram label the symmetry categories of the dual bond algebras while the each arrow implies a duality map induced by gauging. It has been shown in Ref. [24], distinct gaugings of a symmetry category $\mathcal{C}$ are in one-to-one correspondence with the left (or right) module categories over $\mathcal{C}$. Accordingly, we label each arrow in Fig. 11 by the choice of the corresponding module category.[36] Alternatively, each left (or right) module category over $\mathcal{C}$ is equivalent to the category of right (or left) $A$-modules over $\mathcal{C}$ where $A \in \mathcal{C}$ is some algebra object (see Appendix A of Ref. [38]). This correspondence forms the connection between the module categories over $\mathcal{C}$ and the perspective on gauging as summing over

---

[36]Module categories over $\mathsf{Vec}_G$ ($\mathsf{Rep}(G)$) are given by the categories $\mathsf{Vec}_H$ ($\mathsf{Rep}(H)$) where $H \leqslant G$, i.e., $H$ is a subgroup of G [111].

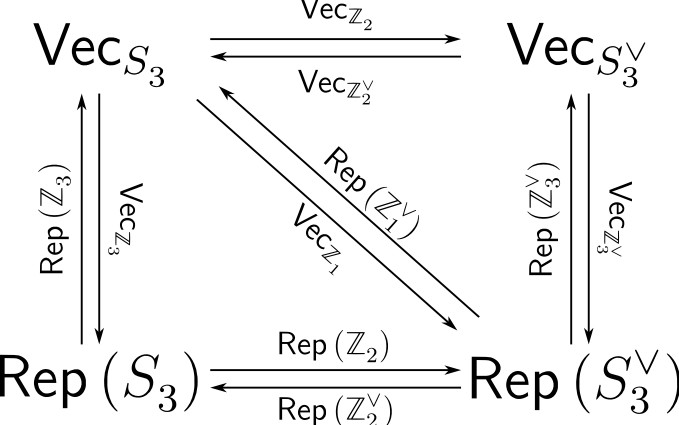

Figure 11: The web of dualities obtained induced by gauging. The four categories $\mathsf{Vec}_{S_3}$, $\mathsf{Vec}_{S_3^\vee}$, $\mathsf{Rep}(S_3)$, and $\mathsf{Rep}(S_3^\vee)$ at the corners of the diagram denote the fusion category symmetry of the bond algebras (2.6), (2.10), (3.5), and (3.14), respectively. Each directed arrow between these fusion categories denote a duality induced by gauging. The label on each arrow denotes the corresponding module category over the category at the source of the arrow.

symmetry defect insertions in two-dimensional spacetime. In the context of fusion category symmetries, gauging can then be understood as summing over all insertions of $A$-defects in the partition function in two-dimensional spacetime.

In Fig. 11, for the fusion category $\mathsf{Vec}_{S_3}$,[37] the module categories label the symmetry category of the subgroups that are not gauged. Note that the module categories are in one-to-one correspondence with (conjugacy classes) of subgroups of $S_3$. In particular, the module categories $\mathsf{Vec}_{S_3}$ and $\mathsf{Vec}_{\mathbb{Z}_2}$ of $\mathsf{Vec}_{S_3}$ correspond to gauging by the algebra objects $e$ (the trivial algebra object, implementing trivial gauging) and $e \oplus r \oplus r^2$, respectively. These gauging maps give back the same symmetry category $\mathsf{Vec}_{S_3}$. The latter, in particular, is what we would ordinarily describe as the $\mathbb{Z}_3$ Kramers-Wannier duality. We provided a recipe for implementing this gauging map in Sec. 2.2. On the other hand, the module categories $\mathsf{Vec}_{\mathbb{Z}_3}$ and $\mathsf{Vec}_{\mathbb{Z}_1}$, where $\mathbb{Z}_1$ is the trivial group, correspond to gauging $\mathsf{Vec}_{S_3}$ by the algebra objects $e \oplus s$ and $e \oplus r \oplus r^2 \oplus s \oplus sr \oplus sr^2$, respectively. These gauging maps lead to the dual $\mathsf{Rep}(S_3)$ symmetry category. The first of these is exactly the gauging map we used to construct the $\mathsf{Rep}(S_3)$ spin chain in Sec. 3, starting from the $S_3$ spin chain of Sec. 2. The second one of these can be implemented by first gauging $\mathbb{Z}_3$ in $S_3$ and then gauging the $\mathbb{Z}_2$ subgroup of the dual $S_3^\vee$, as discussed in Sec. 3.2.

For the fusion category $\mathsf{Rep}(S_3)$, we find that the algebra objects corresponding to three distinct gaugings of $\mathsf{Rep}(S_3)$ labeled by module categories $\mathsf{Rep}(\mathbb{Z}_3)$, $\mathsf{Rep}(\mathbb{Z}_2)$, and $\mathsf{Rep}(\mathbb{Z}_1)$ to be $\mathbf{1} \oplus \mathbf{1}'$, $\mathbf{1} \oplus \mathbf{2}$, and $\mathbf{1} \oplus \mathbf{1}' \oplus 2\,\mathbf{2}$, respectively.[38] Gauging by certain algebra objects can give back the same symmetry category. For other algebra objects, one gets a new (dual) symmetry category. This is a generalization of Kramers-Wannier duality to arbitrary fusion category symmetries. In our example with $\mathsf{Rep}(S_3)$ symmetry, we find that gauging by either of the algebra objects $\mathbf{1} \oplus \mathbf{1}'$ or $\mathbf{1} \oplus \mathbf{1}' \oplus 2\,\mathbf{2}$ (*i.e.*, the regular representation object) of $\mathsf{Rep}(S_3)$ gives rise to $S_3$ symmetry. On the other hand, gauging by either the

---

[37]For any group $G$, the fusion category $\mathsf{Vec}_G$ consists of simple objects that can be thought of as $G$-graded vector spaces, which fuse according to group multiplication law. The morphisms of this category are graded $G$-graded linear maps. In this subsection, we will refer to $S_3$ symmetry as $\mathsf{Vec}_{S_3}$ to emphasize the general language of fusion category symmetry.

[38]These were derived using the internal Hom construction outlined in Appendix A.3 of Ref. [38].

Table 3: Expectation values of $S_3$ order and disorder operators defined in Eq. (5.1) in the fixed point ground state wavefunctions defined in Eqs. (2.22), (2.23), (2.24), and (2.25), respectively. The non-zero (zero) expectation values of order (disorder) operators detect the spontaneous symmetry breaking and long-range order in the ground states.

| GS | $\widehat{C}_{\mathbb{Z}_2}(j,\ell)$ | $\widehat{U}_{[s]}(j,\ell)$ | $\widehat{C}_{\mathbb{Z}_3}(j,\ell)$ | $\widehat{U}_{[r]}(j,\ell)$ |
|---|---|---|---|---|
| $|\mathrm{GS}_{S_3}\rangle$ | 0 | +3 | 0 | +2 |
| $|\mathrm{GS}_{\mathbb{Z}_3}^{\pm}\rangle$ | +1 | 0 | 0 | +2 |
| $|\mathrm{GS}_{\mathbb{Z}_2}^{\alpha}\rangle$ | 0 | +1 | +2 | 0 |
| $|\mathrm{GS}_{\mathbb{Z}_1}^{\pm,\alpha}\rangle$ | +1 | 0 | +2 | 0 |

trivial algebra object $\mathbf{1}$ (*i.e.*, implementing trivial gauging) or the algebra object $\mathbf{1} \oplus \mathbf{2}$ gives back a "dual" $\mathsf{Rep}(S_3^\vee)$ symmetry. Like the familiar KW duality associated with gauging Abelian groups, the gauging by $\mathbf{1} \oplus \mathbf{2}$ implements a duality transformation that exchanges pairs of gapped phases of the system. In our analysis, we did not provide an explicit description of how gauging by algebra objects, via the insertion of defects approach put forward in Ref. [24], works at the level of microscopic Hamiltonians. However, we note that gauging by the $\mathbf{1} \oplus \mathbf{1}'$ algebra object should proceed very identically to gauging of an ordinary $\mathbb{Z}_2$ symmetry since $\widehat{W_{\bar{\mathbf{1}}}}$ indeed generates a $\mathbb{Z}_2$ sub-symmetry of $\mathsf{Rep}(S_3)$, and gauging by the regular representation algebra object should be identical to first gauging by $\mathbf{1} \oplus \mathbf{1}'$ and then gauging the $\mathbb{Z}_3$ subgroup of the resulting $S_3$ symmetry. Finally, we identify a sequential quantum circuit (3.18), that implements a duality transformation of our $\mathsf{Rep}(S_3)$ spin chain, which therefore must correspond to the remaining option of gauging by the $\mathbf{1} \oplus \mathbf{2}$ algebra object.

## 5.2 SSB patterns and order/disorder operators

Ordered phases in which symmetries are spontaneously broken can be detected by non-zero values of appropriate correlation functions of order operators. In contrast, disordered phases can be detected by non-zero values of appropriate correlation functions of disorder operators. The expectation values of correlation functions of order and disorder operators, considered together, have been found to be a tool that can detect gaplessness Ref. [112]. The idea of order and disorder operators can be generalized to non-invertible symmetries in the form of patch operators [29, 53, 107]. Depending on which gapped phase the system is in, different patch operators will get a non-zero expectation value in the ground state(s). This gives a way to detect symmetry-breaking even if we restrict to the symmetric sub-Hilbert space so that ground state degeneracy is no longer a reliable tool.

For the $S_3$ symmetry, we have the patch operators [39]

$$\widehat{C}_{\mathbb{Z}_2}(j,\ell) := \hat{\sigma}_j^z \, \hat{\sigma}_{j+\ell}^z, \qquad \widehat{C}_{\mathbb{Z}_3}(j,\ell) := \widehat{Z}_j \, \widehat{Z}_{j+\ell}^\dagger + \widehat{Z}_j^\dagger \, \widehat{Z}_{j+\ell}, \qquad (5.1\mathrm{a})$$

---

[39]We choose, without loss of generality, $\hat{\sigma}_j^z \, \hat{\sigma}_{j+\ell}^z$ to be the $\mathbb{Z}_2$ order operator. Alternatively, we could have chosen $\hat{\sigma}_j^z \, \hat{\tau}_{j+\ell}^z$, $\hat{\tau}_j^z \, \hat{\sigma}_{j+\ell}^z$, or $\hat{\tau}_j^z \, \hat{\tau}_{j+\ell}^z$ as well. Any of these choices for the order operators produce the same expectation values in the ground states of gapped fixed-points of the Hamiltonian (2.21).

Table 4: Expectation values of $\mathsf{Rep}(S_3)$ order and disorder operators defined in Eq. (5.2) in the fixed point ground state wavefunctions defined in Eqs. (3.27), (3.30), (3.33), (3.36a), and (3.36b), respectively. The non-zero (zero) expectation values of order (disorder) operators detect the spontaneous symmetry breaking and long-range order in the ground states.

| GS | $\widehat{C}_{\mathbf{1}'}(j,\ell)$ | $\widehat{W}_{\mathbf{1}'}(j,\ell)$ | $\widehat{C}_{\mathbf{2}}(j,\ell)$ | $\widehat{W}_{\mathbf{2}}(j,\ell)$ |
|---|---|---|---|---|
| $|\text{GS}_{\mathsf{Rep}(S_3)}\rangle$ | 0 | +1 | 0 | +2 |
| $|\text{GS}_{\mathbf{1}'}^{\alpha}\rangle$ | 0 | +1 | +2 | 0 |
| $|\text{GS}_{\mathbf{2}}^{\pm}\rangle$ | +3 | 0 | 0 | +2 |
| $|\text{GS}_{\mathbf{1}}^{\pm}\rangle$ | +1 | 0 | +2 | 0 |
| $|\text{GS}_{\mathbf{1}}^{3}\rangle$ | +1 | 0 | +2 | 0 |

that are associated with $\mathbb{Z}_2$ and $\mathbb{Z}_3$ order operators, respectively, and the patch operators

$$\widehat{U}_{[s]}(j,\ell) := \sum_{\alpha=0}^{2} \prod_{k=j}^{j+\ell} \hat{\sigma}_k^x \hat{\tau}_k^x \widehat{C}_k \widehat{X}_k^\alpha, \qquad \widehat{U}_{[r]}(j,\ell) := \prod_{k=j}^{j+\ell} \widehat{X}_k + \prod_{k=j}^{j+\ell} \widehat{X}_k^\dagger, \qquad (5.1b)$$

that are associated with $\mathbb{Z}_2$ and $\mathbb{Z}_3$ disorder operators, respectively. We note that both of these classes of patch operators are symmetric under the entire $S_3$ group, i.e., they are constructed out of the generators of $S_3$-symmetric bond algebra (2.6), while the disorder operators are closed under the action of $S_3$. Both order and disorder operators can be thought of as *transparent* patch operators [53] in the sense that they commute all the terms in the $S_3$ Hamiltonian (2.21) that are supported between sites $j+1$ and $j+\ell-1$. On the Hamiltonian their nontrivial actions only appear at their boundaries.

The expectation values attained by the patch operators in the gapped fixed-point ground states (2.22), (2.23), (2.24), and (2.25) are given in Table 3. In the fixed-point ground states, non-zero expectation values of order operators accompany the vanishing expectation values of disorder operators and detect spontaneous symmetry breaking in the ground states. We note that when the $S_3$ symmetry is broken down to $\mathbb{Z}_2$, each of the threefold degenerate ground states preserve a different $\mathbb{Z}_2$ subgroup which reflected in the non-vanishing expectation value of $\mathbb{Z}_2$ disorder operators on only one of the degenerate ground states. Away from the fixed-points the zero expectation values are expected to be replaced by an exponential decay $\propto e^{-|j-\ell|/\xi}$ with a finite non-zero correlation length $\xi$ (with gapped fixed-points corresponding to $\xi \to 0$ limit).

For the $\mathsf{Rep}(S_3)$ symmetry, we can apply the $\mathbb{Z}_2$ gauging map, derived in Sec. 3, to obtain the $\mathsf{Rep}(S_3)$ patch operators. Since we are gauging $\mathbb{Z}_2$ subgroup of $S_3$, we expect the $\mathbb{Z}_2$ order and disorder operators to be mapped to disorder and order operators of the dual $\mathbb{Z}_2$ symmetry generated by $\widehat{W}_{\mathbf{1}'}$ operator. Accordingly, we identify the patch operators

$$\widehat{C}_{\mathbf{1}'}(j,\ell) := \hat{\mu}_j^z \sum_{\alpha=0}^{2} \left( \prod_{k=j}^{j+\ell} \widehat{X}_k^{\prod_{q=j+1}^{k} \hat{\mu}_q^x} \right)^\alpha \hat{\mu}_{j+\ell}^z, \qquad (5.2a)$$

$$\widehat{W}_{\mathbf{1}'}(j,\ell) := \prod_{k=j+1}^{j+\ell} \hat{\mu}_k^x, \qquad (5.2b)$$

that correspond to $\widehat{W}_{\mathbf{1}'}$ order and disorder operators, respectively. In contrast to this, under gauging $\mathbb{Z}_2$ subgroup of $S_3$ symmetry, the $\mathbb{Z}_3$ order and disorder operators are mapped to

$$\widehat{C}_{\mathbf{2}}(j, \ell) := \widehat{Z}_j^{\prod_{k=j+1}^{j+\ell} \hat{\mu}_k^x} \, \widehat{Z}_{j+\ell}^{\dagger} + \widehat{Z}_j^{-\prod_{k=j+1}^{j+\ell} \hat{\mu}_k^x} \, \widehat{Z}_{j+\ell}, \tag{5.2c}$$

$$\widehat{W}_{\mathbf{2}}(j, \ell) := \prod_{k=j}^{j+\ell} \widehat{X}_k^{\prod_{q=j+1}^{k} \hat{\mu}_q^x} + \prod_{k=j}^{j+\ell} \widehat{X}_k^{-\prod_{q=j+1}^{k} \hat{\mu}_q^x}, \tag{5.2d}$$

which are the order and disorder operators, respectively, associated with the non-invertible symmetry operator $\widehat{W}_{\mathbf{2}}$. We note that for $\mathsf{Rep}(S_3)$ symmetry, order operators are non-local string-like objects, as opposed to the case of $S_3$ symmetry for which order operators are bilocal, i.e., products of two local operators. In other words, the spontaneous breaking of non-invertible $\mathsf{Rep}(S_3)$ is detected by non-local string order parameters. The expectation values attained by the patch operators in the gapped fixed-point ground states (3.27), (3.30), (3.33), and (3.36) are given in Table 4. We see that the expectation values of operators (5.2) in these ground states are consistent with interpreting the corresponding phases as $\mathsf{Rep}(S_3)$-symmetric, $\mathsf{Rep}(S_3)/\mathbb{Z}_2$ SSB, $\mathbb{Z}_2$ SSB, and $\mathsf{Rep}(S_3)$ SSB phases, respectively.

## 5.3 Incommensurate phase in a self-dual spin-1/2 chain

In the phase diagram of the $S_3$-symmetric model shown in Fig. 4, we observed an extended gapless phase that is centered around the self-dual line. We have argued that such a gapless phase is an incommensurate phase. We define an incommensurate state as a gapless state that has gapless excitations carrying quasi-momentum that is incommensurate with the size of the Brillouin zone. As a result, an incommensurate state contain gapless excitations whose quasi-momenta form a dense set that covers the whole Brillouin zone. To better understand such an incommensurate phase, we will first consider the following spin-1/2 chain

$$\widehat{H} := -\sum_{j=1}^{L} \left\{ J\, \hat{\sigma}_j^z \, \hat{\sigma}_{j+1}^z + h\, \hat{\sigma}_j^x + \lambda(\hat{\sigma}_j^y \, \hat{\sigma}_{j+1}^z - \hat{\sigma}_j^z \, \hat{\sigma}_{j+1}^y) \right\}, \tag{5.3}$$

which also has an incommensurate phase. Hamiltonian (5.3) describes the transverse field Ising model perturbed by a Dzyaloshinskii-Moriya-type interaction. The latter is one of the simplest two-body interactions that is symmetric under the (non-invertible) $\mathbb{Z}_2$ KW self-duality symmetry of the critical point of the Ising model ($h = J, \lambda = 0$). The Hamiltonian (5.3) then has the self-duality symmetry when $h = J$.

Unlike the $S_3$-symmetric Hamiltonian (2.3), this Hamiltonian is exactly solvable by applying Jordan-Wigner transformation. This maps the Hamiltonian to a fermionic one with only quadratic interactions [113]. Closely related results have also been obtained with other self-dual deformations of the critical Ising model [114–116]. After the JW transformation, one can diagonalize the fermionic Hamiltonian via a Bogoliubov transformation which yields

$$\widehat{H} = \sum_{-\pi < k \leq \pi} E_k \, \widehat{\psi}_k^{\dagger} \widehat{\psi}_k + E_0 \tag{5.4}$$

where

$$E_k = 4\lambda \sin k + 2\sqrt{J^2 \sin^2 k + (h - J \cos k)^2} \tag{5.5}$$

In the ground state of this Hamiltonian, the $k$ modes with negative energy are filled in the ground state. The excitations are hole-like/particle-like for the negative/positive $E_k$ modes.

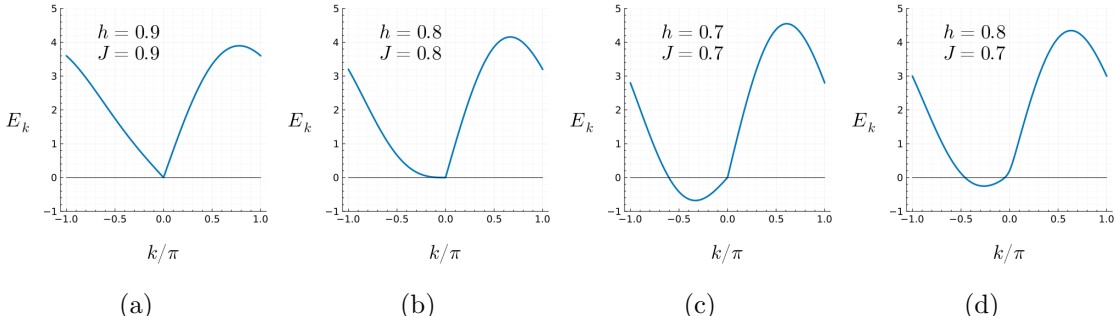

Figure 12: The fermion dispersion (5.5) for $\lambda = 2 - h - J$ and (a) $h = 0.9$, $J = 0.9$ (the $c = \frac{1}{2}$ self-dual critical point), (b) $h = 0.8$, $J = 0.8$ (the $z = 3$ self-dual multi-critical point), (c) $h = 0.7$, $J = 0.7$ (the $c = 1$ self-dual gapless phase), (d) $h = 0.8$, $J = 0.7$ (the $c = 1$ gapless phase)

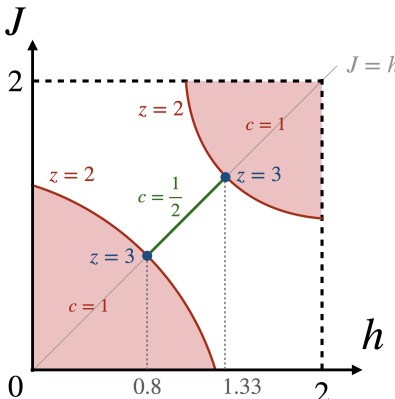

Figure 13: Phase diagram of Hamiltonian (5.3) in the $(h, J)$ plane with $\lambda = 2 - J - h$ at each point. The unshaded regions are gapped. The $h = J$ line has the self-duality symmetry.

The fermion dispersion (5.5) is plotted in Fig. 12 for some values of $h, J, \lambda$. From this spectrum, we can identify the low-energy degrees of freedom and whether the ground state is gapped or not. Along the self-dual line, the central charge computed from DMRG evaluates to $c = 1/2$ for $\lambda \neq 0$. On the other hand, in the large $|\lambda|$ limit, we find $c = 1$. In fact there are extended gapless phases with $c = 1$ even away from the self-dual line in the large $|\lambda|$ limit. The full phase diagram is shown in Fig. 13, with the gapped phases in white and two gapless phases shaded in red. The critical point corresponding to the Ising CFT lies at the $J = h$, $\lambda = 0$ point.

The extended gapless regions labeled $c = 1$ have gapless linearly dispersing modes near a "Fermi momentum" that smoothly varies throughout the regions. This behavior is seen only when $\lambda$ is larger than 0.4 or smaller than $-0.66$. Moreover, we see that the value of $c = 1/2$ is stable to a finite non-zero value of $\lambda$. In other words, the $c = 1/2$ critical point becomes a stable gapless phase in the presence of the self-duality symmetry on the $h = J$ line. This is consistent with the fact that the $\lambda$ term in (5.3) corresponds to an exactly marginal chiral operator $\propto T - \bar{T}$ in the Ising CFT [117].

Let us make a few comments:

(i) Time reversal and reflection symmetries are explicitly broken by the $\lambda$ term in (5.3).

(ii) The self-duality symmetry pins the right-moving particle- and hole-like excitations at the $k = 0$ point.

(iii) In each $c = 1$ gapless region, there is a chiral complex fermion mode that has non-zero Fermi momentum $k_{\mathrm{F}}$. This $k_{\mathrm{F}}$ varies smoothly throughout the region, hence we refer to it as an incommensurate gapless phase.

(iv) Without the self-duality symmetry, there are both left- and right-moving complex fermion modes with non-zero Fermi momenta $k_{\mathrm{F}}$.

(v) The fermion dispersion has a singular change as we go across the $h = J$ line in the gapless incommensurate phase. This suggests that this is a continuous phase transition within the incommensurate phase.

In the incommensurate gapless phase for large $\lambda$, a low-energy chiral (complex) fermion mode near Fermi momentum $k_F$ furnishes a representation of a U(1) symmetry. This U(1) symmetry emanates from the lattice translation symmetry. For instance when $\lambda > 0.4$, there are left-moving fermion modes satisfying

$$\widehat{T}_{\mathrm{lat}}\widehat{\psi}_{L,q}\widehat{T}_{\mathrm{lat}}^{\dagger} = e^{ik_F}e^{iq}\widehat{\psi}_{L,q}\,, \tag{5.6}$$

while the right-moving fermion modes satisfy

$$T_{\mathrm{lat}}\widehat{\psi}_{R,q}T_{\mathrm{lat}}^{\dagger} = e^{iq}\widehat{\psi}_{R,q}\,. \tag{5.7}$$

So this chiral U(1) symmetry acts as $e^{ik_F\widehat{N}_L}$ where $\widehat{N}_L$ is the number operator in the left-moving sector of the IR theory. When $k_F$ is not a rational multiple of $\pi$, the set $\{e^{ik_F\widehat{N}_L}|N_L\in\mathbb{Z}_{\geq 0}\}$ is dense in U(1). However when $k_F = 2\pi p/q$, with $\gcd(p,q) = 1$, the emanent symmetry becomes a finite cyclic group of order $q$. We still refer to the emanent symmetry as $U(1)$ since the finite-order case happens only for a measure 0 subset of values of $k_F \in (-\pi, \pi]$.

The presence of U(1) symmetries emanating from lattice translation may be a general feature of the low energy effective field theory of an incommensurate gapless state. Note that the right-moving particles and holes (for $\lambda > 0.4$) carry zero U(1) charge (see Fig. 12c). The emanant chiral U(1) symmetry is anomalous. This anomaly is reflected in the UV by the fact that the ground state carries momentum

$$k_0 = \sum_{k\in(-\pi,\pi]} k\,\Theta(-E_k) \overset{L\to\infty}{\approx} \frac{k_F^2}{2}\,. \tag{5.8}$$

So the UV translation symmetry acts projectively on the low-energy states. This has close parallels with the examples discussed in Ref. [89]. In addition to the appearance of anomalous chiral U(1) symmetry, the notion of incommensurate state also allows us to make the following conjecture. Some continuous transitions, *e.g.*, Mott insulator to superfluid transition, involve the introduction of gapless modes. If such gapless modes are incommensurate, *i.e.*, if the transition is between commensurate and incommensurate phases, then the dynamical exponent must satisfy $z > 1$. The transition along the self-dual $h = J$ line at $h = J = 0.8$ in Fig. 13 is an example of such a transition, with $z = 3$.

In the $c = 1$ incommensurate phase, away from the $h = J$ line, there are two chiral complex fermion modes, carrying different non-zero U(1) charges (*i.e.*, quasi-momenta). On the $h = J$ line, the U(1) charge of one set of chiral complex fermion modes vanishes, and the associated particle and hole branches acquire unequal velocities. This causes a singularity in the single-particle fermion spectrum, signaling a possible continuous phase transition. This is an unusual kind of phase transition, since it does not have more gapless modes compared to the phases on the two sides of the transition.

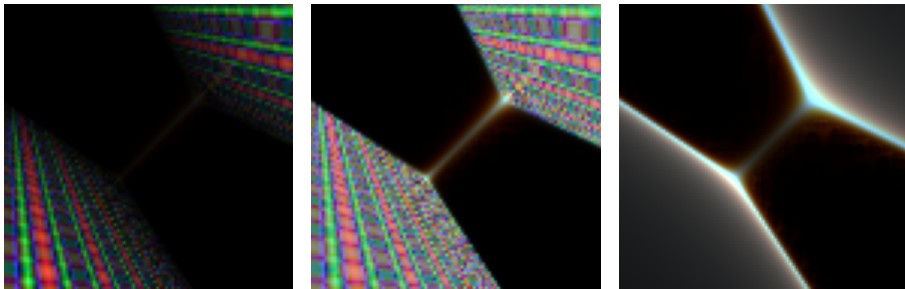

Figure 14: (Middle) Central charge $c$ computed via tensor network numerical approach as a function of $h \in [0, 2]$ (horizontal) and $J \in [0, 2]$ (vertical) with $\lambda = 2 - J - h$, for the Ising chain (5.3) of sizes $L = 256$ (blue), $L = 128$ (green), $L = 64$ (red). The range of the color intensity $[0,1]$ corresponds to the range of the central charge $c \in [0, 2]$. The stripe pattern in the gapless $c \neq 0$ area comes from incommensurate nature of the gapless phase. (Left) A plot of $vc$ for the same range of $h$ and $J$. (Right) A plot of $c/v$ for the same range of $h$ and $J$. Here $v$ is the velocity of the gapless mode.

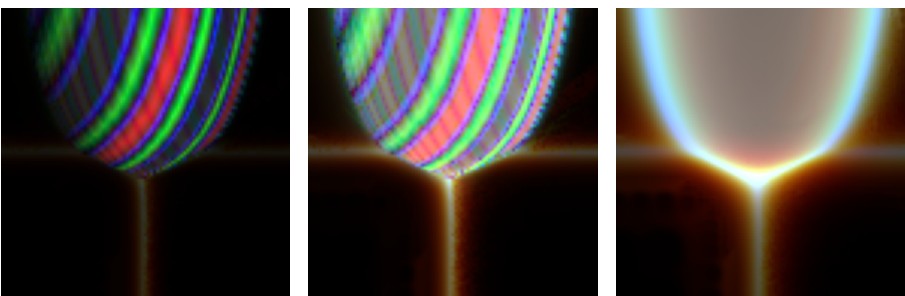

Figure 15: (Middle) Central charge $c$ computed via tensor network numerical approach as a function of $J_1 \in [0.4, 0.6]$ (horizontal) and $J_3 \in [0.4, 0, 6]$ (vertical) for the $S_3$-symmetric model (2.21) of sizes $L = 256$ (blue), $L = 128$ (green), $L = 64$ (red). We have chosen $J_2 = 1 - J_1$, $J_4 = 1 - J_3$ and $\theta = 0.7 \approx \frac{2\pi}{9}$. The $J_1 = J_2 = 0.5$ line is the self-dual line. The range of the color intensity $[0,1]$ corresponds to the range of the central charge $c \in [0, 2]$. The stripe pattern in the gapless $c \neq 0$ area comes from incommensurate nature of the gapless phase. (Left) A plot of $vc$ for the same range of $h$ and $J$. (Right) A plot of $c/v$ for the same range of $h$ and $J$. Here $v$ is the velocity of the gapless mode.

In Fig. 14, we present a tensor network calculation of the central charge for the Hamiltonian (5.3). In the gapless incommensurate phase we find a stripe pattern in central charge $c$. To understand the appearance of stripes, we note that the central charge is contained in the $1/L$ term in the ground state energy:

$$E_{\text{grnd}} = L\epsilon - \frac{cv}{24}\frac{2\pi}{L}. \tag{5.9}$$

By computing the ground state energy at size $L$ and $L/2$, we can extract the $1/L$ and obtain the value of $vc$. By doing the similar calculation with space and time exchanged, we can compute $c/v$. Then the central charge is the geometric mean of $vc$ and $c/v$. Certainly, this approach works only when all the gapless modes have the same velocity $v$. This is how we obtained the Fig. 14. We see that $vc$ is small at the boundary of the incommensurate phase, while $c/v$ is large at the boundary. This is consistent with our exact result, which implies that $v = 0$ at the boundary. As we change parameters, the U(1) charge (*i.e.*, the

quasi-momentum) of the ground state may jump, which cause a change of ground state energy of order $1/L$. This leads to the stripe pattern in the calculated central charge.

Armed with this understanding, we compute the central charge $c$, as well as $vc$ and $c/v$, for the $S_3$ symmetric Hamiltonians (see Fig. 15). The similarity between Figs. 15 and 14 suggests that the gapless phase in Fig. 15 is also incommensurate. The continuous transition between commensurate and incommensurate phases has a dynamical exponent $z > 1$. Also, the gapless incommensurate phase of our $S_3$-symmetric model has a local low energy effective field theory with internal U(1) symmetry (coming from the lattice translation symmetry). Since $S_3$ and $\mathsf{Rep}(S_3)$ symmetries are Morita equivalent as fusion categories, the same results also apply to our $\mathsf{Rep}(S_3)$-symmetric model.

The parameter $\theta$ in our $S_3$ and $\mathsf{Rep}(S_3)$ symmetric models, (2.21) and (3.25), plays a very similar role to that of $\lambda$ in Hamiltonian (5.3). For small $\theta$ and $\lambda$ along the self-dual line, the (multi-)critical point remains gapless and behaves like stable gapless phase protected by the non-invertible self-duality symmetry. At large enough values of $\theta$ and $\lambda$ in respective models, this gapless phase turns into and incommensurate phase via a $z > 1$ continuous transition.

# 6 Conclusion

In this paper, we have studied the consequences of non-invertible symmetries that are realized as genuine UV symmetries (*i.e.*, not emergent IR symmetries) of 1+1d Hamiltonian lattice models defined on Hilbert spaces having tensor product decompositions with finite-dimensional on-site Hilbert spaces; we refer to them as *spin chains* in short. We constructed a spin chains with $\mathsf{Rep}(S_3)$ symmetry by gauging the non-normal $\mathbb{Z}_2$ subgroup symmetry of an $S_3$-symmetric spin chain. This provides an explicit microscopic verification of an instance of the fact that gauging non-normal subgroups in a theory with an ordinary group-like symmetry leads to a dual theory with a non-invertible symmetry, which was known using more abstract methods in the literature. [24,118] We explored SSB phases of the non-invertible $\mathsf{Rep}(S_3)$ symmetry as well as continuous phase transitions between them. Both the $S_3$ and $\mathsf{Rep}(S_3)$ spin chains demonstrate intrinsically non-invertible self-duality symmetries in special subspaces of the full parameter space. We identify the SymTO description of this enhanced symmetry and use it to obtain model-independent constraints on the phase diagrams with this symmetry.

There are various directions for future work. It would be interesting to study how gauging by algebra objects may be implemented at the lattice level. This requires a deeper understanding of how symmetry twists of non-invertible symmetries are implemented in the context of spin chain Hamiltonians. Such an understanding would also be extremely useful in making statements about the gauging-related dualities in the context of non-invertible symmetries, including those discussed in this paper, more precise. We note that similar questions have been addressed in the context of lattice models built on Hilbert spaces that do not necessarily have a tensor product decomposition, such as in Refs. [66, 78, 79]. We believe that our treatment of these questions is complementary to these previous works. In fact, it is not clear that all fusion category symmetries can be realized in spin chain Hamiltonians as strictly internal symmetries (*cf.* footnote 32 of Ref. [76]). Using our lattice model, we also explored KW-symmetric perturbations to the analytically tractable limit of the $S_3$ spin chain and uncovered a stable gapless phase. We find evidence that the numerically computed central charge vanishes at the phase boundaries of this gapless region, which is consistent with a dynamical critical exponent $z > 1$. It would be interesting to characterize this gapless phase and transitions out of it, on or away from the

KW-symmetric line, from a low-energy field theory perspective. Lastly, it is possible to gauge the $\mathbb{Z}_2$ sub-symmetry of the $S_3$ symmetric spin chain using fermionic $\mathbb{Z}_2$ variables instead of bosonic ones. This leads to a theory of fermions coupled to spin variables with a fermionic $\mathsf{SRep}(S_3)$ symmetry. We discuss this in Appendix G as a straightforward generalization of the story presented in Sec. 3.1 to fermionic lattice models with non-invertible symmetries. The fermion parity symmetry $\mathbb{Z}_2^{\mathrm{F}}$ becomes a part of the symmetry category in the gauged model. We leave a discussion of more non- trivial examples for future work.

# Acknowledgements

We thank Liang Kong, Sal Pace, Shu-Heng Shao for helpful comments on the manuscript, and Alex Avdoshkin, Clement Delcamp, Zhihuan Dong, Andreas Läuchli, Frederic Mila, Christopher Mudry, Seth Musser, Sal Pace, Rahul Sahay, Sahand Seifnashri, Tomo Soejima, Ophelia Sommer, Saran Prembabu, Nat Tantivasadakarn, and Carolyn Zhang for many helpful discussions.

**Funding information**  This work is partially supported by NSF DMR-2022428 and by the Simons Collaboration on Ultra-Quantum Matter, which is a grant from the Simons Foundation (651446, XGW). Ö. M. A. is also supported by Swiss National Science Foundation (SNSF) under Grant No. P500PT-214429.

# A  Group $S_3$

The group $S_3$ has cardinality $|S_3| = 6$ and generated by two elements $s$ and $r$ such that $s^2 = r^3 = e$, $s\, r = r^2\, s$, where $e$ is the identity element of the group. It has four non-trivial proper subgroups that we denote by

$$\mathbb{Z}_3^r \coloneqq \left\{e,\, r,\, r^2\right\}, \quad \mathbb{Z}_2^s \coloneqq \left\{e,\, s\right\}, \quad \mathbb{Z}_2^{sr} \coloneqq \left\{e,\, s\,r\right\}, \quad \mathbb{Z}_2^{sr^2} \coloneqq \left\{e,\, s\,r^2\right\}, \tag{A.1}$$

respectively. From now on, we will choose $\mathbb{Z}_2^s$ as the $\mathbb{Z}_2$ subgroup and drop the superscripts $r$ and $s$ when referring to the $\mathbb{Z}_3$ and $\mathbb{Z}_2$ subgroups of $S_3$.

There are 3 irreducible representations (irreps) of the group $S_3$. The trivial one, with all group elements represented by the number 1, is denoted $\mathbf{1}$,

$$U_{\mathbf{1}}(g) = 1, \quad \forall g \in S_3. \tag{A.2}$$

There is a second one-dimensional irrep, denoted $\mathbf{1}'$, with

$$U_{\mathbf{1}'}(e) = U_{\mathbf{1}'}(r) = U_{\mathbf{1}'}(r^2) = 1, \qquad U_{\mathbf{1}'}(s) = U_{\mathbf{1}'}(sr) = U_{\mathbf{1}'}(sr^2) = -1. \tag{A.3}$$

The third irrep is a two-dimensional one, denoted $\mathbf{2}$, with

$$
U_{\mathbf{2}}(e) = \begin{pmatrix} 1 & 0 \\ 0 & 1 \end{pmatrix}, \quad
U_{\mathbf{2}}(r) = \begin{pmatrix} e^{i\frac{2\pi}{3}} & 0 \\ 0 & e^{-i\frac{2\pi}{3}} \end{pmatrix}, \quad
U_{\mathbf{2}}(r^2) = \begin{pmatrix} e^{-i\frac{2\pi}{3}} & 0 \\ 0 & e^{i\frac{2\pi}{3}} \end{pmatrix},
$$
$$
U_{\mathbf{2}}(s) = \begin{pmatrix} 0 & 1 \\ 1 & 0 \end{pmatrix}, \quad
U_{\mathbf{2}}(sr) = \begin{pmatrix} 0 & e^{-i\frac{2\pi}{3}} \\ e^{i\frac{2\pi}{3}} & 0 \end{pmatrix}, \quad
U_{\mathbf{2}}(sr^2) = \begin{pmatrix} 0 & e^{i\frac{2\pi}{3}} \\ e^{-i\frac{2\pi}{3}} & 0 \end{pmatrix}.
$$

The irrep $\mathbf{2}$ is the only faithful irrep of $S_3$. The tensor product of the irreps forms a so-called fusion ring with $\mathbf{1}$ as the identity of the ring. The product operation (or, fusion) is commutative with the following non-trivial fusion rules:

$$\mathbf{1}' \otimes \mathbf{1}' = \mathbf{1}, \quad \mathbf{1}' \otimes \mathbf{2} = \mathbf{2}, \quad \mathbf{2} \otimes \mathbf{2} = \mathbf{1} \oplus \mathbf{1}' \oplus \mathbf{2} \tag{A.4}$$

The group $S_3$ has three conjugacy classes

$$[e] \coloneqq \{e\}, \qquad [s] \coloneqq \{s,\, s\,r,\, s\,r^2\}, \qquad [r] \coloneqq \{r,\, r^2\}, \tag{A.5a}$$

labeled by a representative element. For each of these conjugacy classes, the centralizer of the representative are

$$C_{S_3}(e) = S_3, \qquad C_{S_3}(s) = \mathbb{Z}_2^s, \qquad C_{S_3}(r) = \mathbb{Z}_3^r, \tag{A.5b}$$

respectively. These centralizers have the irreps

$$\pi_e \coloneqq \mathbf{1},\, \mathbf{1}',\, \mathbf{2}, \tag{A.5c}$$
$$\pi_s \coloneqq \mathbf{1},\, \mathbf{1}', \tag{A.5d}$$
$$\pi_r \coloneqq \mathbf{1},\, \mathbf{1}_\omega,\, \mathbf{1}_{\omega^*}, \tag{A.5e}$$

respectively. Here, all irreps are one-dimensional except $\mathbf{2}$. $\mathbf{1}'$ denotes the non-trivial one-dimensional irrep of $\mathbb{Z}_2$, while $\mathbf{1}_\omega$ and $\mathbf{1}_{\omega^*}$ are the non-trivial one-dimensional irreps of $\mathbb{Z}_3^r$ where $\omega = \exp\{\mathrm{i}2\pi/3\}$.

# B Brief review of SymTO

The wide variety of (finite) generalized symmetries considered in the context of quantum field theories and quantum many body physics can be provided a unified description in the language of topological order in one higher dimension. This general philosophy was put forward and discussed in Refs. [8, 28–30, 32, 42, 53, 119, 120], while a related connection with noninvertible gravitational anomaly was explored in Ref. [26, 27]. The correspondence between finite symmetries in $d$ spacetime dimensions and topological order in $d+1$ spacetime dimensions was referred to as Symmetry/Topological Order correspondence in older work of two of the present authors [45]. Closely related constructions have been referred to by various other names in the generalized symmetries literature – SymTFT, topological holography, categorical symmetry, topological symmetry *etc.* Similar ideas were discussed for specialized situations, including for 1+1d systems, for rational conformal field theories, or in the context of duality and gauging in Refs. [1, 2, 10, 39–41, 109, 121–127]. In this appendix, we summarize the aspects of the Sym/TO correspondence that are relevant for the present paper.

## B.1 Algebra of local symmetric operators

The most general way to define generalized symmetry is to start with a subset of local operators, that is closed under addition and multiplication, *i.e.*, to start with a sub-algebra of the local operator algebra. We define the operators in sub-algebra as the symmetric operators of a yet-to-be-determined symmetry. The symmetry transformations are defined as the commutant of the algebra of the local symmetric operators. The symmetry defined this way is very general, which include anomalous, higher-form, higher-group, and/or non-invertible symmetries. What is the mathematical frame work that can describe and classify the generalized symmetry defined this way?

To reveal the underlying mathematical structure of the algebra of the local symmetric operators, Refs. [29, 53] introduced the notion of transparent patch operators to capture the essence of isomorphic algebras of the local symmetric operators. Even though the algebra is generated by local symmetric operators, the algebra must contain extended operators. Patch operators are a type of extended operators, that have an *extended* spatial support and are created by a combination of a *large* number of so-called local symmetric operators.[40] A patch operator has the transparent property if it commutes with all local symmetric operators far away from the patch boundary.

So the bulk of the transparent patch operators is invisible, and can be ignored physically. The boundaries of the transparent patch operators correspond to fractionalized topological excitations, or the super selection sectors of the corresponding generalized symmetry. With this understanding, now it is easy to see that operator algebra of transparent patch operators is equivalent to the braiding and fusion of topological excitations (which is encoded in the algebra of transparent string, membrane, *etc.* operators). This gives a correspondence between the boundaries of transparent patch operators in $d$ dimensional quantum systems and topological excitations of the $d + 1$ dimensional topological order.

Through such a consideration, Refs. [29, 53] reveals a close connection between isomorphic algebras of the local symmetric operators and topological orders in one higher dimension. Such a topological order in one higher dimension is called symmetry topological order (SymTO).[41]

Note that different set of local symmetric operators may generate isomorphic operator algebras. In this case, the corresponding symmetries are called holo-equivalent [8]. The holo-equivalent symmetries give rise to the same local low energy properties and have the same SymTO. The holo-equivalent symmetries are known in the math literature as Morita equivalent symmetries. Many examples of holo-equivalent symmetries are given in Ref. [53].

The algebra of local symmetric operators and their description by transparent patch operators give rise to derivation [53] of the topological holographic principle: *boundary determines bulk* [26, 27], but bulk does not determine boundary. The algebra of local symmetric operators correspond to the "boundary", and the obtained topological order in one higher dimension is the "bulk".

## B.2  Phases and phase transitions from SymTO

Through the Sym/TO correspondence, the gapped boundaries of the SymTO can be mapped to gapped phases of the symmetric systems. The gapped boundaries of topological orders were classified by Lagrangian condensible algebras in Ref. [19], and thus the gapped phases of the symmetric systems can be classified by Lagrangian condensible algebras of the corresponding SymTO.

More generally, Ref. [44] argued that the non-Lagrangian condensable algebras correspond to gapless phases or critical points of the system – see also Ref. [84], for a closely-related discussion. Each non-Lagrangian algebra, in turn, has an associated *reduced SymTO* which constrains the CFT that can describe the corresponding gapless

---

[40]The italicized terms can be made more precise. For a system with linear size $L$, the patch operators have a support on a number of sites that is somewhere between $\mathcal{O}(1)$ and $\mathcal{O}(L)$, say $\mathcal{O}(\sqrt{L})$.

[41]In Refs. [8, 29, 53] SymTO was referred to as "categorical symmetry". The name is motivated by the following consideration: "categorical symmetry" contains conservation of both symmetry charges and symmetry defects, plus the additional braiding structure of the those symmetry objects. Conservation corresponds to "symmetry" and the additional braiding structure corresponds to "categorical" in the name.

state. In more concrete terms, gapped phases described by Lagrangian condensable algebras $\mathcal{A}_1$ and $\mathcal{A}_2$ have a phase transition corresponding to the non-Lagrangian algebra $\mathcal{A}_{12} = \mathcal{A}_1 \cap \mathcal{A}_2$. Anyon permutation symmetries that preserve $\mathcal{A}_{12}$ are associated with emergent symmetries of the IR theory that describes the corresponding phase transition. In Ref. [44], the algebraic structure of the condensable algebras did not play any explicit role. There is a suggestion that the algebraic properties of the order parameters for various gapped phases allowed by the SymTO may be encoded in the algebra product of the corresponding Lagrangian algebra. This connection has not been explored in the literature yet.

### B.3 Holo-equivalence and gauging

Symmetries whose SymTOs are identical were referred to as "holo-equivalent" in previous literature [8]. This is a much more general statement than the Morita equivalence of symmetry (fusion) categories in 1+1d, but they coincide in the latter case [128]. In 1+1d, for instance, the symmetry categories $\mathsf{Vec}_G$ and $\mathsf{Rep}(G)$ are related under gauging. More specifically, gauging the entire group $G$ in $\mathsf{Vec}_G$, which can be achieved by gauging by the algebra object $\mathcal{A}_G = \sum_g a_g \in \mathsf{Vec}_G$, gives the dual symmetry category $\mathsf{Rep}(G)$. Here, by $a_g$ we refer to the simple object of $\mathsf{Vec}_G$ labeled by the group element $g \in G$. On the other hand, gauging by the regular representation algebra object $\mathcal{A}_{\text{reg}} = \sum_R d_R a_R \in \mathsf{Rep}(G)$, where $d_R$ is the dimension of the irreducible representation $R$, gives the dual symmetry category $\mathsf{Vec}_G$. For more details on gauging by algebra objects, the reader is encouraged to refer to Ref. [24].

Following the discussion in Sec. B.2, we conclude that the phases of systems with Morita equivalent symmetries should have a one-to-one correspondence in their local, low energy properties. One way to see this is that since these Morita equivalent symmetries are related to each other by gauging, only certain global features of the corresponding phase diagrams should be altered. In particular, we expect the same CFTs (up to global modifications, *e.g.*, orbifolding) to describe the phase transitions related under this correspondence.

## C  Details of duality transformations

### C.1  $\mathbb{Z}_3$ Kramers-Wannier duality

The duality transformation of the model $\widehat{H}_{S_3}$ is implemented by the operator

$$\widehat{D}_{\text{KW}} := \hat{\mathsf{t}}_{\mathbb{Z}_2}\, \widehat{P}^{U_r=1}\, \widehat{W}\, \left( \widehat{\mathfrak{H}}_1^\dagger\, \widehat{\text{CZ}}_{2,1}^\dagger \right) \left( \widehat{\mathfrak{H}}_2^\dagger\, \widehat{\text{CZ}}_{3,2}^\dagger \right) \cdots \left( \widehat{\mathfrak{H}}_{L-1}^\dagger\, \widehat{\text{CZ}}_{L,L-1}^\dagger \right), \qquad \text{(C.1)}$$

as discussed in the main text. As written in Eq. (C.1), it has the form of a sequential circuit where each operator can be thought of as a unitary quantum gate acting on ket states sequentially starting from the rightmost operator. The unitary operators in $\widehat{D}_{\text{KW}}$ are defined as follows:

(i) We denote by $\widehat{\text{CZ}}_{i,j}^\dagger$ the controlled-Z operator

$$\widehat{\text{CZ}}_{i,j}^\dagger := \sum_{\alpha=0}^{2} \widehat{Z}_j^{-\alpha}\, \widehat{P}_i^{Z=\omega^\alpha}, \qquad \text{(C.2a)}$$

where $\widehat{P}_i^{Z=\omega^\alpha}$ is the projector onto the $\widehat{Z}_i = \omega^\alpha$ subspace.

(ii) We denote by $\widehat{\mathfrak{H}}_i^\dagger$ the Hadamard operator

$$\widehat{\mathfrak{H}}_i^\dagger := \frac{1}{\sqrt{3}} \sum_{\alpha,\beta=0}^{2} \omega^{\alpha\beta}\, \widehat{X}_i^{\alpha-\beta}\, \widehat{P}_i^{Z=\omega^\beta}. \tag{C.2b}$$

(iii) We denote by $\widehat{W}$ the unitary operator

$$\widehat{W} := \sum_{\alpha=0}^{2} \widehat{Z}_L^\alpha\, \widehat{P}^{Z_1^\dagger \widehat{Z}_L = \omega^\alpha}, \tag{C.2c}$$

where $\widehat{P}^{Z_1^\dagger \widehat{Z}_L = \omega^\alpha}$ is the projector onto $\widehat{Z}_1^\dagger \widehat{Z}_L = \omega^\alpha$ subspace. This unitary acts non-trivially only at sites 1 and $L$.

(iv) We denote by $\widehat{P}^{U_r=1}$ the projector

$$\widehat{P}^{U_r=1} := \frac{1}{3} \sum_{\alpha=0}^{2} \prod_{i=1}^{L} \widehat{X}_i^\alpha, \tag{C.2d}$$

which projects onto the $\widehat{U}_r \equiv \prod_i \widehat{X}_i = 1$ subspace.

(v) Finally, we denote by $\hat{\mathfrak{t}}_{\mathbb{Z}_2}$ the unitary operator

$$\hat{\mathfrak{t}}_{\mathbb{Z}_2} := \frac{1+\boldsymbol{\sigma}_1 \cdot \boldsymbol{\tau}_1}{2} \prod_{i=1}^{L-1} \frac{1+\boldsymbol{\tau}_i \cdot \boldsymbol{\sigma}_{i+1}}{2} \frac{1+\boldsymbol{\sigma}_{i+1} \cdot \boldsymbol{\tau}_{i+1}}{2}, \tag{C.2e}$$

which implements a "half-translation" of qubits. We note that as written $\hat{\mathfrak{t}}_{\mathbb{Z}_2}$ also has the form of a sequential quantum circuit.

In what follows, we list the non-trivial action of the $\mathbb{Z}_3$ KW duality operator (C.1) on generators of the bond algebra (2.6) at each step of the sequential circuit.

(i) Step 1: The only non-trivial action of the operator $\widehat{\mathrm{CZ}}_{L,L-1}^\dagger$ by conjugation is

$$\begin{aligned}
\widehat{X}_L &\mapsto \widehat{Z}_{L-1}^\dagger \widehat{X}_L, \\
\widehat{X}_{L-1} &\mapsto \widehat{X}_{L-1}\widehat{Z}_L^\dagger.
\end{aligned} \tag{C.3}$$

Note that the controlled-Z operators commute with all $\widehat{Z}_i$ operators.

(ii) Step 2: The only non-trivial action of the operator $\widehat{\mathfrak{H}}_{L-1}^\dagger$ by conjugation is

$$\begin{aligned}
\widehat{Z}_{L-1}^\dagger \widehat{X}_L &\mapsto \widehat{X}_{L-1}\widehat{X}_L, \\
\widehat{X}_{L-1}\widehat{Z}_L^\dagger &\mapsto \widehat{Z}_{L-1}\widehat{Z}_L^\dagger, \\
\widehat{Z}_{L-1}\widehat{Z}_L^\dagger &\mapsto \widehat{X}_{L-1}^\dagger \widehat{Z}_L^\dagger, \\
\widehat{Z}_{L-2}\widehat{Z}_{L-1}^\dagger &\mapsto \widehat{Z}_{L-2}\widehat{X}_{L-1}.
\end{aligned} \tag{C.4}$$

(iii) Step 3: The only non-trivial action of the operator $\widehat{\mathrm{CZ}}_{L-1,L-2}^\dagger$ by conjugation is

$$\begin{aligned}
\widehat{X}_{L-1}\widehat{X}_L &\mapsto \widehat{Z}_{L-2}^\dagger \widehat{X}_{L-1}\widehat{X}_L, \\
\widehat{X}_{L-1}^\dagger \widehat{Z}_L^\dagger &\mapsto \widehat{Z}_{L-2}\widehat{X}_{L-1}^\dagger \widehat{Z}_L^\dagger, \\
\widehat{Z}_{L-2}\widehat{X}_{L-1} &\mapsto \widehat{X}_{L-1}, \\
\widehat{X}_{L-2} &\mapsto \widehat{X}_{L-2}\widehat{Z}_{L-1}^\dagger.
\end{aligned} \tag{C.5}$$

(iv) Step 4: The only non-trivial action of the operator $\widehat{\mathfrak{H}}_{L-2}^{\dagger}$ by conjugation is given by

$$
\begin{aligned}
\widehat{Z}_{L-2}^{\dagger}\widehat{X}_{L-1}\widehat{X}_L &\mapsto \widehat{X}_{L-2}\widehat{X}_{L-1}\widehat{X}_L\,, \\
\widehat{Z}_{L-2}\widehat{X}_{L-1}^{\dagger}\widehat{Z}_L^{\dagger} &\mapsto \widehat{X}_{L-2}^{\dagger}\widehat{X}_{L-1}^{\dagger}\widehat{Z}_L^{\dagger}\,, \\
\widehat{X}_{L-2}\widehat{Z}_{L-1}^{\dagger} &\mapsto \widehat{Z}_{L-2}\widehat{Z}_{L-1}^{\dagger}\,, \\
\widehat{Z}_{L-3}\widehat{Z}_{L-2}^{\dagger} &\mapsto \widehat{Z}_{L-3}\widehat{X}_{L-2}.
\end{aligned}
\tag{C.6}
$$

Following this pattern for $L-1$ steps maps $\widehat{X}_j$ to $\widehat{Z}_j\widehat{Z}_{j+1}^{\dagger}$ for $j \in \{1,2,\ldots,L-1\}$, and $\widehat{Z}_j\widehat{Z}_{j+1}^{\dagger}$ to $\widehat{X}_{j+1}$ for $j \in \{1,2,\ldots,L-2\}$. This matches the Kramers-Wannier transformation that we set out to achieve but only for all but a few terms around the sites $j=1$ and $j=L$. These terms are

$$
\begin{aligned}
\widehat{X}_L &\mapsto \widehat{X}_1\widehat{X}_2\ldots\widehat{X}_L\,, \\
\widehat{Z}_{L-1}\widehat{Z}_L^{\dagger} &\mapsto \widehat{X}_1^{\dagger}\widehat{X}_2^{\dagger}\ldots\widehat{X}_{L-1}^{\dagger}\widehat{Z}_L^{\dagger}\,, \\
\widehat{Z}_L\widehat{Z}_1^{\dagger} &\mapsto \widehat{Z}_L\widehat{X}_1.
\end{aligned}
\tag{C.7}
$$

We now conjugate by the operator $\widehat{W}$, which acts non-trivially on $\widehat{X}_1$ and $\widehat{X}_L$, and trivially on all other $\widehat{X}_j$. Its action on $\widehat{X}_1$ and $\widehat{X}_L$ delivers

$$
\begin{aligned}
\widehat{X}_1 &\mapsto \widehat{Z}_L^{\dagger}\widehat{X}_1\,, \\
\widehat{X}_L &\mapsto \widehat{Z}_L\widehat{X}_L\,\widehat{Z}_L\widehat{Z}_1^{\dagger}
\end{aligned}
\tag{C.8}
$$

Given this, we find that the three operators on the right hand side of Eq. (C.7) becomes

$$
\begin{aligned}
\widehat{X}_1\widehat{X}_2\ldots\widehat{X}_L &\mapsto \widehat{Z}_L^{\dagger}\widehat{X}_1\widehat{X}_2\ldots\widehat{X}_{L-1} = \left(\prod_{j=1}^{L}\widehat{X}_j\right)\widehat{Z}_L\widehat{Z}_1^{\dagger}\,, \\
\widehat{X}_1^{\dagger}\widehat{X}_2^{\dagger}\ldots\widehat{X}_{L-1}^{\dagger}\widehat{Z}_L^{\dagger} &\mapsto \widehat{Z}_L\widehat{X}_1^{\dagger}\widehat{X}_2^{\dagger}\ldots\widehat{X}_{L-1}^{\dagger}\widehat{Z}_L^{\dagger} = \left(\prod_{j=1}^{L}\widehat{X}_j^{\dagger}\right)\widehat{X}_L\,, \\
\widehat{Z}_L\widehat{X}_1 &\mapsto \widehat{X}_1.
\end{aligned}
\tag{C.9}
$$

In summary, up to the projector $\widehat{P}^{U_r=1}$, conjugation by the unitary operators produce

$$
\begin{aligned}
\widehat{X}_1 &\mapsto \widehat{Z}_1\widehat{Z}_2^{\dagger}\,, & \widehat{Z}_1\widehat{Z}_2^{\dagger} &\mapsto \widehat{X}_2\,, \\
&\;\;\vdots & &\;\;\vdots \\
\widehat{X}_{L-2} &\mapsto \widehat{Z}_{L-2}\widehat{Z}_{L-1}^{\dagger}\,, & \widehat{Z}_{L-2}\widehat{Z}_{L-1}^{\dagger} &\mapsto \widehat{X}_{L-1}\,, \\
\widehat{X}_{L-1} &\mapsto \widehat{Z}_{L-1}\widehat{Z}_L^{\dagger}\,, & \widehat{Z}_{L-1}\widehat{Z}_L^{\dagger} &\mapsto \left(\prod_{j=1}^{L}\widehat{X}_j^{\dagger}\right)\widehat{X}_L\,, \\
\widehat{X}_L &\mapsto \left(\prod_{j=1}^{L}\widehat{X}_j\right)\widehat{Z}_L\widehat{Z}_1^{\dagger}\,, & \widehat{Z}_L\widehat{Z}_1^{\dagger} &\mapsto \widehat{X}_1.
\end{aligned}
\tag{C.10}
$$

Our sequential circuit achieves what one expects from the Kramers-Wannier transformation on the qutrits if if we restrict ourselves to the $\mathbb{Z}_3$ symmetric sector, in which $\prod_{j=1}^{L}\widehat{X}_j = 1$. This achieved by the inclusion of the projector $\widehat{P}^{U_r=1}$ in operator (C.1). Notice that this transformation is not yet a symmetry of the Hamiltonian (2.3) when

$J_1 = J_2$ and $J_5 = J_6 \neq 0$. This is because so far all the operators we considered act on the qutrits, which leads to a relative half-translation of $\mathbb{Z}_3$ and $\widehat{Z}_2$. We correct for this by the final unitary operator $\hat{t}_{\mathbb{Z}_2}$ which implements the transformation

$$\hat{t}_{\mathbb{Z}_2} \begin{pmatrix} \hat{\tau}_i^x & \hat{\tau}_i^z & \hat{\sigma}_i^x & \hat{\sigma}_i^z \end{pmatrix} \hat{t}_{\mathbb{Z}_2}^\dagger = \begin{pmatrix} \hat{\sigma}_{i+1}^x & \hat{\sigma}_{i+1}^z & \hat{\tau}_i^x & \hat{\tau}_i^z \end{pmatrix}. \tag{C.11}$$

This completes our proof that the operator (C.1) is indeed the correct Kramers-Wannier duality operator. This operator, as written in Eq. (C.1), represents a sequential quantum circuit of depth $4L - 2$.[42] We note that this can be straightforwardly generalized to any finite Abelian group. Our sequential circuit is closely related to one that is given in App. A of Ref. [87] for general finite groups. One important difference is that our circuit involves gates (operators) that are not all $\mathbb{Z}_3$-symmetric, while the full circuit is so.

**Algebra of $\widehat{D}_{\mathrm{KW}}$ and other symmetry operators:**

Let us note a few more algebraic properties of the duality operator above. First, we find that the Hermitian conjugate gives

$$\widehat{D}_{\mathrm{KW}}^\dagger = \left( \prod_{j=L-1}^{1} \widehat{\mathrm{CZ}}_{j+1,j} \, \widehat{\mathfrak{H}}_j \right) \widehat{W}^\dagger \, \hat{t}_{\mathbb{Z}_2}^\dagger \, \widehat{P}^{U_r=1}$$

$$= \widehat{P}^{U_r=1} \left( \prod_{j=L-1}^{1} \widehat{\mathrm{CZ}}_{j+1,j} \, \widehat{\mathfrak{H}}_j \right) \widehat{W}^\dagger \, \hat{t}_{\mathbb{Z}_2}^\dagger = \widehat{P}^{U_r=1} \left( \hat{t}_{\mathbb{Z}_2} \, \widehat{W} \prod_{j=1}^{L-1} \widehat{\mathfrak{H}}_j^\dagger \, \widehat{\mathrm{CZ}}_{j+1,j}^\dagger \right)^\dagger, \tag{C.12}$$

which is also a sequential circuit. The $\mathbb{Z}_3$ symmetric local operators are mapped by the unitary part of $\widehat{D}_{\mathrm{KW}}^\dagger$ in exactly the inverse manner as by that of $\widehat{D}_{\mathrm{KW}}$, i.e.,

$$\begin{aligned}
\widehat{Z}_1 \widehat{Z}_2^\dagger &\mapsto \widehat{X}_1, & \widehat{X}_2 &\mapsto \widehat{Z}_1 \widehat{Z}_2^\dagger, \\
&\vdots & &\vdots \\
\widehat{Z}_{L-2} \widehat{Z}_{L-1}^\dagger &\mapsto \widehat{X}_{L-2}, & \widehat{X}_{L-1} &\mapsto \widehat{Z}_{L-2} \widehat{Z}_{L-1}^\dagger, \\
\widehat{Z}_{L-1} \widehat{Z}_L^\dagger &\mapsto \widehat{X}_{L-1}, & \widehat{X}_L &\mapsto \left( \prod_{j=1}^{L} \widehat{X}_j^\dagger \right) \widehat{Z}_{L-1} \widehat{Z}_L^\dagger, \\
\widehat{Z}_L \widehat{Z}_1^\dagger &\mapsto \left( \prod_{j=1}^{L} \widehat{X}_j \right) \widehat{X}_L, & \widehat{X}_1 &\mapsto \widehat{Z}_L \widehat{Z}_1^\dagger.
\end{aligned} \tag{C.13}$$

Now applying the projector $\widehat{P}^{U_r=1}$ and the half-translation operator $\hat{t}_{\mathbb{Z}_2}$ simply produces the transformation

$$\begin{aligned}
\widehat{D}_{\mathrm{KW}}^\dagger \widehat{X}_j &= \widehat{Z}_{j-1} \widehat{Z}_j^\dagger \widehat{D}_{\mathrm{KW}}^\dagger, \quad \widehat{D}_{\mathrm{KW}}^\dagger \widehat{Z}_j \widehat{Z}_{j+1}^\dagger = \widehat{X}_j \widehat{D}_{\mathrm{KW}}^\dagger \\
\widehat{D}_{\mathrm{KW}}^\dagger \begin{pmatrix} \hat{\tau}_i^x & \hat{\tau}_i^z & \hat{\sigma}_i^x & \hat{\sigma}_i^z \end{pmatrix} &= \begin{pmatrix} \hat{\sigma}_i^x & \hat{\sigma}_i^z & \hat{\tau}_{i-1}^x & \hat{\tau}_{i-1}^z \end{pmatrix} \widehat{D}_{\mathrm{KW}}^\dagger,
\end{aligned} \tag{C.14}$$

from which we observe the relation

$$\widehat{D}_{\mathrm{KW}}^\dagger = \widehat{T}^\dagger \widehat{D}_{\mathrm{KW}}. \tag{C.15}$$

---

[42] We note that one can also apply each unitary operator in $\hat{t}_{\mathbb{Z}_2}$ after applying one cycle of controlled-Z and Hadamard operators rendering the sequential circuit of depth $2L - 1$.

Here, the operator $\widehat{T}$ implements the single lattice site translation of both qubits and qutrits, *i.e.*,

$$\widehat{T}\left(\widehat{X}_i \quad \widehat{Z}_i \quad \hat{\tau}_i^x \quad \hat{\tau}_i^z \quad \hat{\sigma}_i^x \quad \hat{\sigma}_i^z\right)\widehat{T}^\dagger = \left(\widehat{X}_{i+1} \quad \widehat{Z}_{i+1} \quad \hat{\tau}_{i+1}^x \quad \hat{\tau}_{i+1}^z \quad \hat{\sigma}_{i+1}^x \quad \hat{\sigma}_{i+1}^z\right). \quad \text{(C.16)}$$

Combining the above results, we obtain the fusion rules

$$\widehat{D}_{\mathrm{KW}}^\dagger \widehat{D}_{\mathrm{KW}} = \widehat{P}^{U_r=1}, \quad \text{(C.17a)}$$

and

$$\widehat{D}_{\mathrm{KW}} \prod_j \widehat{X}_j = \widehat{D}_{\mathrm{KW}} = \left(\prod_j \widehat{X}_j\right) \widehat{D}_{\mathrm{KW}}. \quad \text{(C.17b)}$$

## C.2  Rep($S_3$) self-duality

We found that our $S_3$ symmetric Hamiltonian (2.3a) has a self-duality symmetry in some sub-manifold in the parameter space. We discussed a sequential circuit that performs this transformation above. Recall that gauging a $\mathbb{Z}_2$ subgroup of $S_3$ led us to the Hamiltonian (3.13) with Rep($S_3$) symmetry. We now want to know what symmetry, if any, the above self-duality symmetry gets mapped to. To that end, we would like to follow how each operator in $\widehat{D}_{\mathrm{KW}}$ (recall Eq. (C.1)) change under the gauging map.

(i) We note that acting by $\mathbb{Z}_2$ on half of the system, *i.e.*, on all degrees of freedom to the right of particular site, say $i$, will leave every CZ operator except $\widehat{\mathrm{CZ}}_{i+1,i}$ unaffected. This particular operator gets transformed as

$$\widehat{\mathrm{CZ}}_{i+1,i} = \sum_{\alpha=0}^2 \widehat{Z}_i^\alpha \widehat{P}_{Z_{i+1}=\omega^\alpha} \mapsto \sum_{\alpha=0}^2 \widehat{Z}_i^\alpha \widehat{P}_{Z_{i+1}=\omega^{-\alpha}}$$
$$= \sum_{\alpha=0}^2 \widehat{Z}_i^{-\alpha} \widehat{P}_{Z_{i+1}=\omega^\alpha} = \widehat{\mathrm{CZ}}_{i+1,i}^\dagger. \quad \text{(C.18)}$$

Therefore, if we consider arbitrary $\mathbb{Z}_2$ gauge field configurations, we obtain the minimally coupled CZ operators as

$$\widehat{\mathrm{CZ}}_{i+1,i} \mapsto \widehat{\mathrm{CZ}}_{i+1,i}^{\hat{\mu}_{i+1}^x} \equiv \sum_{\alpha=0}^2 \widehat{Z}_i^{\alpha \hat{\mu}_{i+1}^x} \widehat{P}_{Z_{i+1}=\omega^\alpha}, \quad \text{(C.19)}$$

where we shifted the subscript of the link degrees of freedom by $1/2$, as done in the main text. This minimally coupled operator is the image of $\widehat{\mathrm{CZ}}_{i+1,i}$ under duality map as it is unchanged after the unitary transformation (3.4) and the projection in Eq. (3.7a).

(ii) The Hadamard operator commutes with the charge conjugation operator

$$\widehat{C}_i \widehat{\mathfrak{H}}_i \widehat{C}_i^\dagger = \frac{1}{\sqrt{3}} \sum_{\alpha,\beta=0}^2 \omega^{\alpha\beta} \widehat{X}_i^{\beta-\alpha} \widehat{P}_i^{Z=\omega^{-\beta}}$$
$$= \frac{1}{\sqrt{3}} \sum_{\alpha,\beta=0}^2 \omega^{\alpha\beta} \widehat{X}_i^{\alpha-\beta} \widehat{P}_i^{Z=\omega^\beta} = \widehat{\mathfrak{H}}_i. \quad \text{(C.20)}$$

Hence, the Hadamard operators in $\widehat{D}_{\mathrm{KW}}$ are gauge invariant. Similarly, the Hadamard operator commutes with the unitary transformation (3.4) and is mapped to itself under $\mathbb{Z}_2$ gauging.

(iii) As it was the case for controlled-Z operator $\widehat{CZ}_{i+1,i}$, the unitary $\widehat{W}$ is not gauge invariant. We find its minimally coupled version

$$\widehat{W} = \sum_{\alpha=0}^{2} \widehat{Z}_L^\alpha \, \widehat{P}^{Z_1^\dagger Z_L = \omega^\alpha} \mapsto \widehat{W}_{\mathrm{mc}} = \sum_{\alpha=0}^{2} \widehat{Z}_L^\alpha \, \widehat{P}^{Z_1^{-\mu_1^x} Z_L = \omega^\alpha}, \qquad (\mathrm{C.21})$$

which is the image of the operator $\widehat{W}$ under $\mathbb{Z}_2$ gauging. Note that similar to the discussion of the minimally coupled CZ operators above, $\widehat{W}_{\mathrm{mc}}$ is unchanged by the unitary transformation (3.4) and the projection in Eq. (3.7a).

(iv) The projector $\widehat{P}^{U_r=1}$ in the definition of $\widehat{D}_{\mathrm{KW}}$ (C.1), under minimal coupling takes the following form: [43]

$$\widehat{P}^{U_r=1} = \frac{1}{3} \sum_{\alpha=0}^{2} \prod_{j=1}^{L} \widehat{X}_j^\alpha \mapsto \widehat{P}_{\mathrm{mc}} := \frac{1}{3} \sum_{\alpha=0}^{2} \prod_{j=1}^{L} \widehat{X}_j^{\alpha \prod_{k=1}^{j-1} \hat{\mu}_{k+1}^x}$$

The unitary transformation, (3.4) leaves the operator $\widehat{P}_{\mathrm{mc}}$ unchanged,

$$\widehat{P}_{\mathrm{mc}} \mapsto \frac{1}{3} \sum_{\alpha=0}^{2} \prod_{j=1}^{L} \widehat{X}_j^{\alpha \hat{\sigma}_1^z \prod_{k=1}^{j-1} \hat{\mu}_{k+1}^x}$$

$$= \frac{1}{3} \sum_{\alpha=0}^{2} \left( \frac{\hat{1} + \hat{\sigma}_1^z}{2} \right) \prod_{j=1}^{L} \widehat{X}_j^{\alpha \prod_{k=1}^{j-1} \hat{\mu}_{k+1}^x} + \frac{1}{3} \sum_{\alpha=0}^{2} \left( \frac{\hat{1} - \hat{\sigma}_1^z}{2} \right) \prod_{j=1}^{L} \widehat{X}_j^{-\alpha \prod_{k=1}^{j-1} \hat{\mu}_{k+1}^x}$$

$$= \frac{1}{3} \sum_{\alpha=0}^{2} \left( \frac{\hat{1} + \hat{\sigma}_1^z}{2} \right) \prod_{j=1}^{L} \widehat{X}_j^{\alpha \prod_{k=1}^{j-1} \hat{\mu}_{k+1}^x} + \frac{1}{3} \sum_{\alpha'=0}^{2} \left( \frac{\hat{1} - \hat{\sigma}_1^z}{2} \right) \prod_{j=1}^{L} \widehat{X}_j^{\alpha' \prod_{k=1}^{j-1} \hat{\mu}_{k+1}^x}$$

$$= \frac{1}{3} \sum_{\alpha=0}^{2} \prod_{j=1}^{L} \widehat{X}_j^{\alpha \prod_{k=1}^{j-1} \hat{\mu}_{k+1}^x} = \widehat{P}_{\mathrm{mc}}.$$

Here we used the periodic boundary condition on the $\hat{\sigma}^z$ degrees of freedom, *i.e.*, $\hat{\sigma}_0^z \equiv \hat{\sigma}_L^z$.

(v) Finally, the qubit "half-translation" operator $\hat{t}_{\mathbb{Z}_2}$ after the whole gauging procedure takes the following form:

$$\hat{t}_{\mathsf{Rep}(S_3)} := \widehat{A}_1 \prod_{i=1}^{L-1} \widehat{B}_i \, \widehat{A}_{i+1},$$

$$\widehat{A}_i := \frac{1 + \hat{\tau}_i^z}{2} + \frac{1 - \hat{\tau}_i^z}{2} \, \widehat{C}_i \, \hat{\mu}_i^z \, \hat{\mu}_{i+1}^z, \qquad (\mathrm{C.22})$$

$$\widehat{B}_i := \frac{1 + \hat{\tau}_i^z \, \hat{\mu}_{i+1}^x}{2} + \frac{1 - \hat{\tau}_i^z \, \hat{\mu}_{i+1}^x}{2} \, \hat{\tau}_i^x \, \hat{\tau}_{i+1}^x \, \widehat{C}_{i+1} \, \hat{\mu}_{i+1}^z \, \hat{\mu}_{i+2}^z.$$

The action of this operator on the generators of the $\mathsf{Rep}(S_3)$-symmetric bond algebra (3.5) can be deduced from the action of $\hat{t}_{\mathbb{Z}_2}$ on the generators of $S_3$-symmetric bond

---

[43]Note that the choice to start the string of $\mu_{k+1/2}^z$'s at $k = 1$ is completely arbitrary and unphysical. We could just as well put this "branch cut" anywhere else in the periodic chain.

algebra (2.6). Namely, we find the transformation rules

$$
\hat{\mathfrak{t}}_{\mathsf{Rep}(S_3)}:
\begin{pmatrix}
\hat{\tau}_i^z \\
\hat{\tau}_i^z\,\hat{\mu}_{i+1}^x \\
\hat{\mu}_i^z\,\hat{\tau}_i^x\,\widehat{C}_i\,\hat{\mu}_{i+1}^z \\
\hat{\tau}_i^x \\
\widehat{X}_i + \widehat{X}_i^\dagger \\
\widehat{Z}_i^{\hat{\mu}_{i+1}^x}\,\widehat{Z}_{i+1}^\dagger + \text{H.c.} \\
\widehat{X}_i - \widehat{X}_i^\dagger \\
\hat{\tau}_i^z\,\hat{\mu}_{i+1}^x\left(\widehat{Z}_i^{\hat{\mu}_{i+1}^x}\,\widehat{Z}_{i+1}^\dagger - \text{H.c.}\right)
\end{pmatrix}
\longmapsto
\begin{pmatrix}
\hat{\tau}_i^z\,\hat{\mu}_{i+1}^x \\
\hat{\tau}_{i+1}^z \\
\hat{\tau}_i^x \\
\hat{\mu}_{i+1}^z\,\hat{\tau}_{i+1}^x\,\widehat{C}_{i+1}\,\hat{\mu}_{i+2}^z \\
\widehat{X}_i + \widehat{X}_i^\dagger \\
\widehat{Z}_i^{\hat{\mu}_{i+1}^x}\,\widehat{Z}_{i+1}^\dagger + \text{H.c.} \\
\hat{\tau}_i^z\left(\widehat{X}_i - \widehat{X}_i^\dagger\right) \\
\widehat{Z}_i^{\hat{\mu}_{i+1}^x}\,\widehat{Z}_{i+1}^\dagger - \text{H.c.}
\end{pmatrix},
\qquad (\text{C.23})
$$

on the generators of $\mathsf{Rep}(S_3)$-symmetric bond algebra (3.5).

Since we gauged the original $\mathbb{Z}_2$ symmetry with periodic boundary conditions, we will end up in the symmetric sector of the dual $\mathbb{Z}_2$ symmetry, generated by $\prod_j \hat{\mu}_j^x$. Therefore, we must include a projector to this symmetric sector in the gauged duality operator.[44] So the full gauged operator has the form [45]

$$
\widehat{D}_{\mathsf{Rep}(S_3)} := \hat{\mathfrak{t}}_{\mathsf{Rep}(S_3)}\,\frac{\hat{1} + \prod_j \hat{\mu}_j^x}{2}\,\widehat{P}_{\mathrm{mc}}\,\widehat{D}_0\,, \quad \text{where}
$$

$$
\widehat{D}_0 := \left(\sum_{\alpha=0}^{2} \widehat{Z}_L^\alpha\,\widehat{P}_{Z_1^{\mu_1^x} Z_L = \omega^\alpha}\right)\left(\prod_{j=1}^{L-1} \widehat{\mathfrak{H}}_j^\dagger\,\widehat{\mathrm{CZ}}_{j+1,j}^{-\hat{\mu}_{j+1}^x}\right)
$$

We can further simplify $\widehat{D}_{\mathsf{Rep}(S_3)}$ since

$$
\frac{\hat{1} + \prod_j \hat{\mu}_j^x}{2}\,\widehat{P}_{\mathrm{mc}} = \frac{\hat{1} + \prod_j \hat{\mu}_j^x}{2}\,\frac{\hat{1} + \widehat{W}_{\mathbf{2}}}{3} = \frac{1}{6}\left(\hat{1} + \widehat{W}_{\mathbf{1}'}\right)\left(\hat{1} + \widehat{W}_{\mathbf{2}}\right) = \frac{1}{6}\left(\widehat{W}_{\mathbf{1}} + \widehat{W}_{\mathbf{1}'} + 2\widehat{W}_{\mathbf{2}}\right)
$$

Hence, we have $\widehat{D}_{\mathsf{Rep}(S_3)} = \frac{1}{6}\left(\widehat{W}_{\mathbf{1}} + \widehat{W}_{\mathbf{1}'} + 2\widehat{W}_{\mathbf{2}}\right)\widehat{D}_0$, which we denote in short as $\widehat{D}_{\mathsf{Rep}(S_3)} = \widehat{P}_{\mathrm{reg}}\widehat{D}_0$, where $\widehat{P}_{\mathrm{reg}} := \frac{1}{6}\widehat{W}_{\mathrm{reg}}$ is the projector to the $\mathsf{Rep}(S_3)$-symmetric sector, as discussed in the main text.

Under the action of $\widehat{D}_0$ by conjugation, going through calculations similar to those in Sec. C.1, we find the following operator maps:

$$
\begin{array}{llll}
\widehat{X}_1 & \mapsto \widehat{Z}_1\widehat{Z}_2^{-\hat{\mu}_2^x} & \widehat{Z}_1^{-\hat{\mu}_2^x}\widehat{Z}_2 & \mapsto \widehat{X}_2^\dagger \\
& \vdots & & \vdots \\
\widehat{X}_{L-2} & \mapsto \widehat{Z}_{L-2}\widehat{Z}_{L-1}^{-\hat{\mu}_{L-1}^x} & \widehat{Z}_{L-2}^{-\hat{\mu}_{L-1}^x}\widehat{Z}_{L-1} & \mapsto \widehat{X}_{L-1}^\dagger \\
\widehat{X}_{L-1} & \mapsto \widehat{Z}_{L-1}\widehat{Z}_L^{-\hat{\mu}_L^x} & \widehat{Z}_L^{-\hat{\mu}_1^x}\widehat{Z}_1 & \mapsto \widehat{X}_1^\dagger
\end{array}
$$

and

$$
\widehat{Z}_{L-1}^{-\hat{\mu}_L^x}\widehat{Z}_L \mapsto \widehat{Z}_L^{-\prod_j \hat{\mu}_j^x}\left(\prod_{j=1}^{L-1} \widehat{X}_j^{\prod_{k=j}^{L-1}\hat{\mu}_{k+1}^x}\right)\widehat{Z}_L
$$

---

[44]In fact, without this additional projector, the gauged duality operator would not actually commute with the Hamiltonian (3.13) on the self-dual manifold of parameters described by $J_1 = J_2, J_5 = J_6$.

[45]Note that $\hat{\mathfrak{t}}_{\mathsf{Rep}(S_3)}$ commutes with the projector $\frac{\hat{1} + \prod_j \hat{\mu}_j^x}{2}$.

$$\widehat{X}_L \mapsto \widehat{Z}_L^{-\,\Pi_j\,\hat{\mu}_j^x}\left(\prod_{j=1}^{L-1}\widehat{X}_j^{\Pi_{k=j}^{L-1}\hat{\mu}_{k+1}^x}\right)\widehat{Z}_L\widehat{X}_L\,\widehat{Z}_L\widehat{Z}_1^{-\hat{\mu}_1^x}$$

In the above, we used $\widehat{W}_{\mathrm{mc}}\widehat{X}_1^\dagger\widehat{W}_{\mathrm{mc}}^\dagger = \widehat{X}_1^\dagger\widehat{Z}_L^{\hat{\mu}_1^x}$ and $\quad \widehat{W}_{\mathrm{mc}}\widehat{X}_L\widehat{W}_{\mathrm{mc}}^\dagger = \widehat{Z}_L\widehat{X}_L\widehat{Z}_L\widehat{Z}_1^{-\hat{\mu}_1^x}$ . The last two transformations can be re-written as

$$\widehat{Z}_{L-1}^{-\hat{\mu}_L^x}\widehat{Z}_L \mapsto \widehat{Z}_L^{-\,\Pi_j\,\hat{\mu}_j^x}\left(\prod_{j=1}^{L-1}\widehat{X}_j^{\Pi_{k=0}^{j-1}\hat{\mu}_{k+1}^x}\right)^{\Pi_j\,\hat{\mu}_j^x}\widehat{Z}_L \tag{C.24}$$

$$\widehat{X}_L \mapsto \widehat{Z}_L^{-\,\Pi_j\,\hat{\mu}_j^x}\left(\prod_{j=1}^{L-1}\widehat{X}_j^{\Pi_{k=0}^{j-1}\hat{\mu}_{k+1}^x}\right)^{\Pi_j\,\hat{\mu}_j^x}\widehat{Z}_L\widehat{X}_L\,\widehat{Z}_L\widehat{Z}_1^{-\hat{\mu}_1^x} \tag{C.25}$$

Next, note that $\widehat{P}_{\mathrm{reg}}$ commutes with the first set of operators above, but for the last two we have

$$\widehat{P}_{\mathrm{reg}}\,\widehat{Z}_L^{-\,\Pi_j\,\hat{\mu}_j^x}\left(\prod_{j=1}^{L-1}\widehat{X}_j^{\Pi_{k=0}^{j-1}\hat{\mu}_{k+1}^x}\right)^{\Pi_j\,\hat{\mu}_j^x}\widehat{Z}_L = \widehat{P}_{\mathrm{reg}}\left(\prod_{j=1}^{L-1}\widehat{X}_j^{\Pi_{k=0}^{j-1}\hat{\mu}_{k+1}^x}\right)$$

$$= \widehat{P}_{\mathrm{reg}}\left(\prod_{j=1}^{L-1}\widehat{X}_j^{\Pi_{k=0}^{j-1}\hat{\mu}_{k+1}^x}\right)\widehat{X}_L\widehat{X}_L^\dagger$$

$$= \frac{1+\Pi_j\,\hat{\mu}_j^x}{2}\frac{1}{3}\sum_{\alpha=0}^{2}\prod_{j=1}^{L}\widehat{X}_j^{\alpha\,\Pi_{\ell=1}^{j-1}\hat{\mu}_{\ell+1}^x}\left(\prod_{j=1}^{L-1}\widehat{X}_j^{\Pi_{k=0}^{j-1}\hat{\mu}_{k+1}^x}\right)\widehat{X}_L^{\Pi_j\,\hat{\mu}_j^x}\widehat{X}_L^\dagger$$

$$= \frac{1+\Pi_j\,\hat{\mu}_{j+1}^x}{2}\frac{1}{3}\sum_{\alpha=0}^{2}\prod_{j=1}^{L}\widehat{X}_j^{(\alpha+\hat{\mu}_1^x)\,\Pi_{\ell=1}^{j-1}\hat{\mu}_{\ell+1}^x}\widehat{X}_L^\dagger = \widehat{X}_L^\dagger\,\widehat{P}_{\mathrm{reg}}$$

and

$$\widehat{P}_{\mathrm{reg}}\,\widehat{Z}_L^{-\,\Pi_j\,\hat{\mu}_j^x}\left(\prod_{j=1}^{L-1}\widehat{X}_j^{\Pi_{k=0}^{j-1}\hat{\mu}_{k+1}^x}\right)^{\Pi_j\,\hat{\mu}_j^x}\widehat{Z}_L\widehat{X}_L\,\widehat{Z}_L\widehat{Z}_1^{-\hat{\mu}_1^x}$$

$$= \widehat{P}_{\mathrm{reg}}\left(\prod_{j=1}^{L-1}\widehat{X}_j^{\Pi_{k=0}^{j-1}\hat{\mu}_{k+1}^x}\right)\widehat{X}_L^{\Pi_j\,\hat{\mu}_j^x}\,\widehat{Z}_L\widehat{Z}_1^{-\hat{\mu}_1^x}$$

$$= \frac{1+\Pi_j\,\hat{\mu}_j^x}{2}\frac{1}{3}\sum_{\alpha=0}^{2}\left(\prod_{j=1}^{L}\widehat{X}_j^{(\alpha+\hat{\mu}_1^x)\,\Pi_{\ell=1}^{j-1}\hat{\mu}_{\ell+1/2}^z}\right)\widehat{Z}_L\widehat{Z}_1^{-\hat{\mu}_1^x} = \widehat{Z}_L\widehat{Z}_1^{-\hat{\mu}_1^x}\,\widehat{P}_{\mathrm{reg}}$$

Finally, we act with the unitary $\hat{\mathsf{t}}_{\mathsf{Rep}(S_3)}$, whose action on the $\mathsf{Rep}(S_3)$-symmetric bond algebra is outlined in Eq. (C.23). In all, we have the following transformations of the operators appearing in the $\mathsf{Rep}(S_3)$-symmetric Hamiltonian (3.13):

(i) For operators $\widehat{X}_j + \mathrm{H.c.}$,

$$\widehat{D}_{\mathsf{Rep}(S_3)}\left(\widehat{X}_j + \mathrm{H.c.}\right) = \hat{\mathsf{t}}_{\mathsf{Rep}(S_3)}\left(\widehat{Z}_j\widehat{Z}_{j+1}^{-\hat{\mu}_{j+1}^x} + \widehat{Z}_j^\dagger\widehat{Z}_{j+1}^{\hat{\mu}_{j+1}^x}\right)\widehat{P}_{reg}\widehat{D}_0$$

$$= \hat{\mathsf{t}}_{\mathsf{Rep}(S_3)}\left(\widehat{Z}_j^{\hat{\mu}_{j+1}^x}\widehat{Z}_{j+1}^\dagger + \mathrm{H.c.}\right)\widehat{P}_{reg}\widehat{D}_0 = \left(\widehat{Z}_j^{\hat{\mu}_{j+1}^x}\widehat{Z}_{j+1}^\dagger + \mathrm{H.c.}\right)\widehat{D}_{\mathsf{Rep}(S_3)}.$$

(ii) For operators $\widehat{Z}_j^{\hat{\mu}_{j+1}^x} \widehat{Z}_{j+1}^\dagger + \text{H.c.}$,

$$\widehat{D}_{\mathsf{Rep}(S_3)} \left( \widehat{Z}_j^{\hat{\mu}_{j+1}^x} \widehat{Z}_{j+1}^\dagger + \text{H.c.} \right) = \hat{\mathfrak{t}}_{\mathsf{Rep}(S_3)} \left( \widehat{X}_{j+1} + \text{H.c.} \right) \widehat{P}_{reg} \widehat{D}_0 = \left( \widehat{X}_{j+1} + \text{H.c.} \right) \widehat{D}_{\mathsf{Rep}(S_3)} \,.$$

(iii) For operators $\widehat{X}_j - \text{H.c.}$,

$$\begin{aligned}
\widehat{D}_{\mathsf{Rep}(S_3)} \left( \widehat{X}_j - \text{H.c.} \right) &= \hat{\mathfrak{t}}_{\mathsf{Rep}(S_3)} \left( \widehat{Z}_j \widehat{Z}_{j+1}^{-\hat{\mu}_{j+1}^x} - \widehat{Z}_j^\dagger \widehat{Z}_{j+1}^{\hat{\mu}_{j+1}^x} \right) \widehat{P}_{reg} \widehat{D}_0 \\
&= \hat{\mathfrak{t}}_{\mathsf{Rep}(S_3)} \hat{\mu}_{j+1}^x \left( \widehat{Z}_j^{\hat{\mu}_{j+1}^x} \widehat{Z}_{j+1}^\dagger - \widehat{Z}_j^{-\hat{\mu}_{j+1}^x} \widehat{Z}_{j+1} \right) \widehat{P}_{reg} \widehat{D}_0 \\
&= \hat{\tau}_j^z \hat{\mu}_{j+1}^x \left( \widehat{Z}_j^{\hat{\mu}_{j+1}^x} \widehat{Z}_{j+1}^\dagger - \text{H.c.} \right) \widehat{D}_{\mathsf{Rep}(S_3)} \,.
\end{aligned}$$

(iv) For operators $\hat{\tau}_i^z \hat{\mu}_{i+1}^x \left( \widehat{Z}_i^{\hat{\mu}_{i+1}^x} \widehat{Z}_{i+1}^\dagger - \text{H.c.} \right)$,

$$\begin{aligned}
\widehat{D}_{\mathsf{Rep}(S_3)} \hat{\tau}_i^z \hat{\mu}_{i+1}^x \left( \widehat{Z}_i^{\hat{\mu}_{i+1}^x} \widehat{Z}_{i+1}^\dagger - \text{H.c.} \right) &= \hat{\mathfrak{t}}_{\mathsf{Rep}(S_3)} \hat{\tau}_i^z \hat{\mu}_{i+1}^x \left( \widehat{X}_{j+1} - \widehat{X}_{j+1}^\dagger \right) \widehat{P}_{reg} \widehat{D}_0 \\
&= \left( \widehat{X}_{j+1} - \widehat{X}_{j+1}^\dagger \right) \widehat{D}_{\mathsf{Rep}(S_3)} \,.
\end{aligned}$$

(v) The $J_3$ and $J_4$ terms in the Hamiltonian (3.13) are left invariant, even though some indices get shuffled.

# D   Details about numerical methods

Let us briefly comment on how the TEFR algorithm works. We Trotter-ize the imaginary time path integral (or, partition function) associated with the Hamiltonian to obtain a rank-4 tensor $T$. Since the lattice models have discrete space but continuous time, we implement a renormalization transformation along the time direction to obtain an "isotropic" partition function tensor $T_{\text{iso}}$. The full partition function can then be expressed as a network of these tensors $T_{\text{iso}}$. Applying the TEFR algorithm [97, 98] entails multiple iterations of singular value decomposition and tensor contractions; finally, we reach a fixed point tensor $T_{\text{iso}}^*$, whose largest eigenvalue has the form

$$\lambda^* = \text{GSD} \cdot \exp\left( -E_0 T + \frac{2\pi v}{24L} c + \mathcal{O}\left( \frac{1}{L^2} \right) \right) \tag{D.1}$$

where $T$ is the total length of the compactified imaginary time direction, GSD is the ground state degeneracy, $E_0$ is the ground state energy, and $c$ is the central charge which is only non-zero when the system is in a gapless phase described by a CFT at low energies and $v$ is the "velocity" of the linear-dispersing mode of this CFT. Since $v$ in general depends on details of the microscopic Hamiltonian, the precise numerical value of the central charge is difficult to extract. However, we should note that the algorithm is quite efficient at distinguishing gapless regions of the phase diagram, which have non-zero $c$, from gapped regions where $c = 0$ (within pre-set limits of precision). By benchmarking various known limits, we also find that relative values of $c$ extracted using this approach are in practice reflective of the true central charges of the corresponding CFTs.

In order to extract numerically precise central charges, we used the density matrix renormalization group (DMRG) algorithm from the iTensor library [99, 100]. Specifically, we used the Calabrese-Cardy formula [129] for the entanglement entropy of 1+1d CFTs,

$$S(\ell) = \frac{c}{6} \log\left( \frac{2L}{\pi a} \sin \frac{\pi \ell}{L} \right) + c_1 \,, \tag{D.2}$$

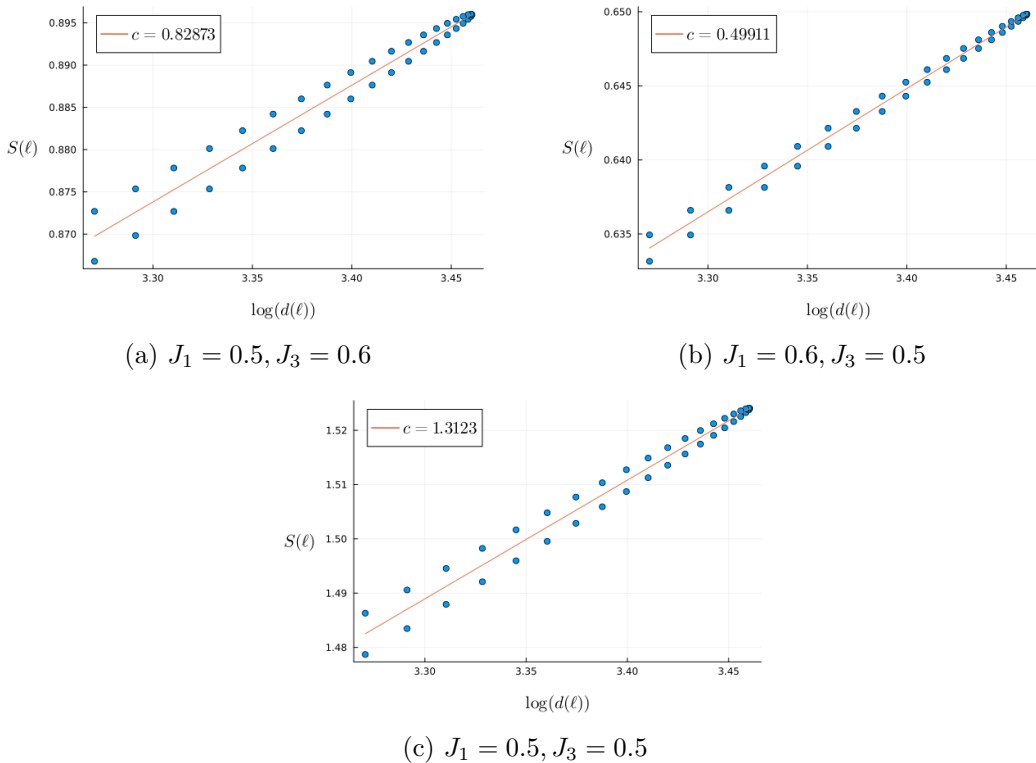

(a) $J_1 = 0.5, J_3 = 0.6$

(b) $J_1 = 0.6, J_3 = 0.5$

(c) $J_1 = 0.5, J_3 = 0.5$

Figure 16: Central charges extracted from fits to the Calabrese-Cardy function, as discussed in the main text. Figure (a) is consistent with Potts criticality, Figure (b) is consistent with Ising criticality, and Figure (c) is consistent with a stacking of these two CFTs. The departures from exact values of $c = 0.8, 0.5, 1.3$, respectively, are due to the effects of finite system size. For these plots, we set the system size as $L = 100$ and performed the central charge fits on the bulk sites by removing 30 sites from each boundary.

where $\ell$ is the size of the subsystem, $L$ is the size of the full system, $S(\ell)$ is the entanglement entropy of the bipartition of the full system into two pieces of size $\ell$ and $L - \ell$, $a$ is the lattice constant of the lattice model described by a CFT at low energies, and $c$ is its central charge. By computing $S(\ell)$ in the ground state for $\ell = 1, \ldots L - 1$ and fitting it to the above functional form, we can quite reliably extract the central charge. We show examples of this method for two points in the phase diagram Fig. 3 of our $S_3$ spin chain (2.21) in Fig. 16, to demonstrate the quality of the fits.

# E    Gapless boundaries of $S_3$ SymTO

## E.1    3-state Potts CFT boundary

The 3-state Potts CFT is a $(6, 5)$ minimal model. This CFT can be realized as a so-called **1**-condensed gapless boundary of the 2+1d $S_3$ topological order. Using the SymTO point of view, this gives us a way to identify operators of the CFT that carry various symmetry charges and symmetry twists. The vacuum sector $Z_1$ contains contributions from local operators that are uncharged under the full $S_3$ symmetry. The **1**$'$ sector contains local operators that carry the sign representation of $S_3$, and hence are charged under the $Z_2$ subgroups of $S_3$ but uncharged under the $Z_3$ subgroup. The **2** sector contains local

operators that carry the 2d irrep of $S_3$. The sectors labeled by $b$ and $c$ contain (uncharged) operators that live on the boundaries of symmetry defects, whereas the sectors labeled by $r_1, r_2, s_1$ contain charged operators of the corresponding symmetry-twisted sectors.

From modular bootstrap calculations, we find the following as the only $\mathbf{1}$-condensed boundary of $\mathcal{D}(S_3)$ constructed out of the conformal characters of the Potts minimal model:

$$
\begin{aligned}
Z_{\mathbf{1}}^{\mathcal{D}(S_3)} &= \chi_{1,0;1,0}^{m6\times\overline{m6}} + \chi_{5,3;5,-3}^{m6\times\overline{m6}} + \chi_{6,\frac{2}{5};6,-\frac{2}{5}}^{m6\times\overline{m6}} + \chi_{10,\frac{7}{5};10,-\frac{7}{5}}^{m6\times\overline{m6}} \\
Z_{\mathbf{1'}}^{\mathcal{D}(S_3)} &= \chi_{1,0;5,-3}^{m6\times\overline{m6}} + \chi_{5,3;1,0}^{m6\times\overline{m6}} + \chi_{6,\frac{2}{5};10,-\frac{7}{5}}^{m6\times\overline{m6}} + \chi_{10,\frac{7}{5};6,-\frac{2}{5}}^{m6\times\overline{m6}} \\
Z_{\mathbf{2}}^{\mathcal{D}(S_3)} &= \chi_{3,\frac{2}{3};3,-\frac{2}{3}}^{m6\times\overline{m6}} + \chi_{8,\frac{1}{15};8,-\frac{1}{15}}^{m6\times\overline{m6}} \\
Z_{r}^{\mathcal{D}(S_3)} &= \chi_{3,\frac{2}{3};3,-\frac{2}{3}}^{m6\times\overline{m6}} + \chi_{8,\frac{1}{15};8,-\frac{1}{15}}^{m6\times\overline{m6}} \\
Z_{r_1}^{\mathcal{D}(S_3)} &= \chi_{1,0;3,-\frac{2}{3}}^{m6\times\overline{m6}} + \chi_{5,3;3,-\frac{2}{3}}^{m6\times\overline{m6}} + \chi_{6,\frac{2}{5};8,-\frac{1}{15}}^{m6\times\overline{m6}} + \chi_{10,\frac{7}{5};8,-\frac{1}{15}}^{m6\times\overline{m6}} \\
Z_{r_2}^{\mathcal{D}(S_3)} &= \chi_{3,\frac{2}{3};1,0}^{m6\times\overline{m6}} + \chi_{3,\frac{2}{3};5,-3}^{m6\times\overline{m6}} + \chi_{8,\frac{1}{15};6,-\frac{2}{5}}^{m6\times\overline{m6}} + \chi_{8,\frac{1}{15};10,-\frac{7}{5}}^{m6\times\overline{m6}} \\
Z_{s}^{\mathcal{D}(S_3)} &= \chi_{2,\frac{1}{8};2,-\frac{1}{8}}^{m6\times\overline{m6}} + \chi_{4,\frac{13}{8};4,-\frac{13}{8}}^{m6\times\overline{m6}} + \chi_{7,\frac{1}{40};7,-\frac{1}{40}}^{m6\times\overline{m6}} + \chi_{9,\frac{21}{40};9,-\frac{21}{40}}^{m6\times\overline{m6}} \\
Z_{s_1}^{\mathcal{D}(S_3)} &= \chi_{2,\frac{1}{8};4,-\frac{13}{8}}^{m6\times\overline{m6}} + \chi_{4,\frac{13}{8};2,-\frac{1}{8}}^{m6\times\overline{m6}} + \chi_{7,\frac{1}{40};9,-\frac{21}{40}}^{m6\times\overline{m6}} + \chi_{9,\frac{21}{40};7,-\frac{1}{40}}^{m6\times\overline{m6}}
\end{aligned}
\tag{E.1}
$$

The various terms in each component of the partition function are conformal characters of the $(6,5)$ minimal model. The expression $\chi_{a,h_a;b,-h_b}^{m6\times\overline{m6}}$ is a short-hand notation for the product of the left moving chiral conformal character associated with the primary operator labeled $a$ (set by an arbitrary indexing convention) with conformal weight $(h_a, 0)$, and the right moving chiral conformal character associated with the primary operator labeled $b$ with conformal weight $(0, h_b)$. The superscript $m6 \times \overline{m6}$ indicates that both the left and right moving chiral conformal characters are picked from the same $(6,5)$ minimal model.

Note that in the above multi-component "SymTO-resolved" partition function, the $\mathbf{1'}$ sector contains the primary operators with odd charge under the $\mathbb{Z}_2$ subgroups of $S_3$. All of these operators have non-zero conformal spin since $h \neq \bar{h}$ for all of them. However, we can construct descendents with zero conformal spin. All such descendents have scaling dimension greater than 2 and hence must be irrelevant perturbations of the CFT.

### E.2   3-state Potts ⊠ Ising CFT boundary

Similar to the previous subsection, we can perform a modular bootstrap numerical calculation to identify gapless boundaries of the 2+1d $S_3$ topological order, considered as a SymTO. The following multi-component partition function shows the various symmetry charge and symmetry twist sectors for the $S_3$ symmetry:

$$
\begin{aligned}
Z_{\mathbf{1}}^{\mathcal{D}(S_3)} ={}& \chi_{1,0;1,0;1,0;1,0}^{m4\times m6\times\overline{m4}\times\overline{m6}} + \chi_{1,0;5,3;1,0;5,-3}^{m4\times m6\times\overline{m4}\times\overline{m6}} + \chi_{1,0;6,\frac{2}{5};1,0;6,-\frac{2}{5}}^{m4\times m6\times\overline{m4}\times\overline{m6}} + \chi_{1,0;10,\frac{7}{5};1,0;10,-\frac{7}{5}}^{m4\times m6\times\overline{m4}\times\overline{m6}} \\
&+ \chi_{2,\frac{1}{16};1,0;2,-\frac{1}{16};5,-3}^{m4\times m6\times\overline{m4}\times\overline{m6}} + \chi_{2,\frac{1}{16};5,3;2,-\frac{1}{16};1,0}^{m4\times m6\times\overline{m4}\times\overline{m6}} + \chi_{2,\frac{1}{16};6,\frac{2}{5};2,-\frac{1}{16};10,-\frac{7}{5}}^{m4\times m6\times\overline{m4}\times\overline{m6}} + \chi_{2,\frac{1}{16};10,\frac{7}{5};2,-\frac{1}{16};6,-\frac{2}{5}}^{m4\times m6\times\overline{m4}\times\overline{m6}} \\
&+ \chi_{3,\frac{1}{2};1,0;3,-\frac{1}{2};1,0}^{m4\times m6\times\overline{m4}\times\overline{m6}} + \chi_{3,\frac{1}{2};5,3;3,-\frac{1}{2};5,-3}^{m4\times m6\times\overline{m4}\times\overline{m6}} + \chi_{3,\frac{1}{2};6,\frac{2}{5};3,-\frac{1}{2};6,-\frac{2}{5}}^{m4\times m6\times\overline{m4}\times\overline{m6}} + \chi_{3,\frac{1}{2};10,\frac{7}{5};3,-\frac{1}{2};10,-\frac{7}{5}}^{m4\times m6\times\overline{m4}\times\overline{m6}}
\end{aligned}
$$

$$
\begin{aligned}
Z_{\mathbf{1'}}^{\mathcal{D}(S_3)} ={}& \chi_{1,0;1,0;1,0;5,-3}^{m4\times m6\times\overline{m4}\times\overline{m6}} + \chi_{1,0;5,3;1,0;1,0}^{m4\times m6\times\overline{m4}\times\overline{m6}} + \chi_{1,0;6,\frac{2}{5};1,0;10,-\frac{7}{5}}^{m4\times m6\times\overline{m4}\times\overline{m6}} + \chi_{1,0;10,\frac{7}{5};1,0;6,-\frac{2}{5}}^{m4\times m6\times\overline{m4}\times\overline{m6}} \\
&+ \chi_{2,\frac{1}{16};1,0;2,-\frac{1}{16};1,0}^{m4\times m6\times\overline{m4}\times\overline{m6}} + \chi_{2,\frac{1}{16};5,3;2,-\frac{1}{16};5,-3}^{m4\times m6\times\overline{m4}\times\overline{m6}} + \chi_{2,\frac{1}{16};6,\frac{2}{5};2,-\frac{1}{16};6,-\frac{2}{5}}^{m4\times m6\times\overline{m4}\times\overline{m6}} + \chi_{2,\frac{1}{16};10,\frac{7}{5};2,-\frac{1}{16};10,-\frac{7}{5}}^{m4\times m6\times\overline{m4}\times\overline{m6}} \\
&+ \chi_{3,\frac{1}{2};1,0;3,-\frac{1}{2};5,-3}^{m4\times m6\times\overline{m4}\times\overline{m6}} + \chi_{3,\frac{1}{2};5,3;3,-\frac{1}{2};1,0}^{m4\times m6\times\overline{m4}\times\overline{m6}} + \chi_{3,\frac{1}{2};6,\frac{2}{5};3,-\frac{1}{2};10,-\frac{7}{5}}^{m4\times m6\times\overline{m4}\times\overline{m6}} + \chi_{3,\frac{1}{2};10,\frac{7}{5};3,-\frac{1}{2};6,-\frac{2}{5}}^{m4\times m6\times\overline{m4}\times\overline{m6}}
\end{aligned}
$$

$$Z_{\mathbf{2}}^{\mathcal{D}(S_3)} = \chi^{m4\times m6\times\overline{m4}\times\overline{m6}}_{1,0,3,\frac{2}{3};1,0,3,-\frac{2}{3}} + \chi^{m4\times m6\times\overline{m4}\times\overline{m6}}_{1,0,8,\frac{1}{15};1,0,8,-\frac{1}{15}} + \chi^{m4\times m6\times\overline{m4}\times\overline{m6}}_{2,\frac{1}{16};3,\frac{2}{3};2,-\frac{1}{16};3,-\frac{2}{3}} + \chi^{m4\times m6\times\overline{m4}\times\overline{m6}}_{2,\frac{1}{16};8,\frac{1}{15};2,-\frac{1}{16};8,-\frac{1}{15}}$$
$$+ \chi^{m4\times m6\times\overline{m4}\times\overline{m6}}_{3,\frac{1}{2};3,\frac{2}{3};3,-\frac{1}{2};3,-\frac{2}{3}} + \chi^{m4\times m6\times\overline{m4}\times\overline{m6}}_{3,\frac{1}{2};8,\frac{1}{15};3,-\frac{1}{2};8,-\frac{1}{15}}$$

$$Z_{r}^{\mathcal{D}(S_3)} = \chi^{m4\times m6\times\overline{m4}\times\overline{m6}}_{1,0,3,\frac{2}{3};1,0,3,-\frac{2}{3}} + \chi^{m4\times m6\times\overline{m4}\times\overline{m6}}_{1,0,8,\frac{1}{15};1,0,8,-\frac{1}{15}} + \chi^{m4\times m6\times\overline{m4}\times\overline{m6}}_{2,\frac{1}{16};3,\frac{2}{3};2,-\frac{1}{16};3,-\frac{2}{3}} + \chi^{m4\times m6\times\overline{m4}\times\overline{m6}}_{2,\frac{1}{16};8,\frac{1}{15};2,-\frac{1}{16};8,-\frac{1}{15}}$$
$$+ \chi^{m4\times m6\times\overline{m4}\times\overline{m6}}_{3,\frac{1}{2};3,\frac{2}{3};3,-\frac{1}{2};3,-\frac{2}{3}} + \chi^{m4\times m6\times\overline{m4}\times\overline{m6}}_{3,\frac{1}{2};8,\frac{1}{15};3,-\frac{1}{2};8,-\frac{1}{15}}$$

$$Z_{r_1}^{\mathcal{D}(S_3)} = \chi^{m4\times m6\times\overline{m4}\times\overline{m6}}_{1,0;1,0;1,0;3,-\frac{2}{3}} + \chi^{m4\times m6\times\overline{m4}\times\overline{m6}}_{1,0;5,3;1,0;3,-\frac{2}{3}} + \chi^{m4\times m6\times\overline{m4}\times\overline{m6}}_{1,0;6,\frac{2}{5};1,0;8,-\frac{1}{15}} + \chi^{m4\times m6\times\overline{m4}\times\overline{m6}}_{1,0;10,\frac{7}{5};1,0;8,-\frac{1}{15}}$$
$$+ \chi^{m4\times m6\times\overline{m4}\times\overline{m6}}_{2,\frac{1}{16};1,0;2,-\frac{1}{16};3,-\frac{2}{3}} + \chi^{m4\times m6\times\overline{m4}\times\overline{m6}}_{2,\frac{1}{16};5,3;2,-\frac{1}{16};3,-\frac{2}{3}} + \chi^{m4\times m6\times\overline{m4}\times\overline{m6}}_{2,\frac{1}{16};6,\frac{2}{5};2,-\frac{1}{16};8,-\frac{1}{15}} + \chi^{m4\times m6\times\overline{m4}\times\overline{m6}}_{2,\frac{1}{16};10,\frac{7}{5};2,-\frac{1}{16};8,-\frac{1}{15}}$$
$$+ \chi^{m4\times m6\times\overline{m4}\times\overline{m6}}_{3,\frac{1}{2};1,0;3,-\frac{1}{2};3,-\frac{2}{3}} + \chi^{m4\times m6\times\overline{m4}\times\overline{m6}}_{3,\frac{1}{2};5,3;3,-\frac{1}{2};3,-\frac{2}{3}} + \chi^{m4\times m6\times\overline{m4}\times\overline{m6}}_{3,\frac{1}{2};6,\frac{2}{5};3,-\frac{1}{2};8,-\frac{1}{15}} + \chi^{m4\times m6\times\overline{m4}\times\overline{m6}}_{3,\frac{1}{2};10,\frac{7}{5};3,-\frac{1}{2};8,-\frac{1}{15}}$$

$$Z_{r_2}^{\mathcal{D}(S_3)} = \chi^{m4\times m6\times\overline{m4}\times\overline{m6}}_{1,0,3,\frac{2}{3};1,0,1,0} + \chi^{m4\times m6\times\overline{m4}\times\overline{m6}}_{1,0,3,\frac{2}{3};1,0,5,-3} + \chi^{m4\times m6\times\overline{m4}\times\overline{m6}}_{1,0,8,\frac{1}{15};1,0,6,-\frac{2}{5}} + \chi^{m4\times m6\times\overline{m4}\times\overline{m6}}_{1,0,8,\frac{1}{15};1,0,10,-\frac{7}{5}}$$
$$+ \chi^{m4\times m6\times\overline{m4}\times\overline{m6}}_{2,\frac{1}{16};3,\frac{2}{3};2,-\frac{1}{16};1,0} + \chi^{m4\times m6\times\overline{m4}\times\overline{m6}}_{2,\frac{1}{16};3,\frac{2}{3};2,-\frac{1}{16};5,-3} + \chi^{m4\times m6\times\overline{m4}\times\overline{m6}}_{2,\frac{1}{16};8,\frac{1}{15};2,-\frac{1}{16};6,-\frac{2}{5}} + \chi^{m4\times m6\times\overline{m4}\times\overline{m6}}_{2,\frac{1}{16};8,\frac{1}{15};2,-\frac{1}{16};10,-\frac{7}{5}}$$
$$+ \chi^{m4\times m6\times\overline{m4}\times\overline{m6}}_{3,\frac{1}{2};3,\frac{2}{3};3,-\frac{1}{2};1,0} + \chi^{m4\times m6\times\overline{m4}\times\overline{m6}}_{3,\frac{1}{2};3,\frac{2}{3};3,-\frac{1}{2};5,-3} + \chi^{m4\times m6\times\overline{m4}\times\overline{m6}}_{3,\frac{1}{2};8,\frac{1}{15};3,-\frac{1}{2};6,-\frac{2}{5}} + \chi^{m4\times m6\times\overline{m4}\times\overline{m6}}_{3,\frac{1}{2};8,\frac{1}{15};3,-\frac{1}{2};10,-\frac{7}{5}}$$

$$Z_{s}^{\mathcal{D}(S_3)} = \chi^{m4\times m6\times\overline{m4}\times\overline{m6}}_{1,0;2,\frac{1}{8};3,-\frac{1}{2};4,-\frac{13}{8}} + \chi^{m4\times m6\times\overline{m4}\times\overline{m6}}_{1,0;4,\frac{13}{8};3,-\frac{1}{2};2,-\frac{1}{8}} + \chi^{m4\times m6\times\overline{m4}\times\overline{m6}}_{1,0;7,\frac{1}{40};3,-\frac{1}{2};9,-\frac{21}{40}} + \chi^{m4\times m6\times\overline{m4}\times\overline{m6}}_{1,0;9,\frac{21}{40};3,-\frac{1}{2};7,-\frac{1}{40}}$$
$$+ \chi^{m4\times m6\times\overline{m4}\times\overline{m6}}_{2,\frac{1}{16};2,\frac{1}{8};2,-\frac{1}{16};2,-\frac{1}{8}} + \chi^{m4\times m6\times\overline{m4}\times\overline{m6}}_{2,\frac{1}{16};4,\frac{13}{8};2,-\frac{1}{16};4,-\frac{13}{8}} + \chi^{m4\times m6\times\overline{m4}\times\overline{m6}}_{2,\frac{1}{16};7,\frac{1}{40};2,-\frac{1}{16};7,-\frac{1}{40}} + \chi^{m4\times m6\times\overline{m4}\times\overline{m6}}_{2,\frac{1}{16};9,\frac{21}{40};2,-\frac{1}{16};9,-\frac{21}{40}}$$
$$+ \chi^{m4\times m6\times\overline{m4}\times\overline{m6}}_{3,\frac{1}{2};2,\frac{1}{8};1,0;4,-\frac{13}{8}} + \chi^{m4\times m6\times\overline{m4}\times\overline{m6}}_{3,\frac{1}{2};4,\frac{13}{8};1,0;2,-\frac{1}{8}} + \chi^{m4\times m6\times\overline{m4}\times\overline{m6}}_{3,\frac{1}{2};7,\frac{1}{40};1,0;9,-\frac{21}{40}} + \chi^{m4\times m6\times\overline{m4}\times\overline{m6}}_{3,\frac{1}{2};9,\frac{21}{40};1,0;7,-\frac{1}{40}}$$

$$Z_{s_1}^{\mathcal{D}(S_3)} = \chi^{m4\times m6\times\overline{m4}\times\overline{m6}}_{1,0;2,\frac{1}{8};3,-\frac{1}{2};2,-\frac{1}{8}} + \chi^{m4\times m6\times\overline{m4}\times\overline{m6}}_{1,0;4,\frac{13}{8};3,-\frac{1}{2};4,-\frac{13}{8}} + \chi^{m4\times m6\times\overline{m4}\times\overline{m6}}_{1,0;7,\frac{1}{40};3,-\frac{1}{2};7,-\frac{1}{40}} + \chi^{m4\times m6\times\overline{m4}\times\overline{m6}}_{1,0;9,\frac{21}{40};3,-\frac{1}{2};9,-\frac{21}{40}}$$
$$+ \chi^{m4\times m6\times\overline{m4}\times\overline{m6}}_{2,\frac{1}{16};2,\frac{1}{8};2,-\frac{1}{16};4,-\frac{13}{8}} + \chi^{m4\times m6\times\overline{m4}\times\overline{m6}}_{2,\frac{1}{16};4,\frac{13}{8};2,-\frac{1}{16};2,-\frac{1}{8}} + \chi^{m4\times m6\times\overline{m4}\times\overline{m6}}_{2,\frac{1}{16};7,\frac{1}{40};2,-\frac{1}{16};9,-\frac{21}{40}} + \chi^{m4\times m6\times\overline{m4}\times\overline{m6}}_{2,\frac{1}{16};9,\frac{21}{40};2,-\frac{1}{16};7,-\frac{1}{40}}$$
$$+ \chi^{m4\times m6\times\overline{m4}\times\overline{m6}}_{3,\frac{1}{2};2,\frac{1}{8};1,0;2,-\frac{1}{8}} + \chi^{m4\times m6\times\overline{m4}\times\overline{m6}}_{3,\frac{1}{2};4,\frac{13}{8};1,0;4,-\frac{13}{8}} + \chi^{m4\times m6\times\overline{m4}\times\overline{m6}}_{3,\frac{1}{2};7,\frac{1}{40};1,0;7,-\frac{1}{40}} + \chi^{m4\times m6\times\overline{m4}\times\overline{m6}}_{3,\frac{1}{2};9,\frac{21}{40};1,0;9,-\frac{21}{40}}$$

This multi-component, SymTO-resolved, partition function is constructed out of the conformal characters of the primary operators of the Ising (labeled as $m4$ indicating it is the (4,3) minimal model) and the Potts (labeled as $m6$ indicating it is the (6,5) minimal model) CFTs. Similar to the convention in Eq. (E.1), the expression $\chi^{m4\times m6\times\overline{m4}\times\overline{m6}}_{a,h_a,\alpha,h_\alpha;b,-h_b,\beta,h_\beta}$ is a short-hand notation for the product of the left moving chiral conformal characters associated with the primary operators of the Ising and Potts CFTs with conformal weights $(h_a, 0)$ and $(h_\alpha, 0)$ respectively, with the right moving chiral conformal characters associated with the primary operators of the Ising and Potts CFTs with conformal weights $(0, h_b)$ and $(0, h_\beta)$ respectively.

The only sector relevant for the purposes of this paper is the "vacuum" sector $Z_{\mathbf{1}}$, which contains the $S_3$ symmetric operators constructed out of the primary operators of Ising and Potts CFTs. Note that there are 3 relevant operators that have zero conformal spin. These have scaling dimensions $\frac{4}{5}$, 1, and $\frac{9}{5}$. They correspond to the three relevant perturbations explored by the couplings $J_1 - J_2$, $J_3 - J_4$, and $J_\perp$ in Sec. 2.3. Only $J_3 - J_4$ is unchanged by the action of the $\mathbb{Z}_3$ KW duality.

# F  Spin chain with $G$ symmetry and its gauged partner with Rep($G$) symmetry

In this appendix, we present a simple manifestation of the fact that upon gauging $G$ symmetry of a spin chain, with on-site Hilbert space identical to the regular representation of $G$, one obtains a dual spin chain, also with on-site Hilbert space identical to the regular representation of $G$, that has Rep($G$) symmetry.

We consider a tensor product Hilbert space in one spatial dimension, where the on-site Hilbert spaces are $|G|$ dimensional, and spanned by orthonormal basis vectors labeled by group elements, *i.e.*,

$$\mathcal{H} = \otimes_i \mathcal{H}_i, \quad \mathcal{H}_i = \mathrm{span}\{|g\rangle \,|g \in G\} \tag{F.1}$$

Then, we can construct a spin chain symmetric under $G$ (0-form) symmetry, using two families of local symmetric operators.

$$\widehat{H}_G = \sum_{i \in \mathrm{sites}} \left( \widehat{L}_i + J\hat{\delta}_{i,i+1} \right) \tag{F.2}$$

where, $\widehat{L}_i = \sum_{h \in G} \widehat{L^h}_i$ and $\widehat{L^h}_i$ acts on a basis vector at site $i$ by left multiplication, *i.e.*,

$$\widehat{L^h}_i |g_i\rangle := |hg_i\rangle. \tag{F.3}$$

and

$$\hat{\delta}_{i,i+1} := \sum_{\{g\}} \delta_{g_i, g_{i+1}} |\{g\}\rangle\langle\{g\}| \tag{F.4}$$

It is straightforward to check that this Hamiltonian has a $G$ symmetry that acts by left multiplication on the basis vectors,[46]

$$\widehat{U_h} := \prod_{i \in \mathrm{sites}} \widehat{L^h}_i, \tag{F.5}$$

where $h$ is any element of $G$. The $G$ symmetry is reflected in the fact that $H_G$ commutes with all $U_h$.

A dual model can be defined in terms of degrees of freedom on links instead of sites. The local Hilbert space on each link is isomorphic to the one described in Eq. (F.1),

$$\mathcal{H} = \otimes_{l \in \mathrm{links}} \mathcal{H}_l, \quad \mathcal{H}_l = \mathrm{span}\{|\widetilde{g}\rangle \,|\, \widetilde{g} \in G\}. \tag{F.6}$$

The Hamiltonian for this dual model is defined as

$$\widehat{H}_{\mathsf{Rep}(G)} = \sum_{i \in \mathrm{sites}} \left( \widehat{Q}_i + J\widehat{\Delta}_{(i,i+1)} \right) \tag{F.7}$$

where $\widehat{Q}_i = \sum_{h \in G} \widehat{Q^h}_i$, with $\widehat{Q^h}_i$ acting as

$$\widehat{Q^h}_i \left| \ldots, \widetilde{g}_{(i-1,i)}, \widetilde{g}_{(i,i+1)}, \ldots \right\rangle := \left| \ldots, \widetilde{g}_{(i-1,i)} h^{-1}, h\widetilde{g}_{(i,i+1)}, \ldots \right\rangle, \tag{F.8}$$

and $\widehat{\Delta}_{(i,i+1)}$ defined as

$$\widehat{\Delta}_{(i,i+1)} := \sum_{\{\widetilde{g}\}} \delta_{\widetilde{g}_{(i,i+1)}, e} |\{\widetilde{g}\}\rangle\langle\{\widetilde{g}\}|, \tag{F.9}$$

---

[46]It also has another independent $G$ symmetry that acts by right-multiplication. We will ignore this symmetry in the present discussion. To be concrete, one can include additional terms in the Hamiltonian that explicitly break this symmetry, while preserving the $G$ symmetry acting by left-multiplications.

where $e$ is the identity element in $G$. In the equations above, we have parametrized the links with pairs of successive site indices $(i, i+1)$.

Let us show that the model (F.7) can be obtained from the model (F.2) by gauging the symmetry $G$. To that end, we introduce link degrees of freedom and enlarge the Hilbert space to

$$\mathcal{H}^{\text{large}} = \mathcal{H}^{\text{sites}} \otimes \mathcal{H}^{\text{links}} \tag{F.10}$$

where $\mathcal{H}^{\text{links}} = \otimes_{l \in \text{links}} \mathcal{H}_l$, with $\mathcal{H}_l = \text{span}\{|g\rangle \,|\, g \in G\}$. Next, we minimally couple our gauge field degrees of freedom (on the links) to the site degrees of freedom by modifying the second term of (F.2) to

$$J\hat{\delta}'_{i,i+1} := \sum_{\{g,\widetilde{g}\}} J\delta_{g_i^{-1}\widetilde{g}_{(i,i+1)}g_{i+1},e} \, |\{g,\widetilde{g}\}\rangle\langle\{g,\widetilde{g}\}|, \tag{F.11}$$

where by $\{g, \widetilde{g}\}$, we refer to the basis vectors of $\mathcal{H}^{\text{large}}$, labeled by $G$-variables on both sites and links. Next, we impose the Gauss law constraints to project down to the smaller physical Hilbert space $\mathcal{H}^{\text{phys}}$. We define the Gauss law operators via its action on the enlarged Hilbert space basis vectors,

$$\widehat{\mathbb{G}^h}_j \left| \ldots, \widetilde{g}_{(j-1,j)}, g_j, \widetilde{g}_{(j,j+1)}, \ldots \right\rangle = \left| \ldots, \widetilde{g}_{(j-1,j)}h^{-1}, hg_j, h\widetilde{g}_{(j,j+1)}, \ldots \right\rangle. \tag{F.12}$$

We can compactly express $\mathbb{G}_j^h$ in terms of the operators $\widehat{L^h}_j$ and $\widehat{Q^h}_j$ introduced in Eq. (F.3) and Eq. (F.8), $\widehat{\mathbb{G}^h}_j = \widehat{L^h}_j \widehat{Q^h}_j$. The physical Hilbert space is given by

$$\mathcal{H}^{\text{phys}} \cong \mathcal{H}^{\text{large}} \big/ _{\{\mathbb{G}_j^h=1 \; \forall j \; \forall h\}}. \tag{F.13}$$

On this reduced Hilbert space, $\widehat{\mathbb{G}^h}_j$ acts as the identity operator, by definition. So as far as states in $\mathcal{H}^{\text{phys}}$ are concerned, $\widehat{L^h}_j$ acts as $\left(\widehat{Q^h}_j\right)^{-1} = \widehat{Q^{h^{-1}}}_j$. Let us now gauge-fix using the unitary operators,

$$\widehat{U}_{\{\gamma_j\}} = \prod_{j \in \text{sites}} \widehat{\mathbb{G}^{\gamma_j}}_j. \tag{F.14}$$

In other words, we start with an arbitrary state

$$\left| \ldots, \widetilde{g}_{(j-1,j)}, g_j, \widetilde{g}_{(j,j+1)}, \ldots \right\rangle \in \mathcal{H}^{\text{large}} \tag{F.15}$$

and gauge-fix by applying $\widehat{U}_{\{\gamma_j\}}$ with $\gamma_j = g_j$, to end up with

$$\left| \ldots, g_{j-1}\widetilde{g}_{(j-1,j)}g_j^{-1}, e, g_j\widetilde{g}_{(j,j+1)}g_{j+1}^{-1}, \ldots \right\rangle. \tag{F.16}$$

Thus our gauge-fixed states $\left| \ldots, \widetilde{g}'_{(j-1,j)}, g'_j, \widetilde{g}'_{(j,j+1)}, \ldots \right\rangle$ are given in terms of the original site and index labels by

$$\widetilde{g}'_{(j,j+1)} = g_j\widetilde{g}_{(j,j+1)}g_{j+1}^{-1}, \quad g'_j = e, \tag{F.17}$$

*i.e.*, the gauge-fixed states all have the site degrees of freedom labeled by the identity element of the group $G$. On these states, our minimal coupling term (F.11) becomes $J\widehat{\Delta}_{(i,i+1)}$ so that the full gauge-fixed gauged Hamiltonian takes the form

$$\widehat{H}_{\text{Rep}(G)} = \sum_{i \in \text{sites}} \widehat{Q}_i + J\widehat{\Delta}_{(i,i+1)},$$

thereby deriving Eq. (F.7). We note that the gauge-fixed Hilbert space does have a tensor product structure, unlike $\mathcal{H}^{\mathrm{phys}}$. Therefore, it makes sense to refer to this model as a "spin chain".

Turns out, the Hamiltonian (F.7) has the non-invertible 0-form symmetry described by the fusion category $\mathsf{Rep}(G)$.[47] The associated symmetry transformations are implemented by the operators

$$\widehat{W}_R = \sum_{\{\widetilde{g}_l\}} \mathrm{Tr}\, R\left(\prod_{l\in\mathrm{links}} \widetilde{g}_l\right) |\{\widetilde{g}_l\}\rangle\langle\{\widetilde{g}_l\}| \tag{F.18}$$

where $R$ takes values in the set of irreducible representations of $G$, namely the simple objects of $\mathsf{Rep}(G)$. The first term in Hamiltonian (F.7) commutes with $\widehat{W}_R$ due to the fact that $R(h^{-1})R(h) = R(h^{-1}h) = R(e) = \mathbf{1}$, on account of $R$ being a representation of $G$. The second term commutes as well since both $\widehat{W}_R$ and this term are diagonal in the product basis of the link degrees of freedom.

# G   Fermionic $\mathsf{SRep}(S_3)$ symmetry

In Sec. 3, we constructed the $\mathsf{Rep}(S_3)$-symmetric Hamiltonian (3.13) form $S_3$-symmetric Hamiltonian (2.3) by gauging the $\mathbb{Z}_2$ subgroup of $S_3$. An alternative way to gauging this $\mathbb{Z}_2$ symmetry is to introduce $\mathbb{Z}_2$ link degrees of freedom that obeys fermionic statistics, which implements the so-called Jordan-Wigner (JW) transformation [23, 130, 131]. Such a JW transformation is viable also for gauging the $\mathbb{Z}_2$ subgroup of $S_3$-symmetry and delivers a super fusion category symmetry $\mathsf{SRep}(S_3)$ (see Ref. [132] for a discussion from topological quantum field theory perspective).

## G.1   Jordan-Wigner duality and constructing $\mathsf{SRep}(S_3)$ symmetry

We follow the strategy employed in Ref. [133] and introduce two Majorana degrees of freedom $\left\{\hat{\eta}_{i+1/2}, \hat{\xi}_{i+1/2}\right\}$ on each link, which satisfy the fermionic anticommutation relations

$$\left\{\hat{\eta}_{i+1/2}, \hat{\xi}_{j+1/2}\right\} = 0, \qquad \left\{\hat{\eta}_{i+1/2}, \hat{\eta}_{j+1/2}\right\} = \left\{\hat{\xi}_{i+1/2}, \hat{\xi}_{j+1/2}\right\} = 2\delta_{ij}. \tag{G.1}$$

Without loss of generality, we impose periodic boundary conditions on the fermionic degrees of freedom, i.e., $\hat{\eta}_{i+L+1/2} = +\hat{\eta}_{i+1/2}$ and $\hat{\xi}_{i+L+1/2} = +\hat{\xi}_{i+1/2}$ and set the cardinality of the lattice to be even, i.e., $L = 0 \bmod 2$. To gauge the $\mathbb{Z}_2$ subgroup of $S_3$, we define the pairwise commuting Gauss operators

$$\widehat{G}_i^{\mathrm{F}} := \mathrm{i}\hat{\xi}_{i-1/2}\, \hat{\sigma}_i^x\, \hat{\tau}_i^x\, \widehat{C}_i\, \hat{\eta}_{i+1/2}, \qquad \left[\widehat{G}_i^{\mathrm{F}}\right]^2 = \hat{\mathbb{1}}, \tag{G.2}$$

where, as opposed to the Gauss operator in Eq. (3.2) the local representative of $\widehat{U}_s$ symmetry is sandwiched between fermionic operators. Just as it was the case before, we define the gauge invariant subspace to be the one for which the Gauss operators are set to identity.

In a similar fashion to Sec. (3), by minimally coupling the bond algebra (2.6) we can construct a gauge invariant bond algebra. To this end, we define the pairwise commuting local operators

$$\hat{p}_{i+1/2} := \mathrm{i}\hat{\xi}_{i+1/2}\, \hat{\eta}_{i+1/2}, \qquad \left[\hat{p}_{i+1/2}, \hat{p}_{j+1/2}\right] = 0, \quad \hat{p}_{i+1/2}^2 = \hat{\mathbb{1}}, \tag{G.3}$$

---

[47]This explains the subscript on the dual Hamiltonian (F.7).

which are local at the links $i + 1/2$. The minimally coupled bond algebra is then

$$\mathfrak{B}_{\mathrm{F}}^{\mathrm{mc}} := \left\langle \hat{\sigma}_i^z \, \hat{\tau}_i^z, \, \hat{\tau}_i^z \, \hat{p}_{i+1/2} \, \hat{\sigma}_{i+1}^z, \, \hat{\sigma}_i^x, \, \hat{\tau}_i^x, \, \left( \widehat{X}_i + \widehat{X}_i^\dagger \right), \, \left( \widehat{Z}_i^{\hat{p}_{i+1/2}} \, \widehat{Z}_{i+1}^\dagger + \widehat{Z}_i^{-\hat{p}_{i+1/2}} \, \widehat{Z}_{i+1} \right), \right.$$
$$\left. \hat{\sigma}_i^z \left( \widehat{X}_i - \widehat{X}_i^\dagger \right), \, \hat{\tau}_i^z \, \hat{p}_{i+1/2} \left( \widehat{Z}_i^{\hat{p}_{i+1/2}} \, \widehat{Z}_{i+1}^\dagger - \widehat{Z}_i^{-\hat{p}_{i+1/2}} \, \widehat{Z}_{i+1} \right) \, \Big| \, \widehat{G}_i^{\mathrm{F}} = 1, \quad i \in \Lambda \right\rangle,$$
(G.4)

where the local operator $\hat{p}_{i+1/2}$ acts as a $\mathbb{Z}_2$-valued bosonic gauge field. Physically, this operator measures the local fermion parity at link $i + 1/2$. We implement an analogue of the unitary transformation (3.4) such that

$$\begin{aligned} \widehat{U} \, \hat{\sigma}_i^x \, \widehat{U}^\dagger &= \mathrm{i} \hat{\xi}_{i-1/2} \, \hat{\sigma}_i^x \, \hat{\tau}_i^x \, \widehat{C}_i \, \hat{\eta}_{i+1/2}, & \widehat{U} \, \hat{\sigma}_i^z \, \widehat{U}^\dagger &= \hat{\sigma}_i^z, \\ \widehat{U} \, \hat{\tau}_i^x \, \widehat{U}^\dagger &= \hat{\tau}_i^x, & \widehat{U} \, \hat{\tau}_i^z \, \widehat{U}^\dagger &= \hat{\tau}_i^z \, \hat{\sigma}_i^z, \\ \widehat{U} \, \widehat{X}_i \, \widehat{U}^\dagger &= \widehat{X}_i^{\hat{\sigma}_i^z}, & \widehat{U} \, \widehat{Z}_i \, \widehat{U}^\dagger &= \widehat{Z}_i^{\hat{\sigma}_i^z}, \\ \widehat{U} \, \hat{\xi}_{i+1/2} \, \widehat{U}^\dagger &= \hat{\xi}_{i+1/2} \, \hat{\sigma}_{i+1}^z, & \widehat{U} \, \hat{\eta}_{i+1/2}, \widehat{U}^\dagger &= \hat{\sigma}_i^z \, \hat{\eta}_{i+1/2}, \end{aligned}$$
(G.5)

which simplifies the Gauss operator to $\widehat{U} \, \widehat{G}_i^{\mathrm{F}} \, \widehat{U}^\dagger = \hat{\sigma}_i^x$. Setting $\hat{\sigma}_i^x = 1$ and shifting the fermionic link degrees of freedom to sites by $i + 1/2 \mapsto i + 1$ delivers the dual bond algebra

$$\mathfrak{B}_{\mathrm{F}} := \left\langle \hat{\tau}_i^z, \, \hat{\tau}_i^z \, \mathrm{i} \hat{\xi}_{i+1} \, \hat{\eta}_{i+1}, \, \mathrm{i} \hat{\xi}_i \, \hat{\tau}_i^x \, \widehat{C}_i \, \hat{\eta}_{i+1}, \, \hat{\tau}_i^x, \, \left( \widehat{X}_i + \widehat{X}_i^\dagger \right), \, \left( \widehat{Z}_i^{\mathrm{i} \hat{\xi}_{i+1} \, \hat{\eta}_{i+1}} \, \widehat{Z}_{i+1}^\dagger + \widehat{Z}_i^{-\mathrm{i} \hat{\xi}_{i+1} \, \hat{\eta}_{i+1}} \, \widehat{Z}_{i+1} \right), \right.$$
$$\left. \left( \widehat{X}_i - \widehat{X}_i^\dagger \right), \, \hat{\tau}_i^z \, \mathrm{i} \hat{\xi}_{i+1} \, \hat{\eta}_{i+1} \left( \widehat{Z}_i^{\mathrm{i} \hat{\xi}_{i+1} \, \hat{\eta}_{i+1}} \, \widehat{Z}_{i+1}^\dagger - \widehat{Z}_i^{-\mathrm{i} \hat{\xi}_{i+1} \, \hat{\eta}_{i+1}} \, \widehat{Z}_{i+1} \right) \, \Big| \, i \in \Lambda \right\rangle.$$
(G.6)

By comparing with the bond algebra (3.5) of $\mathsf{Rep}(S_3)$-symmetric operators, we conclude that the generators of the bond algebra (G.6) commute with the operators

$$\begin{aligned} \widehat{W}_{\mathbf{1}} &:= \hat{\mathbb{1}}, \\ \widehat{W}_p &:= \prod_i^L \mathrm{i} \hat{\xi}_i \, \hat{\eta}_i \equiv \prod_i^L \hat{p}_i, \\ \widehat{W}_{\mathbf{2}} &:= \frac{1}{2} \left( 1 + \prod_{i=1}^L \mathrm{i} \hat{\xi}_i \, \hat{\eta}_i \right) \left[ \prod_{i=1}^L \widehat{X}_i^{\prod_{k=2}^i \mathrm{i} \hat{\xi}_k \, \hat{\eta}_k} + \widehat{X}_i^{-\prod_{k=2}^i \mathrm{i} \hat{\xi}_k \, \hat{\eta}_k} \right]. \end{aligned}$$
(G.7a)

These operator satisfy the fusion rules

$$\widehat{W}_p \, \widehat{W}_p = \widehat{W}_{\mathbf{1}}, \qquad \widehat{W}_p \, \widehat{W}_{\mathbf{2}} = \widehat{W}_{\mathbf{2}}, \qquad \widehat{W}_{\mathbf{2}} \, \widehat{W}_{\mathbf{2}} = \widehat{W}_{\mathbf{1}} + \widehat{W}_p + \widehat{W}_{\mathbf{2}}.$$
(G.7b)

Note that the operator $\widehat{W}_p$ implements the $\mathbb{Z}_2^{\mathrm{F}}$ fermion parity symmetry, which is special in the sense that it cannot be broken explicitly or spontaneously and is a symmetry of any fermionic model. We call the symmetry generated by operators (G.7) the super fusion category $\mathsf{SRep}(S_3)$, where the adjective super signifies the non-trivial inclusion of fermion parity symmetry into the fusion category.

On the one hand, one verifies that the image of the gauged $\mathbb{Z}_2$ symmetry $\widehat{U}_s$ is

$$\prod_i^L \mathrm{i} \hat{\xi}_i \, \hat{\eta}_{i+1} = (-1) \prod_i^L \mathrm{i} \hat{\xi}_i \, \hat{\eta}_i,$$
(G.8)

where we used the facts that $L$ is even and we imposed periodic boundary conditions on the fermionic degrees of freedom. On the other hand, imposing periodic boundary conditions on both fermions and bosons imply that image of $\hat{\mathbb{1}} = \prod_i \hat{\sigma}_i^z \hat{\sigma}_{i+1}^z$ is

$$\prod_i^L \mathrm{i}\hat{\xi}_i \, \hat{\eta}_i \equiv \widehat{W}_p. \tag{G.9}$$

Therefore, we conclude that the duality between the bond algebras (2.6) and (G.6) holds in the subalgebras

$$\mathfrak{B}_{S_3}\Big|_{\widehat{U}_s = -1} \cong \mathfrak{B}_\mathrm{F}\Big|_{\widehat{W}_p = +1}. \tag{G.10}$$

## G.2 Hamiltonian and its phase diagram

Using this duality, we can construct the image of the Hamiltonian (2.3) as

$$
\begin{aligned}
\widehat{H}_{\mathsf{SRep}(S_3)} := & -J_1 \sum_{i=1}^L \left( \widehat{Z}_i^{\mathrm{i}\hat{\xi}_{i+1}\,\hat{\eta}_{i+1}} \widehat{Z}_{i+1}^\dagger + \widehat{Z}_i^{-\mathrm{i}\hat{\xi}_{i+1}\,\hat{\eta}_{i+1}} \widehat{Z}_{i+1} \right) - J_2 \sum_{i=1}^L \left( \widehat{X}_i + \widehat{X}_i^\dagger \right) \\
& -J_3 \sum_{i=1}^L \left( \hat{\tau}_i^z + \hat{\tau}_i^z \, \mathrm{i}\hat{\xi}_{i+1}\,\hat{\eta}_{i+1} \right) - J_4 \sum_{i=1}^L \left( \mathrm{i}\hat{\xi}_i \, \hat{\tau}_i^x \, \widehat{C}_i \, \hat{\eta}_{i+1} + \hat{\tau}_i^x \right) \\
& -J_5 \sum_{i=1}^L \mathrm{i}\, \hat{\tau}_i^z \, \mathrm{i}\hat{\xi}_{i+1}\,\hat{\eta}_{i+1} \left( \widehat{Z}_i^{\mathrm{i}\hat{\xi}_{i+1}\,\hat{\eta}_{i+1}} \widehat{Z}_{i+1}^\dagger - \widehat{Z}_i^{-\mathrm{i}\hat{\xi}_{i+1}\,\hat{\eta}_{i+1}} \widehat{Z}_{i+1} \right) \\
& -J_6 \sum_{i=1}^L \mathrm{i}\left( \widehat{X}_i - \widehat{X}_i^\dagger \right),
\end{aligned}
\tag{G.11}
$$

which is symmetric under the $\mathsf{SRep}(S_3)$ symmetry generated by operators (G.7). By duality, the phase diagram of this Hamiltonian has the same shape as that of Hamiltonian (2.21). Without loss of generality, we set $J_5 = J_6 = 0$ and identify the following ground states corresponding to four fixed-point gapped phases.

(i) When $J_1 = J_4 = 0$, Hamiltonian (G.11) becomes

$$\widehat{H}_{\mathsf{SRep}(S_3);2,3} := -J_2 \sum_{i=1}^L \left( \widehat{X}_i + \widehat{X}_i^\dagger \right) - J_3 \sum_{i=1}^L \left( \hat{\tau}_i^z + \hat{\tau}_i^z \, \mathrm{i}\hat{\xi}_{i+1}\,\hat{\eta}_{i+1} \right) \tag{G.12}$$

There is a single nondegenerate gapped ground state

$$|\mathrm{GS}_{\mathrm{Triv}}\rangle := \bigotimes_{i=1}^L |\tau_i^z = 1,\ \mathrm{i}\xi_i\,\eta_i = 1,\ X_i = 1\rangle, \tag{G.13}$$

which is symmetric under the entire $\mathsf{SRep}(S_3)$ symmetry. This ground state carries even fermion parity and is a trivial invertible fermionic topological state. For that reason we call the phase trivial $\mathsf{SRep}(S_3)$-symmetric phase.

(ii) When $J_2 = J_4 = 0$, Hamiltonian (G.11) becomes

$$\widehat{H}_{\mathsf{Rep}(S_3);1,3} := -J_1 \sum_{i=1}^L \left( \widehat{Z}_i^{\mathrm{i}\hat{\xi}_{i+1}\,\hat{\eta}_{i+1}} \widehat{Z}_{i+1}^\dagger + \widehat{Z}_i^{-\mathrm{i}\hat{\xi}_{i+1}\,\hat{\eta}_{i+1}} \widehat{Z}_{i+1} \right)$$

$$- J_3 \sum_{i=1}^{L} \left( \hat{\tau}_i^z + \hat{\tau}_i^z \, \mathrm{i}\hat{\xi}_{i+1} \, \hat{\eta}_{i+1} \right). \tag{G.14}$$

There are three degenerate ground states

$$|\mathrm{GS}_{\mathrm{Triv}}^\alpha\rangle := \bigotimes_{i=1}^{L} |\tau_i^z = 1, \, \mathrm{i}\xi_i \, \eta_i = 1, \, Z_i = \omega^\alpha\rangle. \tag{G.15}$$

These ground states preserve the $\mathbb{Z}_2^{\mathrm{F}}$ fermion parity symmetry generated by $\widehat{W}_p$ while they break the non-invertible $\widehat{W}_{\mathbf{2}}$ symmetry. Under the latter each ground state is mapped to equal superposition of the the the other two, i.e.,

$$\begin{aligned}
\widehat{W}_{\mathbf{2}} \, |\mathrm{GS}_p^1\rangle &= |\mathrm{GS}_p^2\rangle + |\mathrm{GS}_p^3\rangle, \\
\widehat{W}_{\mathbf{2}} \, |\mathrm{GS}_p^2\rangle &= |\mathrm{GS}_p^3\rangle + |\mathrm{GS}_p^1\rangle, \\
\widehat{W}_{\mathbf{2}} \, |\mathrm{GS}_p^3\rangle &= |\mathrm{GS}_p^1\rangle + |\mathrm{GS}_p^2\rangle.
\end{aligned} \tag{G.16}$$

Each ground state realize a trivial invertible fermionic state. We call this the trivial $\mathsf{SRep}(S_3)/\mathbb{Z}_2^{\mathrm{F}}$ SSB phase.

(iii) When $J_1 = J_3 = 0$, the Hamiltonian (G.11) becomes

$$\widehat{H}_{\mathsf{SRep}(S_3);2,4} := -J_2 \sum_{i=1}^{L} \left( \widehat{X}_i + \widehat{X}_i^\dagger \right) - J_4 \sum_{i=1}^{L} \left( \mathrm{i}\hat{\xi}_i \, \hat{\tau}_i^x \, \widehat{C}_i \, \hat{\eta}_{i+1} + \hat{\tau}_i^x \right). \tag{G.17}$$

There is a single nondegenerate ground state

$$|\mathrm{GS}_{\mathrm{Kitaev}}\rangle := \bigotimes_{i=1}^{L} |\tau_i^x = 1, \, \mathrm{i}\xi_i \, \eta_{i+1} = 1, \, X_i = 1\rangle. \tag{G.18}$$

As opposed to the ground state (G.13), the fermions in this ground state realize a non-trivial invertible phase of matter, *i.e.*, the ground state of the Kitaev chain [134]. When open boundary conditions are imposed, the ground states become twofold degenerate with unpaired Majorana degrees of freedom at each end of the chain.

Ground state is symmetric under $\mathbb{Z}_2^{\mathrm{F}}$ subgroup and carries odd fermion parity (G.18), *i.e.*,

$$\widehat{W}_p \, |\mathrm{GS}_{\mathrm{Kitaev}}\rangle = - \, |\mathrm{GS}_{\mathrm{Kitaev}}\rangle. \tag{G.19a}$$

Because of this, it is annihilated by the non-invertible $\widehat{W}_{\mathbf{2}}$ symmetry

$$\widehat{W}_{\mathbf{2}} \, |\mathrm{GS}_{\mathrm{Kitaev}}\rangle = 0. \tag{G.19b}$$

Interestingly, Hamiltonian (G.12) has a non-degenerate and gapped ground state on which the non-invertible symmetry operator does not act. Since the expectation value of the non-invertible symmetry operator vanishes in this ground state, we call this the Kitaev $\mathsf{SRep}(S_3)/\mathbb{Z}_2^{\mathrm{F}}$ SSB phase.

(iv) When $J_2 = J_3 = 0$, the Hamiltonian (G.11) becomes

$$\widehat{H}_{\mathsf{SRep}(S_3);1,4} := - J_1 \sum_{i=1}^{L} \left( \widehat{Z}_i^{\mathrm{i}\hat{\xi}_{i+1} \, \hat{\eta}_{i+1}} \, \widehat{Z}_{i+1}^\dagger + \widehat{Z}_i^{-\mathrm{i}\hat{\xi}_{i+1} \, \hat{\eta}_{i+1}} \, \widehat{Z}_{i+1} \right)$$

$$- J_4 \sum_{i=1}^{L} \left( \mathrm{i}\hat{\xi}_i \, \hat{\tau}_i^x \, \widehat{C}_i \, \hat{\eta}_{i+1} + \hat{\tau}_i^x \right). \tag{G.20}$$

There are two degenerate ground states. First, there is a ground state obtained by setting $\widehat{Z}_i = 1$ for all sites that is given by

$$\left| \mathrm{GS}_{\mathbf{1}}^{\mathrm{K}} \right\rangle := \bigotimes_{i=1}^{L} \left| \tau_i^x = 1, \mathrm{i}\xi_i \, \eta_{i+1} = 1, Z_i = 1 \right\rangle. \tag{G.21a}$$

The second ground state is [48]

$$\left| \mathrm{GS}_{\mathbf{1}}^{\mathrm{Triv}} \right\rangle := \frac{1}{2^{L/2}} \sum_{\{s_i = \pm 1\}} s_1 \left( \bigotimes_{i=1}^{L} \left| \tau_i^x = 1, \mathrm{i}\xi_i \, \eta_i = s_i \, s_{i-1}, Z_i = \omega^{s_i} \right\rangle \right). \tag{G.21b}$$

Fermionic degrees of freedom in the former ground state (G.21a) realize the Kitaev phase, while they are in the trivial phase for the latter ground state (G.21b). These two ground states transform under $\mathsf{SRep}(S_3)$ symmetry as

$$\begin{aligned}
\widehat{W}_p \left| \mathrm{GS}_{\mathbf{1}}^{\mathrm{K}} \right\rangle &= - \left| \mathrm{GS}_{\mathbf{1}}^{\mathrm{K}} \right\rangle, & \widehat{W}_p \left| \mathrm{GS}_{\mathbf{1}}^{\mathrm{Triv}} \right\rangle &= + \left| \mathrm{GS}_{\mathbf{1}}^{\mathrm{Triv}} \right\rangle, \\
\widehat{W}_{\mathbf{2}} \left| \mathrm{GS}_{\mathbf{1}}^{\mathrm{K}} \right\rangle &= 0, & \widehat{W}_{\mathbf{2}} \left| \mathrm{GS}_{\mathbf{1}}^{\mathrm{Triv}} \right\rangle &= - \left| \mathrm{GS}_{\mathbf{1}}^{\mathrm{Triv}} \right\rangle.
\end{aligned} \tag{G.22}$$

We note that while ground state $\left| \mathrm{GS}_{\mathbf{1}}^{\mathrm{K}} \right\rangle$ breaks $\widehat{W}_{\mathbf{2}}$ symmetry, $\left| \mathrm{GS}_{\mathbf{1}}^{\mathrm{Triv}} \right\rangle$ preserves the entire symmetry group. Since fermionic degrees of freedom realize trivial and non-trivial invertible fermionic states, we call this phase mixed $\mathsf{SRep}(S_3)/\mathbb{Z}_2^{\mathrm{F}}$ SSB phase.

We identified four gapped fixed-points and constructed the corresponding ground states. We can deduce the shape of the phase diagram using the duality between Hamiltonians (2.3) and (G.11). In Fig. 17a, we show the phase diagram of (G.11) when $J_5 = J_6 = 0$. We deduce the continuous phase transitions also using the fact that the duality transformation does not change the central charge. We replace the Ising CFT with Majorana CFT which are known to be dual to each other under JW transformation we implemented.

We note that since fermion parity symmetry $\mathbb{Z}_2^{\mathrm{F}}$ cannot be spontaneously broken, in all gapped phase the only possible symmetry that can be broken is $\mathsf{SRep}(S_3)/\mathbb{Z}_2^{\mathrm{F}}$. We observe this in three of the four gapped phases. In each of the these three gapped phases, there is a distinct $\mathsf{SRep}(S_3)/\mathbb{Z}_2^{\mathrm{F}}$ SSB pattern showcasing a rich possibility of phase diagrams when non-invertible symmetries are spontaneously broken. We distinguish these symmetry breaking patterns by ground state degeneracy and whether the degenerate states realize trivial state or Kitaev state (see Fig. 17a).

## G.3    Alternative Jordan-Wigner duality

There is a second way to gauge the $\mathbb{Z}_2$ subgroup of $S_3$ symmetry using fermionic gauge fields, which also delivers an $\mathsf{SRep}(S_3)$. This alternative way differs from the discussion in

---

[48]Much like the ground state (3.36b), it is not obvious that $\left| \mathrm{GS}_{\mathbf{1}}^{\mathrm{Triv}} \right\rangle$ is shot-range entangled. However, there exists a finite depth local unitary circuit that prepares this state from the product state $\bigotimes_{i=1}^{L} \left[ \left| \tau_i^x = 1, \mathrm{i}\hat{\xi}_i \, \hat{\eta}_i = 1 \right\rangle \otimes \frac{1}{\sqrt{2}} \left( |Z_i = \omega\rangle + |Z_i = \omega^*\rangle \right) \right]$. Namely, $\left| \mathrm{GS}_{\mathbf{1}}^{\mathrm{Triv}} \right\rangle =$ $\prod_{j=1}^{L} \widehat{C}_j^{\mathrm{F}} \bigotimes_{i=1}^{L} \left[ \left| \tau_i^x = 1, \mathrm{i}\hat{\xi}_i \, \hat{\eta}_{i+1} = 1 \right\rangle \otimes \frac{1}{\sqrt{2}} \left( |Z_i = \omega\rangle + |Z_i = \omega^*\rangle \right) \right]$ where $\widehat{C}_j^{\mathrm{F}}$ is a kind of CZ operator that acts as the identity operator if $Z_j = \omega$ and as $\mathrm{i}\hat{\xi}_{j-1}\hat{\eta}_j$ if $Z_j = \omega^*$.

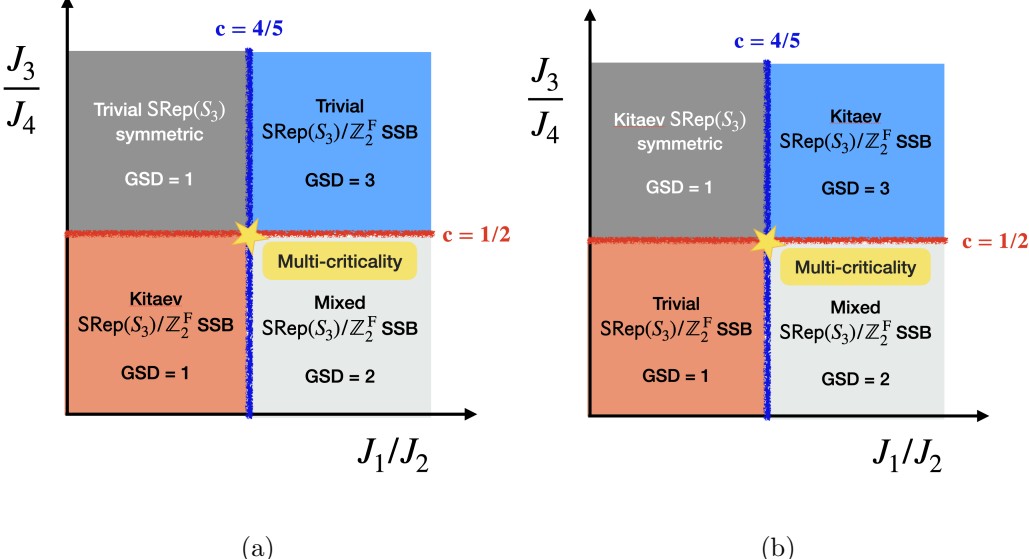

(a)                                                          (b)

Figure 17: (a) The phase diagram of Hamiltonian (G.11). Since the fermion parity symmetry $\mathsf{SRep}(S_3)$ cannot be broken (explicitly or spontaneously), in each gapped phase $\mathsf{SRep}(S_3)/\mathbb{Z}_2^F$ is broken if SSB takes place. There are three distinct SSB patterns that are distinguished by ground state degeneracies and whether if the fermionic degrees of freedom are in trivial or Kitaev phase. The corresponding fixed-point ground states are given in Eqs. (G.13), (G.15), (G.18), and (G.21). (b) The phase diagram of Hamiltonian (G.28) that is equivalent to Hamiltonian (G.11) by half translation (G.24). Because of the unitary equivalence the phase diagram of the two Hamiltonians and corresponding SSB patterns are the same. Half-translation (G.24) corresponds to stacking each ground state with the Kitaev state, which results in an exchange of labels trivial and Kitaev.

Sec. G.1 by the choice of Gauss operator. Namely, we can define

$$\widehat{G}_i^{\mathrm{F}'} := \hat{\sigma}_i^x \, \hat{\tau}_i^x \, \widehat{C}_i \, \mathrm{i}\hat{\eta}_{i+1/2} \, \hat{\xi}_{i+1/2}, \qquad \left[\widehat{G}_i^{\mathrm{F}'}\right]^2 = \hat{\mathbb{1}}, \tag{G.23}$$

which, as opposed to the Gauss operator (G.2) acts on only a single link. The two ways of gauging are related by a half-translation on Majorana operators

$$\left(\hat{\xi}_{i+1/2}, \, \hat{\eta}_{i+1/2}\right) \mapsto \left(\hat{\eta}_{i+3/2}, \, \hat{\xi}_{i+1/2}\right) \tag{G.24}$$

which can also be interpreted as "stacking" the resulting fermionic theory with Kitaev chain [74, 132, 133]. Alternatively, this half-translation corresponds to first gauging $\mathbb{Z}_2$ subgroup using bosonic gauge fields, as we have done in Sec. 3, and then gauging the $\mathbb{Z}_2$ subgroup of resulting $\mathsf{Rep}(S_3)$ symmetry.

Under this second way of implementing JW duality, the $\mathsf{Rep}(S_3)$ symmetric bond algebra is

$$\mathfrak{B}_{\mathrm{F}}' := \Big\langle \hat{\tau}_i^z, \, \hat{\tau}_i^z \, \mathrm{i}\hat{\xi}_{i+1}\hat{\eta}_{i+2}, \, \hat{\tau}_i^x \, \widehat{C}_i \, \mathrm{i}\hat{\xi}_{i+1} \, \hat{\eta}_{i+1}, \, \hat{\tau}_i^x, \, \left(\widehat{X}_i + \widehat{X}_i^\dagger\right), \, \left(\widehat{Z}_i^{-\mathrm{i}\hat{\xi}_{i+1} \, \hat{\eta}_{i+2}} \, \widehat{Z}_{i+1}^\dagger + \widehat{Z}_i^{+\mathrm{i}\hat{\xi}_{i+1} \, \hat{\eta}_{i+2}} \, \widehat{Z}_{i+1}\right),$$

$$\left(\widehat{X}_i - \widehat{X}_i^\dagger\right), \, \hat{\tau}_i^z \, \mathrm{i}\hat{\xi}_{i+1} \, \hat{\eta}_{i+2} \left(\widehat{Z}_i^{-\mathrm{i}\hat{\xi}_{i+1} \, \hat{\eta}_{i+2}} \, \widehat{Z}_{i+1}^\dagger - \widehat{Z}_i^{+\mathrm{i}\hat{\xi}_{i+1} \, \hat{\eta}_{i+2}} \, \widehat{Z}_{i+1}\right) \Big| \, i \in \Lambda \Big\rangle. \tag{G.25}$$

Since half-translation operator anticommutes with the fermion parity, we find the symme-

tries of this bond algebra to be generated by

$$
\widehat{W}'_{\mathbf{1}} := \hat{\mathbb{1}},
$$

$$
\widehat{W}'_p := \prod_i^L \mathrm{i}\hat{\xi}_i\,\hat{\eta}_i \equiv \prod_i^L \hat{p}_i,
$$

(G.26a)

$$
\widehat{W}'_{\mathbf{2}} := \frac{1}{2}\left(1 - \prod_{i=1}^L \mathrm{i}\hat{\xi}_i\,\hat{\eta}_i\right)\left[\prod_{i=1}^L \widehat{X}_i^{\prod_{k=2}^i \mathrm{i}\hat{\eta}_{k+1}\,\hat{\xi}_k} + \widehat{X}_i^{-\prod_{k=2}^i \mathrm{i}\hat{\eta}_k\,\hat{\xi}_k}\right].
$$

which satisfy the fusion rules

$$
\widehat{W}'_p\,\widehat{W}'_p = \widehat{W}'_{\mathbf{1}}, \qquad \widehat{W}'_p\,\widehat{W}'_{\mathbf{2}} = \widehat{W}'_{\mathbf{2}}, \qquad \widehat{W}'_{\mathbf{2}}\,\widehat{W}'_{\mathbf{2}} = \widehat{W}'_{\mathbf{1}} - \widehat{W}'_p + \widehat{W}'_{\mathbf{2}}.
$$

(G.26b)

The duality between bond algebras (2.6) and (G.25) holds in the subalgebras

$$
\mathfrak{B}_{S_3}\Big|_{\widehat{U}_s=-1} \cong \mathfrak{B}'_{\mathrm{F}}\Big|_{\widehat{W}_p=-1}.
$$

(G.27)

Under the half-translation (G.24), the Hamiltonian (G.11) becomes

$$
\begin{aligned}
\widehat{H}'_{\mathsf{SRep}(S_3)} :=& - J_1 \sum_{i=1}^L \left(\widehat{Z}_i^{-\mathrm{i}\hat{\xi}_{i+1}\,\hat{\eta}_{i+2}}\,\widehat{Z}_{i+1}^\dagger + \widehat{Z}_i^{+\mathrm{i}\hat{\xi}_{i+1}\,\hat{\eta}_{i+2}}\,\widehat{Z}_{i+1}\right) - J_2 \sum_{i=1}^L \left(\widehat{X}_i + \widehat{X}_i^\dagger\right)\\
& - J_3 \sum_{i=1}^L \left(\hat{\tau}_i^z + \hat{\tau}_i^z\,\mathrm{i}\hat{\eta}_{i+1}\,\hat{\xi}_{i+2}\right) - J_4 \sum_{i=1}^L \left(\hat{\tau}_i^x\,\widehat{C}_i\,\mathrm{i}\hat{\eta}_{i+1}\,\hat{\xi}_{i+1} + \hat{\tau}_i^x\right)\\
& + J_5 \sum_{i=1}^L \mathrm{i}\,\hat{\tau}_i^z\,\mathrm{i}\hat{\xi}_{i+1}\,\hat{\eta}_{i+2}\left(\widehat{Z}_i^{-\mathrm{i}\hat{\xi}_{i+1}\,\hat{\eta}_{i+2}}\,\widehat{Z}_{i+1}^\dagger - \widehat{Z}_i^{+\mathrm{i}\hat{\xi}_{i+1}\,\hat{\eta}_{i+2}}\,\widehat{Z}_{i+1}\right)\\
& - J_6 \sum_{i=1}^L \mathrm{i}\left(\widehat{X}_i - \widehat{X}_i^\dagger\right).
\end{aligned}
$$

(G.28)

Begin unitarily equivalent to Hamiltonian (G.11), this Hamiltonian shares the same phase diagram, which is shown in Fig. 17b when $J_5 = J_6 = 0$. The only difference between the two Hamiltonians is that the labels trivial and Kitaev that denote the fermionic sector of the ground states are exchanged. This is expected since half-translation (G.24) effectively stacks a Kitaev chain on top of each ground state. Since this is a unitary transformation the SSB patterns do not change. The two Hamiltonians are no longer unitarily equivalent when open boundary conditions are imposed, since the unitary equivalence under half-translation (G.24) relies on the translation invariance which is broken by open boundary conditions. This inequivalence is reflected by the differing ground state degeneracies of the two Hamiltonians with open boundary conditions.

# H  The gapless states for the self-dual $S_3$-symmetric model

To obtain the potential gapless states for the self-dual $S_3$-symmetric model, we ask, for example, do we have gapless states described by $m4 \times \overline{m4}$ minimal model CFT? To answer this question, we use the holographic modular bootstrap [29, 44, 109], *i.e.*, try to use the conformal characters of $m4 \times \overline{m4}$ model CFT to construct the 25-component partition functions that transform covariantly under the modular transformation generated by the $S, T$-matrices of the $\mathrm{JK}_4 \boxtimes \overline{\mathrm{JK}_4}$ SymTO. We obtain the following solution

$$Z_{\mathbf{1};\mathbf{1}}^{JK_4 \boxtimes \overline{JK_4}} = \chi_{1,0;1,0}^{m4 \times \overline{m4}} + \chi_{3,\frac{1}{2};3,-\frac{1}{2}}^{m4 \times \overline{m4}}$$

$$Z_{\mathbf{1};\bar{e}}^{JK_4 \boxtimes \overline{JK_4}} = \chi_{2,\frac{1}{16};2,-\frac{1}{16}}^{m4 \times \overline{m4}}$$

$$Z_{e;\mathbf{1}}^{JK_4 \boxtimes \overline{JK_4}} = \chi_{2,\frac{1}{16};2,-\frac{1}{16}}^{m4 \times \overline{m4}}$$

$$Z_{e;\bar{e}}^{JK_4 \boxtimes \overline{JK_4}} = \chi_{1,0;1,0}^{m4 \times \overline{m4}} + \chi_{3,\frac{1}{2};3,-\frac{1}{2}}^{m4 \times \overline{m4}}$$

$$Z_{m;\bar{m}}^{JK_4 \boxtimes \overline{JK_4}} = \chi_{2,\frac{1}{16};2,-\frac{1}{16}}^{m4 \times \overline{m4}}$$

$$Z_{m;\bar{m}_1}^{JK_4 \boxtimes \overline{JK_4}} = \chi_{1,0;3,-\frac{1}{2}}^{m4 \times \overline{m4}} + \chi_{3,\frac{1}{2};1,0}^{m4 \times \overline{m4}}$$

$$Z_{m_1;\bar{m}}^{JK_4 \boxtimes \overline{JK_4}} = \chi_{1,0;3,-\frac{1}{2}}^{m4 \times \overline{m4}} + \chi_{3,\frac{1}{2};1,0}^{m4 \times \overline{m4}}$$

$$Z_{m_1;\bar{m}_1}^{JK_4 \boxtimes \overline{JK_4}} = \chi_{2,\frac{1}{16};2,-\frac{1}{16}}^{m4 \times \overline{m4}}$$

$$Z_{q;\bar{q}}^{JK_4 \boxtimes \overline{JK_4}} = \chi_{1,0;1,0}^{m4 \times \overline{m4}} + \chi_{2,\frac{1}{16};2,-\frac{1}{16}}^{m4 \times \overline{m4}} + \chi_{3,\frac{1}{2};3,-\frac{1}{2}}^{m4 \times \overline{m4}}$$

$$\text{others} = 0. \tag{H.1}$$

From $Z_{\mathbf{1};\mathbf{1}}^{JK_4 \boxtimes \overline{JK_4}}$, we see that there is one symmetric relevant perturbation of scaling dimension $(h, \bar{h}) = (\frac{1}{2}, \frac{1}{2})$. From the position of $\chi_{1,0;1,0}^{m4 \times \overline{m4}}$, we see that the above gapless state is induced by condensation $\mathcal{A}_1 = \mathbf{1} \otimes e\bar{e} \otimes q\bar{q}$.

In the following, we list the gapless phases (*i.e.*, the gapless states with no symmetric relevant operators) of self-dual $S_3$ symmetric model that we find using the above method. The self-dual $\mathsf{Rep}(S_3)$ symmetric model have the same gapless phases. The SymTO $JK_4 \boxtimes \overline{JK_4}$ may be spontaneously broken in those gapless phases. We also list the correspond condensation $\mathcal{A}$ that reduces the SymTO $JK_4 \boxtimes \overline{JK_4}$ to a smaller one $\mathcal{M}_{/\mathcal{A}}$.

$$Z_{\mathbf{1};\mathbf{1}}^{JK_4 \boxtimes \overline{JK_4}} = \chi_{1,0;1,0}^{m6 \times \overline{m6}} + \chi_{10,\frac{7}{5};10,-\frac{7}{5}}^{m6 \times \overline{m6}}$$

$$Z_{\mathbf{1};\bar{e}}^{JK_4 \boxtimes \overline{JK_4}} = \chi_{1,0;5,-3}^{m6 \times \overline{m6}} + \chi_{10,\frac{7}{5};6,-\frac{2}{5}}^{m6 \times \overline{m6}}$$

$$Z_{\mathbf{1};\bar{m}}^{JK_4 \boxtimes \overline{JK_4}} = \chi_{5,3;2,-\frac{1}{8}}^{m6 \times \overline{m6}} + \chi_{6,\frac{2}{5};9,-\frac{21}{40}}^{m6 \times \overline{m6}}$$

$$Z_{\mathbf{1};\bar{m}_1}^{JK_4 \boxtimes \overline{JK_4}} = \chi_{5,3;4,-\frac{13}{8}}^{m6 \times \overline{m6}} + \chi_{6,\frac{2}{5};7,-\frac{1}{40}}^{m6 \times \overline{m6}}$$

$$Z_{\mathbf{1};\bar{q}}^{JK_4 \boxtimes \overline{JK_4}} = \chi_{1,0;3,-\frac{2}{3}}^{m6 \times \overline{m6}} + \chi_{10,\frac{7}{5};8,-\frac{1}{15}}^{m6 \times \overline{m6}}$$

$$Z_{e;\mathbf{1}}^{JK_4 \boxtimes \overline{JK_4}} = \chi_{5,3;1,0}^{m6 \times \overline{m6}} + \chi_{6,\frac{2}{5};10,-\frac{7}{5}}^{m6 \times \overline{m6}}$$

$$Z_{e;\bar{e}}^{JK_4 \boxtimes \overline{JK_4}} = \chi_{5,3;5,-3}^{m6 \times \overline{m6}} + \chi_{6,\frac{2}{5};6,-\frac{2}{5}}^{m6 \times \overline{m6}}$$

$$Z_{e;\bar{m}}^{JK_4 \boxtimes \overline{JK_4}} = \chi_{1,0;2,-\frac{1}{8}}^{m6 \times \overline{m6}} + \chi_{10,\frac{7}{5};9,-\frac{21}{40}}^{m6 \times \overline{m6}}$$

$$Z_{e;\bar{m}_1}^{JK_4 \boxtimes \overline{JK_4}} = \chi_{1,0;4,-\frac{13}{8}}^{m6 \times \overline{m6}} + \chi_{10,\frac{7}{5};7,-\frac{1}{40}}^{m6 \times \overline{m6}}$$

$$Z_{e;\bar{q}}^{JK_4 \boxtimes \overline{JK_4}} = \chi_{5,3;3,-\frac{2}{3}}^{m6 \times \overline{m6}} + \chi_{6,\frac{2}{5};8,-\frac{1}{15}}^{m6 \times \overline{m6}}$$

$$Z_{m;\mathbf{1}}^{JK_4 \boxtimes \overline{JK_4}} = \chi_{2,\frac{1}{8};5,-3}^{m6 \times \overline{m6}} + \chi_{9,\frac{21}{40};6,-\frac{2}{5}}^{m6 \times \overline{m6}}$$

$$Z_{m;\bar{e}}^{JK_4 \boxtimes \overline{JK_4}} = \chi_{2,\frac{1}{8};1,0}^{m6 \times \overline{m6}} + \chi_{9,\frac{21}{40};10,-\frac{7}{5}}^{m6 \times \overline{m6}}$$

$$Z_{m;\bar{m}}^{JK_4 \boxtimes \overline{JK_4}} = \chi_{4,\frac{13}{8};4,-\frac{13}{8}}^{m6 \times \overline{m6}} + \chi_{7,\frac{1}{40};7,-\frac{1}{40}}^{m6 \times \overline{m6}}$$

$$Z_{m;\bar{m}_1}^{JK_4 \boxtimes \overline{JK_4}} = \chi_{4,\frac{13}{8};2,-\frac{1}{8}}^{m6 \times \overline{m6}} + \chi_{7,\frac{1}{40};9,-\frac{21}{40}}^{m6 \times \overline{m6}}$$

$$Z_{m;\bar{q}}^{JK_4 \boxtimes \overline{JK_4}} = \chi_{2,\frac{1}{8};3,-\frac{2}{3}}^{m6 \times \overline{m6}} + \chi_{9,\frac{21}{40};8,-\frac{1}{15}}^{m6 \times \overline{m6}}$$

$$Z_{m_1;\mathbf{1}}^{JK_4 \boxtimes \overline{JK_4}} = \chi_{4,\frac{13}{8};5,-3}^{m6 \times \overline{m6}} + \chi_{7,\frac{1}{40};6,-\frac{2}{5}}^{m6 \times \overline{m6}}$$

$$Z_{m_1;\bar{e}}^{JK_4 \boxtimes \overline{JK_4}} = \chi_{4,\frac{13}{8};1,0}^{m6 \times \overline{m6}} + \chi_{7,\frac{1}{40};10,-\frac{7}{5}}^{m6 \times \overline{m6}}$$

$$Z_{m_1;\bar{m}}^{JK_4 \boxtimes \overline{JK_4}} = \chi_{2,\frac{1}{8};4,-\frac{13}{8}}^{m6 \times \overline{m6}} + \chi_{9,\frac{21}{40};7,-\frac{1}{40}}^{m6 \times \overline{m6}}$$

$$Z_{m_1;\bar{m}_1}^{JK_4 \boxtimes \overline{JK_4}} = \chi_{2,\frac{1}{8};2,-\frac{1}{8}}^{m6 \times \overline{m6}} + \chi_{9,\frac{21}{40};9,-\frac{21}{40}}^{m6 \times \overline{m6}}$$

$$Z_{m_1;\bar{q}}^{JK_4 \boxtimes \overline{JK_4}} = \chi_{4,\frac{13}{8};3,-\frac{2}{3}}^{m6 \times \overline{m6}} + \chi_{7,\frac{1}{40};8,-\frac{1}{15}}^{m6 \times \overline{m6}}$$

$$Z_{q;\mathbf{1}}^{JK_4 \boxtimes \overline{JK_4}} = \chi_{3,\frac{2}{3};1,0}^{m6 \times \overline{m6}} + \chi_{8,\frac{1}{15};10,-\frac{7}{5}}^{m6 \times \overline{m6}}$$

$$Z_{q;\bar{e}}^{JK_4 \boxtimes \overline{JK_4}} = \chi_{3,\frac{2}{3};5,-3}^{m6 \times \overline{m6}} + \chi_{8,\frac{1}{15};6,-\frac{2}{5}}^{m6 \times \overline{m6}}$$

$$Z_{q;\bar{m}}^{JK_4 \boxtimes \overline{JK_4}} = \chi_{3,\frac{2}{3};2,-\frac{1}{8}}^{m6 \times \overline{m6}} + \chi_{8,\frac{1}{15};9,-\frac{21}{40}}^{m6 \times \overline{m6}}$$

$$Z_{q;\bar{m}_1}^{JK_4 \boxtimes \overline{JK_4}} = \chi_{3,\frac{2}{3};4,-\frac{13}{8}}^{m6 \times \overline{m6}} + \chi_{8,\frac{1}{15};7,-\frac{1}{40}}^{m6 \times \overline{m6}}$$

$$Z_{q;\bar{q}}^{JK_4 \boxtimes \overline{JK_4}} = \chi_{3,\frac{2}{3};3,-\frac{2}{3}}^{m6 \times \overline{m6}} + \chi_{8,\frac{1}{15};8,-\frac{1}{15}}^{m6 \times \overline{m6}}$$

$$\mathcal{A} = \mathbf{1}. \tag{H.2}$$

$$Z_{\mathbf{1};\mathbf{1}}^{JK_4 \boxtimes \overline{JK_4}} = \color{blue}{\chi_{1,0;1,0}^{m6 \times \overline{m6}}} + \chi_{10,\frac{7}{5};10,-\frac{7}{5}}^{m6 \times \overline{m6}}$$

$$Z_{\mathbf{1};\bar{e}}^{JK_4 \boxtimes \overline{JK_4}} = \chi_{1,0;5,-3}^{m6 \times \overline{m6}} + \chi_{10,\frac{7}{5};6,-\frac{2}{5}}^{m6 \times \overline{m6}}$$

$$Z_{\mathbf{1};\bar{m}}^{JK_4 \boxtimes \overline{JK_4}} = \chi_{1,0;2,-\frac{1}{8}}^{m6 \times \overline{m6}} + \chi_{10,\frac{7}{5};9,-\frac{21}{40}}^{m6 \times \overline{m6}}$$

$$Z_{\mathbf{1};\bar{m}_1}^{JK_4 \boxtimes \overline{JK_4}} = \chi_{1,0;4,-\frac{13}{8}}^{m6 \times \overline{m6}} + \chi_{10,\frac{7}{5};7,-\frac{1}{40}}^{m6 \times \overline{m6}}$$

$$Z_{\mathbf{1};\bar{q}}^{JK_4 \boxtimes \overline{JK_4}} = \chi_{1,0;3,-\frac{2}{3}}^{m6 \times \overline{m6}} + \chi_{10,\frac{7}{5};8,-\frac{1}{15}}^{m6 \times \overline{m6}}$$

$$Z_{e;\mathbf{1}}^{JK_4 \boxtimes \overline{JK_4}} = \chi_{5,3;1,0}^{m6 \times \overline{m6}} + \chi_{6,\frac{2}{5};10,-\frac{7}{5}}^{m6 \times \overline{m6}}$$

$$Z_{e;\bar{e}}^{JK_4 \boxtimes \overline{JK_4}} = \chi_{5,3;5,-3}^{m6 \times \overline{m6}} + \chi_{6,\frac{2}{5};6,-\frac{2}{5}}^{m6 \times \overline{m6}}$$

$$Z_{e;\bar{m}}^{JK_4 \boxtimes \overline{JK_4}} = \chi_{5,3;2,-\frac{1}{8}}^{m6 \times \overline{m6}} + \chi_{6,\frac{2}{5};9,-\frac{21}{40}}^{m6 \times \overline{m6}}$$

$$Z_{e;\bar{m}_1}^{JK_4 \boxtimes \overline{JK_4}} = \chi_{5,3;4,-\frac{13}{8}}^{m6 \times \overline{m6}} + \chi_{6,\frac{2}{5};7,-\frac{1}{40}}^{m6 \times \overline{m6}}$$

$$Z_{e;\bar{q}}^{JK_4 \boxtimes \overline{JK_4}} = \chi_{5,3;3,-\frac{2}{3}}^{m6 \times \overline{m6}} + \chi_{6,\frac{2}{5};8,-\frac{1}{15}}^{m6 \times \overline{m6}}$$

$$Z_{m;\mathbf{1}}^{JK_4 \boxtimes \overline{JK_4}} = \chi_{2,\frac{1}{8};1,0}^{m6 \times \overline{m6}} + \chi_{9,\frac{21}{40};10,-\frac{7}{5}}^{m6 \times \overline{m6}}$$

$$Z_{m;\bar{e}}^{JK_4 \boxtimes \overline{JK_4}} = \chi_{2,\frac{1}{8};5,-3}^{m6 \times \overline{m6}} + \chi_{9,\frac{21}{40};6,-\frac{2}{5}}^{m6 \times \overline{m6}}$$

$$Z_{m;\bar{m}}^{JK_4 \boxtimes \overline{JK_4}} = \chi_{2,\frac{1}{8};2,-\frac{1}{8}}^{m6 \times \overline{m6}} + \chi_{9,\frac{21}{40};9,-\frac{21}{40}}^{m6 \times \overline{m6}}$$

$$Z_{m;\bar{m}_1}^{JK_4 \boxtimes \overline{JK_4}} = \chi_{2,\frac{1}{8};4,-\frac{13}{8}}^{m6 \times \overline{m6}} + \chi_{9,\frac{21}{40};7,-\frac{1}{40}}^{m6 \times \overline{m6}}$$

$$Z_{m;\bar{q}}^{JK_4 \boxtimes \overline{JK_4}} = \chi_{2,\frac{1}{8};3,-\frac{2}{3}}^{m6 \times \overline{m6}} + \chi_{9,\frac{21}{40};8,-\frac{1}{15}}^{m6 \times \overline{m6}}$$

$$Z_{m_1;\mathbf{1}}^{JK_4 \boxtimes \overline{JK_4}} = \chi_{4,\frac{13}{8};1,0}^{m6 \times \overline{m6}} + \chi_{7,\frac{1}{40};10,-\frac{7}{5}}^{m6 \times \overline{m6}}$$

$$Z_{m_1;\bar{e}}^{JK_4 \boxtimes \overline{JK_4}} = \chi_{4,\frac{13}{8};5,-3}^{m6 \times \overline{m6}} + \chi_{7,\frac{1}{40};6,-\frac{2}{5}}^{m6 \times \overline{m6}}$$

$$Z_{m_1;\bar{m}}^{JK_4 \boxtimes \overline{JK_4}} = \chi_{4,\frac{13}{8};2,-\frac{1}{8}}^{m6 \times \overline{m6}} + \chi_{7,\frac{1}{40};9,-\frac{21}{40}}^{m6 \times \overline{m6}}$$

$$Z_{m_1;\bar{m}_1}^{JK_4 \boxtimes \overline{JK_4}} = \chi_{4,\frac{13}{8};4,-\frac{13}{8}}^{m6 \times \overline{m6}} + \chi_{7,\frac{1}{40};7,-\frac{1}{40}}^{m6 \times \overline{m6}}$$

$$Z_{m_1;\bar{q}}^{JK_4 \boxtimes \overline{JK_4}} = \chi_{4,\frac{13}{8};3,-\frac{2}{3}}^{m6 \times \overline{m6}} + \chi_{7,\frac{1}{40};8,-\frac{1}{15}}^{m6 \times \overline{m6}}$$

$$Z_{q;\mathbf{1}}^{JK_4 \boxtimes \overline{JK_4}} = \chi_{3,\frac{2}{3};1,0}^{m6 \times \overline{m6}} + \chi_{8,\frac{1}{15};10,-\frac{7}{5}}^{m6 \times \overline{m6}}$$

$$Z_{q;\bar{e}}^{JK_4 \boxtimes \overline{JK_4}} = \chi_{3,\frac{2}{3};5,-3}^{m6 \times \overline{m6}} + \chi_{8,\frac{1}{15};6,-\frac{2}{5}}^{m6 \times \overline{m6}}$$

$$Z_{q;\bar{m}}^{JK_4 \boxtimes \overline{JK_4}} = \chi_{3,\frac{2}{3};2,-\frac{1}{8}}^{m6 \times \overline{m6}} + \chi_{8,\frac{1}{15};9,-\frac{21}{40}}^{m6 \times \overline{m6}}$$

$$Z_{q;\bar{m}_1}^{JK_4 \boxtimes \overline{JK_4}} = \chi_{3,\frac{2}{3};4,-\frac{13}{8}}^{m6 \times \overline{m6}} + \chi_{8,\frac{1}{15};7,-\frac{1}{40}}^{m6 \times \overline{m6}}$$

$$Z_{q;\bar{q}}^{JK_4 \boxtimes \overline{JK_4}} = \chi_{3,\frac{2}{3};3,-\frac{2}{3}}^{m6 \times \overline{m6}} + \chi_{8,\frac{1}{15};8,-\frac{1}{15}}^{m6 \times \overline{m6}}$$

$$\mathcal{A} = \mathbf{1}. \tag{H.3}$$

$$Z_{\mathbf{1};\mathbf{1}}^{JK_4 \boxtimes \overline{JK_4}} = \chi_{1,0;1,0}^{m6 \times \overline{m6}} + \chi_{5,3;1,0}^{m6 \times \overline{m6}} + \chi_{6,\frac{2}{5};10,-\frac{7}{5}}^{m6 \times \overline{m6}} + \chi_{10,\frac{7}{5};10,-\frac{7}{5}}^{m6 \times \overline{m6}}$$

$$Z_{\mathbf{1};\bar{e}}^{JK_4 \boxtimes \overline{JK_4}} = \chi_{1,0;5,-3}^{m6 \times \overline{m6}} + \chi_{5,3;5,-3}^{m6 \times \overline{m6}} + \chi_{6,\frac{2}{5};6,-\frac{2}{5}}^{m6 \times \overline{m6}} + \chi_{10,\frac{7}{5};6,-\frac{2}{5}}^{m6 \times \overline{m6}}$$

$$Z_{\mathbf{1};\bar{m}}^{JK_4 \boxtimes \overline{JK_4}} = \chi_{1,0;2,-\frac{1}{8}}^{m6 \times \overline{m6}} + \chi_{5,3;2,-\frac{1}{8}}^{m6 \times \overline{m6}} + \chi_{6,\frac{2}{5};9,-\frac{21}{40}}^{m6 \times \overline{m6}} + \chi_{10,\frac{7}{5};9,-\frac{21}{40}}^{m6 \times \overline{m6}}$$

$$Z_{\mathbf{1};\bar{m}_1}^{JK_4 \boxtimes \overline{JK_4}} = \chi_{1,0;4,-\frac{13}{8}}^{m6 \times \overline{m6}} + \chi_{5,3;4,-\frac{13}{8}}^{m6 \times \overline{m6}} + \chi_{6,\frac{2}{5};7,-\frac{1}{40}}^{m6 \times \overline{m6}} + \chi_{10,\frac{7}{5};7,-\frac{1}{40}}^{m6 \times \overline{m6}}$$

$$Z_{\mathbf{1};\bar{q}}^{JK_4 \boxtimes \overline{JK_4}} = \chi_{1,0;3,-\frac{2}{3}}^{m6 \times \overline{m6}} + \chi_{5,3;3,-\frac{2}{3}}^{m6 \times \overline{m6}} + \chi_{6,\frac{2}{5};8,-\frac{1}{15}}^{m6 \times \overline{m6}} + \chi_{10,\frac{7}{5};8,-\frac{1}{15}}^{m6 \times \overline{m6}}$$

$$Z_{e;\mathbf{1}}^{JK_4 \boxtimes \overline{JK_4}} = \chi_{1,0;1,0}^{m6 \times \overline{m6}} + \chi_{5,3;1,0}^{m6 \times \overline{m6}} + \chi_{6,\frac{2}{5};10,-\frac{7}{5}}^{m6 \times \overline{m6}} + \chi_{10,\frac{7}{5};10,-\frac{7}{5}}^{m6 \times \overline{m6}}$$

$$Z_{e;\bar{e}}^{JK_4 \boxtimes \overline{JK_4}} = \chi_{1,0;5,-3}^{m6 \times \overline{m6}} + \chi_{5,3;5,-3}^{m6 \times \overline{m6}} + \chi_{6,\frac{2}{5};6,-\frac{2}{5}}^{m6 \times \overline{m6}} + \chi_{10,\frac{7}{5};6,-\frac{2}{5}}^{m6 \times \overline{m6}}$$

$$Z_{e;\bar{m}}^{JK_4 \boxtimes \overline{JK_4}} = \chi_{1,0;2,-\frac{1}{8}}^{m6 \times \overline{m6}} + \chi_{5,3;2,-\frac{1}{8}}^{m6 \times \overline{m6}} + \chi_{6,\frac{2}{5};9,-\frac{21}{40}}^{m6 \times \overline{m6}} + \chi_{10,\frac{7}{5};9,-\frac{21}{40}}^{m6 \times \overline{m6}}$$

$$Z_{e;\bar{m}_1}^{JK_4 \boxtimes \overline{JK_4}} = \chi_{1,0;4,-\frac{13}{8}}^{m6 \times \overline{m6}} + \chi_{5,3;4,-\frac{13}{8}}^{m6 \times \overline{m6}} + \chi_{6,\frac{2}{5};7,-\frac{1}{40}}^{m6 \times \overline{m6}} + \chi_{10,\frac{7}{5};7,-\frac{1}{40}}^{m6 \times \overline{m6}}$$

$$Z_{e;\bar{q}}^{JK_4 \boxtimes \overline{JK_4}} = \chi_{1,0;3,-\frac{2}{3}}^{m6 \times \overline{m6}} + \chi_{5,3;3,-\frac{2}{3}}^{m6 \times \overline{m6}} + \chi_{6,\frac{2}{5};8,-\frac{1}{15}}^{m6 \times \overline{m6}} + \chi_{10,\frac{7}{5};8,-\frac{1}{15}}^{m6 \times \overline{m6}}$$

$$Z_{m;\mathbf{1}}^{JK_4 \boxtimes \overline{JK_4}} = 0$$

$$Z_{m;\bar{e}}^{JK_4 \boxtimes \overline{JK_4}} = 0$$

$$Z_{m;\bar{m}}^{JK_4 \boxtimes \overline{JK_4}} = 0$$

$$Z_{m;\bar{m}_1}^{JK_4 \boxtimes \overline{JK_4}} = 0$$

$$Z_{m;\bar{q}}^{JK_4 \boxtimes \overline{JK_4}} = 0$$

$$Z_{m_1;\mathbf{1}}^{JK_4 \boxtimes \overline{JK_4}} = 0$$

$$Z_{m_1;\bar{e}}^{JK_4 \boxtimes \overline{JK_4}} = 0$$

$$Z^{JK_4 \boxtimes \overline{JK_4}}_{m_1;\bar{m}} = 0$$

$$Z^{JK_4 \boxtimes \overline{JK_4}}_{m_1;\bar{m}_1} = 0$$

$$Z^{JK_4 \boxtimes \overline{JK_4}}_{m_1;\bar{q}} = 0$$

$$Z^{JK_4 \boxtimes \overline{JK_4}}_{q;\mathbf{1}} = 2\chi^{m6\times\overline{m6}}_{3,\frac{2}{3};1,0} + 2\chi^{m6\times\overline{m6}}_{8,\frac{1}{15};10,-\frac{7}{5}}$$

$$Z^{JK_4 \boxtimes \overline{JK_4}}_{q;\bar{e}} = 2\chi^{m6\times\overline{m6}}_{3,\frac{2}{3};5,-3} + 2\chi^{m6\times\overline{m6}}_{8,\frac{1}{15};6,-\frac{2}{5}}$$

$$Z^{JK_4 \boxtimes \overline{JK_4}}_{q;\bar{m}} = 2\chi^{m6\times\overline{m6}}_{3,\frac{2}{3};2,-\frac{1}{8}} + 2\chi^{m6\times\overline{m6}}_{8,\frac{1}{15};9,-\frac{21}{40}}$$

$$Z^{JK_4 \boxtimes \overline{JK_4}}_{q;\bar{m}_1} = 2\chi^{m6\times\overline{m6}}_{3,\frac{2}{3};4,-\frac{13}{8}} + 2\chi^{m6\times\overline{m6}}_{8,\frac{1}{15};7,-\frac{1}{40}}$$

$$Z^{JK_4 \boxtimes \overline{JK_4}}_{q;\bar{q}} = 2\chi^{m6\times\overline{m6}}_{3,\frac{2}{3};3,-\frac{2}{3}} + 2\chi^{m6\times\overline{m6}}_{8,\frac{1}{15};8,-\frac{1}{15}}$$

$$\mathcal{A} = \mathbf{1} \oplus e. \tag{H.4}$$

$$Z^{JK_4 \boxtimes \overline{JK_4}}_{\mathbf{1};\mathbf{1}} = \textcolor{blue}{\chi^{m6\times\overline{m6}}_{1,0;1,0}} + \chi^{m6\times\overline{m6}}_{1,0;5,-3} + \chi^{m6\times\overline{m6}}_{10,\frac{7}{5};6,-\frac{2}{5}} + \chi^{m6\times\overline{m6}}_{10,\frac{7}{5};10,-\frac{7}{5}}$$

$$Z^{JK_4 \boxtimes \overline{JK_4}}_{\mathbf{1};\bar{e}} = \textcolor{blue}{\chi^{m6\times\overline{m6}}_{1,0;1,0}} + \chi^{m6\times\overline{m6}}_{1,0;5,-3} + \chi^{m6\times\overline{m6}}_{10,\frac{7}{5};6,-\frac{2}{5}} + \chi^{m6\times\overline{m6}}_{10,\frac{7}{5};10,-\frac{7}{5}}$$

$$Z^{JK_4 \boxtimes \overline{JK_4}}_{\mathbf{1};\bar{m}} = 0$$

$$Z^{JK_4 \boxtimes \overline{JK_4}}_{\mathbf{1};\bar{m}_1} = 0$$

$$Z^{JK_4 \boxtimes \overline{JK_4}}_{\mathbf{1};\bar{q}} = 2\chi^{m6\times\overline{m6}}_{1,0;3,-\frac{2}{3}} + 2\chi^{m6\times\overline{m6}}_{10,\frac{7}{5};8,-\frac{1}{15}}$$

$$Z^{JK_4 \boxtimes \overline{JK_4}}_{e;\mathbf{1}} = \chi^{m6\times\overline{m6}}_{5,3;1,0} + \chi^{m6\times\overline{m6}}_{5,3;5,-3} + \chi^{m6\times\overline{m6}}_{6,\frac{2}{5};6,-\frac{2}{5}} + \chi^{m6\times\overline{m6}}_{6,\frac{2}{5};10,-\frac{7}{5}}$$

$$Z^{JK_4 \boxtimes \overline{JK_4}}_{e;\bar{e}} = \chi^{m6\times\overline{m6}}_{5,3;1,0} + \chi^{m6\times\overline{m6}}_{5,3;5,-3} + \chi^{m6\times\overline{m6}}_{6,\frac{2}{5};6,-\frac{2}{5}} + \chi^{m6\times\overline{m6}}_{6,\frac{2}{5};10,-\frac{7}{5}}$$

$$Z^{JK_4 \boxtimes \overline{JK_4}}_{e;\bar{m}} = 0$$

$$Z^{JK_4 \boxtimes \overline{JK_4}}_{e;\bar{m}_1} = 0$$

$$Z^{JK_4 \boxtimes \overline{JK_4}}_{e;\bar{q}} = 2\chi^{m6\times\overline{m6}}_{5,3;3,-\frac{2}{3}} + 2\chi^{m6\times\overline{m6}}_{6,\frac{2}{5};8,-\frac{1}{15}}$$

$$Z^{JK_4 \boxtimes \overline{JK_4}}_{m;\mathbf{1}} = \chi^{m6\times\overline{m6}}_{2,\frac{1}{8};1,0} + \chi^{m6\times\overline{m6}}_{2,\frac{1}{8};5,-3} + \chi^{m6\times\overline{m6}}_{9,\frac{21}{40};6,-\frac{2}{5}} + \chi^{m6\times\overline{m6}}_{9,\frac{21}{40};10,-\frac{7}{5}}$$

$$Z^{JK_4 \boxtimes \overline{JK_4}}_{m;\bar{e}} = \chi^{m6\times\overline{m6}}_{2,\frac{1}{8};1,0} + \chi^{m6\times\overline{m6}}_{2,\frac{1}{8};5,-3} + \chi^{m6\times\overline{m6}}_{9,\frac{21}{40};6,-\frac{2}{5}} + \chi^{m6\times\overline{m6}}_{9,\frac{21}{40};10,-\frac{7}{5}}$$

$$Z^{JK_4 \boxtimes \overline{JK_4}}_{m;\bar{m}} = 0$$

$$Z^{JK_4 \boxtimes \overline{JK_4}}_{m;\bar{m}_1} = 0$$

$$Z^{JK_4 \boxtimes \overline{JK_4}}_{m;\bar{q}} = 2\chi^{m6\times\overline{m6}}_{2,\frac{1}{8};3,-\frac{2}{3}} + 2\chi^{m6\times\overline{m6}}_{9,\frac{21}{40};8,-\frac{1}{15}}$$

$$Z^{JK_4 \boxtimes \overline{JK_4}}_{m_1;\mathbf{1}} = \chi^{m6\times\overline{m6}}_{4,\frac{13}{8};1,0} + \chi^{m6\times\overline{m6}}_{4,\frac{13}{8};5,-3} + \chi^{m6\times\overline{m6}}_{7,\frac{1}{40};6,-\frac{2}{5}} + \chi^{m6\times\overline{m6}}_{7,\frac{1}{40};10,-\frac{7}{5}}$$

$$Z^{JK_4 \boxtimes \overline{JK_4}}_{m_1;\bar{e}} = \chi^{m6\times\overline{m6}}_{4,\frac{13}{8};1,0} + \chi^{m6\times\overline{m6}}_{4,\frac{13}{8};5,-3} + \chi^{m6\times\overline{m6}}_{7,\frac{1}{40};6,-\frac{2}{5}} + \chi^{m6\times\overline{m6}}_{7,\frac{1}{40};10,-\frac{7}{5}}$$

$$Z^{JK_4 \boxtimes \overline{JK_4}}_{m_1;\bar{m}} = 0$$

$$Z^{JK_4 \boxtimes \overline{JK_4}}_{m_1;\bar{m}_1} = 0$$

$$Z^{JK_4 \boxtimes \overline{JK_4}}_{m_1;\bar{q}} = 2\chi^{m6\times\overline{m6}}_{4,\frac{13}{8};3,-\frac{2}{3}} + 2\chi^{m6\times\overline{m6}}_{7,\frac{1}{40};8,-\frac{1}{15}}$$

$$Z_{q;\mathbf{1}}^{JK_4\boxtimes\overline{JK}_4} = \chi_{3,\frac{2}{3};1,0}^{m6\times\overline{m6}} + \chi_{3,\frac{2}{3};5,-3}^{m6\times\overline{m6}} + \chi_{8,\frac{1}{15};6,-\frac{2}{5}}^{m6\times\overline{m6}} + \chi_{8,\frac{1}{15};10,-\frac{7}{5}}^{m6\times\overline{m6}}$$

$$Z_{q;\bar{e}}^{JK_4\boxtimes\overline{JK}_4} = \chi_{3,\frac{2}{3};1,0}^{m6\times\overline{m6}} + \chi_{3,\frac{2}{3};5,-3}^{m6\times\overline{m6}} + \chi_{8,\frac{1}{15};6,-\frac{2}{5}}^{m6\times\overline{m6}} + \chi_{8,\frac{1}{15};10,-\frac{7}{5}}^{m6\times\overline{m6}}$$

$$Z_{q;\bar{m}}^{JK_4\boxtimes\overline{JK}_4} = 0$$

$$Z_{q;\bar{m}_1}^{JK_4\boxtimes\overline{JK}_4} = 0$$

$$Z_{q;\bar{q}}^{JK_4\boxtimes\overline{JK}_4} = 2\chi_{3,\frac{2}{3};3,-\frac{2}{3}}^{m6\times\overline{m6}} + 2\chi_{8,\frac{1}{15};8,-\frac{1}{15}}^{m6\times\overline{m6}}$$

$$\mathcal{A} = \mathbf{1} \oplus \bar{e}. \tag{H.5}$$

$$Z_{\mathbf{1};\mathbf{1}}^{JK_4\boxtimes\overline{JK}_4} = \textcolor{blue}{\chi_{1,0;1,0;1,0}^{m4\times m4\times\overline{U1}_4}} + \textcolor{red}{\chi_{1,0;3,\frac{1}{2};3,-\frac{1}{2}}^{m4\times m4\times\overline{U1}_4}} + \textcolor{red}{\chi_{3,\frac{1}{2};1,0;3,-\frac{1}{2}}^{m4\times m4\times\overline{U1}_4}} + \chi_{3,\frac{1}{2};3,\frac{1}{2};1,0}^{m4\times m4\times\overline{U1}_4}$$

$$Z_{\mathbf{1};\bar{e}}^{JK_4\boxtimes\overline{JK}_4} = \chi_{2,\frac{1}{16};2,\frac{1}{16};2,-\frac{1}{8}}^{m4\times m4\times\overline{U1}_4} + \chi_{2,\frac{1}{16};2,\frac{1}{16};4,-\frac{9}{8}}^{m4\times m4\times\overline{U1}_4}$$

$$Z_{\mathbf{1};\bar{m}}^{JK_4\boxtimes\overline{JK}_4} = 0$$

$$Z_{\mathbf{1};\bar{m}_1}^{JK_4\boxtimes\overline{JK}_4} = 0$$

$$Z_{\mathbf{1};\bar{q}}^{JK_4\boxtimes\overline{JK}_4} = 0$$

$$Z_{e;\mathbf{1}}^{JK_4\boxtimes\overline{JK}_4} = \chi_{2,\frac{1}{16};2,\frac{1}{16};2,-\frac{1}{8}}^{m4\times m4\times\overline{U1}_4} + \chi_{2,\frac{1}{16};2,\frac{1}{16};4,-\frac{9}{8}}^{m4\times m4\times\overline{U1}_4}$$

$$Z_{e;\bar{e}}^{JK_4\boxtimes\overline{JK}_4} = \textcolor{blue}{\chi_{1,0;1,0;1,0}^{m4\times m4\times\overline{U1}_4}} + \chi_{1,0;3,\frac{1}{2};3,-\frac{1}{2}}^{m4\times m4\times\overline{U1}_4} + \chi_{3,\frac{1}{2};1,0;3,-\frac{1}{2}}^{m4\times m4\times\overline{U1}_4} + \chi_{3,\frac{1}{2};3,\frac{1}{2};1,0}^{m4\times m4\times\overline{U1}_4}$$

$$Z_{e;\bar{m}}^{JK_4\boxtimes\overline{JK}_4} = 0$$

$$Z_{e;\bar{m}_1}^{JK_4\boxtimes\overline{JK}_4} = 0$$

$$Z_{e;\bar{q}}^{JK_4\boxtimes\overline{JK}_4} = 0$$

$$Z_{m;\mathbf{1}}^{JK_4\boxtimes\overline{JK}_4} = 0$$

$$Z_{m;\bar{e}}^{JK_4\boxtimes\overline{JK}_4} = 0$$

$$Z_{m;\bar{m}}^{JK_4\boxtimes\overline{JK}_4} = \chi_{2,\frac{1}{16};2,\frac{1}{16};2,-\frac{1}{8}}^{m4\times m4\times\overline{U1}_4} + \chi_{2,\frac{1}{16};2,\frac{1}{16};4,-\frac{9}{8}}^{m4\times m4\times\overline{U1}_4}$$

$$Z_{m;\bar{m}_1}^{JK_4\boxtimes\overline{JK}_4} = \chi_{1,0;1,0;3,-\frac{1}{2}}^{m4\times m4\times\overline{U1}_4} + \chi_{1,0;3,\frac{1}{2};1,0}^{m4\times m4\times\overline{U1}_4} + \chi_{3,\frac{1}{2};1,0;1,0}^{m4\times m4\times\overline{U1}_4} + \chi_{3,\frac{1}{2};3,\frac{1}{2};3,-\frac{1}{2}}^{m4\times m4\times\overline{U1}_4}$$

$$Z_{m;\bar{q}}^{JK_4\boxtimes\overline{JK}_4} = 0$$

$$Z_{m_1;\mathbf{1}}^{JK_4\boxtimes\overline{JK}_4} = 0$$

$$Z_{m_1;\bar{e}}^{JK_4\boxtimes\overline{JK}_4} = 0$$

$$Z_{m_1;\bar{m}}^{JK_4\boxtimes\overline{JK}_4} = \chi_{1,0;1,0;3,-\frac{1}{2}}^{m4\times m4\times\overline{U1}_4} + \chi_{1,0;3,\frac{1}{2};1,0}^{m4\times m4\times\overline{U1}_4} + \chi_{3,\frac{1}{2};1,0;1,0}^{m4\times m4\times\overline{U1}_4} + \chi_{3,\frac{1}{2};3,\frac{1}{2};3,-\frac{1}{2}}^{m4\times m4\times\overline{U1}_4}$$

$$Z_{m_1;\bar{m}_1}^{JK_4\boxtimes\overline{JK}_4} = \chi_{2,\frac{1}{16};2,\frac{1}{16};2,-\frac{1}{8}}^{m4\times m4\times\overline{U1}_4} + \chi_{2,\frac{1}{16};2,\frac{1}{16};4,-\frac{9}{8}}^{m4\times m4\times\overline{U1}_4}$$

$$Z_{m_1;\bar{q}}^{JK_4\boxtimes\overline{JK}_4} = 0$$

$$Z_{q;\mathbf{1}}^{JK_4\boxtimes\overline{JK}_4} = 0$$

$$Z_{q;\bar{e}}^{JK_4\boxtimes\overline{JK}_4} = 0$$

$$Z_{q;\bar{m}}^{JK_4\boxtimes\overline{JK}_4} = 0$$

$$Z_{q;\bar{m}_1}^{JK_4\boxtimes\overline{JK}_4} = 0$$

$$Z_{q;\bar{q}}^{JK_4\boxtimes\overline{JK}_4} = \textcolor{blue}{\chi_{1,0;1,0;1,0}^{m4\times m4\times\overline{U1}_4}} + \chi_{1,0;3,\frac{1}{2};3,-\frac{1}{2}}^{m4\times m4\times\overline{U1}_4} + \chi_{2,\frac{1}{16};2,\frac{1}{16};2,-\frac{1}{8}}^{m4\times m4\times\overline{U1}_4} + \chi_{2,\frac{1}{16};2,\frac{1}{16};4,-\frac{9}{8}}^{m4\times m4\times\overline{U1}_4}$$

$$+ \chi^{m4\times m4\times \overline{U1}_4}_{3,\frac{1}{2};1,0;3,-\frac{1}{2}} + \chi^{m4\times m4\times \overline{U1}_4}_{3,\frac{1}{2};3,\frac{1}{2};1,0}$$
$$\mathcal{A} = \mathbf{1} \oplus e\bar{e} \oplus q\bar{q}. \tag{H.6}$$

In the following, we list the gapless critical states with only one symmetric relevant operators of self-dual $S_3$ symmetric model or the self-dual $\mathsf{Rep}(S_3)$ symmetric model.

$$Z^{JK_4\boxtimes\overline{JK}_4}_{\mathbf{1};\mathbf{1}} = \chi^{m4\times\overline{m4}}_{1,0;1,0} + \chi^{m4\times\overline{m4}}_{3,\frac{1}{2};3,-\frac{1}{2}}$$

$$Z^{JK_4\boxtimes\overline{JK}_4}_{\mathbf{1};\bar{e}} = \chi^{m4\times\overline{m4}}_{2,\frac{1}{16};2,-\frac{1}{16}}$$

$$Z^{JK_4\boxtimes\overline{JK}_4}_{\mathbf{1};\bar{m}} = 0$$

$$Z^{JK_4\boxtimes\overline{JK}_4}_{\mathbf{1};\bar{m}_1} = 0$$

$$Z^{JK_4\boxtimes\overline{JK}_4}_{\mathbf{1};\bar{q}} = 0$$

$$Z^{JK_4\boxtimes\overline{JK}_4}_{e;\mathbf{1}} = \chi^{m4\times\overline{m4}}_{2,\frac{1}{16};2,-\frac{1}{16}}$$

$$Z^{JK_4\boxtimes\overline{JK}_4}_{e;\bar{e}} = \chi^{m4\times\overline{m4}}_{1,0;1,0} + \chi^{m4\times\overline{m4}}_{3,\frac{1}{2};3,-\frac{1}{2}}$$

$$Z^{JK_4\boxtimes\overline{JK}_4}_{e;\bar{m}} = 0$$

$$Z^{JK_4\boxtimes\overline{JK}_4}_{e;\bar{m}_1} = 0$$

$$Z^{JK_4\boxtimes\overline{JK}_4}_{e;\bar{q}} = 0$$

$$Z^{JK_4\boxtimes\overline{JK}_4}_{m;\mathbf{1}} = 0$$

$$Z^{JK_4\boxtimes\overline{JK}_4}_{m;\bar{e}} = 0$$

$$Z^{JK_4\boxtimes\overline{JK}_4}_{m;\bar{m}} = \chi^{m4\times\overline{m4}}_{2,\frac{1}{16};2,-\frac{1}{16}}$$

$$Z^{JK_4\boxtimes\overline{JK}_4}_{m;\bar{m}_1} = \chi^{m4\times\overline{m4}}_{1,0;3,-\frac{1}{2}} + \chi^{m4\times\overline{m4}}_{3,\frac{1}{2};1,0}$$

$$Z^{JK_4\boxtimes\overline{JK}_4}_{m;\bar{q}} = 0$$

$$Z^{JK_4\boxtimes\overline{JK}_4}_{m_1;\mathbf{1}} = 0$$

$$Z^{JK_4\boxtimes\overline{JK}_4}_{m_1;\bar{e}} = 0$$

$$Z^{JK_4\boxtimes\overline{JK}_4}_{m_1;\bar{m}} = \chi^{m4\times\overline{m4}}_{1,0;3,-\frac{1}{2}} + \chi^{m4\times\overline{m4}}_{3,\frac{1}{2};1,0}$$

$$Z^{JK_4\boxtimes\overline{JK}_4}_{m_1;\bar{m}_1} = \chi^{m4\times\overline{m4}}_{2,\frac{1}{16};2,-\frac{1}{16}}$$

$$Z^{JK_4\boxtimes\overline{JK}_4}_{m_1;\bar{q}} = 0$$

$$Z^{JK_4\boxtimes\overline{JK}_4}_{q;\mathbf{1}} = 0$$

$$Z^{JK_4\boxtimes\overline{JK}_4}_{q;\bar{e}} = 0$$

$$Z^{JK_4\boxtimes\overline{JK}_4}_{q;\bar{m}} = 0$$

$$Z^{JK_4\boxtimes\overline{JK}_4}_{q;\bar{m}_1} = 0$$

$$Z^{JK_4\boxtimes\overline{JK}_4}_{q;\bar{q}} = \chi^{m4\times\overline{m4}}_{1,0;1,0} + \chi^{m4\times\overline{m4}}_{2,\frac{1}{16};2,-\frac{1}{16}} + \chi^{m4\times\overline{m4}}_{3,\frac{1}{2};3,-\frac{1}{2}}$$

$$\mathcal{A} = \mathbf{1} \oplus e\bar{e} \oplus q\bar{q}. \tag{H.7}$$

$$Z^{JK_4\boxtimes\overline{JK}_4}_{\mathbf{1};\mathbf{1}} = \chi^{m6\times\overline{m6}}_{1,0;1,0} + \chi^{m6\times\overline{m6}}_{5,3;5,-3} + \chi^{m6\times\overline{m6}}_{6,\frac{2}{5};6,-\frac{2}{5}} + \chi^{m6\times\overline{m6}}_{10,\frac{7}{5};10,-\frac{7}{5}}$$

$$Z^{JK_4 \boxtimes \overline{JK}_4}_{1;\bar{e}} = \chi^{m6 \times \overline{m6}}_{1,0;5,-3} + \chi^{m6 \times \overline{m6}}_{5,3;1,0} + \chi^{m6 \times \overline{m6}}_{6,\frac{2}{5};10,-\frac{7}{5}} + \chi^{m6 \times \overline{m6}}_{10,\frac{7}{5};6,-\frac{2}{5}}$$

$$Z^{JK_4 \boxtimes \overline{JK}_4}_{1;\bar{m}} = 0$$

$$Z^{JK_4 \boxtimes \overline{JK}_4}_{1;\bar{m}_1} = 0$$

$$Z^{JK_4 \boxtimes \overline{JK}_4}_{1;\bar{q}} = \chi^{m6 \times \overline{m6}}_{1,0;3,-\frac{2}{3}} + \chi^{m6 \times \overline{m6}}_{5,3;3,-\frac{2}{3}} + \chi^{m6 \times \overline{m6}}_{6,\frac{2}{5};8,-\frac{1}{15}} + \chi^{m6 \times \overline{m6}}_{10,\frac{7}{5};8,-\frac{1}{15}}$$

$$Z^{JK_4 \boxtimes \overline{JK}_4}_{e;1} = \chi^{m6 \times \overline{m6}}_{1,0;5,-3} + \chi^{m6 \times \overline{m6}}_{5,3;1,0} + \chi^{m6 \times \overline{m6}}_{6,\frac{2}{5};10,-\frac{7}{5}} + \chi^{m6 \times \overline{m6}}_{10,\frac{7}{5};6,-\frac{2}{5}}$$

$$Z^{JK_4 \boxtimes \overline{JK}_4}_{e;\bar{e}} = \color{blue}{\chi^{m6 \times \overline{m6}}_{1,0;1,0}} + \chi^{m6 \times \overline{m6}}_{5,3;5,-3} + \chi^{m6 \times \overline{m6}}_{6,\frac{2}{5};6,-\frac{2}{5}} + \chi^{m6 \times \overline{m6}}_{10,\frac{7}{5};10,-\frac{7}{5}}$$

$$Z^{JK_4 \boxtimes \overline{JK}_4}_{e;\bar{m}} = 0$$

$$Z^{JK_4 \boxtimes \overline{JK}_4}_{e;\bar{m}_1} = 0$$

$$Z^{JK_4 \boxtimes \overline{JK}_4}_{e;\bar{q}} = \chi^{m6 \times \overline{m6}}_{1,0;3,-\frac{2}{3}} + \chi^{m6 \times \overline{m6}}_{5,3;3,-\frac{2}{3}} + \chi^{m6 \times \overline{m6}}_{6,\frac{2}{5};8,-\frac{1}{15}} + \chi^{m6 \times \overline{m6}}_{10,\frac{7}{5};8,-\frac{1}{15}}$$

$$Z^{JK_4 \boxtimes \overline{JK}_4}_{m;1} = 0$$

$$Z^{JK_4 \boxtimes \overline{JK}_4}_{m;\bar{e}} = 0$$

$$Z^{JK_4 \boxtimes \overline{JK}_4}_{m;\bar{m}} = \chi^{m6 \times \overline{m6}}_{2,\frac{1}{8};2,-\frac{1}{8}} + \chi^{m6 \times \overline{m6}}_{4,\frac{13}{8};4,-\frac{13}{8}} + \chi^{m6 \times \overline{m6}}_{7,\frac{1}{40};7,-\frac{1}{40}} + \chi^{m6 \times \overline{m6}}_{9,\frac{21}{40};9,-\frac{21}{40}}$$

$$Z^{JK_4 \boxtimes \overline{JK}_4}_{m;\bar{m}_1} = \chi^{m6 \times \overline{m6}}_{2,\frac{1}{8};4,-\frac{13}{8}} + \chi^{m6 \times \overline{m6}}_{4,\frac{13}{8};2,-\frac{1}{8}} + \chi^{m6 \times \overline{m6}}_{7,\frac{1}{40};9,-\frac{21}{40}} + \chi^{m6 \times \overline{m6}}_{9,\frac{21}{40};7,-\frac{1}{40}}$$

$$Z^{JK_4 \boxtimes \overline{JK}_4}_{m;\bar{q}} = 0$$

$$Z^{JK_4 \boxtimes \overline{JK}_4}_{m_1;1} = 0$$

$$Z^{JK_4 \boxtimes \overline{JK}_4}_{m_1;\bar{e}} = 0$$

$$Z^{JK_4 \boxtimes \overline{JK}_4}_{m_1;\bar{m}} = \chi^{m6 \times \overline{m6}}_{2,\frac{1}{8};4,-\frac{13}{8}} + \chi^{m6 \times \overline{m6}}_{4,\frac{13}{8};2,-\frac{1}{8}} + \chi^{m6 \times \overline{m6}}_{7,\frac{1}{40};9,-\frac{21}{40}} + \chi^{m6 \times \overline{m6}}_{9,\frac{21}{40};7,-\frac{1}{40}}$$

$$Z^{JK_4 \boxtimes \overline{JK}_4}_{m_1;\bar{m}_1} = \chi^{m6 \times \overline{m6}}_{2,\frac{1}{8};2,-\frac{1}{8}} + \chi^{m6 \times \overline{m6}}_{4,\frac{13}{8};4,-\frac{13}{8}} + \chi^{m6 \times \overline{m6}}_{7,\frac{1}{40};7,-\frac{1}{40}} + \chi^{m6 \times \overline{m6}}_{9,\frac{21}{40};9,-\frac{21}{40}}$$

$$Z^{JK_4 \boxtimes \overline{JK}_4}_{m_1;\bar{q}} = 0$$

$$Z^{JK_4 \boxtimes \overline{JK}_4}_{q;1} = \chi^{m6 \times \overline{m6}}_{3,\frac{2}{3};1,0} + \chi^{m6 \times \overline{m6}}_{3,\frac{2}{3};5,-3} + \chi^{m6 \times \overline{m6}}_{8,\frac{1}{15};6,-\frac{2}{5}} + \chi^{m6 \times \overline{m6}}_{8,\frac{1}{15};10,-\frac{7}{5}}$$

$$Z^{JK_4 \boxtimes \overline{JK}_4}_{q;\bar{e}} = \chi^{m6 \times \overline{m6}}_{3,\frac{2}{3};1,0} + \chi^{m6 \times \overline{m6}}_{3,\frac{2}{3};5,-3} + \chi^{m6 \times \overline{m6}}_{8,\frac{1}{15};6,-\frac{2}{5}} + \chi^{m6 \times \overline{m6}}_{8,\frac{1}{15};10,-\frac{7}{5}}$$

$$Z^{JK_4 \boxtimes \overline{JK}_4}_{q;\bar{m}} = 0$$

$$Z^{JK_4 \boxtimes \overline{JK}_4}_{q;\bar{m}_1} = 0$$

$$Z^{JK_4 \boxtimes \overline{JK}_4}_{q;\bar{q}} = 2\chi^{m6 \times \overline{m6}}_{3,\frac{2}{3};3,-\frac{2}{3}} + 2\chi^{m6 \times \overline{m6}}_{8,\frac{1}{15};8,-\frac{1}{15}}$$

$$\mathcal{A} = \mathbf{1} \oplus e\bar{e}. \tag{H.8}$$

$$Z^{JK_4 \boxtimes \overline{JK}_4}_{1;1} = \color{blue}{\chi^{m6 \times \overline{m6}}_{1,0;1,0}} + \chi^{m6 \times \overline{m6}}_{1,0;5,-3} + \chi^{m6 \times \overline{m6}}_{5,3;1,0} + \chi^{m6 \times \overline{m6}}_{5,3;5,-3} + \color{red}{\chi^{m6 \times \overline{m6}}_{6,\frac{2}{5};6,-\frac{2}{5}}}$$
$$+ \chi^{m6 \times \overline{m6}}_{6,\frac{2}{5};10,-\frac{7}{5}} + \chi^{m6 \times \overline{m6}}_{10,\frac{7}{5};6,-\frac{2}{5}} + \chi^{m6 \times \overline{m6}}_{10,\frac{7}{5};10,-\frac{7}{5}}$$

$$Z^{JK_4 \boxtimes \overline{JK}_4}_{1;\bar{e}} = \color{blue}{\chi^{m6 \times \overline{m6}}_{1,0;1,0}} + \chi^{m6 \times \overline{m6}}_{1,0;5,-3} + \chi^{m6 \times \overline{m6}}_{5,3;1,0} + \chi^{m6 \times \overline{m6}}_{5,3;5,-3} + \chi^{m6 \times \overline{m6}}_{6,\frac{2}{5};6,-\frac{2}{5}}$$
$$+ \chi^{m6 \times \overline{m6}}_{6,\frac{2}{5};10,-\frac{7}{5}} + \chi^{m6 \times \overline{m6}}_{10,\frac{7}{5};6,-\frac{2}{5}} + \chi^{m6 \times \overline{m6}}_{10,\frac{7}{5};10,-\frac{7}{5}}$$

$$Z^{JK_4 \boxtimes \overline{JK}_4}_{1;\bar{m}} = 0$$

$$Z_{\mathbf{1};\bar{m}_1}^{JK_4\boxtimes\overline{JK}_4} = 0$$

$$Z_{\mathbf{1};\bar{q}}^{JK_4\boxtimes\overline{JK}_4} = 2\chi_{1,0;3,-\frac{2}{3}}^{m6\times\overline{m6}} + 2\chi_{5,3;3,-\frac{2}{3}}^{m6\times\overline{m6}} + 2\chi_{6,\frac{2}{5};8,-\frac{1}{15}}^{m6\times\overline{m6}} + 2\chi_{10,\frac{7}{5};8,-\frac{1}{15}}^{m6\times\overline{m6}}$$

$$Z_{e;\mathbf{1}}^{JK_4\boxtimes\overline{JK}_4} = \color{blue}{\chi_{1,0;1,0}^{m6\times\overline{m6}}} \color{black}{+ \chi_{1,0;5,-3}^{m6\times\overline{m6}} + \chi_{5,3;1,0}^{m6\times\overline{m6}} + \chi_{5,3;5,-3}^{m6\times\overline{m6}} + \chi_{6,\frac{2}{5};6,-\frac{2}{5}}^{m6\times\overline{m6}}}$$
$$+ \chi_{6,\frac{2}{5};10,-\frac{7}{5}}^{m6\times\overline{m6}} + \chi_{10,\frac{7}{5};6,-\frac{2}{5}}^{m6\times\overline{m6}} + \chi_{10,\frac{7}{5};10,-\frac{7}{5}}^{m6\times\overline{m6}}$$

$$Z_{e;\bar{e}}^{JK_4\boxtimes\overline{JK}_4} = \color{blue}{\chi_{1,0;1,0}^{m6\times\overline{m6}}} \color{black}{+ \chi_{1,0;5,-3}^{m6\times\overline{m6}} + \chi_{5,3;1,0}^{m6\times\overline{m6}} + \chi_{5,3;5,-3}^{m6\times\overline{m6}} + \chi_{6,\frac{2}{5};6,-\frac{2}{5}}^{m6\times\overline{m6}}}$$
$$+ \chi_{6,\frac{2}{5};10,-\frac{7}{5}}^{m6\times\overline{m6}} + \chi_{10,\frac{7}{5};6,-\frac{2}{5}}^{m6\times\overline{m6}} + \chi_{10,\frac{7}{5};10,-\frac{7}{5}}^{m6\times\overline{m6}}$$

$$Z_{e;\bar{m}}^{JK_4\boxtimes\overline{JK}_4} = 0$$

$$Z_{e;\bar{m}_1}^{JK_4\boxtimes\overline{JK}_4} = 0$$

$$Z_{e;\bar{q}}^{JK_4\boxtimes\overline{JK}_4} = 2\chi_{1,0;3,-\frac{2}{3}}^{m6\times\overline{m6}} + 2\chi_{5,3;3,-\frac{2}{3}}^{m6\times\overline{m6}} + 2\chi_{6,\frac{2}{5};8,-\frac{1}{15}}^{m6\times\overline{m6}} + 2\chi_{10,\frac{7}{5};8,-\frac{1}{15}}^{m6\times\overline{m6}}$$

$$Z_{m;\mathbf{1}}^{JK_4\boxtimes\overline{JK}_4} = 0$$

$$Z_{m;\bar{e}}^{JK_4\boxtimes\overline{JK}_4} = 0$$

$$Z_{m;\bar{m}}^{JK_4\boxtimes\overline{JK}_4} = 0$$

$$Z_{m;\bar{m}_1}^{JK_4\boxtimes\overline{JK}_4} = 0$$

$$Z_{m;\bar{q}}^{JK_4\boxtimes\overline{JK}_4} = 0$$

$$Z_{m_1;\mathbf{1}}^{JK_4\boxtimes\overline{JK}_4} = 0$$

$$Z_{m_1;\bar{e}}^{JK_4\boxtimes\overline{JK}_4} = 0$$

$$Z_{m_1;\bar{m}}^{JK_4\boxtimes\overline{JK}_4} = 0$$

$$Z_{m_1;\bar{m}_1}^{JK_4\boxtimes\overline{JK}_4} = 0$$

$$Z_{m_1;\bar{q}}^{JK_4\boxtimes\overline{JK}_4} = 0$$

$$Z_{q;\mathbf{1}}^{JK_4\boxtimes\overline{JK}_4} = 2\chi_{3,\frac{2}{3};1,0}^{m6\times\overline{m6}} + 2\chi_{3,\frac{2}{3};5,-3}^{m6\times\overline{m6}} + 2\chi_{8,\frac{1}{15};6,-\frac{2}{5}}^{m6\times\overline{m6}} + 2\chi_{8,\frac{1}{15};10,-\frac{7}{5}}^{m6\times\overline{m6}}$$

$$Z_{q;\bar{e}}^{JK_4\boxtimes\overline{JK}_4} = 2\chi_{3,\frac{2}{3};1,0}^{m6\times\overline{m6}} + 2\chi_{3,\frac{2}{3};5,-3}^{m6\times\overline{m6}} + 2\chi_{8,\frac{1}{15};6,-\frac{2}{5}}^{m6\times\overline{m6}} + 2\chi_{8,\frac{1}{15};10,-\frac{7}{5}}^{m6\times\overline{m6}}$$

$$Z_{q;\bar{m}}^{JK_4\boxtimes\overline{JK}_4} = 0$$

$$Z_{q;\bar{m}_1}^{JK_4\boxtimes\overline{JK}_4} = 0$$

$$Z_{q;\bar{q}}^{JK_4\boxtimes\overline{JK}_4} = 4\chi_{3,\frac{2}{3};3,-\frac{2}{3}}^{m6\times\overline{m6}} + 4\chi_{8,\frac{1}{15};8,-\frac{1}{15}}^{m6\times\overline{m6}}$$

$$\mathcal{A} = \mathbf{1} \oplus e \oplus \bar{e} \oplus e\bar{e}. \tag{H.9}$$

$$Z_{\mathbf{1};\mathbf{1}}^{JK_4\boxtimes\overline{JK}_4} = \color{blue}{\chi_{1,0;1,0}^{m7\times\overline{m7}}} \color{black}{+} \color{red}{\chi_{3,\frac{5}{7};3,-\frac{5}{7}}^{m7\times\overline{m7}}} \color{black}{+ \chi_{5,\frac{22}{7};5,-\frac{22}{7}}^{m7\times\overline{m7}}}$$

$$Z_{\mathbf{1};\bar{e}}^{JK_4\boxtimes\overline{JK}_4} = \chi_{2,\frac{1}{7};5,-\frac{22}{7}}^{m7\times\overline{m7}} + \chi_{4,\frac{12}{7};3,-\frac{5}{7}}^{m7\times\overline{m7}} + \chi_{6,5;1,0}^{m7\times\overline{m7}}$$

$$Z_{\mathbf{1};\bar{m}}^{JK_4\boxtimes\overline{JK}_4} = \chi_{8,\frac{1}{56};2,-\frac{1}{7}}^{m7\times\overline{m7}} + \chi_{10,\frac{33}{56};4,-\frac{12}{7}}^{m7\times\overline{m7}} + \chi_{12,\frac{23}{8};6,-5}^{m7\times\overline{m7}}$$

$$Z_{\mathbf{1};\bar{m}_1}^{JK_4\boxtimes\overline{JK}_4} = \chi_{7,\frac{3}{8};6,-5}^{m7\times\overline{m7}} + \chi_{9,\frac{5}{56};4,-\frac{12}{7}}^{m7\times\overline{m7}} + \chi_{11,\frac{85}{56};2,-\frac{1}{7}}^{m7\times\overline{m7}}$$

$$Z_{\mathbf{1};\bar{q}}^{JK_4\boxtimes\overline{JK}_4} = \chi_{13,\frac{4}{3};1,0}^{m7\times\overline{m7}} + \chi_{14,\frac{10}{21};5,-\frac{22}{7}}^{m7\times\overline{m7}} + \chi_{15,\frac{1}{21};3,-\frac{5}{7}}^{m7\times\overline{m7}}$$

$$Z^{JK_4\boxtimes\overline{JK_4}}_{e;\mathbf{1}} = \chi^{m7\times\overline{m7}}_{1,0;6,-5} + \chi^{m7\times\overline{m7}}_{3,\frac{5}{7};4,-\frac{12}{7}} + \chi^{m7\times\overline{m7}}_{5,\frac{22}{7};2,-\frac{1}{7}}$$

$$Z^{JK_4\boxtimes\overline{JK_4}}_{e;\bar{e}} = \chi^{m7\times\overline{m7}}_{2,\frac{1}{7};2,-\frac{1}{7}} + \chi^{m7\times\overline{m7}}_{4,\frac{12}{7};4,-\frac{12}{7}} + \chi^{m7\times\overline{m7}}_{6,5;6,-5}$$

$$Z^{JK_4\boxtimes\overline{JK_4}}_{e;\bar{m}} = \chi^{m7\times\overline{m7}}_{8,\frac{1}{56};5,-\frac{22}{7}} + \chi^{m7\times\overline{m7}}_{10,\frac{33}{56};3,-\frac{5}{7}} + \chi^{m7\times\overline{m7}}_{12,\frac{23}{8};1,0}$$

$$Z^{JK_4\boxtimes\overline{JK_4}}_{e;\bar{m}_1} = \chi^{m7\times\overline{m7}}_{7,\frac{3}{8};1,0} + \chi^{m7\times\overline{m7}}_{9,\frac{5}{56};3,-\frac{5}{7}} + \chi^{m7\times\overline{m7}}_{11,\frac{85}{56};5,-\frac{22}{7}}$$

$$Z^{JK_4\boxtimes\overline{JK_4}}_{e;\bar{q}} = \chi^{m7\times\overline{m7}}_{13,\frac{4}{3};6,-5} + \chi^{m7\times\overline{m7}}_{14,\frac{10}{21};2,-\frac{1}{7}} + \chi^{m7\times\overline{m7}}_{15,\frac{1}{21};4,-\frac{12}{7}}$$

$$Z^{JK_4\boxtimes\overline{JK_4}}_{m;\mathbf{1}} = \chi^{m7\times\overline{m7}}_{2,\frac{1}{7};8,-\frac{1}{56}} + \chi^{m7\times\overline{m7}}_{4,\frac{12}{7};10,-\frac{33}{56}} + \chi^{m7\times\overline{m7}}_{6,5;12,-\frac{23}{8}}$$

$$Z^{JK_4\boxtimes\overline{JK_4}}_{m;\bar{e}} = \chi^{m7\times\overline{m7}}_{1,0;12,-\frac{23}{8}} + \chi^{m7\times\overline{m7}}_{3,\frac{5}{7};10,-\frac{33}{56}} + \chi^{m7\times\overline{m7}}_{5,\frac{22}{7};8,-\frac{1}{56}}$$

$$Z^{JK_4\boxtimes\overline{JK_4}}_{m;\bar{m}} = \chi^{m7\times\overline{m7}}_{7,\frac{3}{8};7,-\frac{3}{8}} + \chi^{m7\times\overline{m7}}_{9,\frac{5}{56};9,-\frac{5}{56}} + \chi^{m7\times\overline{m7}}_{11,\frac{85}{56};11,-\frac{85}{56}}$$

$$Z^{JK_4\boxtimes\overline{JK_4}}_{m;\bar{m}_1} = \chi^{m7\times\overline{m7}}_{8,\frac{1}{56};11,-\frac{85}{56}} + \chi^{m7\times\overline{m7}}_{10,\frac{33}{56};9,-\frac{5}{56}} + \chi^{m7\times\overline{m7}}_{12,\frac{23}{8};7,-\frac{3}{8}}$$

$$Z^{JK_4\boxtimes\overline{JK_4}}_{m;\bar{q}} = \chi^{m7\times\overline{m7}}_{13,\frac{4}{3};12,-\frac{23}{8}} + \chi^{m7\times\overline{m7}}_{14,\frac{10}{21};8,-\frac{1}{56}} + \chi^{m7\times\overline{m7}}_{15,\frac{1}{21};10,-\frac{33}{56}}$$

$$Z^{JK_4\boxtimes\overline{JK_4}}_{m_1;\mathbf{1}} = \chi^{m7\times\overline{m7}}_{2,\frac{1}{7};11,-\frac{85}{56}} + \chi^{m7\times\overline{m7}}_{4,\frac{12}{7};9,-\frac{5}{56}} + \chi^{m7\times\overline{m7}}_{6,5;7,-\frac{3}{8}}$$

$$Z^{JK_4\boxtimes\overline{JK_4}}_{m_1;\bar{e}} = \chi^{m7\times\overline{m7}}_{1,0;7,-\frac{3}{8}} + \chi^{m7\times\overline{m7}}_{3,\frac{5}{7};9,-\frac{5}{56}} + \chi^{m7\times\overline{m7}}_{5,\frac{22}{7};11,-\frac{85}{56}}$$

$$Z^{JK_4\boxtimes\overline{JK_4}}_{m_1;\bar{m}} = \chi^{m7\times\overline{m7}}_{7,\frac{3}{8};12,-\frac{23}{8}} + \chi^{m7\times\overline{m7}}_{9,\frac{5}{56};10,-\frac{33}{56}} + \chi^{m7\times\overline{m7}}_{11,\frac{85}{56};8,-\frac{1}{56}}$$

$$Z^{JK_4\boxtimes\overline{JK_4}}_{m_1;\bar{m}_1} = \chi^{m7\times\overline{m7}}_{8,\frac{1}{56};8,-\frac{1}{56}} + \chi^{m7\times\overline{m7}}_{10,\frac{33}{56};10,-\frac{33}{56}} + \chi^{m7\times\overline{m7}}_{12,\frac{23}{8};12,-\frac{23}{8}}$$

$$Z^{JK_4\boxtimes\overline{JK_4}}_{m_1;\bar{q}} = \chi^{m7\times\overline{m7}}_{13,\frac{4}{3};7,-\frac{3}{8}} + \chi^{m7\times\overline{m7}}_{14,\frac{10}{21};11,-\frac{85}{56}} + \chi^{m7\times\overline{m7}}_{15,\frac{1}{21};9,-\frac{5}{56}}$$

$$Z^{JK_4\boxtimes\overline{JK_4}}_{q;\mathbf{1}} = \chi^{m7\times\overline{m7}}_{1,0;13,-\frac{4}{3}} + \chi^{m7\times\overline{m7}}_{3,\frac{5}{7};15,-\frac{1}{21}} + \chi^{m7\times\overline{m7}}_{5,\frac{22}{7};14,-\frac{10}{21}}$$

$$Z^{JK_4\boxtimes\overline{JK_4}}_{q;\bar{e}} = \chi^{m7\times\overline{m7}}_{2,\frac{1}{7};14,-\frac{10}{21}} + \chi^{m7\times\overline{m7}}_{4,\frac{12}{7};15,-\frac{1}{21}} + \chi^{m7\times\overline{m7}}_{6,5;13,-\frac{4}{3}}$$

$$Z^{JK_4\boxtimes\overline{JK_4}}_{q;\bar{m}} = \chi^{m7\times\overline{m7}}_{8,\frac{1}{56};14,-\frac{10}{21}} + \chi^{m7\times\overline{m7}}_{10,\frac{33}{56};15,-\frac{1}{21}} + \chi^{m7\times\overline{m7}}_{12,\frac{23}{8};13,-\frac{4}{3}}$$

$$Z^{JK_4\boxtimes\overline{JK_4}}_{q;\bar{m}_1} = \chi^{m7\times\overline{m7}}_{7,\frac{3}{8};13,-\frac{4}{3}} + \chi^{m7\times\overline{m7}}_{9,\frac{5}{56};15,-\frac{1}{21}} + \chi^{m7\times\overline{m7}}_{11,\frac{85}{56};14,-\frac{10}{21}}$$

$$Z^{JK_4\boxtimes\overline{JK_4}}_{q;\bar{q}} = \chi^{m7\times\overline{m7}}_{13,\frac{4}{3};13,-\frac{4}{3}} + \chi^{m7\times\overline{m7}}_{14,\frac{10}{21};14,-\frac{10}{21}} + \chi^{m7\times\overline{m7}}_{15,\frac{1}{21};15,-\frac{1}{21}}$$

$$\mathcal{A} = \mathbf{1}. \tag{H.10}$$

$$Z^{JK_4\boxtimes\overline{JK_4}}_{\mathbf{1};\mathbf{1}} = \color{blue}{\chi^{m7\times\overline{m7}}_{1,0;1,0}} + \color{red}{\chi^{m7\times\overline{m7}}_{3,\frac{5}{7};3,-\frac{5}{7}}} + \chi^{m7\times\overline{m7}}_{5,\frac{22}{7};5,-\frac{22}{7}}$$

$$Z^{JK_4\boxtimes\overline{JK_4}}_{\mathbf{1};\bar{e}} = \chi^{m7\times\overline{m7}}_{2,\frac{1}{7};5,-\frac{22}{7}} + \chi^{m7\times\overline{m7}}_{4,\frac{12}{7};3,-\frac{5}{7}} + \chi^{m7\times\overline{m7}}_{6,5;1,0}$$

$$Z^{JK_4\boxtimes\overline{JK_4}}_{\mathbf{1};\bar{m}} = \chi^{m7\times\overline{m7}}_{8,\frac{1}{56};5,-\frac{22}{7}} + \chi^{m7\times\overline{m7}}_{10,\frac{33}{56};3,-\frac{5}{7}} + \chi^{m7\times\overline{m7}}_{12,\frac{23}{8};1,0}$$

$$Z^{JK_4\boxtimes\overline{JK_4}}_{\mathbf{1};\bar{m}_1} = \chi^{m7\times\overline{m7}}_{7,\frac{3}{8};1,0} + \chi^{m7\times\overline{m7}}_{9,\frac{5}{56};3,-\frac{5}{7}} + \chi^{m7\times\overline{m7}}_{11,\frac{85}{56};5,-\frac{22}{7}}$$

$$Z^{JK_4\boxtimes\overline{JK_4}}_{\mathbf{1};\bar{q}} = \chi^{m7\times\overline{m7}}_{13,\frac{4}{3};1,0} + \chi^{m7\times\overline{m7}}_{14,\frac{10}{21};5,-\frac{22}{7}} + \chi^{m7\times\overline{m7}}_{15,\frac{1}{21};3,-\frac{5}{7}}$$

$$Z^{JK_4\boxtimes\overline{JK_4}}_{e;\mathbf{1}} = \chi^{m7\times\overline{m7}}_{1,0;6,-5} + \chi^{m7\times\overline{m7}}_{3,\frac{5}{7};4,-\frac{12}{7}} + \chi^{m7\times\overline{m7}}_{5,\frac{22}{7};2,-\frac{1}{7}}$$

$$Z^{JK_4\boxtimes\overline{JK_4}}_{e;\bar{e}} = \chi^{m7\times\overline{m7}}_{2,\frac{1}{7};2,-\frac{1}{7}} + \chi^{m7\times\overline{m7}}_{4,\frac{12}{7};4,-\frac{12}{7}} + \chi^{m7\times\overline{m7}}_{6,5;6,-5}$$

$$Z_{e;\bar{m}}^{JK_4\boxtimes\overline{JK_4}} = \chi_{8,\frac{1}{56};2,-\frac{1}{7}}^{m7\times\overline{m7}} + \chi_{10,\frac{33}{56};4,-\frac{12}{7}}^{m7\times\overline{m7}} + \chi_{12,\frac{23}{8};6,-5}^{m7\times\overline{m7}}$$

$$Z_{e;\bar{m}_1}^{JK_4\boxtimes\overline{JK_4}} = \chi_{7,\frac{3}{8};6,-5}^{m7\times\overline{m7}} + \chi_{9,\frac{5}{56};4,-\frac{12}{7}}^{m7\times\overline{m7}} + \chi_{11,\frac{85}{56};2,-\frac{1}{7}}^{m7\times\overline{m7}}$$

$$Z_{e;\bar{q}}^{JK_4\boxtimes\overline{JK_4}} = \chi_{13,\frac{4}{3};6,-5}^{m7\times\overline{m7}} + \chi_{14,\frac{10}{21};2,-\frac{1}{7}}^{m7\times\overline{m7}} + \chi_{15,\frac{1}{21};4,-\frac{12}{7}}^{m7\times\overline{m7}}$$

$$Z_{m;\mathbf{1}}^{JK_4\boxtimes\overline{JK_4}} = \chi_{1,0;12,-\frac{23}{8}}^{m7\times\overline{m7}} + \chi_{3,\frac{5}{7};10,-\frac{33}{56}}^{m7\times\overline{m7}} + \chi_{5,\frac{22}{7};8,-\frac{1}{56}}^{m7\times\overline{m7}}$$

$$Z_{m;\bar{e}}^{JK_4\boxtimes\overline{JK_4}} = \chi_{2,\frac{1}{7};8,-\frac{1}{56}}^{m7\times\overline{m7}} + \chi_{4,\frac{12}{7};10,-\frac{33}{56}}^{m7\times\overline{m7}} + \chi_{6,5;12,-\frac{23}{8}}^{m7\times\overline{m7}}$$

$$Z_{m;\bar{m}}^{JK_4\boxtimes\overline{JK_4}} = \chi_{8,\frac{1}{56};8,-\frac{1}{56}}^{m7\times\overline{m7}} + \chi_{10,\frac{33}{56};10,-\frac{33}{56}}^{m7\times\overline{m7}} + \chi_{12,\frac{23}{8};12,-\frac{23}{8}}^{m7\times\overline{m7}}$$

$$Z_{m;\bar{m}_1}^{JK_4\boxtimes\overline{JK_4}} = \chi_{7,\frac{3}{8};12,-\frac{23}{8}}^{m7\times\overline{m7}} + \chi_{9,\frac{5}{56};10,-\frac{33}{56}}^{m7\times\overline{m7}} + \chi_{11,\frac{85}{56};8,-\frac{1}{56}}^{m7\times\overline{m7}}$$

$$Z_{m;\bar{q}}^{JK_4\boxtimes\overline{JK_4}} = \chi_{13,\frac{4}{3};12,-\frac{23}{8}}^{m7\times\overline{m7}} + \chi_{14,\frac{10}{21};8,-\frac{1}{56}}^{m7\times\overline{m7}} + \chi_{15,\frac{1}{21};10,-\frac{33}{56}}^{m7\times\overline{m7}}$$

$$Z_{m_1;\mathbf{1}}^{JK_4\boxtimes\overline{JK_4}} = \chi_{1,0;7,-\frac{3}{8}}^{m7\times\overline{m7}} + \chi_{3,\frac{5}{7};9,-\frac{5}{56}}^{m7\times\overline{m7}} + \chi_{5,\frac{22}{7};11,-\frac{85}{56}}^{m7\times\overline{m7}}$$

$$Z_{m_1;\bar{e}}^{JK_4\boxtimes\overline{JK_4}} = \chi_{2,\frac{1}{7};11,-\frac{85}{56}}^{m7\times\overline{m7}} + \chi_{4,\frac{12}{7};9,-\frac{5}{56}}^{m7\times\overline{m7}} + \chi_{6,5;7,-\frac{3}{8}}^{m7\times\overline{m7}}$$

$$Z_{m_1;\bar{m}}^{JK_4\boxtimes\overline{JK_4}} = \chi_{8,\frac{1}{56};11,-\frac{85}{56}}^{m7\times\overline{m7}} + \chi_{10,\frac{33}{56};9,-\frac{5}{56}}^{m7\times\overline{m7}} + \chi_{12,\frac{23}{8};7,-\frac{3}{8}}^{m7\times\overline{m7}}$$

$$Z_{m_1;\bar{m}_1}^{JK_4\boxtimes\overline{JK_4}} = \chi_{7,\frac{3}{8};7,-\frac{3}{8}}^{m7\times\overline{m7}} + \chi_{9,\frac{5}{56};9,-\frac{5}{56}}^{m7\times\overline{m7}} + \chi_{11,\frac{85}{56};11,-\frac{85}{56}}^{m7\times\overline{m7}}$$

$$Z_{m_1;\bar{q}}^{JK_4\boxtimes\overline{JK_4}} = \chi_{13,\frac{4}{3};7,-\frac{3}{8}}^{m7\times\overline{m7}} + \chi_{14,\frac{10}{21};11,-\frac{85}{56}}^{m7\times\overline{m7}} + \chi_{15,\frac{1}{21};9,-\frac{5}{56}}^{m7\times\overline{m7}}$$

$$Z_{q;\mathbf{1}}^{JK_4\boxtimes\overline{JK_4}} = \chi_{1,0;13,-\frac{4}{3}}^{m7\times\overline{m7}} + \chi_{3,\frac{5}{7};15,-\frac{1}{21}}^{m7\times\overline{m7}} + \chi_{5,\frac{22}{7};14,-\frac{10}{21}}^{m7\times\overline{m7}}$$

$$Z_{q;\bar{e}}^{JK_4\boxtimes\overline{JK_4}} = \chi_{2,\frac{1}{7};14,-\frac{10}{21}}^{m7\times\overline{m7}} + \chi_{4,\frac{12}{7};15,-\frac{1}{21}}^{m7\times\overline{m7}} + \chi_{6,5;13,-\frac{4}{3}}^{m7\times\overline{m7}}$$

$$Z_{q;\bar{m}}^{JK_4\boxtimes\overline{JK_4}} = \chi_{8,\frac{1}{56};14,-\frac{10}{21}}^{m7\times\overline{m7}} + \chi_{10,\frac{33}{56};15,-\frac{1}{21}}^{m7\times\overline{m7}} + \chi_{12,\frac{23}{8};13,-\frac{4}{3}}^{m7\times\overline{m7}}$$

$$Z_{q;\bar{m}_1}^{JK_4\boxtimes\overline{JK_4}} = \chi_{7,\frac{3}{8};13,-\frac{4}{3}}^{m7\times\overline{m7}} + \chi_{9,\frac{5}{56};15,-\frac{1}{21}}^{m7\times\overline{m7}} + \chi_{11,\frac{85}{56};14,-\frac{10}{21}}^{m7\times\overline{m7}}$$

$$Z_{q;\bar{q}}^{JK_4\boxtimes\overline{JK_4}} = \chi_{13,\frac{4}{3};13,-\frac{4}{3}}^{m7\times\overline{m7}} + \chi_{14,\frac{10}{21};14,-\frac{10}{21}}^{m7\times\overline{m7}} + \chi_{15,\frac{1}{21};15,-\frac{1}{21}}^{m7\times\overline{m7}}$$

$$\mathcal{A} = \mathbf{1}. \tag{H.11}$$

$$Z_{\mathbf{1};\mathbf{1}}^{JK_4\boxtimes\overline{JK_4}} = \chi_{1,0;1,0}^{m7\times\overline{m7}} + \chi_{2,\frac{1}{7};5,-\frac{22}{7}}^{m7\times\overline{m7}} + \chi_{3,\frac{5}{7};3,-\frac{5}{7}}^{m7\times\overline{m7}} + \chi_{4,\frac{12}{7};3,-\frac{5}{7}}^{m7\times\overline{m7}} + \chi_{5,\frac{22}{7};5,-\frac{22}{7}}^{m7\times\overline{m7}} + \chi_{6,5;1,0}^{m7\times\overline{m7}}$$

$$Z_{\mathbf{1};\bar{e}}^{JK_4\boxtimes\overline{JK_4}} = \chi_{1,0;1,0}^{m7\times\overline{m7}} + \chi_{2,\frac{1}{7};5,-\frac{22}{7}}^{m7\times\overline{m7}} + \chi_{3,\frac{5}{7};3,-\frac{5}{7}}^{m7\times\overline{m7}} + \chi_{4,\frac{12}{7};3,-\frac{5}{7}}^{m7\times\overline{m7}} + \chi_{5,\frac{22}{7};5,-\frac{22}{7}}^{m7\times\overline{m7}} + \chi_{6,5;1,0}^{m7\times\overline{m7}}$$

$$Z_{\mathbf{1};\bar{m}}^{JK_4\boxtimes\overline{JK_4}} = 0$$

$$Z_{\mathbf{1};\bar{m}_1}^{JK_4\boxtimes\overline{JK_4}} = 0$$

$$Z_{\mathbf{1};\bar{q}}^{JK_4\boxtimes\overline{JK_4}} = 2\chi_{13,\frac{4}{3};1,0}^{m7\times\overline{m7}} + 2\chi_{14,\frac{10}{21};5,-\frac{22}{7}}^{m7\times\overline{m7}} + 2\chi_{15,\frac{1}{21};3,-\frac{5}{7}}^{m7\times\overline{m7}}$$

$$Z_{e;\mathbf{1}}^{JK_4\boxtimes\overline{JK_4}} = \chi_{1,0;6,-5}^{m7\times\overline{m7}} + \chi_{2,\frac{1}{7};2,-\frac{1}{7}}^{m7\times\overline{m7}} + \chi_{3,\frac{5}{7};4,-\frac{12}{7}}^{m7\times\overline{m7}} + \chi_{4,\frac{12}{7};4,-\frac{12}{7}}^{m7\times\overline{m7}} + \chi_{5,\frac{22}{7};2,-\frac{1}{7}}^{m7\times\overline{m7}} + \chi_{6,5;6,-5}^{m7\times\overline{m7}}$$

$$Z_{e;\bar{e}}^{JK_4\boxtimes\overline{JK_4}} = \chi_{1,0;6,-5}^{m7\times\overline{m7}} + \chi_{2,\frac{1}{7};2,-\frac{1}{7}}^{m7\times\overline{m7}} + \chi_{3,\frac{5}{7};4,-\frac{12}{7}}^{m7\times\overline{m7}} + \chi_{4,\frac{12}{7};4,-\frac{12}{7}}^{m7\times\overline{m7}} + \chi_{5,\frac{22}{7};2,-\frac{1}{7}}^{m7\times\overline{m7}} + \chi_{6,5;6,-5}^{m7\times\overline{m7}}$$

$$Z_{e;\bar{m}}^{JK_4\boxtimes\overline{JK_4}} = 0$$

$$Z_{e;\bar{m}_1}^{JK_4\boxtimes\overline{JK_4}} = 0$$

$$Z_{e;\bar{q}}^{JK_4 \boxtimes \overline{JK_4}} = 2\chi_{13,\frac{4}{3};6,-5}^{m7\times\overline{m7}} + 2\chi_{14,\frac{10}{21};2,-\frac{1}{7}}^{m7\times\overline{m7}} + 2\chi_{15,\frac{1}{21};4,-\frac{12}{7}}^{m7\times\overline{m7}}$$

$$Z_{m;\mathbf{1}}^{JK_4 \boxtimes \overline{JK_4}} = \chi_{1,0;12,-\frac{23}{8}}^{m7\times\overline{m7}} + \chi_{2,\frac{1}{7};8,-\frac{1}{56}}^{m7\times\overline{m7}} + \chi_{3,\frac{5}{7};10,-\frac{33}{56}}^{m7\times\overline{m7}} + \chi_{4,\frac{12}{7};10,-\frac{33}{56}}^{m7\times\overline{m7}} + \chi_{5,\frac{22}{7};8,-\frac{1}{56}}^{m7\times\overline{m7}} + \chi_{6,5;12,-\frac{23}{8}}^{m7\times\overline{m7}}$$

$$Z_{m;\bar{e}}^{JK_4 \boxtimes \overline{JK_4}} = \chi_{1,0;12,-\frac{23}{8}}^{m7\times\overline{m7}} + \chi_{2,\frac{1}{7};8,-\frac{1}{56}}^{m7\times\overline{m7}} + \chi_{3,\frac{5}{7};10,-\frac{33}{56}}^{m7\times\overline{m7}} + \chi_{4,\frac{12}{7};10,-\frac{33}{56}}^{m7\times\overline{m7}} + \chi_{5,\frac{22}{7};8,-\frac{1}{56}}^{m7\times\overline{m7}} + \chi_{6,5;12,-\frac{23}{8}}^{m7\times\overline{m7}}$$

$$Z_{m;\bar{m}}^{JK_4 \boxtimes \overline{JK_4}} = 0$$

$$Z_{m;\bar{m}_1}^{JK_4 \boxtimes \overline{JK_4}} = 0$$

$$Z_{m;\bar{q}}^{JK_4 \boxtimes \overline{JK_4}} = 2\chi_{13,\frac{4}{3};12,-\frac{23}{8}}^{m7\times\overline{m7}} + 2\chi_{14,\frac{10}{21};8,-\frac{1}{56}}^{m7\times\overline{m7}} + 2\chi_{15,\frac{1}{21};10,-\frac{33}{56}}^{m7\times\overline{m7}}$$

$$Z_{m_1;\mathbf{1}}^{JK_4 \boxtimes \overline{JK_4}} = \chi_{1,0;7,-\frac{3}{8}}^{m7\times\overline{m7}} + \chi_{2,\frac{1}{7};11,-\frac{85}{56}}^{m7\times\overline{m7}} + \chi_{3,\frac{5}{7};9,-\frac{5}{56}}^{m7\times\overline{m7}} + \chi_{4,\frac{12}{7};9,-\frac{5}{56}}^{m7\times\overline{m7}} + \chi_{5,\frac{22}{7};11,-\frac{85}{56}}^{m7\times\overline{m7}} + \chi_{6,5;7,-\frac{3}{8}}^{m7\times\overline{m7}}$$

$$Z_{m_1;\bar{e}}^{JK_4 \boxtimes \overline{JK_4}} = \chi_{1,0;7,-\frac{3}{8}}^{m7\times\overline{m7}} + \chi_{2,\frac{1}{7};11,-\frac{85}{56}}^{m7\times\overline{m7}} + \chi_{3,\frac{5}{7};9,-\frac{5}{56}}^{m7\times\overline{m7}} + \chi_{4,\frac{12}{7};9,-\frac{5}{56}}^{m7\times\overline{m7}} + \chi_{5,\frac{22}{7};11,-\frac{85}{56}}^{m7\times\overline{m7}} + \chi_{6,5;7,-\frac{3}{8}}^{m7\times\overline{m7}}$$

$$Z_{m_1;\bar{m}}^{JK_4 \boxtimes \overline{JK_4}} = 0$$

$$Z_{m_1;\bar{m}_1}^{JK_4 \boxtimes \overline{JK_4}} = 0$$

$$Z_{m_1;\bar{q}}^{JK_4 \boxtimes \overline{JK_4}} = 2\chi_{13,\frac{4}{3};7,-\frac{3}{8}}^{m7\times\overline{m7}} + 2\chi_{14,\frac{10}{21};11,-\frac{85}{56}}^{m7\times\overline{m7}} + 2\chi_{15,\frac{1}{21};9,-\frac{5}{56}}^{m7\times\overline{m7}}$$

$$Z_{q;\mathbf{1}}^{JK_4 \boxtimes \overline{JK_4}} = \chi_{1,0;13,-\frac{4}{3}}^{m7\times\overline{m7}} + \chi_{2,\frac{1}{7};14,-\frac{10}{21}}^{m7\times\overline{m7}} + \chi_{3,\frac{5}{7};15,-\frac{1}{21}}^{m7\times\overline{m7}} + \chi_{4,\frac{12}{7};15,-\frac{1}{21}}^{m7\times\overline{m7}} + \chi_{5,\frac{22}{7};14,-\frac{10}{21}}^{m7\times\overline{m7}} + \chi_{6,5;13,-\frac{4}{3}}^{m7\times\overline{m7}}$$

$$Z_{q;\bar{e}}^{JK_4 \boxtimes \overline{JK_4}} = \chi_{1,0;13,-\frac{4}{3}}^{m7\times\overline{m7}} + \chi_{2,\frac{1}{7};14,-\frac{10}{21}}^{m7\times\overline{m7}} + \chi_{3,\frac{5}{7};15,-\frac{1}{21}}^{m7\times\overline{m7}} + \chi_{4,\frac{12}{7};15,-\frac{1}{21}}^{m7\times\overline{m7}} + \chi_{5,\frac{22}{7};14,-\frac{10}{21}}^{m7\times\overline{m7}} + \chi_{6,5;13,-\frac{4}{3}}^{m7\times\overline{m7}}$$

$$Z_{q;\bar{m}}^{JK_4 \boxtimes \overline{JK_4}} = 0$$

$$Z_{q;\bar{m}_1}^{JK_4 \boxtimes \overline{JK_4}} = 0$$

$$Z_{q;\bar{q}}^{JK_4 \boxtimes \overline{JK_4}} = 2\chi_{13,\frac{4}{3};13,-\frac{4}{3}}^{m7\times\overline{m7}} + 2\chi_{14,\frac{10}{21};14,-\frac{10}{21}}^{m7\times\overline{m7}} + 2\chi_{15,\frac{1}{21};15,-\frac{1}{21}}^{m7\times\overline{m7}}$$

$$\mathcal{A} = \mathbf{1} \oplus \bar{e}. \tag{H.12}$$

$$Z_{\mathbf{1};\mathbf{1}}^{JK_4 \boxtimes \overline{JK_4}} = \textcolor{blue}{\chi_{1,0;1,0}^{m7\times\overline{m7}}} + \chi_{1,0;6,-5}^{m7\times\overline{m7}} + \textcolor{red}{\chi_{3,\frac{5}{7};3,-\frac{5}{7}}^{m7\times\overline{m7}}} + \chi_{3,\frac{5}{7};4,-\frac{12}{7}}^{m7\times\overline{m7}} + \chi_{5,\frac{22}{7};2,-\frac{1}{7}}^{m7\times\overline{m7}} + \chi_{5,\frac{22}{7};5,-\frac{22}{7}}^{m7\times\overline{m7}}$$

$$Z_{\mathbf{1};\bar{e}}^{JK_4 \boxtimes \overline{JK_4}} = \chi_{2,\frac{1}{7};2,-\frac{1}{7}}^{m7\times\overline{m7}} + \chi_{2,\frac{1}{7};5,-\frac{22}{7}}^{m7\times\overline{m7}} + \chi_{4,\frac{12}{7};3,-\frac{5}{7}}^{m7\times\overline{m7}} + \chi_{4,\frac{12}{7};4,-\frac{12}{7}}^{m7\times\overline{m7}} + \chi_{6,5;1,0}^{m7\times\overline{m7}} + \chi_{6,5;6,-5}^{m7\times\overline{m7}}$$

$$Z_{\mathbf{1};\bar{m}}^{JK_4 \boxtimes \overline{JK_4}} = \chi_{8,\frac{1}{56};2,-\frac{1}{7}}^{m7\times\overline{m7}} + \chi_{8,\frac{1}{56};5,-\frac{22}{7}}^{m7\times\overline{m7}} + \chi_{10,\frac{33}{56};3,-\frac{5}{7}}^{m7\times\overline{m7}} + \chi_{10,\frac{33}{56};4,-\frac{12}{7}}^{m7\times\overline{m7}} + \chi_{12,\frac{23}{8};1,0}^{m7\times\overline{m7}} + \chi_{12,\frac{23}{8};6,-5}^{m7\times\overline{m7}}$$

$$Z_{\mathbf{1};\bar{m}_1}^{JK_4 \boxtimes \overline{JK_4}} = \chi_{7,\frac{3}{8};1,0}^{m7\times\overline{m7}} + \chi_{7,\frac{3}{8};6,-5}^{m7\times\overline{m7}} + \chi_{9,\frac{5}{56};3,-\frac{5}{7}}^{m7\times\overline{m7}} + \chi_{9,\frac{5}{56};4,-\frac{12}{7}}^{m7\times\overline{m7}} + \chi_{11,\frac{85}{56};2,-\frac{1}{7}}^{m7\times\overline{m7}} + \chi_{11,\frac{85}{56};5,-\frac{22}{7}}^{m7\times\overline{m7}}$$

$$Z_{\mathbf{1};\bar{q}}^{JK_4 \boxtimes \overline{JK_4}} = \chi_{13,\frac{4}{3};1,0}^{m7\times\overline{m7}} + \chi_{13,\frac{4}{3};6,-5}^{m7\times\overline{m7}} + \chi_{14,\frac{10}{21};2,-\frac{1}{7}}^{m7\times\overline{m7}} + \chi_{14,\frac{10}{21};5,-\frac{22}{7}}^{m7\times\overline{m7}} + \chi_{15,\frac{1}{21};3,-\frac{5}{7}}^{m7\times\overline{m7}} + \chi_{15,\frac{1}{21};4,-\frac{12}{7}}^{m7\times\overline{m7}}$$

$$Z_{e;\mathbf{1}}^{JK_4 \boxtimes \overline{JK_4}} = \textcolor{blue}{\chi_{1,0;1,0}^{m7\times\overline{m7}}} + \chi_{1,0;6,-5}^{m7\times\overline{m7}} + \chi_{3,\frac{5}{7};3,-\frac{5}{7}}^{m7\times\overline{m7}} + \chi_{3,\frac{5}{7};4,-\frac{12}{7}}^{m7\times\overline{m7}} + \chi_{5,\frac{22}{7};2,-\frac{1}{7}}^{m7\times\overline{m7}} + \chi_{5,\frac{22}{7};5,-\frac{22}{7}}^{m7\times\overline{m7}}$$

$$Z_{e;\bar{e}}^{JK_4 \boxtimes \overline{JK_4}} = \chi_{2,\frac{1}{7};2,-\frac{1}{7}}^{m7\times\overline{m7}} + \chi_{2,\frac{1}{7};5,-\frac{22}{7}}^{m7\times\overline{m7}} + \chi_{4,\frac{12}{7};3,-\frac{5}{7}}^{m7\times\overline{m7}} + \chi_{4,\frac{12}{7};4,-\frac{12}{7}}^{m7\times\overline{m7}} + \chi_{6,5;1,0}^{m7\times\overline{m7}} + \chi_{6,5;6,-5}^{m7\times\overline{m7}}$$

$$Z_{e;\bar{m}}^{JK_4 \boxtimes \overline{JK_4}} = \chi_{8,\frac{1}{56};2,-\frac{1}{7}}^{m7\times\overline{m7}} + \chi_{8,\frac{1}{56};5,-\frac{22}{7}}^{m7\times\overline{m7}} + \chi_{10,\frac{33}{56};3,-\frac{5}{7}}^{m7\times\overline{m7}} + \chi_{10,\frac{33}{56};4,-\frac{12}{7}}^{m7\times\overline{m7}} + \chi_{12,\frac{23}{8};1,0}^{m7\times\overline{m7}} + \chi_{12,\frac{23}{8};6,-5}^{m7\times\overline{m7}}$$

$$Z_{e;\bar{m}_1}^{JK_4 \boxtimes \overline{JK_4}} = \chi_{7,\frac{3}{8};1,0}^{m7\times\overline{m7}} + \chi_{7,\frac{3}{8};6,-5}^{m7\times\overline{m7}} + \chi_{9,\frac{5}{56};3,-\frac{5}{7}}^{m7\times\overline{m7}} + \chi_{9,\frac{5}{56};4,-\frac{12}{7}}^{m7\times\overline{m7}} + \chi_{11,\frac{85}{56};2,-\frac{1}{7}}^{m7\times\overline{m7}} + \chi_{11,\frac{85}{56};5,-\frac{22}{7}}^{m7\times\overline{m7}}$$

$$Z_{e;\bar{q}}^{JK_4 \boxtimes \overline{JK_4}} = \chi_{13,\frac{4}{3};1,0}^{m7\times\overline{m7}} + \chi_{13,\frac{4}{3};6,-5}^{m7\times\overline{m7}} + \chi_{14,\frac{10}{21};2,-\frac{1}{7}}^{m7\times\overline{m7}} + \chi_{14,\frac{10}{21};5,-\frac{22}{7}}^{m7\times\overline{m7}} + \chi_{15,\frac{1}{21};3,-\frac{5}{7}}^{m7\times\overline{m7}} + \chi_{15,\frac{1}{21};4,-\frac{12}{7}}^{m7\times\overline{m7}}$$

$$Z_{m;\mathbf{1}}^{JK_4 \boxtimes \overline{JK_4}} = 0$$

$$Z_{m;\bar{e}}^{JK_4 \boxtimes \overline{JK_4}} = 0$$

$$Z^{JK_4 \boxtimes \overline{JK}_4}_{m;\bar{m}} = 0$$

$$Z^{JK_4 \boxtimes \overline{JK}_4}_{m;\bar{m}_1} = 0$$

$$Z^{JK_4 \boxtimes \overline{JK}_4}_{m;\bar{q}} = 0$$

$$Z^{JK_4 \boxtimes \overline{JK}_4}_{m_1;\mathbf{1}} = 0$$

$$Z^{JK_4 \boxtimes \overline{JK}_4}_{m_1;\bar{e}} = 0$$

$$Z^{JK_4 \boxtimes \overline{JK}_4}_{m_1;\bar{m}} = 0$$

$$Z^{JK_4 \boxtimes \overline{JK}_4}_{m_1;\bar{m}_1} = 0$$

$$Z^{JK_4 \boxtimes \overline{JK}_4}_{m_1;\bar{q}} = 0$$

$$Z^{JK_4 \boxtimes \overline{JK}_4}_{q;\mathbf{1}} = 2\chi^{m7 \times \overline{m7}}_{1,0;13,-\frac{4}{3}} + 2\chi^{m7 \times \overline{m7}}_{3,\frac{5}{7};15,-\frac{1}{21}} + 2\chi^{m7 \times \overline{m7}}_{5,\frac{22}{7};14,-\frac{10}{21}}$$

$$Z^{JK_4 \boxtimes \overline{JK}_4}_{q;\bar{e}} = 2\chi^{m7 \times \overline{m7}}_{2,\frac{1}{7};14,-\frac{10}{21}} + 2\chi^{m7 \times \overline{m7}}_{4,\frac{12}{7};15,-\frac{1}{21}} + 2\chi^{m7 \times \overline{m7}}_{6,5;13,-\frac{4}{3}}$$

$$Z^{JK_4 \boxtimes \overline{JK}_4}_{q;\bar{m}} = 2\chi^{m7 \times \overline{m7}}_{8,\frac{1}{56};14,-\frac{10}{21}} + 2\chi^{m7 \times \overline{m7}}_{10,\frac{33}{56};15,-\frac{1}{21}} + 2\chi^{m7 \times \overline{m7}}_{12,\frac{23}{8};13,-\frac{4}{3}}$$

$$Z^{JK_4 \boxtimes \overline{JK}_4}_{q;\bar{m}_1} = 2\chi^{m7 \times \overline{m7}}_{7,\frac{3}{8};13,-\frac{4}{3}} + 2\chi^{m7 \times \overline{m7}}_{9,\frac{5}{56};15,-\frac{1}{21}} + 2\chi^{m7 \times \overline{m7}}_{11,\frac{85}{56};14,-\frac{10}{21}}$$

$$Z^{JK_4 \boxtimes \overline{JK}_4}_{q;\bar{q}} = 2\chi^{m7 \times \overline{m7}}_{13,\frac{4}{3};13,-\frac{4}{3}} + 2\chi^{m7 \times \overline{m7}}_{14,\frac{10}{21};14,-\frac{10}{21}} + 2\chi^{m7 \times \overline{m7}}_{15,\frac{1}{21};15,-\frac{1}{21}}$$

$$\mathcal{A} = \mathbf{1} \oplus e. \tag{H.13}$$

$$Z^{JK_4 \boxtimes \overline{JK}_4}_{\mathbf{1};\mathbf{1}} = \color{blue}{\chi^{m4 \times m6 \times \overline{m4} \times \overline{m6}}_{1,0;1,0;1,0;1,0}} + \chi^{m4 \times m6 \times \overline{m4} \times \overline{m6}}_{1,0;10,\frac{7}{5};1,0;10,-\frac{7}{5}} + \chi^{m4 \times m6 \times \overline{m4} \times \overline{m6}}_{2,\frac{1}{16};5,3;2,-\frac{1}{16};1,0} + \chi^{m4 \times m6 \times \overline{m4} \times \overline{m6}}_{2,\frac{1}{16};6,\frac{2}{5};2,-\frac{1}{16};10,-\frac{7}{5}}$$
$$+ \color{red}{\chi^{m4 \times m6 \times \overline{m4} \times \overline{m6}}_{3,\frac{1}{2};1,0;3,-\frac{1}{2};1,0}} + \chi^{m4 \times m6 \times \overline{m4} \times \overline{m6}}_{3,\frac{1}{2};10,\frac{7}{5};3,-\frac{1}{2};10,-\frac{7}{5}}$$

$$Z^{JK_4 \boxtimes \overline{JK}_4}_{\mathbf{1};\bar{e}} = \chi^{m4 \times m6 \times \overline{m4} \times \overline{m6}}_{1,0;1,0;1,0;5,-3} + \chi^{m4 \times m6 \times \overline{m4} \times \overline{m6}}_{1,0;10,\frac{7}{5};1,0;6,-\frac{2}{5}} + \chi^{m4 \times m6 \times \overline{m4} \times \overline{m6}}_{2,\frac{1}{16};5,3;2,-\frac{1}{16};5,-3} + \chi^{m4 \times m6 \times \overline{m4} \times \overline{m6}}_{2,\frac{1}{16};6,\frac{2}{5};2,-\frac{1}{16};6,-\frac{2}{5}}$$
$$+ \chi^{m4 \times m6 \times \overline{m4} \times \overline{m6}}_{3,\frac{1}{2};1,0;3,-\frac{1}{2};5,-3} + \chi^{m4 \times m6 \times \overline{m4} \times \overline{m6}}_{3,\frac{1}{2};10,\frac{7}{5};3,-\frac{1}{2};6,-\frac{2}{5}}$$

$$Z^{JK_4 \boxtimes \overline{JK}_4}_{\mathbf{1};\bar{m}} = \chi^{m4 \times m6 \times \overline{m4} \times \overline{m6}}_{1,0;1,0;1,0;2,-\frac{1}{8}} + \chi^{m4 \times m6 \times \overline{m4} \times \overline{m6}}_{1,0;10,\frac{7}{5};1,0;9,-\frac{21}{40}} + \chi^{m4 \times m6 \times \overline{m4} \times \overline{m6}}_{2,\frac{1}{16};5,3;2,-\frac{1}{16};2,-\frac{1}{8}} + \chi^{m4 \times m6 \times \overline{m4} \times \overline{m6}}_{2,\frac{1}{16};6,\frac{2}{5};2,-\frac{1}{16};9,-\frac{21}{40}}$$
$$+ \chi^{m4 \times m6 \times \overline{m4} \times \overline{m6}}_{3,\frac{1}{2};1,0;3,-\frac{1}{2};2,-\frac{1}{8}} + \chi^{m4 \times m6 \times \overline{m4} \times \overline{m6}}_{3,\frac{1}{2};10,\frac{7}{5};3,-\frac{1}{2};9,-\frac{21}{40}}$$

$$Z^{JK_4 \boxtimes \overline{JK}_4}_{\mathbf{1};\bar{m}_1} = \chi^{m4 \times m6 \times \overline{m4} \times \overline{m6}}_{1,0;1,0;1,0;4,-\frac{13}{8}} + \chi^{m4 \times m6 \times \overline{m4} \times \overline{m6}}_{1,0;10,\frac{7}{5};1,0;7,-\frac{1}{40}} + \chi^{m4 \times m6 \times \overline{m4} \times \overline{m6}}_{2,\frac{1}{16};5,3;2,-\frac{1}{16};4,-\frac{13}{8}} + \chi^{m4 \times m6 \times \overline{m4} \times \overline{m6}}_{2,\frac{1}{16};6,\frac{2}{5};2,-\frac{1}{16};7,-\frac{1}{40}}$$
$$+ \chi^{m4 \times m6 \times \overline{m4} \times \overline{m6}}_{3,\frac{1}{2};1,0;3,-\frac{1}{2};4,-\frac{13}{8}} + \chi^{m4 \times m6 \times \overline{m4} \times \overline{m6}}_{3,\frac{1}{2};10,\frac{7}{5};3,-\frac{1}{2};7,-\frac{1}{40}}$$

$$Z^{JK_4 \boxtimes \overline{JK}_4}_{\mathbf{1};\bar{q}} = \chi^{m4 \times m6 \times \overline{m4} \times \overline{m6}}_{1,0;1,0;1,0;3,-\frac{2}{3}} + \chi^{m4 \times m6 \times \overline{m4} \times \overline{m6}}_{1,0;10,\frac{7}{5};1,0;8,-\frac{1}{15}} + \chi^{m4 \times m6 \times \overline{m4} \times \overline{m6}}_{2,\frac{1}{16};5,3;2,-\frac{1}{16};3,-\frac{2}{3}} + \chi^{m4 \times m6 \times \overline{m4} \times \overline{m6}}_{2,\frac{1}{16};6,\frac{2}{5};2,-\frac{1}{16};8,-\frac{1}{15}}$$
$$+ \chi^{m4 \times m6 \times \overline{m4} \times \overline{m6}}_{3,\frac{1}{2};1,0;3,-\frac{1}{2};3,-\frac{2}{3}} + \chi^{m4 \times m6 \times \overline{m4} \times \overline{m6}}_{3,\frac{1}{2};10,\frac{7}{5};3,-\frac{1}{2};8,-\frac{1}{15}}$$

$$Z^{JK_4 \boxtimes \overline{JK}_4}_{e;\mathbf{1}} = \chi^{m4 \times m6 \times \overline{m4} \times \overline{m6}}_{1,0;5,3;1,0;1,0} + \chi^{m4 \times m6 \times \overline{m4} \times \overline{m6}}_{1,0;6,\frac{2}{5};1,0;10,-\frac{7}{5}} + \chi^{m4 \times m6 \times \overline{m4} \times \overline{m6}}_{2,\frac{1}{16};1,0;2,-\frac{1}{16};1,0} + \chi^{m4 \times m6 \times \overline{m4} \times \overline{m6}}_{2,\frac{1}{16};10,\frac{7}{5};2,-\frac{1}{16};10,-\frac{7}{5}}$$
$$+ \chi^{m4 \times m6 \times \overline{m4} \times \overline{m6}}_{3,\frac{1}{2};5,3;3,-\frac{1}{2};1,0} + \chi^{m4 \times m6 \times \overline{m4} \times \overline{m6}}_{3,\frac{1}{2};6,\frac{2}{5};3,-\frac{1}{2};10,-\frac{7}{5}}$$

$$Z^{JK_4 \boxtimes \overline{JK}_4}_{e;\bar{e}} = \chi^{m4 \times m6 \times \overline{m4} \times \overline{m6}}_{1,0;5,3;1,0;5,-3} + \chi^{m4 \times m6 \times \overline{m4} \times \overline{m6}}_{1,0;6,\frac{2}{5};1,0;6,-\frac{2}{5}} + \chi^{m4 \times m6 \times \overline{m4} \times \overline{m6}}_{2,\frac{1}{16};1,0;2,-\frac{1}{16};5,-3} + \chi^{m4 \times m6 \times \overline{m4} \times \overline{m6}}_{2,\frac{1}{16};10,\frac{7}{5};2,-\frac{1}{16};6,-\frac{2}{5}}$$
$$+ \chi^{m4 \times m6 \times \overline{m4} \times \overline{m6}}_{3,\frac{1}{2};5,3;3,-\frac{1}{2};5,-3} + \chi^{m4 \times m6 \times \overline{m4} \times \overline{m6}}_{3,\frac{1}{2};6,\frac{2}{5};3,-\frac{1}{2};6,-\frac{2}{5}}$$

$$Z^{JK_4 \boxtimes \overline{JK}_4}_{e;\bar{m}} = \chi^{m4 \times m6 \times \overline{m4} \times \overline{m6}}_{1,0;5,3;1,0;2,-\frac{1}{8}} + \chi^{m4 \times m6 \times \overline{m4} \times \overline{m6}}_{1,0;6,\frac{2}{5};1,0;9,-\frac{21}{40}} + \chi^{m4 \times m6 \times \overline{m4} \times \overline{m6}}_{2,\frac{1}{16};1,0;2,-\frac{1}{16};2,-\frac{1}{8}} + \chi^{m4 \times m6 \times \overline{m4} \times \overline{m6}}_{2,\frac{1}{16};10,\frac{7}{5};2,-\frac{1}{16};9,-\frac{21}{40}}$$

$$+ \chi^{m4\times m6\times \overline{m4}\times \overline{m6}}_{3,\frac{1}{2};5,3;3,-\frac{1}{2};2,-\frac{1}{8}} + \chi^{m4\times m6\times \overline{m4}\times \overline{m6}}_{3,\frac{1}{2};6,\frac{2}{5};3,-\frac{1}{2};9,-\frac{21}{40}}$$

$$Z^{JK_4\boxtimes\overline{JK_4}}_{e;\overline{m}_1} = \chi^{m4\times m6\times \overline{m4}\times \overline{m6}}_{1,0;5,3;1,0;4,-\frac{13}{8}} + \chi^{m4\times m6\times \overline{m4}\times \overline{m6}}_{1,0;6,\frac{2}{5};1,0;7,-\frac{1}{40}} + \chi^{m4\times m6\times \overline{m4}\times \overline{m6}}_{2,\frac{1}{16};1,0;2,-\frac{1}{16};4,-\frac{13}{8}} + \chi^{m4\times m6\times \overline{m4}\times \overline{m6}}_{2,\frac{1}{16};10,\frac{7}{5};2,-\frac{1}{16};7,-\frac{1}{40}}$$
$$+ \chi^{m4\times m6\times \overline{m4}\times \overline{m6}}_{3,\frac{1}{2};5,3;3,-\frac{1}{2};4,-\frac{13}{8}} + \chi^{m4\times m6\times \overline{m4}\times \overline{m6}}_{3,\frac{1}{2};6,\frac{2}{5};3,-\frac{1}{2};7,-\frac{1}{40}}$$

$$Z^{JK_4\boxtimes\overline{JK_4}}_{e;\overline{q}} = \chi^{m4\times m6\times \overline{m4}\times \overline{m6}}_{1,0;5,3;1,0;3,-\frac{2}{3}} + \chi^{m4\times m6\times \overline{m4}\times \overline{m6}}_{1,0;6,\frac{2}{5};1,0;8,-\frac{1}{15}} + \chi^{m4\times m6\times \overline{m4}\times \overline{m6}}_{2,\frac{1}{16};1,0;2,-\frac{1}{16};3,-\frac{2}{3}} + \chi^{m4\times m6\times \overline{m4}\times \overline{m6}}_{2,\frac{1}{16};10,\frac{7}{5};2,-\frac{1}{16};8,-\frac{1}{15}}$$
$$+ \chi^{m4\times m6\times \overline{m4}\times \overline{m6}}_{3,\frac{1}{2};5,3;3,-\frac{1}{2};3,-\frac{2}{3}} + \chi^{m4\times m6\times \overline{m4}\times \overline{m6}}_{3,\frac{1}{2};6,\frac{2}{5};3,-\frac{1}{2};8,-\frac{1}{15}}$$

$$Z^{JK_4\boxtimes\overline{JK_4}}_{m;\mathbf{1}} = \chi^{m4\times m6\times \overline{m4}\times \overline{m6}}_{1,0;4,\frac{13}{8};3,-\frac{1}{2};1,0} + \chi^{m4\times m6\times \overline{m4}\times \overline{m6}}_{1,0;7,\frac{1}{40};3,-\frac{1}{2};10,-\frac{7}{5}} + \chi^{m4\times m6\times \overline{m4}\times \overline{m6}}_{2,\frac{1}{16};2,\frac{1}{8};2,-\frac{1}{16};1,0} + \chi^{m4\times m6\times \overline{m4}\times \overline{m6}}_{2,\frac{1}{16};9,\frac{21}{40};2,-\frac{1}{16};10,-\frac{7}{5}}$$
$$+ \chi^{m4\times m6\times \overline{m4}\times \overline{m6}}_{3,\frac{1}{2};4,\frac{13}{8};1,0;1,0} + \chi^{m4\times m6\times \overline{m4}\times \overline{m6}}_{3,\frac{1}{2};7,\frac{1}{40};1,0;10,-\frac{7}{5}}$$

$$Z^{JK_4\boxtimes\overline{JK_4}}_{m;\overline{e}} = \chi^{m4\times m6\times \overline{m4}\times \overline{m6}}_{1,0;4,\frac{13}{8};3,-\frac{1}{2};5,-3} + \chi^{m4\times m6\times \overline{m4}\times \overline{m6}}_{1,0;7,\frac{1}{40};3,-\frac{1}{2};6,-\frac{2}{5}} + \chi^{m4\times m6\times \overline{m4}\times \overline{m6}}_{2,\frac{1}{16};2,\frac{1}{8};2,-\frac{1}{16};5,-3} + \chi^{m4\times m6\times \overline{m4}\times \overline{m6}}_{2,\frac{1}{16};9,\frac{21}{40};2,-\frac{1}{16};6,-\frac{2}{5}}$$
$$+ \chi^{m4\times m6\times \overline{m4}\times \overline{m6}}_{3,\frac{1}{2};4,\frac{13}{8};1,0;5,-3} + \chi^{m4\times m6\times \overline{m4}\times \overline{m6}}_{3,\frac{1}{2};7,\frac{1}{40};1,0;6,-\frac{2}{5}}$$

$$Z^{JK_4\boxtimes\overline{JK_4}}_{m;\overline{m}} = \chi^{m4\times m6\times \overline{m4}\times \overline{m6}}_{1,0;4,\frac{13}{8};3,-\frac{1}{2};2,-\frac{1}{8}} + \chi^{m4\times m6\times \overline{m4}\times \overline{m6}}_{1,0;7,\frac{1}{40};3,-\frac{1}{2};9,-\frac{21}{40}} + \chi^{m4\times m6\times \overline{m4}\times \overline{m6}}_{2,\frac{1}{16};2,\frac{1}{8};2,-\frac{1}{16};2,-\frac{1}{8}} + \chi^{m4\times m6\times \overline{m4}\times \overline{m6}}_{2,\frac{1}{16};9,\frac{21}{40};2,-\frac{1}{16};9,-\frac{21}{40}}$$
$$+ \chi^{m4\times m6\times \overline{m4}\times \overline{m6}}_{3,\frac{1}{2};4,\frac{13}{8};1,0;2,-\frac{1}{8}} + \chi^{m4\times m6\times \overline{m4}\times \overline{m6}}_{3,\frac{1}{2};7,\frac{1}{40};1,0;9,-\frac{21}{40}}$$

$$Z^{JK_4\boxtimes\overline{JK_4}}_{m;\overline{m}_1} = \chi^{m4\times m6\times \overline{m4}\times \overline{m6}}_{1,0;4,\frac{13}{8};3,-\frac{1}{2};4,-\frac{13}{8}} + \chi^{m4\times m6\times \overline{m4}\times \overline{m6}}_{1,0;7,\frac{1}{40};3,-\frac{1}{2};7,-\frac{1}{40}} + \chi^{m4\times m6\times \overline{m4}\times \overline{m6}}_{2,\frac{1}{16};2,\frac{1}{8};2,-\frac{1}{16};4,-\frac{13}{8}} + \chi^{m4\times m6\times \overline{m4}\times \overline{m6}}_{2,\frac{1}{16};9,\frac{21}{40};2,-\frac{1}{16};7,-\frac{1}{40}}$$
$$+ \chi^{m4\times m6\times \overline{m4}\times \overline{m6}}_{3,\frac{1}{2};4,\frac{13}{8};1,0;4,-\frac{13}{8}} + \chi^{m4\times m6\times \overline{m4}\times \overline{m6}}_{3,\frac{1}{2};7,\frac{1}{40};1,0;7,-\frac{1}{40}}$$

$$Z^{JK_4\boxtimes\overline{JK_4}}_{m;\overline{q}} = \chi^{m4\times m6\times \overline{m4}\times \overline{m6}}_{1,0;4,\frac{13}{8};3,-\frac{1}{2};3,-\frac{2}{3}} + \chi^{m4\times m6\times \overline{m4}\times \overline{m6}}_{1,0;7,\frac{1}{40};3,-\frac{1}{2};8,-\frac{1}{15}} + \chi^{m4\times m6\times \overline{m4}\times \overline{m6}}_{2,\frac{1}{16};2,\frac{1}{8};2,-\frac{1}{16};3,-\frac{2}{3}} + \chi^{m4\times m6\times \overline{m4}\times \overline{m6}}_{2,\frac{1}{16};9,\frac{21}{40};2,-\frac{1}{16};8,-\frac{1}{15}}$$
$$+ \chi^{m4\times m6\times \overline{m4}\times \overline{m6}}_{3,\frac{1}{2};4,\frac{13}{8};1,0;3,-\frac{2}{3}} + \chi^{m4\times m6\times \overline{m4}\times \overline{m6}}_{3,\frac{1}{2};7,\frac{1}{40};1,0;8,-\frac{1}{15}}$$

$$Z^{JK_4\boxtimes\overline{JK_4}}_{m_1;\mathbf{1}} = \chi^{m4\times m6\times \overline{m4}\times \overline{m6}}_{1,0;2,\frac{1}{8};3,-\frac{1}{2};1,0} + \chi^{m4\times m6\times \overline{m4}\times \overline{m6}}_{1,0;9,\frac{21}{40};3,-\frac{1}{2};10,-\frac{7}{5}} + \chi^{m4\times m6\times \overline{m4}\times \overline{m6}}_{2,\frac{1}{16};4,\frac{13}{8};2,-\frac{1}{16};1,0} + \chi^{m4\times m6\times \overline{m4}\times \overline{m6}}_{2,\frac{1}{16};7,\frac{1}{40};2,-\frac{1}{16};10,-\frac{7}{5}}$$
$$+ \chi^{m4\times m6\times \overline{m4}\times \overline{m6}}_{3,\frac{1}{2};2,\frac{1}{8};1,0;1,0} + \chi^{m4\times m6\times \overline{m4}\times \overline{m6}}_{3,\frac{1}{2};9,\frac{21}{40};1,0;10,-\frac{7}{5}}$$

$$Z^{JK_4\boxtimes\overline{JK_4}}_{m_1;\overline{e}} = \chi^{m4\times m6\times \overline{m4}\times \overline{m6}}_{1,0;2,\frac{1}{8};3,-\frac{1}{2};5,-3} + \chi^{m4\times m6\times \overline{m4}\times \overline{m6}}_{1,0;9,\frac{21}{40};3,-\frac{1}{2};6,-\frac{2}{5}} + \chi^{m4\times m6\times \overline{m4}\times \overline{m6}}_{2,\frac{1}{16};4,\frac{13}{8};2,-\frac{1}{16};5,-3} + \chi^{m4\times m6\times \overline{m4}\times \overline{m6}}_{2,\frac{1}{16};7,\frac{1}{40};2,-\frac{1}{16};6,-\frac{2}{5}}$$
$$+ \chi^{m4\times m6\times \overline{m4}\times \overline{m6}}_{3,\frac{1}{2};2,\frac{1}{8};1,0;5,-3} + \chi^{m4\times m6\times \overline{m4}\times \overline{m6}}_{3,\frac{1}{2};9,\frac{21}{40};1,0;6,-\frac{2}{5}}$$

$$Z^{JK_4\boxtimes\overline{JK_4}}_{m_1;\overline{m}} = \chi^{m4\times m6\times \overline{m4}\times \overline{m6}}_{1,0;2,\frac{1}{8};3,-\frac{1}{2};2,-\frac{1}{8}} + \chi^{m4\times m6\times \overline{m4}\times \overline{m6}}_{1,0;9,\frac{21}{40};3,-\frac{1}{2};9,-\frac{21}{40}} + \chi^{m4\times m6\times \overline{m4}\times \overline{m6}}_{2,\frac{1}{16};4,\frac{13}{8};2,-\frac{1}{16};2,-\frac{1}{8}} + \chi^{m4\times m6\times \overline{m4}\times \overline{m6}}_{2,\frac{1}{16};7,\frac{1}{40};2,-\frac{1}{16};9,-\frac{21}{40}}$$
$$+ \chi^{m4\times m6\times \overline{m4}\times \overline{m6}}_{3,\frac{1}{2};2,\frac{1}{8};1,0;2,-\frac{1}{8}} + \chi^{m4\times m6\times \overline{m4}\times \overline{m6}}_{3,\frac{1}{2};9,\frac{21}{40};1,0;9,-\frac{21}{40}}$$

$$Z^{JK_4\boxtimes\overline{JK_4}}_{m_1;\overline{m}_1} = \chi^{m4\times m6\times \overline{m4}\times \overline{m6}}_{1,0;2,\frac{1}{8};3,-\frac{1}{2};4,-\frac{13}{8}} + \chi^{m4\times m6\times \overline{m4}\times \overline{m6}}_{1,0;9,\frac{21}{40};3,-\frac{1}{2};7,-\frac{1}{40}} + \chi^{m4\times m6\times \overline{m4}\times \overline{m6}}_{2,\frac{1}{16};4,\frac{13}{8};2,-\frac{1}{16};4,-\frac{13}{8}} + \chi^{m4\times m6\times \overline{m4}\times \overline{m6}}_{2,\frac{1}{16};7,\frac{1}{40};2,-\frac{1}{16};7,-\frac{1}{40}}$$
$$+ \chi^{m4\times m6\times \overline{m4}\times \overline{m6}}_{3,\frac{1}{2};2,\frac{1}{8};1,0;4,-\frac{13}{8}} + \chi^{m4\times m6\times \overline{m4}\times \overline{m6}}_{3,\frac{1}{2};9,\frac{21}{40};1,0;7,-\frac{1}{40}}$$

$$Z^{JK_4\boxtimes\overline{JK_4}}_{m_1;\overline{q}} = \chi^{m4\times m6\times \overline{m4}\times \overline{m6}}_{1,0;2,\frac{1}{8};3,-\frac{1}{2};3,-\frac{2}{3}} + \chi^{m4\times m6\times \overline{m4}\times \overline{m6}}_{1,0;9,\frac{21}{40};3,-\frac{1}{2};8,-\frac{1}{15}} + \chi^{m4\times m6\times \overline{m4}\times \overline{m6}}_{2,\frac{1}{16};4,\frac{13}{8};2,-\frac{1}{16};3,-\frac{2}{3}} + \chi^{m4\times m6\times \overline{m4}\times \overline{m6}}_{2,\frac{1}{16};7,\frac{1}{40};2,-\frac{1}{16};8,-\frac{1}{15}}$$
$$+ \chi^{m4\times m6\times \overline{m4}\times \overline{m6}}_{3,\frac{1}{2};2,\frac{1}{8};1,0;3,-\frac{2}{3}} + \chi^{m4\times m6\times \overline{m4}\times \overline{m6}}_{3,\frac{1}{2};9,\frac{21}{40};1,0;8,-\frac{1}{15}}$$

$$Z^{JK_4\boxtimes\overline{JK_4}}_{q;\mathbf{1}} = \chi^{m4\times m6\times \overline{m4}\times \overline{m6}}_{1,0;3,\frac{2}{3};1,0;1,0} + \chi^{m4\times m6\times \overline{m4}\times \overline{m6}}_{1,0;8,\frac{1}{15};1,0;10,-\frac{7}{5}} + \chi^{m4\times m6\times \overline{m4}\times \overline{m6}}_{2,\frac{1}{16};3,\frac{2}{3};2,-\frac{1}{16};1,0} + \chi^{m4\times m6\times \overline{m4}\times \overline{m6}}_{2,\frac{1}{16};8,\frac{1}{15};2,-\frac{1}{16};10,-\frac{7}{5}}$$
$$+ \chi^{m4\times m6\times \overline{m4}\times \overline{m6}}_{3,\frac{1}{2};3,\frac{2}{3};3,-\frac{1}{2};1,0} + \chi^{m4\times m6\times \overline{m4}\times \overline{m6}}_{3,\frac{1}{2};8,\frac{1}{15};3,-\frac{1}{2};10,-\frac{7}{5}}$$

$$Z^{JK_4\boxtimes\overline{JK_4}}_{q;\overline{e}} = \chi^{m4\times m6\times \overline{m4}\times \overline{m6}}_{1,0;3,\frac{2}{3};1,0;5,-3} + \chi^{m4\times m6\times \overline{m4}\times \overline{m6}}_{1,0;8,\frac{1}{15};1,0;6,-\frac{2}{5}} + \chi^{m4\times m6\times \overline{m4}\times \overline{m6}}_{2,\frac{1}{16};3,\frac{2}{3};2,-\frac{1}{16};5,-3} + \chi^{m4\times m6\times \overline{m4}\times \overline{m6}}_{2,\frac{1}{16};8,\frac{1}{15};2,-\frac{1}{16};6,-\frac{2}{5}}$$

$$+ \chi^{m4\times m6\times\overline{m4}\times\overline{m6}}_{3,\frac{1}{2};3,\frac{2}{3};3,-\frac{1}{2};5,-3} + \chi^{m4\times m6\times\overline{m4}\times\overline{m6}}_{3,\frac{1}{2};8,\frac{1}{15};3,-\frac{1}{2};6,-\frac{2}{5}}$$

$$Z^{JK_4\boxtimes\overline{JK_4}}_{q;\bar{m}} = \chi^{m4\times m6\times\overline{m4}\times\overline{m6}}_{1,0;3,\frac{2}{3};1,0;2,-\frac{1}{8}} + \chi^{m4\times m6\times\overline{m4}\times\overline{m6}}_{1,0;8,\frac{1}{15};1,0;9,-\frac{21}{40}} + \chi^{m4\times m6\times\overline{m4}\times\overline{m6}}_{2,\frac{1}{16};3,\frac{2}{3};2,-\frac{1}{16};2,-\frac{1}{8}} + \chi^{m4\times m6\times\overline{m4}\times\overline{m6}}_{2,\frac{1}{16};8,\frac{1}{15};2,-\frac{1}{16};9,-\frac{21}{40}}$$
$$+ \chi^{m4\times m6\times\overline{m4}\times\overline{m6}}_{3,\frac{1}{2};3,\frac{2}{3};3,-\frac{1}{2};2,-\frac{1}{8}} + \chi^{m4\times m6\times\overline{m4}\times\overline{m6}}_{3,\frac{1}{2};8,\frac{1}{15};3,-\frac{1}{2};9,-\frac{21}{40}}$$

$$Z^{JK_4\boxtimes\overline{JK_4}}_{q;\bar{m}_1} = \chi^{m4\times m6\times\overline{m4}\times\overline{m6}}_{1,0;3,\frac{2}{3};1,0;4,-\frac{13}{8}} + \chi^{m4\times m6\times\overline{m4}\times\overline{m6}}_{1,0;8,\frac{1}{15};1,0;7,-\frac{1}{40}} + \chi^{m4\times m6\times\overline{m4}\times\overline{m6}}_{2,\frac{1}{16};3,\frac{2}{3};2,-\frac{1}{16};4,-\frac{13}{8}} + \chi^{m4\times m6\times\overline{m4}\times\overline{m6}}_{2,\frac{1}{16};8,\frac{1}{15};2,-\frac{1}{16};7,-\frac{1}{40}}$$
$$+ \chi^{m4\times m6\times\overline{m4}\times\overline{m6}}_{3,\frac{1}{2};3,\frac{2}{3};3,-\frac{1}{2};4,-\frac{13}{8}} + \chi^{m4\times m6\times\overline{m4}\times\overline{m6}}_{3,\frac{1}{2};8,\frac{1}{15};3,-\frac{1}{2};7,-\frac{1}{40}}$$

$$Z^{JK_4\boxtimes\overline{JK_4}}_{q;\bar{q}} = \chi^{m4\times m6\times\overline{m4}\times\overline{m6}}_{1,0;3,\frac{2}{3};1,0;3,-\frac{2}{3}} + \chi^{m4\times m6\times\overline{m4}\times\overline{m6}}_{1,0;8,\frac{1}{15};1,0;8,-\frac{1}{15}} + \chi^{m4\times m6\times\overline{m4}\times\overline{m6}}_{2,\frac{1}{16};3,\frac{2}{3};2,-\frac{1}{16};3,-\frac{2}{3}} + \chi^{m4\times m6\times\overline{m4}\times\overline{m6}}_{2,\frac{1}{16};8,\frac{1}{15};2,-\frac{1}{16};8,-\frac{1}{15}}$$
$$+ \chi^{m4\times m6\times\overline{m4}\times\overline{m6}}_{3,\frac{1}{2};3,\frac{2}{3};3,-\frac{1}{2};3,-\frac{2}{3}} + \chi^{m4\times m6\times\overline{m4}\times\overline{m6}}_{3,\frac{1}{2};8,\frac{1}{15};3,-\frac{1}{2};8,-\frac{1}{15}}$$

$$\mathcal{A} = \mathbf{1}. \tag{H.14}$$

$$Z^{JK_4\boxtimes\overline{JK_4}}_{\mathbf{1};\mathbf{1}} = {\color{blue}\chi^{m4\times m6\times\overline{m4}\times\overline{m6}}_{1,0;1,0;1,0;1,0}} + \chi^{m4\times m6\times\overline{m4}\times\overline{m6}}_{1,0;10,\frac{7}{5};1,0;10,-\frac{7}{5}} + \chi^{m4\times m6\times\overline{m4}\times\overline{m6}}_{2,\frac{1}{16};5,3;2,-\frac{1}{16};1,0} + \chi^{m4\times m6\times\overline{m4}\times\overline{m6}}_{2,\frac{1}{16};6,\frac{2}{5};2,-\frac{1}{16};10,-\frac{7}{5}}$$
$$+ {\color{red}\chi^{m4\times m6\times\overline{m4}\times\overline{m6}}_{3,\frac{1}{2};1,0;3,-\frac{1}{2};1,0}} + \chi^{m4\times m6\times\overline{m4}\times\overline{m6}}_{3,\frac{1}{2};10,\frac{7}{5};3,-\frac{1}{2};10,-\frac{7}{5}}$$

$$Z^{JK_4\boxtimes\overline{JK_4}}_{\mathbf{1};\bar{e}} = \chi^{m4\times m6\times\overline{m4}\times\overline{m6}}_{1,0;1,0;1,0;5,-3} + \chi^{m4\times m6\times\overline{m4}\times\overline{m6}}_{1,0;10,\frac{7}{5};1,0;6,-\frac{2}{5}} + \chi^{m4\times m6\times\overline{m4}\times\overline{m6}}_{2,\frac{1}{16};5,3;2,-\frac{1}{16};5,-3}$$
$$+ \chi^{m4\times m6\times\overline{m4}\times\overline{m6}}_{2,\frac{1}{16};6,\frac{2}{5};2,-\frac{1}{16};6,-\frac{2}{5}} + \chi^{m4\times m6\times\overline{m4}\times\overline{m6}}_{3,\frac{1}{2};1,0;3,-\frac{1}{2};5,-3} + \chi^{m4\times m6\times\overline{m4}\times\overline{m6}}_{3,\frac{1}{2};10,\frac{7}{5};3,-\frac{1}{2};6,-\frac{2}{5}}$$

$$Z^{JK_4\boxtimes\overline{JK_4}}_{\mathbf{1};\bar{m}} = \chi^{m4\times m6\times\overline{m4}\times\overline{m6}}_{1,0;5,3;1,0;2,-\frac{1}{8}} + \chi^{m4\times m6\times\overline{m4}\times\overline{m6}}_{1,0;6,\frac{2}{5};1,0;9,-\frac{21}{40}} + \chi^{m4\times m6\times\overline{m4}\times\overline{m6}}_{2,\frac{1}{16};1,0;2,-\frac{1}{16};2,-\frac{1}{8}} + \chi^{m4\times m6\times\overline{m4}\times\overline{m6}}_{2,\frac{1}{16};10,\frac{7}{5};2,-\frac{1}{16};9,-\frac{21}{40}}$$
$$+ \chi^{m4\times m6\times\overline{m4}\times\overline{m6}}_{3,\frac{1}{2};5,3;3,-\frac{1}{2};2,-\frac{1}{8}} + \chi^{m4\times m6\times\overline{m4}\times\overline{m6}}_{3,\frac{1}{2};6,\frac{2}{5};3,-\frac{1}{2};9,-\frac{21}{40}}$$

$$Z^{JK_4\boxtimes\overline{JK_4}}_{\mathbf{1};\bar{m}_1} = \chi^{m4\times m6\times\overline{m4}\times\overline{m6}}_{1,0;5,3;1,0;4,-\frac{13}{8}} + \chi^{m4\times m6\times\overline{m4}\times\overline{m6}}_{1,0;6,\frac{2}{5};1,0;7,-\frac{1}{40}} + \chi^{m4\times m6\times\overline{m4}\times\overline{m6}}_{2,\frac{1}{16};1,0;2,-\frac{1}{16};4,-\frac{13}{8}} + \chi^{m4\times m6\times\overline{m4}\times\overline{m6}}_{2,\frac{1}{16};10,\frac{7}{5};2,-\frac{1}{16};7,-\frac{1}{40}}$$
$$+ \chi^{m4\times m6\times\overline{m4}\times\overline{m6}}_{3,\frac{1}{2};5,3;3,-\frac{1}{2};4,-\frac{13}{8}} + \chi^{m4\times m6\times\overline{m4}\times\overline{m6}}_{3,\frac{1}{2};6,\frac{2}{5};3,-\frac{1}{2};7,-\frac{1}{40}}$$

$$Z^{JK_4\boxtimes\overline{JK_4}}_{\mathbf{1};\bar{q}} = \chi^{m4\times m6\times\overline{m4}\times\overline{m6}}_{1,0;1,0;1,0;3,-\frac{2}{3}} + \chi^{m4\times m6\times\overline{m4}\times\overline{m6}}_{1,0;10,\frac{7}{5};1,0;8,-\frac{1}{15}} + \chi^{m4\times m6\times\overline{m4}\times\overline{m6}}_{2,\frac{1}{16};5,3;2,-\frac{1}{16};3,-\frac{2}{3}} + \chi^{m4\times m6\times\overline{m4}\times\overline{m6}}_{2,\frac{1}{16};6,\frac{2}{5};2,-\frac{1}{16};8,-\frac{1}{15}}$$
$$+ \chi^{m4\times m6\times\overline{m4}\times\overline{m6}}_{3,\frac{1}{2};1,0;3,-\frac{1}{2};3,-\frac{2}{3}} + \chi^{m4\times m6\times\overline{m4}\times\overline{m6}}_{3,\frac{1}{2};10,\frac{7}{5};3,-\frac{1}{2};8,-\frac{1}{15}}$$

$$Z^{JK_4\boxtimes\overline{JK_4}}_{e;\mathbf{1}} = \chi^{m4\times m6\times\overline{m4}\times\overline{m6}}_{1,0;5,3;1,0;1,0} + \chi^{m4\times m6\times\overline{m4}\times\overline{m6}}_{1,0;6,\frac{2}{5};1,0;10,-\frac{7}{5}} + \chi^{m4\times m6\times\overline{m4}\times\overline{m6}}_{2,\frac{1}{16};1,0;2,-\frac{1}{16};1,0} + \chi^{m4\times m6\times\overline{m4}\times\overline{m6}}_{2,\frac{1}{16};10,\frac{7}{5};2,-\frac{1}{16};10,-\frac{7}{5}}$$
$$+ \chi^{m4\times m6\times\overline{m4}\times\overline{m6}}_{3,\frac{1}{2};5,3;3,-\frac{1}{2};1,0} + \chi^{m4\times m6\times\overline{m4}\times\overline{m6}}_{3,\frac{1}{2};6,\frac{2}{5};3,-\frac{1}{2};10,-\frac{7}{5}}$$

$$Z^{JK_4\boxtimes\overline{JK_4}}_{e;\bar{e}} = \chi^{m4\times m6\times\overline{m4}\times\overline{m6}}_{1,0;5,3;1,0;5,-3} + \chi^{m4\times m6\times\overline{m4}\times\overline{m6}}_{1,0;6,\frac{2}{5};1,0;6,-\frac{2}{5}} + \chi^{m4\times m6\times\overline{m4}\times\overline{m6}}_{2,\frac{1}{16};1,0;2,-\frac{1}{16};5,-3} + \chi^{m4\times m6\times\overline{m4}\times\overline{m6}}_{2,\frac{1}{16};10,\frac{7}{5};2,-\frac{1}{16};6,-\frac{2}{5}}$$
$$+ \chi^{m4\times m6\times\overline{m4}\times\overline{m6}}_{3,\frac{1}{2};5,3;3,-\frac{1}{2};5,-3} + \chi^{m4\times m6\times\overline{m4}\times\overline{m6}}_{3,\frac{1}{2};6,\frac{2}{5};3,-\frac{1}{2};6,-\frac{2}{5}}$$

$$Z^{JK_4\boxtimes\overline{JK_4}}_{e;\bar{m}} = \chi^{m4\times m6\times\overline{m4}\times\overline{m6}}_{1,0;1,0;1,0;2,-\frac{1}{8}} + \chi^{m4\times m6\times\overline{m4}\times\overline{m6}}_{1,0;10,\frac{7}{5};1,0;9,-\frac{21}{40}} + \chi^{m4\times m6\times\overline{m4}\times\overline{m6}}_{2,\frac{1}{16};5,3;2,-\frac{1}{16};2,-\frac{1}{8}} + \chi^{m4\times m6\times\overline{m4}\times\overline{m6}}_{2,\frac{1}{16};6,\frac{2}{5};2,-\frac{1}{16};9,-\frac{21}{40}}$$
$$+ \chi^{m4\times m6\times\overline{m4}\times\overline{m6}}_{3,\frac{1}{2};1,0;3,-\frac{1}{2};2,-\frac{1}{8}} + \chi^{m4\times m6\times\overline{m4}\times\overline{m6}}_{3,\frac{1}{2};10,\frac{7}{5};3,-\frac{1}{2};9,-\frac{21}{40}}$$

$$Z^{JK_4\boxtimes\overline{JK_4}}_{e;\bar{m}_1} = \chi^{m4\times m6\times\overline{m4}\times\overline{m6}}_{1,0;1,0;1,0;4,-\frac{13}{8}} + \chi^{m4\times m6\times\overline{m4}\times\overline{m6}}_{1,0;10,\frac{7}{5};1,0;7,-\frac{1}{40}} + \chi^{m4\times m6\times\overline{m4}\times\overline{m6}}_{2,\frac{1}{16};5,3;2,-\frac{1}{16};4,-\frac{13}{8}} + \chi^{m4\times m6\times\overline{m4}\times\overline{m6}}_{2,\frac{1}{16};6,\frac{2}{5};2,-\frac{1}{16};7,-\frac{1}{40}}$$
$$+ \chi^{m4\times m6\times\overline{m4}\times\overline{m6}}_{3,\frac{1}{2};1,0;3,-\frac{1}{2};4,-\frac{13}{8}} + \chi^{m4\times m6\times\overline{m4}\times\overline{m6}}_{3,\frac{1}{2};10,\frac{7}{5};3,-\frac{1}{2};7,-\frac{1}{40}}$$

$$Z^{JK_4\boxtimes\overline{JK_4}}_{e;\bar{q}} = \chi^{m4\times m6\times\overline{m4}\times\overline{m6}}_{1,0;5,3;1,0;3,-\frac{2}{3}} + \chi^{m4\times m6\times\overline{m4}\times\overline{m6}}_{1,0;6,\frac{2}{5};1,0;8,-\frac{1}{15}} + \chi^{m4\times m6\times\overline{m4}\times\overline{m6}}_{2,\frac{1}{16};1,0;2,-\frac{1}{16};3,-\frac{2}{3}} + \chi^{m4\times m6\times\overline{m4}\times\overline{m6}}_{2,\frac{1}{16};10,\frac{7}{5};2,-\frac{1}{16};8,-\frac{1}{15}}$$
$$+ \chi^{m4\times m6\times\overline{m4}\times\overline{m6}}_{3,\frac{1}{2};5,3;3,-\frac{1}{2};3,-\frac{2}{3}} + \chi^{m4\times m6\times\overline{m4}\times\overline{m6}}_{3,\frac{1}{2};6,\frac{2}{5};3,-\frac{1}{2};8,-\frac{1}{15}}$$

$$Z_{m;\mathbf{1}}^{JK_4 \boxtimes \overline{JK}_4} = \chi^{m4\times m6\times\overline{m4}\times\overline{m6}}_{1,0,4,\frac{13}{8};3,-\frac{1}{2};5,-3} + \chi^{m4\times m6\times\overline{m4}\times\overline{m6}}_{1,0,7,\frac{1}{40};3,-\frac{1}{2};6,-\frac{2}{5}} + \chi^{m4\times m6\times\overline{m4}\times\overline{m6}}_{2,\frac{1}{16};2,\frac{1}{8};2,-\frac{1}{16};5,-3} + \chi^{m4\times m6\times\overline{m4}\times\overline{m6}}_{2,\frac{1}{16};9,\frac{21}{40};2,-\frac{1}{16};6,-\frac{2}{5}}$$
$$+ \chi^{m4\times m6\times\overline{m4}\times\overline{m6}}_{3,\frac{1}{2};4,\frac{13}{8};1,0;5,-3} + \chi^{m4\times m6\times\overline{m4}\times\overline{m6}}_{3,\frac{1}{2};7,\frac{1}{40};1,0;6,-\frac{2}{5}}$$

$$Z_{m;\bar{e}}^{JK_4 \boxtimes \overline{JK}_4} = \chi^{m4\times m6\times\overline{m4}\times\overline{m6}}_{1,0,4,\frac{13}{8};3,-\frac{1}{2};1,0} + \chi^{m4\times m6\times\overline{m4}\times\overline{m6}}_{1,0,7,\frac{1}{40};3,-\frac{1}{2};10,-\frac{7}{5}} + \chi^{m4\times m6\times\overline{m4}\times\overline{m6}}_{2,\frac{1}{16};2,\frac{1}{8};2,-\frac{1}{16};1,0} + \chi^{m4\times m6\times\overline{m4}\times\overline{m6}}_{2,\frac{1}{16};9,\frac{21}{40};2,-\frac{1}{16};10,-\frac{7}{5}}$$
$$+ \chi^{m4\times m6\times\overline{m4}\times\overline{m6}}_{3,\frac{1}{2};4,\frac{13}{8};1,0;1,0} + \chi^{m4\times m6\times\overline{m4}\times\overline{m6}}_{3,\frac{1}{2};7,\frac{1}{40};1,0;10,-\frac{7}{5}}$$

$$Z_{m;\bar{m}}^{JK_4 \boxtimes \overline{JK}_4} = \chi^{m4\times m6\times\overline{m4}\times\overline{m6}}_{1,0,2,\frac{1}{8};3,-\frac{1}{2};4,-\frac{13}{8}} + \chi^{m4\times m6\times\overline{m4}\times\overline{m6}}_{1,0,9,\frac{21}{40};3,-\frac{1}{2};7,-\frac{1}{40}} + \chi^{m4\times m6\times\overline{m4}\times\overline{m6}}_{2,\frac{1}{16};4,\frac{13}{8};2,-\frac{1}{16};4,-\frac{13}{8}} + \chi^{m4\times m6\times\overline{m4}\times\overline{m6}}_{2,\frac{1}{16};7,\frac{1}{40};2,-\frac{1}{16};7,-\frac{1}{40}}$$
$$+ \chi^{m4\times m6\times\overline{m4}\times\overline{m6}}_{3,\frac{1}{2};2,\frac{1}{8};1,0;4,-\frac{13}{8}} + \chi^{m4\times m6\times\overline{m4}\times\overline{m6}}_{3,\frac{1}{2};9,\frac{21}{40};1,0;7,-\frac{1}{40}}$$

$$Z_{m;\bar{m}_1}^{JK_4 \boxtimes \overline{JK}_4} = \chi^{m4\times m6\times\overline{m4}\times\overline{m6}}_{1,0,2,\frac{1}{8};3,-\frac{1}{2};2,-\frac{1}{8}} + \chi^{m4\times m6\times\overline{m4}\times\overline{m6}}_{1,0,9,\frac{21}{40};3,-\frac{1}{2};9,-\frac{21}{40}} + \chi^{m4\times m6\times\overline{m4}\times\overline{m6}}_{2,\frac{1}{16};4,\frac{13}{8};2,-\frac{1}{16};2,-\frac{1}{8}} + \chi^{m4\times m6\times\overline{m4}\times\overline{m6}}_{2,\frac{1}{16};7,\frac{1}{40};2,-\frac{1}{16};9,-\frac{21}{40}}$$
$$+ \chi^{m4\times m6\times\overline{m4}\times\overline{m6}}_{3,\frac{1}{2};2,\frac{1}{8};1,0;2,-\frac{1}{8}} + \chi^{m4\times m6\times\overline{m4}\times\overline{m6}}_{3,\frac{1}{2};9,\frac{21}{40};1,0;9,-\frac{21}{40}}$$

$$Z_{m;\bar{q}}^{JK_4 \boxtimes \overline{JK}_4} = \chi^{m4\times m6\times\overline{m4}\times\overline{m6}}_{1,0,4,\frac{13}{8};3,-\frac{1}{2};3,-\frac{2}{3}} + \chi^{m4\times m6\times\overline{m4}\times\overline{m6}}_{1,0,7,\frac{1}{40};3,-\frac{1}{2};8,-\frac{1}{15}} + \chi^{m4\times m6\times\overline{m4}\times\overline{m6}}_{2,\frac{1}{16};2,\frac{1}{8};2,-\frac{1}{16};3,-\frac{2}{3}} + \chi^{m4\times m6\times\overline{m4}\times\overline{m6}}_{2,\frac{1}{16};9,\frac{21}{40};2,-\frac{1}{16};8,-\frac{1}{15}}$$
$$+ \chi^{m4\times m6\times\overline{m4}\times\overline{m6}}_{3,\frac{1}{2};4,\frac{13}{8};1,0;3,-\frac{2}{3}} + \chi^{m4\times m6\times\overline{m4}\times\overline{m6}}_{3,\frac{1}{2};7,\frac{1}{40};1,0;8,-\frac{1}{15}}$$

$$Z_{m_1;\mathbf{1}}^{JK_4 \boxtimes \overline{JK}_4} = \chi^{m4\times m6\times\overline{m4}\times\overline{m6}}_{1,0,2,\frac{1}{8};3,-\frac{1}{2};5,-3} + \chi^{m4\times m6\times\overline{m4}\times\overline{m6}}_{1,0,9,\frac{21}{40};3,-\frac{1}{2};6,-\frac{2}{5}} + \chi^{m4\times m6\times\overline{m4}\times\overline{m6}}_{2,\frac{1}{16};4,\frac{13}{8};2,-\frac{1}{16};5,-3} + \chi^{m4\times m6\times\overline{m4}\times\overline{m6}}_{2,\frac{1}{16};7,\frac{1}{40};2,-\frac{1}{16};6,-\frac{2}{5}}$$
$$+ \chi^{m4\times m6\times\overline{m4}\times\overline{m6}}_{3,\frac{1}{2};2,\frac{1}{8};1,0;5,-3} + \chi^{m4\times m6\times\overline{m4}\times\overline{m6}}_{3,\frac{1}{2};9,\frac{21}{40};1,0;6,-\frac{2}{5}}$$

$$Z_{m_1;\bar{e}}^{JK_4 \boxtimes \overline{JK}_4} = \chi^{m4\times m6\times\overline{m4}\times\overline{m6}}_{1,0,2,\frac{1}{8};3,-\frac{1}{2};1,0} + \chi^{m4\times m6\times\overline{m4}\times\overline{m6}}_{1,0,9,\frac{21}{40};3,-\frac{1}{2};10,-\frac{7}{5}} + \chi^{m4\times m6\times\overline{m4}\times\overline{m6}}_{2,\frac{1}{16};4,\frac{13}{8};2,-\frac{1}{16};1,0} + \chi^{m4\times m6\times\overline{m4}\times\overline{m6}}_{2,\frac{1}{16};7,\frac{1}{40};2,-\frac{1}{16};10,-\frac{7}{5}}$$
$$+ \chi^{m4\times m6\times\overline{m4}\times\overline{m6}}_{3,\frac{1}{2};2,\frac{1}{8};1,0;1,0} + \chi^{m4\times m6\times\overline{m4}\times\overline{m6}}_{3,\frac{1}{2};9,\frac{21}{40};1,0;10,-\frac{7}{5}}$$

$$Z_{m_1;\bar{m}}^{JK_4 \boxtimes \overline{JK}_4} = \chi^{m4\times m6\times\overline{m4}\times\overline{m6}}_{1,0,4,\frac{13}{8};3,-\frac{1}{2};4,-\frac{13}{8}} + \chi^{m4\times m6\times\overline{m4}\times\overline{m6}}_{1,0,7,\frac{1}{40};3,-\frac{1}{2};7,-\frac{1}{40}} + \chi^{m4\times m6\times\overline{m4}\times\overline{m6}}_{2,\frac{1}{16};2,\frac{1}{8};2,-\frac{1}{16};4,-\frac{13}{8}} + \chi^{m4\times m6\times\overline{m4}\times\overline{m6}}_{2,\frac{1}{16};9,\frac{21}{40};2,-\frac{1}{16};7,-\frac{1}{40}}$$
$$+ \chi^{m4\times m6\times\overline{m4}\times\overline{m6}}_{3,\frac{1}{2};4,\frac{13}{8};1,0;4,-\frac{13}{8}} + \chi^{m4\times m6\times\overline{m4}\times\overline{m6}}_{3,\frac{1}{2};7,\frac{1}{40};1,0;7,-\frac{1}{40}}$$

$$Z_{m_1;\bar{m}_1}^{JK_4 \boxtimes \overline{JK}_4} = \chi^{m4\times m6\times\overline{m4}\times\overline{m6}}_{1,0,4,\frac{13}{8};3,-\frac{1}{2};2,-\frac{1}{8}} + \chi^{m4\times m6\times\overline{m4}\times\overline{m6}}_{1,0,7,\frac{1}{40};3,-\frac{1}{2};9,-\frac{21}{40}} + \chi^{m4\times m6\times\overline{m4}\times\overline{m6}}_{2,\frac{1}{16};2,\frac{1}{8};2,-\frac{1}{16};2,-\frac{1}{8}} + \chi^{m4\times m6\times\overline{m4}\times\overline{m6}}_{2,\frac{1}{16};9,\frac{21}{40};2,-\frac{1}{16};9,-\frac{21}{40}}$$
$$+ \chi^{m4\times m6\times\overline{m4}\times\overline{m6}}_{3,\frac{1}{2};4,\frac{13}{8};1,0;2,-\frac{1}{8}} + \chi^{m4\times m6\times\overline{m4}\times\overline{m6}}_{3,\frac{1}{2};7,\frac{1}{40};1,0;9,-\frac{21}{40}}$$

$$Z_{m_1;\bar{q}}^{JK_4 \boxtimes \overline{JK}_4} = \chi^{m4\times m6\times\overline{m4}\times\overline{m6}}_{1,0,2,\frac{1}{8};3,-\frac{1}{2};3,-\frac{2}{3}} + \chi^{m4\times m6\times\overline{m4}\times\overline{m6}}_{1,0,9,\frac{21}{40};3,-\frac{1}{2};8,-\frac{1}{15}} + \chi^{m4\times m6\times\overline{m4}\times\overline{m6}}_{2,\frac{1}{16};4,\frac{13}{8};2,-\frac{1}{16};3,-\frac{2}{3}} + \chi^{m4\times m6\times\overline{m4}\times\overline{m6}}_{2,\frac{1}{16};7,\frac{1}{40};2,-\frac{1}{16};8,-\frac{1}{15}}$$
$$+ \chi^{m4\times m6\times\overline{m4}\times\overline{m6}}_{3,\frac{1}{2};2,\frac{1}{8};1,0;3,-\frac{2}{3}} + \chi^{m4\times m6\times\overline{m4}\times\overline{m6}}_{3,\frac{1}{2};9,\frac{21}{40};1,0;8,-\frac{1}{15}}$$

$$Z_{q;\mathbf{1}}^{JK_4 \boxtimes \overline{JK}_4} = \chi^{m4\times m6\times\overline{m4}\times\overline{m6}}_{1,0,3,\frac{2}{3};1,0;1,0} + \chi^{m4\times m6\times\overline{m4}\times\overline{m6}}_{1,0,8,\frac{1}{15};1,0;10,-\frac{7}{5}} + \chi^{m4\times m6\times\overline{m4}\times\overline{m6}}_{2,\frac{1}{16};3,\frac{2}{3};2,-\frac{1}{16};1,0} + \chi^{m4\times m6\times\overline{m4}\times\overline{m6}}_{2,\frac{1}{16};8,\frac{1}{15};2,-\frac{1}{16};10,-\frac{7}{5}}$$
$$+ \chi^{m4\times m6\times\overline{m4}\times\overline{m6}}_{3,\frac{1}{2};3,\frac{2}{3};3,-\frac{1}{2};1,0} + \chi^{m4\times m6\times\overline{m4}\times\overline{m6}}_{3,\frac{1}{2};8,\frac{1}{15};3,-\frac{1}{2};10,-\frac{7}{5}}$$

$$Z_{q;\bar{e}}^{JK_4 \boxtimes \overline{JK}_4} = \chi^{m4\times m6\times\overline{m4}\times\overline{m6}}_{1,0,3,\frac{2}{3};1,0;5,-3} + \chi^{m4\times m6\times\overline{m4}\times\overline{m6}}_{1,0,8,\frac{1}{15};1,0;6,-\frac{2}{5}} + \chi^{m4\times m6\times\overline{m4}\times\overline{m6}}_{2,\frac{1}{16};3,\frac{2}{3};2,-\frac{1}{16};5,-3} + \chi^{m4\times m6\times\overline{m4}\times\overline{m6}}_{2,\frac{1}{16};8,\frac{1}{15};2,-\frac{1}{16};6,-\frac{2}{5}}$$
$$+ \chi^{m4\times m6\times\overline{m4}\times\overline{m6}}_{3,\frac{1}{2};3,\frac{2}{3};3,-\frac{1}{2};5,-3} + \chi^{m4\times m6\times\overline{m4}\times\overline{m6}}_{3,\frac{1}{2};8,\frac{1}{15};3,-\frac{1}{2};6,-\frac{2}{5}}$$

$$Z_{q;\bar{m}}^{JK_4 \boxtimes \overline{JK}_4} = \chi^{m4\times m6\times\overline{m4}\times\overline{m6}}_{1,0,3,\frac{2}{3};1,0;2,-\frac{1}{8}} + \chi^{m4\times m6\times\overline{m4}\times\overline{m6}}_{1,0,8,\frac{1}{15};1,0;9,-\frac{21}{40}} + \chi^{m4\times m6\times\overline{m4}\times\overline{m6}}_{2,\frac{1}{16};3,\frac{2}{3};2,-\frac{1}{16};2,-\frac{1}{8}} + \chi^{m4\times m6\times\overline{m4}\times\overline{m6}}_{2,\frac{1}{16};8,\frac{1}{15};2,-\frac{1}{16};9,-\frac{21}{40}}$$
$$+ \chi^{m4\times m6\times\overline{m4}\times\overline{m6}}_{3,\frac{1}{2};3,\frac{2}{3};3,-\frac{1}{2};2,-\frac{1}{8}} + \chi^{m4\times m6\times\overline{m4}\times\overline{m6}}_{3,\frac{1}{2};8,\frac{1}{15};3,-\frac{1}{2};9,-\frac{21}{40}}$$

$$Z_{q;\bar{m}_1}^{JK_4 \boxtimes \overline{JK}_4} = \chi^{m4\times m6\times\overline{m4}\times\overline{m6}}_{1,0,3,\frac{2}{3};1,0;4,-\frac{13}{8}} + \chi^{m4\times m6\times\overline{m4}\times\overline{m6}}_{1,0,8,\frac{1}{15};1,0;7,-\frac{1}{40}} + \chi^{m4\times m6\times\overline{m4}\times\overline{m6}}_{2,\frac{1}{16};3,\frac{2}{3};2,-\frac{1}{16};4,-\frac{13}{8}} + \chi^{m4\times m6\times\overline{m4}\times\overline{m6}}_{2,\frac{1}{16};8,\frac{1}{15};2,-\frac{1}{16};7,-\frac{1}{40}}$$
$$+ \chi^{m4\times m6\times\overline{m4}\times\overline{m6}}_{3,\frac{1}{2};3,\frac{2}{3};3,-\frac{1}{2};4,-\frac{13}{8}} + \chi^{m4\times m6\times\overline{m4}\times\overline{m6}}_{3,\frac{1}{2};8,\frac{1}{15};3,-\frac{1}{2};7,-\frac{1}{40}}$$

$$Z^{JK_4\boxtimes\overline{JK_4}}_{q;\bar{q}} = \chi^{m4\times m6\times\overline{m4}\times\overline{m6}}_{1,0;3,\frac{2}{3};1,0;3,-\frac{2}{3}} + \chi^{m4\times m6\times\overline{m4}\times\overline{m6}}_{1,0;8,\frac{1}{15};1,0;8,-\frac{1}{15}} + \chi^{m4\times m6\times\overline{m4}\times\overline{m6}}_{2,\frac{1}{16};3,\frac{2}{3};2,-\frac{1}{16};3,-\frac{2}{3}} + \chi^{m4\times m6\times\overline{m4}\times\overline{m6}}_{2,\frac{1}{16};8,\frac{1}{15};2,-\frac{1}{16};8,-\frac{1}{15}}$$
$$+ \chi^{m4\times m6\times\overline{m4}\times\overline{m6}}_{3,\frac{1}{2};3,\frac{2}{3};3,-\frac{1}{2};3,-\frac{2}{3}} + \chi^{m4\times m6\times\overline{m4}\times\overline{m6}}_{3,\frac{1}{2};8,\frac{1}{15};3,-\frac{1}{2};8,-\frac{1}{15}}$$
$$\mathcal{A} = \mathbf{1}. \tag{H.15}$$

$$Z^{JK_4\boxtimes\overline{JK_4}}_{\mathbf{1};\mathbf{1}} = \textcolor{blue}{\chi^{m4\times m6\times\overline{m4}\times\overline{m6}}_{1,0;1,0;1,0;1,0}} + \chi^{m4\times m6\times\overline{m4}\times\overline{m6}}_{1,0;10,\frac{7}{5};1,0;10,-\frac{7}{5}} + \chi^{m4\times m6\times\overline{m4}\times\overline{m6}}_{2,\frac{1}{16};1,0;2,-\frac{1}{16};5,-3} + \chi^{m4\times m6\times\overline{m4}\times\overline{m6}}_{2,\frac{1}{16};10,\frac{7}{5};2,-\frac{1}{16};6,-\frac{2}{5}}$$
$$+ \textcolor{red}{\chi^{m4\times m6\times\overline{m4}\times\overline{m6}}_{3,\frac{1}{2};1,0;3,-\frac{1}{2};1,0}} + \chi^{m4\times m6\times\overline{m4}\times\overline{m6}}_{3,\frac{1}{2};10,\frac{7}{5};3,-\frac{1}{2};10,-\frac{7}{5}}$$

$$Z^{JK_4\boxtimes\overline{JK_4}}_{\mathbf{1};\bar{e}} = \chi^{m4\times m6\times\overline{m4}\times\overline{m6}}_{1,0;1,0;1,0;5,-3} + \chi^{m4\times m6\times\overline{m4}\times\overline{m6}}_{1,0;10,\frac{7}{5};1,0;6,-\frac{2}{5}} + \chi^{m4\times m6\times\overline{m4}\times\overline{m6}}_{2,\frac{1}{16};1,0;2,-\frac{1}{16};1,0} + \chi^{m4\times m6\times\overline{m4}\times\overline{m6}}_{2,\frac{1}{16};10,\frac{7}{5};2,-\frac{1}{16};10,-\frac{7}{5}}$$
$$+ \chi^{m4\times m6\times\overline{m4}\times\overline{m6}}_{3,\frac{1}{2};1,0;3,-\frac{1}{2};5,-3} + \chi^{m4\times m6\times\overline{m4}\times\overline{m6}}_{3,\frac{1}{2};10,\frac{7}{5};3,-\frac{1}{2};6,-\frac{2}{5}}$$

$$Z^{JK_4\boxtimes\overline{JK_4}}_{\mathbf{1};\bar{m}} = \chi^{m4\times m6\times\overline{m4}\times\overline{m6}}_{1,0;5,3;3,-\frac{1}{2};4,-\frac{13}{8}} + \chi^{m4\times m6\times\overline{m4}\times\overline{m6}}_{1,0;6,\frac{2}{5};3,-\frac{1}{2};7,-\frac{1}{40}} + \chi^{m4\times m6\times\overline{m4}\times\overline{m6}}_{2,\frac{1}{16};5,3;2,-\frac{1}{16};2,-\frac{1}{8}} + \chi^{m4\times m6\times\overline{m4}\times\overline{m6}}_{2,\frac{1}{16};6,\frac{2}{5};2,-\frac{1}{16};9,-\frac{21}{40}}$$
$$+ \chi^{m4\times m6\times\overline{m4}\times\overline{m6}}_{3,\frac{1}{2};5,3;1,0;4,-\frac{13}{8}} + \chi^{m4\times m6\times\overline{m4}\times\overline{m6}}_{3,\frac{1}{2};6,\frac{2}{5};1,0;7,-\frac{1}{40}}$$

$$Z^{JK_4\boxtimes\overline{JK_4}}_{\mathbf{1};\bar{m}_1} = \chi^{m4\times m6\times\overline{m4}\times\overline{m6}}_{1,0;5,3;3,-\frac{1}{2};2,-\frac{1}{8}} + \chi^{m4\times m6\times\overline{m4}\times\overline{m6}}_{1,0;6,\frac{2}{5};3,-\frac{1}{2};9,-\frac{21}{40}} + \chi^{m4\times m6\times\overline{m4}\times\overline{m6}}_{2,\frac{1}{16};5,3;2,-\frac{1}{16};4,-\frac{13}{8}} + \chi^{m4\times m6\times\overline{m4}\times\overline{m6}}_{2,\frac{1}{16};6,\frac{2}{5};2,-\frac{1}{16};7,-\frac{1}{40}}$$
$$+ \chi^{m4\times m6\times\overline{m4}\times\overline{m6}}_{3,\frac{1}{2};5,3;1,0;2,-\frac{1}{8}} + \chi^{m4\times m6\times\overline{m4}\times\overline{m6}}_{3,\frac{1}{2};6,\frac{2}{5};1,0;9,-\frac{21}{40}}$$

$$Z^{JK_4\boxtimes\overline{JK_4}}_{\mathbf{1};\bar{q}} = \chi^{m4\times m6\times\overline{m4}\times\overline{m6}}_{1,0;1,0;1,0;3,-\frac{2}{3}} + \chi^{m4\times m6\times\overline{m4}\times\overline{m6}}_{1,0;10,\frac{7}{5};1,0;8,-\frac{1}{15}} + \chi^{m4\times m6\times\overline{m4}\times\overline{m6}}_{2,\frac{1}{16};1,0;2,-\frac{1}{16};3,-\frac{2}{3}} + \chi^{m4\times m6\times\overline{m4}\times\overline{m6}}_{2,\frac{1}{16};10,\frac{7}{5};2,-\frac{1}{16};8,-\frac{1}{15}}$$
$$+ \chi^{m4\times m6\times\overline{m4}\times\overline{m6}}_{3,\frac{1}{2};1,0;3,-\frac{1}{2};3,-\frac{2}{3}} + \chi^{m4\times m6\times\overline{m4}\times\overline{m6}}_{3,\frac{1}{2};10,\frac{7}{5};3,-\frac{1}{2};8,-\frac{1}{15}}$$

$$Z^{JK_4\boxtimes\overline{JK_4}}_{e;\mathbf{1}} = \chi^{m4\times m6\times\overline{m4}\times\overline{m6}}_{1,0;5,3;1,0;1,0} + \chi^{m4\times m6\times\overline{m4}\times\overline{m6}}_{1,0;6,\frac{2}{5};1,0;10,-\frac{7}{5}} + \chi^{m4\times m6\times\overline{m4}\times\overline{m6}}_{2,\frac{1}{16};5,3;2,-\frac{1}{16};5,-3} + \chi^{m4\times m6\times\overline{m4}\times\overline{m6}}_{2,\frac{1}{16};6,\frac{2}{5};2,-\frac{1}{16};6,-\frac{2}{5}}$$
$$+ \chi^{m4\times m6\times\overline{m4}\times\overline{m6}}_{3,\frac{1}{2};5,3;3,-\frac{1}{2};1,0} + \chi^{m4\times m6\times\overline{m4}\times\overline{m6}}_{3,\frac{1}{2};6,\frac{2}{5};3,-\frac{1}{2};10,-\frac{7}{5}}$$

$$Z^{JK_4\boxtimes\overline{JK_4}}_{e;\bar{e}} = \chi^{m4\times m6\times\overline{m4}\times\overline{m6}}_{1,0;5,3;1,0;5,-3} + \chi^{m4\times m6\times\overline{m4}\times\overline{m6}}_{1,0;6,\frac{2}{5};1,0;6,-\frac{2}{5}} + \chi^{m4\times m6\times\overline{m4}\times\overline{m6}}_{2,\frac{1}{16};5,3;2,-\frac{1}{16};1,0} + \chi^{m4\times m6\times\overline{m4}\times\overline{m6}}_{2,\frac{1}{16};6,\frac{2}{5};2,-\frac{1}{16};10,-\frac{7}{5}}$$
$$+ \chi^{m4\times m6\times\overline{m4}\times\overline{m6}}_{3,\frac{1}{2};5,3;3,-\frac{1}{2};5,-3} + \chi^{m4\times m6\times\overline{m4}\times\overline{m6}}_{3,\frac{1}{2};6,\frac{2}{5};3,-\frac{1}{2};6,-\frac{2}{5}}$$

$$Z^{JK_4\boxtimes\overline{JK_4}}_{e;\bar{m}} = \chi^{m4\times m6\times\overline{m4}\times\overline{m6}}_{1,0;1,0;3,-\frac{1}{2};4,-\frac{13}{8}} + \chi^{m4\times m6\times\overline{m4}\times\overline{m6}}_{1,0;10,\frac{7}{5};3,-\frac{1}{2};7,-\frac{1}{40}} + \chi^{m4\times m6\times\overline{m4}\times\overline{m6}}_{2,\frac{1}{16};1,0;2,-\frac{1}{16};2,-\frac{1}{8}} + \chi^{m4\times m6\times\overline{m4}\times\overline{m6}}_{2,\frac{1}{16};10,\frac{7}{5};2,-\frac{1}{16};9,-\frac{21}{40}}$$
$$+ \chi^{m4\times m6\times\overline{m4}\times\overline{m6}}_{3,\frac{1}{2};1,0;1,0;4,-\frac{13}{8}} + \chi^{m4\times m6\times\overline{m4}\times\overline{m6}}_{3,\frac{1}{2};10,\frac{7}{5};1,0;7,-\frac{1}{40}}$$

$$Z^{JK_4\boxtimes\overline{JK_4}}_{e;\bar{m}_1} = \chi^{m4\times m6\times\overline{m4}\times\overline{m6}}_{1,0;1,0;3,-\frac{1}{2};2,-\frac{1}{8}} + \chi^{m4\times m6\times\overline{m4}\times\overline{m6}}_{1,0;10,\frac{7}{5};3,-\frac{1}{2};9,-\frac{21}{40}} + \chi^{m4\times m6\times\overline{m4}\times\overline{m6}}_{2,\frac{1}{16};1,0;2,-\frac{1}{16};4,-\frac{13}{8}} + \chi^{m4\times m6\times\overline{m4}\times\overline{m6}}_{2,\frac{1}{16};10,\frac{7}{5};2,-\frac{1}{16};7,-\frac{1}{40}}$$
$$+ \chi^{m4\times m6\times\overline{m4}\times\overline{m6}}_{3,\frac{1}{2};1,0;1,0;2,-\frac{1}{8}} + \chi^{m4\times m6\times\overline{m4}\times\overline{m6}}_{3,\frac{1}{2};10,\frac{7}{5};1,0;9,-\frac{21}{40}}$$

$$Z^{JK_4\boxtimes\overline{JK_4}}_{e;\bar{q}} = \chi^{m4\times m6\times\overline{m4}\times\overline{m6}}_{1,0;5,3;1,0;3,-\frac{2}{3}} + \chi^{m4\times m6\times\overline{m4}\times\overline{m6}}_{1,0;6,\frac{2}{5};1,0;8,-\frac{1}{15}} + \chi^{m4\times m6\times\overline{m4}\times\overline{m6}}_{2,\frac{1}{16};5,3;2,-\frac{1}{16};3,-\frac{2}{3}} + \chi^{m4\times m6\times\overline{m4}\times\overline{m6}}_{2,\frac{1}{16};6,\frac{2}{5};2,-\frac{1}{16};8,-\frac{1}{15}}$$
$$+ \chi^{m4\times m6\times\overline{m4}\times\overline{m6}}_{3,\frac{1}{2};5,3;3,-\frac{1}{2};3,-\frac{2}{3}} + \chi^{m4\times m6\times\overline{m4}\times\overline{m6}}_{3,\frac{1}{2};6,\frac{2}{5};3,-\frac{1}{2};8,-\frac{1}{15}}$$

$$Z^{JK_4\boxtimes\overline{JK_4}}_{m;\mathbf{1}} = \chi^{m4\times m6\times\overline{m4}\times\overline{m6}}_{1,0;2,\frac{1}{8};1,0;5,-3} + \chi^{m4\times m6\times\overline{m4}\times\overline{m6}}_{1,0;9,\frac{21}{40};1,0;6,-\frac{2}{5}} + \chi^{m4\times m6\times\overline{m4}\times\overline{m6}}_{2,\frac{1}{16};2,\frac{1}{8};2,-\frac{1}{16};1,0} + \chi^{m4\times m6\times\overline{m4}\times\overline{m6}}_{2,\frac{1}{16};9,\frac{21}{40};2,-\frac{1}{16};10,-\frac{7}{5}}$$
$$+ \chi^{m4\times m6\times\overline{m4}\times\overline{m6}}_{3,\frac{1}{2};2,\frac{1}{8};3,-\frac{1}{2};5,-3} + \chi^{m4\times m6\times\overline{m4}\times\overline{m6}}_{3,\frac{1}{2};9,\frac{21}{40};3,-\frac{1}{2};6,-\frac{2}{5}}$$

$$Z^{JK_4\boxtimes\overline{JK_4}}_{m;\bar{e}} = \chi^{m4\times m6\times\overline{m4}\times\overline{m6}}_{1,0;2,\frac{1}{8};1,0;1,0} + \chi^{m4\times m6\times\overline{m4}\times\overline{m6}}_{1,0;9,\frac{21}{40};1,0;10,-\frac{7}{5}} + \chi^{m4\times m6\times\overline{m4}\times\overline{m6}}_{2,\frac{1}{16};2,\frac{1}{8};2,-\frac{1}{16};5,-3} + \chi^{m4\times m6\times\overline{m4}\times\overline{m6}}_{2,\frac{1}{16};9,\frac{21}{40};2,-\frac{1}{16};6,-\frac{2}{5}}$$
$$+ \chi^{m4\times m6\times\overline{m4}\times\overline{m6}}_{3,\frac{1}{2};2,\frac{1}{8};3,-\frac{1}{2};1,0} + \chi^{m4\times m6\times\overline{m4}\times\overline{m6}}_{3,\frac{1}{2};9,\frac{21}{40};3,-\frac{1}{2};10,-\frac{7}{5}}$$

$$Z^{JK_4\boxtimes\overline{JK_4}}_{m;\bar{m}} = \chi^{m4\times m6\times\overline{m4}\times\overline{m6}}_{1,0;4,\frac{13}{8};3,-\frac{1}{2};2,-\frac{1}{8}} + \chi^{m4\times m6\times\overline{m4}\times\overline{m6}}_{1,0;7,\frac{1}{40};3,-\frac{1}{2};9,-\frac{21}{40}} + \chi^{m4\times m6\times\overline{m4}\times\overline{m6}}_{2,\frac{1}{16};4,\frac{13}{8};2,-\frac{1}{16};4,-\frac{13}{8}} + \chi^{m4\times m6\times\overline{m4}\times\overline{m6}}_{2,\frac{1}{16};7,\frac{1}{40};2,-\frac{1}{16};7,-\frac{1}{40}}$$

$$+ \chi^{m4\times m6\times \overline{m4}\times \overline{m6}}_{3,\frac{1}{2};4,\frac{13}{8};1,0;2,-\frac{1}{8}} + \chi^{m4\times m6\times \overline{m4}\times \overline{m6}}_{3,\frac{1}{2};7,\frac{1}{40};1,0;9,-\frac{21}{40}}$$

$$Z^{JK_4\boxtimes \overline{JK_4}}_{m;\bar{m}_1} = \chi^{m4\times m6\times \overline{m4}\times \overline{m6}}_{1,0;4,\frac{13}{8};3,-\frac{1}{2};4,-\frac{13}{8}} + \chi^{m4\times m6\times \overline{m4}\times \overline{m6}}_{1,0;7,\frac{1}{40};3,-\frac{1}{2};7,-\frac{1}{40}} + \chi^{m4\times m6\times \overline{m4}\times \overline{m6}}_{2,\frac{1}{16};4,\frac{13}{8};2,-\frac{1}{16};2,-\frac{1}{8}} + \chi^{m4\times m6\times \overline{m4}\times \overline{m6}}_{2,\frac{1}{16};7,\frac{1}{40};2,-\frac{1}{16};9,-\frac{21}{40}}$$
$$+ \chi^{m4\times m6\times \overline{m4}\times \overline{m6}}_{3,\frac{1}{2};4,\frac{13}{8};1,0;4,-\frac{13}{8}} + \chi^{m4\times m6\times \overline{m4}\times \overline{m6}}_{3,\frac{1}{2};7,\frac{1}{40};1,0;7,-\frac{1}{40}}$$

$$Z^{JK_4\boxtimes \overline{JK_4}}_{m;\bar{q}} = \chi^{m4\times m6\times \overline{m4}\times \overline{m6}}_{1,0;2,\frac{1}{8};1,0;3,-\frac{2}{3}} + \chi^{m4\times m6\times \overline{m4}\times \overline{m6}}_{1,0;9,\frac{21}{40};1,0;8,-\frac{1}{15}} + \chi^{m4\times m6\times \overline{m4}\times \overline{m6}}_{2,\frac{1}{16};2,\frac{1}{8};2,-\frac{1}{16};3,-\frac{2}{3}} + \chi^{m4\times m6\times \overline{m4}\times \overline{m6}}_{2,\frac{1}{16};9,\frac{21}{40};2,-\frac{1}{16};8,-\frac{1}{15}}$$
$$+ \chi^{m4\times m6\times \overline{m4}\times \overline{m6}}_{3,\frac{1}{2};2,\frac{1}{8};3,-\frac{1}{2};3,-\frac{2}{3}} + \chi^{m4\times m6\times \overline{m4}\times \overline{m6}}_{3,\frac{1}{2};9,\frac{21}{40};3,-\frac{1}{2};8,-\frac{1}{15}}$$

$$Z^{JK_4\boxtimes \overline{JK_4}}_{m_1;\mathbf{1}} = \chi^{m4\times m6\times \overline{m4}\times \overline{m6}}_{1,0;4,\frac{13}{8};1,0;5,-3} + \chi^{m4\times m6\times \overline{m4}\times \overline{m6}}_{1,0;7,\frac{1}{40};1,0;6,-\frac{2}{5}} + \chi^{m4\times m6\times \overline{m4}\times \overline{m6}}_{2,\frac{1}{16};4,\frac{13}{8};2,-\frac{1}{16};1,0} + \chi^{m4\times m6\times \overline{m4}\times \overline{m6}}_{2,\frac{1}{16};7,\frac{1}{40};2,-\frac{1}{16};10,-\frac{7}{5}}$$
$$+ \chi^{m4\times m6\times \overline{m4}\times \overline{m6}}_{3,\frac{1}{2};4,\frac{13}{8};3,-\frac{1}{2};5,-3} + \chi^{m4\times m6\times \overline{m4}\times \overline{m6}}_{3,\frac{1}{2};7,\frac{1}{40};3,-\frac{1}{2};6,-\frac{2}{5}}$$

$$Z^{JK_4\boxtimes \overline{JK_4}}_{m_1;\bar{e}} = \chi^{m4\times m6\times \overline{m4}\times \overline{m6}}_{1,0;4,\frac{13}{8};1,0;1,0} + \chi^{m4\times m6\times \overline{m4}\times \overline{m6}}_{1,0;7,\frac{1}{40};1,0;10,-\frac{7}{5}} + \chi^{m4\times m6\times \overline{m4}\times \overline{m6}}_{2,\frac{1}{16};4,\frac{13}{8};2,-\frac{1}{16};5,-3} + \chi^{m4\times m6\times \overline{m4}\times \overline{m6}}_{2,\frac{1}{16};7,\frac{1}{40};2,-\frac{1}{16};6,-\frac{2}{5}}$$
$$+ \chi^{m4\times m6\times \overline{m4}\times \overline{m6}}_{3,\frac{1}{2};4,\frac{13}{8};3,-\frac{1}{2};1,0} + \chi^{m4\times m6\times \overline{m4}\times \overline{m6}}_{3,\frac{1}{2};7,\frac{1}{40};3,-\frac{1}{2};10,-\frac{7}{5}}$$

$$Z^{JK_4\boxtimes \overline{JK_4}}_{m_1;\bar{m}} = \chi^{m4\times m6\times \overline{m4}\times \overline{m6}}_{1,0;2,\frac{1}{8};3,-\frac{1}{2};2,-\frac{1}{8}} + \chi^{m4\times m6\times \overline{m4}\times \overline{m6}}_{1,0;9,\frac{21}{40};3,-\frac{1}{2};9,-\frac{21}{40}} + \chi^{m4\times m6\times \overline{m4}\times \overline{m6}}_{2,\frac{1}{16};2,\frac{1}{8};2,-\frac{1}{16};4,-\frac{13}{8}} + \chi^{m4\times m6\times \overline{m4}\times \overline{m6}}_{2,\frac{1}{16};9,\frac{21}{40};2,-\frac{1}{16};7,-\frac{1}{40}}$$
$$+ \chi^{m4\times m6\times \overline{m4}\times \overline{m6}}_{3,\frac{1}{2};2,\frac{1}{8};1,0;2,-\frac{1}{8}} + \chi^{m4\times m6\times \overline{m4}\times \overline{m6}}_{3,\frac{1}{2};9,\frac{21}{40};1,0;9,-\frac{21}{40}}$$

$$Z^{JK_4\boxtimes \overline{JK_4}}_{m_1;\bar{m}_1} = \chi^{m4\times m6\times \overline{m4}\times \overline{m6}}_{1,0;2,\frac{1}{8};3,-\frac{1}{2};4,-\frac{13}{8}} + \chi^{m4\times m6\times \overline{m4}\times \overline{m6}}_{1,0;9,\frac{21}{40};3,-\frac{1}{2};7,-\frac{1}{40}} + \chi^{m4\times m6\times \overline{m4}\times \overline{m6}}_{2,\frac{1}{16};2,\frac{1}{8};2,-\frac{1}{16};2,-\frac{1}{8}} + \chi^{m4\times m6\times \overline{m4}\times \overline{m6}}_{2,\frac{1}{16};9,\frac{21}{40};2,-\frac{1}{16};9,-\frac{21}{40}}$$
$$+ \chi^{m4\times m6\times \overline{m4}\times \overline{m6}}_{3,\frac{1}{2};2,\frac{1}{8};1,0;4,-\frac{13}{8}} + \chi^{m4\times m6\times \overline{m4}\times \overline{m6}}_{3,\frac{1}{2};9,\frac{21}{40};1,0;7,-\frac{1}{40}}$$

$$Z^{JK_4\boxtimes \overline{JK_4}}_{m_1;\bar{q}} = \chi^{m4\times m6\times \overline{m4}\times \overline{m6}}_{1,0;4,\frac{13}{8};1,0;3,-\frac{2}{3}} + \chi^{m4\times m6\times \overline{m4}\times \overline{m6}}_{1,0;7,\frac{1}{40};1,0;8,-\frac{1}{15}} + \chi^{m4\times m6\times \overline{m4}\times \overline{m6}}_{2,\frac{1}{16};4,\frac{13}{8};2,-\frac{1}{16};3,-\frac{2}{3}} + \chi^{m4\times m6\times \overline{m4}\times \overline{m6}}_{2,\frac{1}{16};7,\frac{1}{40};2,-\frac{1}{16};8,-\frac{1}{15}}$$
$$+ \chi^{m4\times m6\times \overline{m4}\times \overline{m6}}_{3,\frac{1}{2};4,\frac{13}{8};3,-\frac{1}{2};3,-\frac{2}{3}} + \chi^{m4\times m6\times \overline{m4}\times \overline{m6}}_{3,\frac{1}{2};7,\frac{1}{40};3,-\frac{1}{2};8,-\frac{1}{15}}$$

$$Z^{JK_4\boxtimes \overline{JK_4}}_{q;\mathbf{1}} = \chi^{m4\times m6\times \overline{m4}\times \overline{m6}}_{1,0;3,\frac{2}{3};1,0;1,0} + \chi^{m4\times m6\times \overline{m4}\times \overline{m6}}_{1,0;8,\frac{1}{15};1,0;10,-\frac{7}{5}} + \chi^{m4\times m6\times \overline{m4}\times \overline{m6}}_{2,\frac{1}{16};3,\frac{2}{3};2,-\frac{1}{16};5,-3} + \chi^{m4\times m6\times \overline{m4}\times \overline{m6}}_{2,\frac{1}{16};8,\frac{1}{15};2,-\frac{1}{16};6,-\frac{2}{5}}$$
$$+ \chi^{m4\times m6\times \overline{m4}\times \overline{m6}}_{3,\frac{1}{2};3,\frac{2}{3};3,-\frac{1}{2};1,0} + \chi^{m4\times m6\times \overline{m4}\times \overline{m6}}_{3,\frac{1}{2};8,\frac{1}{15};3,-\frac{1}{2};10,-\frac{7}{5}}$$

$$Z^{JK_4\boxtimes \overline{JK_4}}_{q;\bar{e}} = \chi^{m4\times m6\times \overline{m4}\times \overline{m6}}_{1,0;3,\frac{2}{3};1,0;5,-3} + \chi^{m4\times m6\times \overline{m4}\times \overline{m6}}_{1,0;8,\frac{1}{15};1,0;6,-\frac{2}{5}} + \chi^{m4\times m6\times \overline{m4}\times \overline{m6}}_{2,\frac{1}{16};3,\frac{2}{3};2,-\frac{1}{16};1,0} + \chi^{m4\times m6\times \overline{m4}\times \overline{m6}}_{2,\frac{1}{16};8,\frac{1}{15};2,-\frac{1}{16};10,-\frac{7}{5}}$$
$$+ \chi^{m4\times m6\times \overline{m4}\times \overline{m6}}_{3,\frac{1}{2};3,\frac{2}{3};3,-\frac{1}{2};5,-3} + \chi^{m4\times m6\times \overline{m4}\times \overline{m6}}_{3,\frac{1}{2};8,\frac{1}{15};3,-\frac{1}{2};6,-\frac{2}{5}}$$

$$Z^{JK_4\boxtimes \overline{JK_4}}_{q;\bar{m}} = \chi^{m4\times m6\times \overline{m4}\times \overline{m6}}_{1,0;3,\frac{2}{3};3,-\frac{1}{2};4,-\frac{13}{8}} + \chi^{m4\times m6\times \overline{m4}\times \overline{m6}}_{1,0;8,\frac{1}{15};3,-\frac{1}{2};7,-\frac{1}{40}} + \chi^{m4\times m6\times \overline{m4}\times \overline{m6}}_{2,\frac{1}{16};3,\frac{2}{3};2,-\frac{1}{16};2,-\frac{1}{8}} + \chi^{m4\times m6\times \overline{m4}\times \overline{m6}}_{2,\frac{1}{16};8,\frac{1}{15};2,-\frac{1}{16};9,-\frac{21}{40}}$$
$$+ \chi^{m4\times m6\times \overline{m4}\times \overline{m6}}_{3,\frac{1}{2};3,\frac{2}{3};1,0;4,-\frac{13}{8}} + \chi^{m4\times m6\times \overline{m4}\times \overline{m6}}_{3,\frac{1}{2};8,\frac{1}{15};1,0;7,-\frac{1}{40}}$$

$$Z^{JK_4\boxtimes \overline{JK_4}}_{q;\bar{m}_1} = \chi^{m4\times m6\times \overline{m4}\times \overline{m6}}_{1,0;3,\frac{2}{3};3,-\frac{1}{2};2,-\frac{1}{8}} + \chi^{m4\times m6\times \overline{m4}\times \overline{m6}}_{1,0;8,\frac{1}{15};3,-\frac{1}{2};9,-\frac{21}{40}} + \chi^{m4\times m6\times \overline{m4}\times \overline{m6}}_{2,\frac{1}{16};3,\frac{2}{3};2,-\frac{1}{16};4,-\frac{13}{8}} + \chi^{m4\times m6\times \overline{m4}\times \overline{m6}}_{2,\frac{1}{16};8,\frac{1}{15};2,-\frac{1}{16};7,-\frac{1}{40}}$$
$$+ \chi^{m4\times m6\times \overline{m4}\times \overline{m6}}_{3,\frac{1}{2};3,\frac{2}{3};1,0;2,-\frac{1}{8}} + \chi^{m4\times m6\times \overline{m4}\times \overline{m6}}_{3,\frac{1}{2};8,\frac{1}{15};1,0;9,-\frac{21}{40}}$$

$$Z^{JK_4\boxtimes \overline{JK_4}}_{q;\bar{q}} = \chi^{m4\times m6\times \overline{m4}\times \overline{m6}}_{1,0;3,\frac{2}{3};1,0;3,-\frac{2}{3}} + \chi^{m4\times m6\times \overline{m4}\times \overline{m6}}_{1,0;8,\frac{1}{15};1,0;8,-\frac{1}{15}} + \chi^{m4\times m6\times \overline{m4}\times \overline{m6}}_{2,\frac{1}{16};3,\frac{2}{3};2,-\frac{1}{16};3,-\frac{2}{3}} + \chi^{m4\times m6\times \overline{m4}\times \overline{m6}}_{2,\frac{1}{16};8,\frac{1}{15};2,-\frac{1}{16};8,-\frac{1}{15}}$$
$$+ \chi^{m4\times m6\times \overline{m4}\times \overline{m6}}_{3,\frac{1}{2};3,\frac{2}{3};3,-\frac{1}{2};3,-\frac{2}{3}} + \chi^{m4\times m6\times \overline{m4}\times \overline{m6}}_{3,\frac{1}{2};8,\frac{1}{15};3,-\frac{1}{2};8,-\frac{1}{15}}$$

$$\mathcal{A} = \mathbf{1}. \tag{H.16}$$

$$Z^{JK_4\boxtimes \overline{JK_4}}_{\mathbf{1};\mathbf{1}} = \color{blue}{\chi^{m4\times m6\times \overline{m4}\times \overline{m6}}_{1,0;1,0;1,0;1,0}} + \chi^{m4\times m6\times \overline{m4}\times \overline{m6}}_{1,0;10,\frac{7}{5};1,0;10,-\frac{7}{5}} + \chi^{m4\times m6\times \overline{m4}\times \overline{m6}}_{2,\frac{1}{16};1,0;2,-\frac{1}{16};5,-3} + \chi^{m4\times m6\times \overline{m4}\times \overline{m6}}_{2,\frac{1}{16};10,\frac{7}{5};2,-\frac{1}{16};6,-\frac{2}{5}}$$
$$+ \color{red}{\chi^{m4\times m6\times \overline{m4}\times \overline{m6}}_{3,\frac{1}{2};1,0;3,-\frac{1}{2};1,0}} + \chi^{m4\times m6\times \overline{m4}\times \overline{m6}}_{3,\frac{1}{2};10,\frac{7}{5};3,-\frac{1}{2};10,-\frac{7}{5}}$$

$$Z_{\mathbf{1};\bar{e}}^{JK_4 \boxtimes \overline{JK_4}} = \chi_{1,0;1,0;1,0;5,-3}^{m4\times m6\times\overline{m4}\times\overline{m6}} + \chi_{1,0;10,\frac{7}{5};1,0;6,-\frac{2}{5}}^{m4\times m6\times\overline{m4}\times\overline{m6}} + \chi_{2,\frac{1}{16};1,0;2,-\frac{1}{16};1,0}^{m4\times m6\times\overline{m4}\times\overline{m6}} + \chi_{2,\frac{1}{16};10,\frac{7}{5};2,-\frac{1}{16};10,-\frac{7}{5}}^{m4\times m6\times\overline{m4}\times\overline{m6}}$$
$$+ \chi_{3,\frac{1}{2};1,0;3,-\frac{1}{2};5,-3}^{m4\times m6\times\overline{m4}\times\overline{m6}} + \chi_{3,\frac{1}{2};10,\frac{7}{5};3,-\frac{1}{2};6,-\frac{2}{5}}^{m4\times m6\times\overline{m4}\times\overline{m6}}$$

$$Z_{\mathbf{1};\bar{m}}^{JK_4 \boxtimes \overline{JK_4}} = \chi_{1,0;1,0;3,-\frac{1}{2};4,-\frac{13}{8}}^{m4\times m6\times\overline{m4}\times\overline{m6}} + \chi_{1,0;10,\frac{7}{5};3,-\frac{1}{2};7,-\frac{1}{40}}^{m4\times m6\times\overline{m4}\times\overline{m6}} + \chi_{2,\frac{1}{16};1,0;2,-\frac{1}{16};2,-\frac{1}{8}}^{m4\times m6\times\overline{m4}\times\overline{m6}} + \chi_{2,\frac{1}{16};10,\frac{7}{5};2,-\frac{1}{16};9,-\frac{21}{40}}^{m4\times m6\times\overline{m4}\times\overline{m6}}$$
$$+ \chi_{3,\frac{1}{2};1,0;1,0;4,-\frac{13}{8}}^{m4\times m6\times\overline{m4}\times\overline{m6}} + \chi_{3,\frac{1}{2};10,\frac{7}{5};1,0;7,-\frac{1}{40}}^{m4\times m6\times\overline{m4}\times\overline{m6}}$$

$$Z_{\mathbf{1};\bar{m}_1}^{JK_4 \boxtimes \overline{JK_4}} = \chi_{1,0;1,0;3,-\frac{1}{2};2,-\frac{1}{8}}^{m4\times m6\times\overline{m4}\times\overline{m6}} + \chi_{1,0;10,\frac{7}{5};3,-\frac{1}{2};9,-\frac{21}{40}}^{m4\times m6\times\overline{m4}\times\overline{m6}} + \chi_{2,\frac{1}{16};1,0;2,-\frac{1}{16};4,-\frac{13}{8}}^{m4\times m6\times\overline{m4}\times\overline{m6}} + \chi_{2,\frac{1}{16};10,\frac{7}{5};2,-\frac{1}{16};7,-\frac{1}{40}}^{m4\times m6\times\overline{m4}\times\overline{m6}}$$
$$+ \chi_{3,\frac{1}{2};1,0;1,0;2,-\frac{1}{8}}^{m4\times m6\times\overline{m4}\times\overline{m6}} + \chi_{3,\frac{1}{2};10,\frac{7}{5};1,0;9,-\frac{21}{40}}^{m4\times m6\times\overline{m4}\times\overline{m6}}$$

$$Z_{\mathbf{1};\bar{q}}^{JK_4 \boxtimes \overline{JK_4}} = \chi_{1,0;1,0;1,0;3,-\frac{2}{3}}^{m4\times m6\times\overline{m4}\times\overline{m6}} + \chi_{1,0;10,\frac{7}{5};1,0;8,-\frac{1}{15}}^{m4\times m6\times\overline{m4}\times\overline{m6}} + \chi_{2,\frac{1}{16};1,0;2,-\frac{1}{16};3,-\frac{2}{3}}^{m4\times m6\times\overline{m4}\times\overline{m6}} + \chi_{2,\frac{1}{16};10,\frac{7}{5};2,-\frac{1}{16};8,-\frac{1}{15}}^{m4\times m6\times\overline{m4}\times\overline{m6}}$$
$$+ \chi_{3,\frac{1}{2};1,0;3,-\frac{1}{2};3,-\frac{2}{3}}^{m4\times m6\times\overline{m4}\times\overline{m6}} + \chi_{3,\frac{1}{2};10,\frac{7}{5};3,-\frac{1}{2};8,-\frac{1}{15}}^{m4\times m6\times\overline{m4}\times\overline{m6}}$$

$$Z_{e;\mathbf{1}}^{JK_4 \boxtimes \overline{JK_4}} = \chi_{1,0;5,3;1,0;1,0}^{m4\times m6\times\overline{m4}\times\overline{m6}} + \chi_{1,0;6,\frac{2}{5};1,0;10,-\frac{7}{5}}^{m4\times m6\times\overline{m4}\times\overline{m6}} + \chi_{2,\frac{1}{16};5,3;2,-\frac{1}{16};5,-3}^{m4\times m6\times\overline{m4}\times\overline{m6}} + \chi_{2,\frac{1}{16};6,\frac{2}{5};2,-\frac{1}{16};6,-\frac{2}{5}}^{m4\times m6\times\overline{m4}\times\overline{m6}}$$
$$+ \chi_{3,\frac{1}{2};5,3;3,-\frac{1}{2};1,0}^{m4\times m6\times\overline{m4}\times\overline{m6}} + \chi_{3,\frac{1}{2};6,\frac{2}{5};3,-\frac{1}{2};10,-\frac{7}{5}}^{m4\times m6\times\overline{m4}\times\overline{m6}}$$

$$Z_{e;\bar{e}}^{JK_4 \boxtimes \overline{JK_4}} = \chi_{1,0;5,3;1,0;5,-3}^{m4\times m6\times\overline{m4}\times\overline{m6}} + \chi_{1,0;6,\frac{2}{5};1,0;6,-\frac{2}{5}}^{m4\times m6\times\overline{m4}\times\overline{m6}} + \chi_{2,\frac{1}{16};5,3;2,-\frac{1}{16};1,0}^{m4\times m6\times\overline{m4}\times\overline{m6}} + \chi_{2,\frac{1}{16};6,\frac{2}{5};2,-\frac{1}{16};10,-\frac{7}{5}}^{m4\times m6\times\overline{m4}\times\overline{m6}}$$
$$+ \chi_{3,\frac{1}{2};5,3;3,-\frac{1}{2};5,-3}^{m4\times m6\times\overline{m4}\times\overline{m6}} + \chi_{3,\frac{1}{2};6,\frac{2}{5};3,-\frac{1}{2};6,-\frac{2}{5}}^{m4\times m6\times\overline{m4}\times\overline{m6}}$$

$$Z_{e;\bar{m}}^{JK_4 \boxtimes \overline{JK_4}} = \chi_{1,0;5,3;3,-\frac{1}{2};4,-\frac{13}{8}}^{m4\times m6\times\overline{m4}\times\overline{m6}} + \chi_{1,0;6,\frac{2}{5};3,-\frac{1}{2};7,-\frac{1}{40}}^{m4\times m6\times\overline{m4}\times\overline{m6}} + \chi_{2,\frac{1}{16};5,3;2,-\frac{1}{16};2,-\frac{1}{8}}^{m4\times m6\times\overline{m4}\times\overline{m6}} + \chi_{2,\frac{1}{16};6,\frac{2}{5};2,-\frac{1}{16};9,-\frac{21}{40}}^{m4\times m6\times\overline{m4}\times\overline{m6}}$$
$$+ \chi_{3,\frac{1}{2};5,3;1,0;4,-\frac{13}{8}}^{m4\times m6\times\overline{m4}\times\overline{m6}} + \chi_{3,\frac{1}{2};6,\frac{2}{5};1,0;7,-\frac{1}{40}}^{m4\times m6\times\overline{m4}\times\overline{m6}}$$

$$Z_{e;\bar{m}_1}^{JK_4 \boxtimes \overline{JK_4}} = \chi_{1,0;5,3;3,-\frac{1}{2};2,-\frac{1}{8}}^{m4\times m6\times\overline{m4}\times\overline{m6}} + \chi_{1,0;6,\frac{2}{5};3,-\frac{1}{2};9,-\frac{21}{40}}^{m4\times m6\times\overline{m4}\times\overline{m6}} + \chi_{2,\frac{1}{16};5,3;2,-\frac{1}{16};4,-\frac{13}{8}}^{m4\times m6\times\overline{m4}\times\overline{m6}} + \chi_{2,\frac{1}{16};6,\frac{2}{5};2,-\frac{1}{16};7,-\frac{1}{40}}^{m4\times m6\times\overline{m4}\times\overline{m6}}$$
$$+ \chi_{3,\frac{1}{2};5,3;1,0;2,-\frac{1}{8}}^{m4\times m6\times\overline{m4}\times\overline{m6}} + \chi_{3,\frac{1}{2};6,\frac{2}{5};1,0;9,-\frac{21}{40}}^{m4\times m6\times\overline{m4}\times\overline{m6}}$$

$$Z_{e;\bar{q}}^{JK_4 \boxtimes \overline{JK_4}} = \chi_{1,0;5,3;1,0;3,-\frac{2}{3}}^{m4\times m6\times\overline{m4}\times\overline{m6}} + \chi_{1,0;6,\frac{2}{5};1,0;8,-\frac{1}{15}}^{m4\times m6\times\overline{m4}\times\overline{m6}} + \chi_{2,\frac{1}{16};5,3;2,-\frac{1}{16};3,-\frac{2}{3}}^{m4\times m6\times\overline{m4}\times\overline{m6}} + \chi_{2,\frac{1}{16};6,\frac{2}{5};2,-\frac{1}{16};8,-\frac{1}{15}}^{m4\times m6\times\overline{m4}\times\overline{m6}}$$
$$+ \chi_{3,\frac{1}{2};5,3;3,-\frac{1}{2};3,-\frac{2}{3}}^{m4\times m6\times\overline{m4}\times\overline{m6}} + \chi_{3,\frac{1}{2};6,\frac{2}{5};3,-\frac{1}{2};8,-\frac{1}{15}}^{m4\times m6\times\overline{m4}\times\overline{m6}}$$

$$Z_{m;\mathbf{1}}^{JK_4 \boxtimes \overline{JK_4}} = \chi_{1,0;2,\frac{1}{8};1,0;1,0}^{m4\times m6\times\overline{m4}\times\overline{m6}} + \chi_{1,0;9,\frac{21}{40};1,0;10,-\frac{7}{5}}^{m4\times m6\times\overline{m4}\times\overline{m6}} + \chi_{2,\frac{1}{16};2,\frac{1}{8};2,-\frac{1}{16};5,-3}^{m4\times m6\times\overline{m4}\times\overline{m6}} + \chi_{2,\frac{1}{16};9,\frac{21}{40};2,-\frac{1}{16};6,-\frac{2}{5}}^{m4\times m6\times\overline{m4}\times\overline{m6}}$$
$$+ \chi_{3,\frac{1}{2};2,\frac{1}{8};3,-\frac{1}{2};1,0}^{m4\times m6\times\overline{m4}\times\overline{m6}} + \chi_{3,\frac{1}{2};9,\frac{21}{40};3,-\frac{1}{2};10,-\frac{7}{5}}^{m4\times m6\times\overline{m4}\times\overline{m6}}$$

$$Z_{m;\bar{e}}^{JK_4 \boxtimes \overline{JK_4}} = \chi_{1,0;2,\frac{1}{8};1,0;5,-3}^{m4\times m6\times\overline{m4}\times\overline{m6}} + \chi_{1,0;9,\frac{21}{40};1,0;6,-\frac{2}{5}}^{m4\times m6\times\overline{m4}\times\overline{m6}} + \chi_{2,\frac{1}{16};2,\frac{1}{8};2,-\frac{1}{16};1,0}^{m4\times m6\times\overline{m4}\times\overline{m6}} + \chi_{2,\frac{1}{16};9,\frac{21}{40};2,-\frac{1}{16};10,-\frac{7}{5}}^{m4\times m6\times\overline{m4}\times\overline{m6}}$$
$$+ \chi_{3,\frac{1}{2};2,\frac{1}{8};3,-\frac{1}{2};5,-3}^{m4\times m6\times\overline{m4}\times\overline{m6}} + \chi_{3,\frac{1}{2};9,\frac{21}{40};3,-\frac{1}{2};6,-\frac{2}{5}}^{m4\times m6\times\overline{m4}\times\overline{m6}}$$

$$Z_{m;\bar{m}}^{JK_4 \boxtimes \overline{JK_4}} = \chi_{1,0;2,\frac{1}{8};3,-\frac{1}{2};4,-\frac{13}{8}}^{m4\times m6\times\overline{m4}\times\overline{m6}} + \chi_{1,0;9,\frac{21}{40};3,-\frac{1}{2};7,-\frac{1}{40}}^{m4\times m6\times\overline{m4}\times\overline{m6}} + \chi_{2,\frac{1}{16};2,\frac{1}{8};2,-\frac{1}{16};2,-\frac{1}{8}}^{m4\times m6\times\overline{m4}\times\overline{m6}} + \chi_{2,\frac{1}{16};9,\frac{21}{40};2,-\frac{1}{16};9,-\frac{21}{40}}^{m4\times m6\times\overline{m4}\times\overline{m6}}$$
$$+ \chi_{3,\frac{1}{2};2,\frac{1}{8};1,0;4,-\frac{13}{8}}^{m4\times m6\times\overline{m4}\times\overline{m6}} + \chi_{3,\frac{1}{2};9,\frac{21}{40};1,0;7,-\frac{1}{40}}^{m4\times m6\times\overline{m4}\times\overline{m6}}$$

$$Z_{m;\bar{m}_1}^{JK_4 \boxtimes \overline{JK_4}} = \chi_{1,0;2,\frac{1}{8};3,-\frac{1}{2};2,-\frac{1}{8}}^{m4\times m6\times\overline{m4}\times\overline{m6}} + \chi_{1,0;9,\frac{21}{40};3,-\frac{1}{2};9,-\frac{21}{40}}^{m4\times m6\times\overline{m4}\times\overline{m6}} + \chi_{2,\frac{1}{16};2,\frac{1}{8};2,-\frac{1}{16};4,-\frac{13}{8}}^{m4\times m6\times\overline{m4}\times\overline{m6}} + \chi_{2,\frac{1}{16};9,\frac{21}{40};2,-\frac{1}{16};7,-\frac{1}{40}}^{m4\times m6\times\overline{m4}\times\overline{m6}}$$
$$+ \chi_{3,\frac{1}{2};2,\frac{1}{8};1,0;2,-\frac{1}{8}}^{m4\times m6\times\overline{m4}\times\overline{m6}} + \chi_{3,\frac{1}{2};9,\frac{21}{40};1,0;9,-\frac{21}{40}}^{m4\times m6\times\overline{m4}\times\overline{m6}}$$

$$Z_{m;\bar{q}}^{JK_4 \boxtimes \overline{JK_4}} = \chi_{1,0;2,\frac{1}{8};1,0;3,-\frac{2}{3}}^{m4\times m6\times\overline{m4}\times\overline{m6}} + \chi_{1,0;9,\frac{21}{40};1,0;8,-\frac{1}{15}}^{m4\times m6\times\overline{m4}\times\overline{m6}} + \chi_{2,\frac{1}{16};2,\frac{1}{8};2,-\frac{1}{16};3,-\frac{2}{3}}^{m4\times m6\times\overline{m4}\times\overline{m6}} + \chi_{2,\frac{1}{16};9,\frac{21}{40};2,-\frac{1}{16};8,-\frac{1}{15}}^{m4\times m6\times\overline{m4}\times\overline{m6}}$$
$$+ \chi_{3,\frac{1}{2};2,\frac{1}{8};3,-\frac{1}{2};3,-\frac{2}{3}}^{m4\times m6\times\overline{m4}\times\overline{m6}} + \chi_{3,\frac{1}{2};9,\frac{21}{40};3,-\frac{1}{2};8,-\frac{1}{15}}^{m4\times m6\times\overline{m4}\times\overline{m6}}$$

$$Z_{m_1;\mathbf{1}}^{JK_4 \boxtimes \overline{JK_4}} = \chi_{1,0;4,\frac{13}{8};1,0;1,0}^{m4\times m6\times \overline{m4}\times \overline{m6}} + \chi_{1,0;7,\frac{1}{40};1,0;10,-\frac{7}{5}}^{m4\times m6\times \overline{m4}\times \overline{m6}} + \chi_{2,\frac{1}{16};4,\frac{13}{8};2,-\frac{1}{16};5,-3}^{m4\times m6\times \overline{m4}\times \overline{m6}} + \chi_{2,\frac{1}{16};7,\frac{1}{40};2,-\frac{1}{16};6,-\frac{2}{5}}^{m4\times m6\times \overline{m4}\times \overline{m6}}$$
$$+ \chi_{3,\frac{1}{2};4,\frac{13}{8};3,-\frac{1}{2};1,0}^{m4\times m6\times \overline{m4}\times \overline{m6}} + \chi_{3,\frac{1}{2};7,\frac{1}{40};3,-\frac{1}{2};10,-\frac{7}{5}}^{m4\times m6\times \overline{m4}\times \overline{m6}}$$

$$Z_{m_1;\bar{e}}^{JK_4 \boxtimes \overline{JK_4}} = \chi_{1,0;4,\frac{13}{8};1,0;5,-3}^{m4\times m6\times \overline{m4}\times \overline{m6}} + \chi_{1,0;7,\frac{1}{40};1,0;6,-\frac{2}{5}}^{m4\times m6\times \overline{m4}\times \overline{m6}} + \chi_{2,\frac{1}{16};4,\frac{13}{8};2,-\frac{1}{16};1,0}^{m4\times m6\times \overline{m4}\times \overline{m6}} + \chi_{2,\frac{1}{16};7,\frac{1}{40};2,-\frac{1}{16};10,-\frac{7}{5}}^{m4\times m6\times \overline{m4}\times \overline{m6}}$$
$$+ \chi_{3,\frac{1}{2};4,\frac{13}{8};3,-\frac{1}{2};5,-3}^{m4\times m6\times \overline{m4}\times \overline{m6}} + \chi_{3,\frac{1}{2};7,\frac{1}{40};3,-\frac{1}{2};6,-\frac{2}{5}}^{m4\times m6\times \overline{m4}\times \overline{m6}}$$

$$Z_{m_1;\bar{m}}^{JK_4 \boxtimes \overline{JK_4}} = \chi_{1,0;4,\frac{13}{8};3,-\frac{1}{2};4,-\frac{13}{8}}^{m4\times m6\times \overline{m4}\times \overline{m6}} + \chi_{1,0;7,\frac{1}{40};3,-\frac{1}{2};7,-\frac{1}{40}}^{m4\times m6\times \overline{m4}\times \overline{m6}} + \chi_{2,\frac{1}{16};4,\frac{13}{8};2,-\frac{1}{16};2,-\frac{1}{8}}^{m4\times m6\times \overline{m4}\times \overline{m6}} + \chi_{2,\frac{1}{16};7,\frac{1}{40};2,-\frac{1}{16};9,-\frac{21}{40}}^{m4\times m6\times \overline{m4}\times \overline{m6}}$$
$$+ \chi_{3,\frac{1}{2};4,\frac{13}{8};1,0;4,-\frac{13}{8}}^{m4\times m6\times \overline{m4}\times \overline{m6}} + \chi_{3,\frac{1}{2};7,\frac{1}{40};1,0;7,-\frac{1}{40}}^{m4\times m6\times \overline{m4}\times \overline{m6}}$$

$$Z_{m_1;\bar{m}_1}^{JK_4 \boxtimes \overline{JK_4}} = \chi_{1,0;4,\frac{13}{8};3,-\frac{1}{2};2,-\frac{1}{8}}^{m4\times m6\times \overline{m4}\times \overline{m6}} + \chi_{1,0;7,\frac{1}{40};3,-\frac{1}{2};9,-\frac{21}{40}}^{m4\times m6\times \overline{m4}\times \overline{m6}} + \chi_{2,\frac{1}{16};4,\frac{13}{8};2,-\frac{1}{16};4,-\frac{13}{8}}^{m4\times m6\times \overline{m4}\times \overline{m6}} + \chi_{2,\frac{1}{16};7,\frac{1}{40};2,-\frac{1}{16};7,-\frac{1}{40}}^{m4\times m6\times \overline{m4}\times \overline{m6}}$$
$$+ \chi_{3,\frac{1}{2};4,\frac{13}{8};1,0;2,-\frac{1}{8}}^{m4\times m6\times \overline{m4}\times \overline{m6}} + \chi_{3,\frac{1}{2};7,\frac{1}{40};1,0;9,-\frac{21}{40}}^{m4\times m6\times \overline{m4}\times \overline{m6}}$$

$$Z_{m_1;\bar{q}}^{JK_4 \boxtimes \overline{JK_4}} = \chi_{1,0;4,\frac{13}{8};1,0;3,-\frac{2}{3}}^{m4\times m6\times \overline{m4}\times \overline{m6}} + \chi_{1,0;7,\frac{1}{40};1,0;8,-\frac{1}{15}}^{m4\times m6\times \overline{m4}\times \overline{m6}} + \chi_{2,\frac{1}{16};4,\frac{13}{8};2,-\frac{1}{16};3,-\frac{2}{3}}^{m4\times m6\times \overline{m4}\times \overline{m6}} + \chi_{2,\frac{1}{16};7,\frac{1}{40};2,-\frac{1}{16};8,-\frac{1}{15}}^{m4\times m6\times \overline{m4}\times \overline{m6}}$$
$$+ \chi_{3,\frac{1}{2};4,\frac{13}{8};3,-\frac{1}{2};3,-\frac{2}{3}}^{m4\times m6\times \overline{m4}\times \overline{m6}} + \chi_{3,\frac{1}{2};7,\frac{1}{40};3,-\frac{1}{2};8,-\frac{1}{15}}^{m4\times m6\times \overline{m4}\times \overline{m6}}$$

$$Z_{q;\mathbf{1}}^{JK_4 \boxtimes \overline{JK_4}} = \chi_{1,0;3,\frac{2}{3};1,0;1,0}^{m4\times m6\times \overline{m4}\times \overline{m6}} + \chi_{1,0;8,\frac{1}{15};1,0;10,-\frac{7}{5}}^{m4\times m6\times \overline{m4}\times \overline{m6}} + \chi_{2,\frac{1}{16};3,\frac{2}{3};2,-\frac{1}{16};5,-3}^{m4\times m6\times \overline{m4}\times \overline{m6}} + \chi_{2,\frac{1}{16};8,\frac{1}{15};2,-\frac{1}{16};6,-\frac{2}{5}}^{m4\times m6\times \overline{m4}\times \overline{m6}}$$
$$+ \chi_{3,\frac{1}{2};3,\frac{2}{3};3,-\frac{1}{2};1,0}^{m4\times m6\times \overline{m4}\times \overline{m6}} + \chi_{3,\frac{1}{2};8,\frac{1}{15};3,-\frac{1}{2};10,-\frac{7}{5}}^{m4\times m6\times \overline{m4}\times \overline{m6}}$$

$$Z_{q;\bar{e}}^{JK_4 \boxtimes \overline{JK_4}} = \chi_{1,0;3,\frac{2}{3};1,0;5,-3}^{m4\times m6\times \overline{m4}\times \overline{m6}} + \chi_{1,0;8,\frac{1}{15};1,0;6,-\frac{2}{5}}^{m4\times m6\times \overline{m4}\times \overline{m6}} + \chi_{2,\frac{1}{16};3,\frac{2}{3};2,-\frac{1}{16};1,0}^{m4\times m6\times \overline{m4}\times \overline{m6}} + \chi_{2,\frac{1}{16};8,\frac{1}{15};2,-\frac{1}{16};10,-\frac{7}{5}}^{m4\times m6\times \overline{m4}\times \overline{m6}}$$
$$+ \chi_{3,\frac{1}{2};3,\frac{2}{3};3,-\frac{1}{2};5,-3}^{m4\times m6\times \overline{m4}\times \overline{m6}} + \chi_{3,\frac{1}{2};8,\frac{1}{15};3,-\frac{1}{2};6,-\frac{2}{5}}^{m4\times m6\times \overline{m4}\times \overline{m6}}$$

$$Z_{q;\bar{m}}^{JK_4 \boxtimes \overline{JK_4}} = \chi_{1,0;3,\frac{2}{3};3,-\frac{1}{2};4,-\frac{13}{8}}^{m4\times m6\times \overline{m4}\times \overline{m6}} + \chi_{1,0;8,\frac{1}{15};3,-\frac{1}{2};7,-\frac{1}{40}}^{m4\times m6\times \overline{m4}\times \overline{m6}} + \chi_{2,\frac{1}{16};3,\frac{2}{3};2,-\frac{1}{16};2,-\frac{1}{8}}^{m4\times m6\times \overline{m4}\times \overline{m6}} + \chi_{2,\frac{1}{16};8,\frac{1}{15};2,-\frac{1}{16};9,-\frac{21}{40}}^{m4\times m6\times \overline{m4}\times \overline{m6}}$$
$$+ \chi_{3,\frac{1}{2};3,\frac{2}{3};1,0;4,-\frac{13}{8}}^{m4\times m6\times \overline{m4}\times \overline{m6}} + \chi_{3,\frac{1}{2};8,\frac{1}{15};1,0;7,-\frac{1}{40}}^{m4\times m6\times \overline{m4}\times \overline{m6}}$$

$$Z_{q;\bar{m}_1}^{JK_4 \boxtimes \overline{JK_4}} = \chi_{1,0;3,\frac{2}{3};3,-\frac{1}{2};2,-\frac{1}{8}}^{m4\times m6\times \overline{m4}\times \overline{m6}} + \chi_{1,0;8,\frac{1}{15};3,-\frac{1}{2};9,-\frac{21}{40}}^{m4\times m6\times \overline{m4}\times \overline{m6}} + \chi_{2,\frac{1}{16};3,\frac{2}{3};2,-\frac{1}{16};4,-\frac{13}{8}}^{m4\times m6\times \overline{m4}\times \overline{m6}} + \chi_{2,\frac{1}{16};8,\frac{1}{15};2,-\frac{1}{16};7,-\frac{1}{40}}^{m4\times m6\times \overline{m4}\times \overline{m6}}$$
$$+ \chi_{3,\frac{1}{2};3,\frac{2}{3};1,0;2,-\frac{1}{8}}^{m4\times m6\times \overline{m4}\times \overline{m6}} + \chi_{3,\frac{1}{2};8,\frac{1}{15};1,0;9,-\frac{21}{40}}^{m4\times m6\times \overline{m4}\times \overline{m6}}$$

$$Z_{q;\bar{q}}^{JK_4 \boxtimes \overline{JK_4}} = \chi_{1,0;3,\frac{2}{3};1,0;3,-\frac{2}{3}}^{m4\times m6\times \overline{m4}\times \overline{m6}} + \chi_{1,0;8,\frac{1}{15};1,0;8,-\frac{1}{15}}^{m4\times m6\times \overline{m4}\times \overline{m6}} + \chi_{2,\frac{1}{16};3,\frac{2}{3};2,-\frac{1}{16};3,-\frac{2}{3}}^{m4\times m6\times \overline{m4}\times \overline{m6}} + \chi_{2,\frac{1}{16};8,\frac{1}{15};2,-\frac{1}{16};8,-\frac{1}{15}}^{m4\times m6\times \overline{m4}\times \overline{m6}}$$
$$+ \chi_{3,\frac{1}{2};3,\frac{2}{3};3,-\frac{1}{2};3,-\frac{2}{3}}^{m4\times m6\times \overline{m4}\times \overline{m6}} + \chi_{3,\frac{1}{2};8,\frac{1}{15};3,-\frac{1}{2};8,-\frac{1}{15}}^{m4\times m6\times \overline{m4}\times \overline{m6}}$$

$$\mathcal{A} = \mathbf{1}. \tag{H.17}$$

Self-dual $S_3$ symmetric model also has the gapless states described by $m5 \times \overline{m5}$ CFT. But those gapless states have two or more symmetric relevant operators.

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
