# Peer review of "Quantum Phases and Transitions in Spin Chains with Non-Invertible Symmetries"

_SciPost Physics_

## Round 1 · Referee Report · Anonymous · 2024-9-4

Report

This is a good paper which should be published. The goal is to find and study examples of explicit lattice models that have exact realizations of non-invertible symmetry, which can then be used to understand the spontaneous breaking of such symmetries, and transitions between such phases. Despite the generally forbidding subject of non-invertible symmetry, this paper is mostly clear and concrete and readable -- although it touches on some mathematical machinery that is generally unfamiliar to physicists, it is concrete enough to be understandable. The paper is very long, but meaningful contact is made between the numerical parts of the paper (2 and 3) and the later formal section (4).

-- from the abstract:
"Such symmetries appear generically in gapless states of quantum matter constraining the low-energy dynamics. "

I expect that this statement about non-invertible symmetries might not be true.
We know that they appear in rational CFT and in a few other examples of gapless field theories (including nonlinear sigma models of goldstone modes [64]). But I don't see any reason to believe that they will appear even in generic 1+1d CFTs (such as Calabi-Yau sigma models at generic points in the moduli space).

-- The section called "relation to previous work" is very helpful to the reader and much appreciated.

-- When one says that a category is "anomaly-free " (e.g. top of page 4)
I guess this means just that all the quantum dimensions are integers? Or is it a stronger condition? Some explanation would be helpful.
[I eventually see that this notion is defined in footnote 33.]

-- I would like to discourage the use of the adjective "holographic" to describe the ideas related to "SymTO" or "SymTFT". The word "holographic" has been extensively used in the context of gauge/gravity duality, which is (conjectured to be) an exact equivalence between a d-dimensional theory without gravity and a d+1 (or more) dimenisonal theory with gravity (the Hilbert spaces are supposed to be the same; the dynamics are supposed to be the same). As far as I understand, the relation between SymTO and the lower-dimensional theory whose symmetries it encodes is not such an equivalence at all.

-- It would be good if the authors explain their use of the term "incommensurate" early on in the paper (around page 7). I see that a definition appears eventually on page 44. Is there actually spontaneous breaking of internal and translation symmetry in these states?

-- The generalization of the work of Ref [74], making completely explicit duality operators for the Z_3 case, is nice.

-- I didn't understand what is being shown in Figure 9.

-- I didn't understand yet the explanation on page 47 for why, in this "incommensurate phase", the numerically-inferred central charge to varies across the phase diagram. I see that a variation with parameters of the $1/L$ term in the ground state energy would lead to a variation of the inferred $cv$, but wouldn't that simply contradict equation 5.9? What does the groundstate quasimomentum have to do with it?

-- "A related discussion on the classification of gravitational anomalies and anomalies of group- like symmetries can be found in Refs. [26,36])"

I think there is no open parenthesis.

-- page 11: "Therefore, the duality holds when both conditions (2.12a) and (2.12b)."
should be
"Therefore, the duality holds when both conditions (2.12a) and (2.12b) hold."

-- page 48: "this gapless phase turns into and incommensurate phase"
should be
"this gapless phase turns into an incommensurate phase"

Recommendation

Publish (easily meets expectations and criteria for this Journal; among top 50%)

---

## Round 1 · Referee Report · Anonymous · 2024-9-8

Report

The manuscript construct concrete examples of 1+1d lattice models with non-invertible symmetry, such as Rep(S3) , by gauging invertible symmetries, and discuss their phase diagrams. In addition, the authors construct the corresponding 2+1d bulk topological orders describing the symmetries. I recommend the manuscript for publication.

Recommendation

Publish (meets expectations and criteria for this Journal)

---

## Editorial Decision

awaiting_resubmission